

**Simulation of ozone-vegetation coupling and feedback in**
**China using multiple ozone damage schemes**
Jiachen Cao[1], Xu Yue[1*], Mingrui Ma[2]
1. Jiangsu Key Laboratory of Atmospheric Environment Monitoring and Pollution
Control, Collaborative Innovation Center of Atmospheric Environment and
Equipment Technology, School of Environmental Science and Engineering, Nanjing
University of Information Science & Technology (NUIST), Nanjing, 210044, China
2. State Key Laboratory of Pollution Control and Resource Reuse, School of the
Environment, Nanjing University, Nanjing, 210044, China
*Corresponding author: Xu Yue
email: *yuexu@nuist.edu.cn*





**Abstract**

As a phytotoxic pollutant, surface ozone ($O_3$) not only affects plant physiology but
also influences meteorological fields and air quality by altering leaf stomatal
functions. Previous studies revealed strong feedbacks of $O_3$-vegetation coupling in
China but with large uncertainties due to the applications of varied $O_3$ damage
schemes and chemistry-vegetation models. In this study, we quantify the $O_3$
vegetation damage and the consequent feedbacks to surface meteorology and air
quality in China by coupling two $O_3$ damage schemes (S2007 vs. L2013) into a fully
coupled regional meteorology-chemistry model. With different schemes and
damaging sensitivities, surface $O_3$ is predicted to decrease summertime gross primary
productivity by 5.5%-21.4% and transpiration by 5.4%-23.2% in China, in which the
L2013 scheme yields 2.5-4 times of losses relative to the S2007 scheme. The damages
to photosynthesis of sunlit leaves are ~2.6 times that of shaded leaves in the S2007
scheme but show limited differences in the L2013 scheme. Though with large
discrepancies in offline responses, the two schemes yield similar magnitude of
feedback to surface meteorology and $O_3$ air quality. The $O_3$-induced damage to
transpiration increases national sensible heat by 3.2-6.0 W m$^{-2}$ (8.9% to 16.2%) while
reduces latent heat by 3.3-6.4 W m$^{-2}$ (-5.6% to -17.4%), leading to a 0.2-0.51 °C
increase in surface air temperature and a 2.2-3.9% reduction in relative humidity.
Meanwhile, surface $O_3$ concentrations on average increase by 1.3-3.3 μg m$^{-3}$ due to
the inhibitions of stomatal uptake and the anomalous enhancement in isoprene
emissions, the latter of which is attributed to the surface warming by $O_3$-vegetaion
coupling. Our results highlight the importance of $O_3$ control in China due to its
adverse effects on ecosystem functions, deterioration of global warming, and
exacerbation of $O_3$ pollution through the $O_3$-vegetation coupling.

**Keywords:** Ozone, vegetation, feedback, meteorology, air quality, regional model



**1 Introduction**

Surface ozone ($O_3$) is one of the most enduring air pollutants affecting air quality in China, with detrimental effects on human health and ecosystem functions (Monk et al., 2015). Long-term observations and numerical simulations have shown that $O_3$ affects stomatal conductance (Li et al., 2017), accelerates vegetation aging (Feng et al., 2015), and reduces photosynthesis (Wittig et al., 2007). These negative effects altered carbon allocation (Yue and Unger, 2014; Lombardozzi et al., 2015) and inhibited plant growth (Li et al., 2016), leading to a decreased strength of ecosystem carbon uptake (Ainsworth, 2012). Moreover, these effects have profound implications for global/regional climate and atmospheric environment. Given the significant ecological impacts, a systematic quantification of the $O_3$ vegetation damage effect in China is of great importance for the better understanding of the side effects of $O_3$ pollution on both regional carbon uptake and climate change.

At present, field experiments on $O_3$-induced vegetation damage have been conducted in China but were mostly confined to individual monitoring sites. For instance, Su et al. (2017) conducted experiments on grassland in Inner Mongolia and found that elevated $O_3$ concentrations resulted in a decrease of approximately 20% in the photosynthetic rate of herbaceous plants. Meta-analysis of tropical, subtropical, and temperate tree species in China found that increased $O_3$ concentrations reduced net photosynthesis and total biomass of Chinese woody plants by 28% and 14%, respectively (Li et al., 2017). However, most of these experiments were conducted using open-top chambers with artificially controlled $O_3$ concentrations, rather than actual surface $O_3$ concentrations, making it difficult to quantitatively estimate the impact of ambient $O_3$ on vegetation productivity. Furthermore, the spatial coverage of field experiments is limited, which hinders the direct use of observational data for assessing $O_3$ vegetation damage in different regions of China.

Alternatively, numerical models provide a more feasible approach to quantify the $O_3$-induced vegetation damage from the regional to global scales. Currently, there are three main parameterizations for the calculation of ozone vegetation damage. Felzer et



al. (2004) established an empirical scheme based on the Accumulated Ozone exposure
over a Threshold of 40 ppb (AOT40) within the framework of a terrestrial ecosystem
model. They further estimated that $O_3$ pollution in the United States led to a decrease
in net primary productivity (NPP) by 2.6% to 6.8% during the period of 1980-1990.
However, the AOT40 is related to $O_3$ concentrations alone and ignores the biological
regulations on the $O_3$ stomatal uptake, leading to inconsistent tendencies between $O_3$
pollution level and plant damage at the drought conditions (Gong et al., 2021). In
acknowledge of such deficit, Sitch et al. (2007) proposed a semi-mechanistic scheme
calculating $O_3$ vegetation damage based on the stomatal uptake of $O_3$ fluxes and the
coupling between stomatal conductance and leaf photosynthesis. Yue and Unger
(2014) implemented this scheme into the Yale Interactive terrestrial Biosphere (YIBs)
model. Taking into account varied $O_3$ sensitivities of different vegetation types, they
estimated that surface $O_3$ led to reductions of 2-5% in the summer gross primary
productivity (GPP) in eastern U.S. from 1998 to 2007. Later, Lombardozzi et al.
(2013) conducted a meta-analysis using published chamber data and found different
levels of responses to $O_3$ exposure between stomatal conductance and photosynthesis.
They further implemented the independent response relationships into the Community
Land Model (CLM) and estimated that current ozone levels led to a reduction in
global GPP by 8%-12% (Lombardozzi et al., 2015).

The $O_3$ stress on vegetation physiology can feed back to affect regional climate.

Lombardozzi et al. (2015) employed the CLM model and found that current $O_3$
exposure reduced transpiration by 2%-2.4% globally and up to 15% regionally over
eastern U.S., Europe, and Southeast Asia, leading to further perturbations in surface
energy and runoff. In U.S., Li et al. (2016) found that the $O_3$ vegetation damage
reduced latent heat (LH) flux, precipitation, and runoff by 10-27 W $m^{-2}$, 0.9-1.4 mm
$d^{-1}$, and 0.1-0.17 mm $d^{-1}$, respectively, but increased surface air temperature by
0.6-2.0 °C during the summer of 2007-2012. In China, Zhu et al. (2022) performed
simulations and found that the inclusion of $O_3$-vegetation interaction caused a 5-30 W
$m^{-2}$ decrease in LH, 0.2-0.8 °C increase in surface air temperature, and 3% reduction



in relative humidity during summers of 2014-2017. Recently, Jin et al. (2023) applied
a different regional model and estimated that $O_3$ exposure weakened plant
transpiration and altered surface heat flux in China, resulting in significant increase of
up to 0.16 °C in maximum daytime temperature and decrease of -0.74% in relative
humidity. However, all these previous estimates of $O_3$-induced feedback to climate
were derived using the empirical $O_3$ damage scheme proposed by Lombardozzi et al.
(2013), which assumed fixed damage ratios independent of $O_3$ dose for some
vegetation species and as a result may have biases in the further estimated feedback to
climate.

The $O_3$-vegetation coupling also has intricate implications for air quality. On one

hand, $O_3$-vegetation coupling can influence meteorological conditions that affect $O_3$
generation, ultimately influencing the $O_3$ level (Sadiq et al., 2017). On the other hand,
it can also influence biogenic emissions and dry deposition, thereby affecting $O_3$
concentrations (Gong et al., 2020). Sadiq et al. (2017) implemented $O_3$-vegetation
coupling in the Community Earth System Model (CESM) and estimated that surface
$O_3$ concentrations increased 4-6 ppb in Europe, North America, and China due to
$O_3$-vegetation coupling. By using the CLM model with the empirical scheme of
Lombardozzi et al. (2013), Zhou et al. (2018) found that $O_3$-induced damage on leaf
area index (LAI) could lead to changes in global $O_3$ concentrations by -1.8 to +3 ppb
in boreal summer. Gong et al., (2020) used the $O_3$ damage scheme from Sitch et al.
(2007) embedded in a global climate-chemistry-carbon coupled model and estimated
that $O_3$-induced stomatal inhibition led to an average surface $O_3$ increase of 1.2-2.1
ppb in eastern China and 1.0-1.3 ppb in western Europe. Different from the above
global simulations with coarse resolutions, regional modeling with fine resolution can
reveal more details about $O_3$-vegetation coupling and feedback to surface $O_3$
concentrations in China (Zhu et al., 2022; Jin et al., 2023). However, all these regional
simulations were carried out using $O_3$ damage scheme of Lombardozzi et al. (2013),
limiting the exploration of model uncertainties due to varied $O_3$ vegetation damage
schemes.



In this study, we implemented O₃ vegetation damage schemes from both Sitch et
al. (2007) and Lombardozzi et al. (2013) into the widely-used regional
meteorology-chemistry model WRF-Chem. We validated the simulated meteorology
and O₃ concentrations, and performed sensitivity experiments to explore the O₃
damage to GPP and consequent feedbacks to regional climate and air quality in China.
Within the same framework, we compared the differences of O₃-vegetation coupling
from two schemes and explored the causes for the discrepancies. We aimed to
quantify the modeling uncertainties in the up-to-date estimates of O₃ impact on
regional carbon fluxes and its feedback to regional climate and air quality in China.

**2 Method**
**2.1 WRF-Chem model**
We used WRF-Chem model version 3.9.1 to simulate meteorological fields and
O₃ concentration in China. The model includes atmospheric physics and dynamical
processes, atmospheric chemistry, and biophysical and biochemical processes (Grell
et al., 2005, Skamarock et al., 2008). The model domain is configured with 196×160
gird cells at 27 km horizontal resolution on the Lambert conformal projection, and
covers the entire mainland China. In the vertical direction, 28 layers are set extending
from surface to 50 hPa. The meteorological initial and boundary conditions were
adopted from ERA5 reanalysis produced by the European Centre for Medium-Range
Weather Forecasts (ECMWF) at a horizontal resolution of 0.25°×0.25° (Hersbach et al.,
2020). The chemical initial and boundary conditions were generated from the Model
for Ozone and Related Chemical Tracer version 4 (MOZART-4), which is available at
a horizontal resolution of 1.9°×2.5° with 56 vertical layers (Emmons et al., 2010).
Anthropogenic emissions are adopted from the 0.25° Multi-resolution Emission
Inventory for China (MEIC) and MIX Asian emission inventory for the other regions
(available at http://meicmodel.org). Biogenic emissions are calculated online using
the Model of Emissions of Gases and Aerosols from Nature (Guenther et al., 2006),
which considers the impacts of plant types, weather conditions, and leaf area on





vegetation emissions. Atmospheric chemistry is simulated using the Carbon Bond
Mechanism version Z (CBMZ) (Zaveri and Peters, 1999) gas-phase chemistry module
coupled with a four-bin sectional Model for Simulating Aerosol Interactions and
Chemistry (MOSAIC) (Zaveri et al., 2008). The photolysis scheme is based on the
Madronich Fast-TUV photolysis module (Tie et al., 2003). The physical
configurations include the Morrison double-moment microphysics scheme (Morrison
et al., 2009), the Grell-3 cumulus scheme (Grell et al., 2002), the Rapid Radiative
Transfer Model longwave radiation scheme (Mlawer et al., 1997), the Goddard
short-wave radiation scheme (Chou and Suarez, 1994), the Yonsei University
planetary boundary layer scheme (Hong et al., 2006), and the revised MM5 (Fifth
generation Mesoscale Model) Monin–Obukhov surface layer scheme.

**181 2.2 Noah-MP model**

Noah-MP is a land surface model coupled to WRF-Chem with multiple options
for key land-atmosphere interaction processes (Niu et al., 2011). Noah-MP considers
canopy structure with canopy height and crown radius, and depicts leaves with
prescribed dimensions, orientation, density, and radiometric properties. The model
employs a two-stream radiative transfer approach for surface energy and water
transfer processes (Dickinson, 1983). Noah-MP is capable of distinguishing
photosynthesis pathways between $C_3$ and $C_4$ plants, and defines vegetation-specific
parameters for leaf photosynthesis and respiration.
Noah-MP considers prognostic vegetation growth through the coupling between
photosynthesis and stomatal conductance (Farquhar et al., 1980; Ball et al., 1987).
The photosynthesis rate, $A$ ($\mu molCO_2$ m$^{-2}$ s$^{-1}$), is calculated as one of three limiting
factors as follows:
$A_{tot} = min\,(W_c, W_j, W_e)I_{gs}$

(1)

where $W_c$ is the RuBisco-limited photosynthesis rate, $W_j$ is the light-limited
photosynthesis rate, and $W_e$ is the export-limited photosynthesis rate. $I_{gs}$ is the





growing season index with values ranging from 0 to 1. Stomatal conductance ($g_s$) is
computed based on photosynthetic rate as follows:
$$g_s = \frac{1}{r_s} = m \frac{A_{net}}{C_s} RH + b \qquad (2)$$
where $b$ is the minimum stomatal conductance; $m$ is the Ball-Berry slope of the
conductance-photosynthesis relationship; $A_{net}$ is the net photosynthesis by subtracting
dark respiration from $A_{tot}$; $C_s$ is the ambient $CO_2$ concentration at the leaf surface.
The assimilated carbon is allocated to various parts of vegetation (leaf, stem, wood,
and root) and soil carbon pools (fast and slow), which determines the variations of
LAI and canopy height. Plant transpiration rate is then estimated using the dynamic
LAI and stomatal conductance. Noah-MP also distinguishes the photosynthesis of
sunlit and shaded leaves. Sunlit leaves are more limited by $CO_2$ concentration while
shaded leaves are more constrained by insolation, leading to varied responses to $O_3$
damage.

**212 2.3 Scheme for ozone damage on vegetation**

We implemented the $O_3$ vegetation damage schemes proposed by Sitch et al.
(2007) (thereafter S2007) and Lombardozzi et al. (2013) (thereafter L2013) into the
Noah-MP. In S2007 scheme, the undamaged fraction $F$ for net photosynthesis is
dependent on the sensitivity parameter $a_{PFT}$ and excessive area-based stomatal $O_3$ flux,
which is calculated as the difference between $f_{O_3}$ and threshold $y_{PFT}$:
$$F = 1 - a_{PFT} \times max\{f_{O_3} - y_{PFT}, 0\} \qquad (3)$$
where $a_{PFT}$ and $y_{PFT}$ are specifically determined for individual plant functional types
(PFTs) based on measurements (Table 1). The stomatal $O_3$ flux $f_{O_3}$ is calculated as
$$f_{O_3} = \frac{[O_3]}{r_a + k_{O_3} \cdot r_s} \qquad (4)$$
where $[O_3]$ is the $O_3$ concentration at the reference level (nmol m$^{-3}$), $r_a$ is the
aerodynamic and boundary layer resistance between leaf surface and reference level
(s m$^{-1}$). $k_{O_3} = 1.67$ represents the ratio of leaf resistance for $O_3$ to that for water vapor.
$r_s$ represents stomatal resistance (s m$^{-1}$). For S2007 scheme, stomatal conductance is



damaged with the same ratio (1-*F*) as photosynthesis and further affects $O_3$ uptake.
As a comparison, the L2013 scheme applies separate $O_3$ damaging relationships
for photosynthetic rate and stomatal conductance. These independent relationships
account for different plant groups and are calculated based on the cumulative uptake
of $O_3$ (CUO) under different levels of chronic $O_3$ exposure. The leaf-level CUO
(mmol m$^{-2}$) over the growing season is calculated as follows:
$CUO = \sum (k_{O_3}/r_s + 1/r_a) \times [O_3]$

(5)

The physical parameters in Equation (5) are the same as those in Equation (4). $O_3$
uptake is accumulated over time steps during the growing season with mean LAI >
0.5 (Lombardozzi et al., 2012), when vegetation is most vulnerable to air pollution
episodes. $O_3$ uptake is only accumulated when $O_3$ flux is above an instantaneous
threshold of 0.8 nmol $O_3$ m$^{-2}$ s$^{-1}$ to account for ozone detoxification by vegetation at
low $O_3$ levels (Lombardozzi et al., 2015). We also include a leaf-turnover rate for
evergreen plants so that the accumulation of $O_3$ flux does not last beyond the average
foliar lifetime. The $O_3$ damaging ratios depend on CUO with empirical linear
relationships as follows:
$F_{PO3} = a_p \times CUO + b_p$                                                   (6)
$F_{cO3} = a_c \times CUO + b_c$                                                    (7)
where $F_{pO3}$ and $F_{cO3}$ are the ozone damage ratios for photosynthesis and stomatal
conductance, respectively. The slopes ($a_p$ for photosynthesis and $a_c$ for stomatal
conductance) and intercepts ($b_p$ for photosynthesis and $b_c$ for stomatal conductance)
of regression functions are determined based on the meta-analysis of hundreds of
measurements (Table 2). The ratios predicted in Equations (6) and (7) are applied to
photosynthesis and stomatal conductance, respectively, to account for their
independent responses to $O_3$ damages.

**2.4 Observational data**
We validated the simulated meteorology and air pollutants with observations.



The meteorological data were downloaded from the National Meteorological
Information Center of China Meteorological Administration (CMA Meteorological
Data Centre, 2022, http://data.cma.cn/data/detail/dataCode/A.0012.0001.html). The
daily averaged surface pressure (PRES), wind speed at a height of 10 m (WS10),
relative humidity (RH) and temperature at a height of 2 m (T2) were collected from
839 ground stations. Hourly surface $O_3$ concentrations at 1597 sites in China were
collected from Chinese National Environmental Monitoring Center (CNEMC,
http://websearch.mep.gov.cn/).

**2.5. Simulations**
We performed seven experiments to quantify the damaging effects of ambient $O_3$
on GPP and the feedbacks to regional climate and air quality (Table 3). All
simulations are conducted from 1st May to 31st August of 2017 with the first month
excluded from the analysis as the spin-up. The control simulations (CRTL) excluded
the impact of ozone on vegetation. Three offline simulations were performed with the
same settings as the CTRL run, except that $O_3$ vegetation damages were calculated
and output without feedback to affect vegetation growth. These offline runs were
established using either the S2007 scheme (Offline_SH07 for high sensitivity and
Offline_SL07 for low sensitivity) or the L2013 scheme (Offline_L13). As a
comparison, three online simulations applied the S2007 scheme (Online_SH07 for
high sensitivity and Online_SL07 for low sensitivity) and the L2013 scheme
(Online_L13) to estimate the $O_3$ damages to GPP, which further influenced LAI
development, leaf transpiration, and dry deposition. The differences between CTRL
and Online runs indicated the responses of surface meteorology and $O_3$ concentrations
to the $O_3$-induced vegetation damages.

**3. Results**
**3.1 Model evaluations**
We compared the simulated summer near-surface temperature, relative humidity,



wind speed, and surface $O_3$ concentrations to observations. The model reasonably
reproduces the spatial pattern of near-surface temperature with warmings in the
Southeast and Northwest but coolings over the Tibetan Plateau (Figure 1a). On the
national scale, the near-surface temperature is underestimated with a mean bias (MB)
of 1.04 ℃ and a spatial R of 0.96. Unlike temperature, simulated relative humidity is
overestimated with a MB of 5.04 % but a high R of 0.93 (Figure 1b). Due to the
modeling biases in the topographic effects, simulated wind speed is overestimated by
more than 1.06 m s$^{-1}$ on the national scale (Figure 1c). Such overestimation was also
reported in other studies (Hu et al., 2016, Liu et al., 2020, Zhu et al., 2022).
Comparisons with the measurements from air quality sites show that the
simulated $O_3$ deviates from the observed mean concentrations by 5.42 μg m$^{-3}$ with a
spatial R of 0.68. The model reasonably captures the hotspots over North China Plain
though with some overestimations. Such elevated bias in summer $O_3$ is a common
issue for both global and regional models over Asia. For example, Zhu et al. (2022)
overestimated summer average ozone concentration by 13.82 μg m$^{-3}$ in China. Liu et
al. (2020) reached positive biases ranging from 3.7 μg m$^{-3}$ to 13.32 μg m$^{-3}$ using the
WRF-CMAQ model. Overall, the WRF-Chem model shows reasonable performance
in the simulation of surface meteorology and $O_3$ concentrations in China.

**3.2 Offline $O_3$ damage**
We compared the offline $O_3$ damage to photosynthesis between sunlit (PSNSUN)
and shaded (PSNSHA) leaves during the summer. The S2007 scheme is dependent on
instantaneous $O_3$ uptake, which peaks when both $O_3$ concentrations and stomatal
conductance are high. For the same $O_3$ pollution level, the damages are much higher
for the sunlit leaves (Figures 2a-2b) than that for the shaded leaves (Figures 2d-2e),
because of the higher stomatal conductance linked with the more active
photosynthesis for the sunlit leaves. In contrast, the L2013 scheme depends on the
accumulated $O_3$ flux, which results in vegetation damage even at lower instant $O_3$
concentrations. As a result, we found limited differences in the $O_3$ damages between



sunlit (Figure 2c) and shaded (Figure 2f) leaves with L2013 scheme. Observations
have reported that surface $O_3$ has limited impacts on the shaded leaves (Wan et al.,
2014), consistent with the results simulated by the S2007 scheme. Furthermore,
surface $O_3$ concentrations are low in southwest during summer (Figure 1d),
suggesting a low $O_3$ vegetation damage over Tibetan Plateau and the more reasonable
performance with the S2007 scheme.

Figure 3 shows the effect of $O_3$ damage to stomatal resistance of sunlit (RSSUN)

and shaded (RSSHA) leaves. Overall, the spatial pattern of the changes in stomatal
resistance is consistent with those of photosynthesis (Figure 2) but with opposite signs.
Both RSSUN and RSSHA are enhanced by $O_3$ damage so as to prevent more $O_3$
uptake. For S2007 scheme, RSSUN with high and low sensitivities respectively
increases by 13.43% (Figure 3a) and 8.35% (Figure 3b), higher than the rates of
4.71% (Figure 3d) and 2.97% (Figure 3e) for RSSHA. These ratios are inversely
connected to the changes of photosynthesis (Figure 2), suggesting the full coupling of
damages between leaf photosynthesis and stomatal conductance. For L2013 scheme,
predicted changes in RSSUN (Figure 3c) and RSSHA (Figure 3f) are very similar
with the magnitude of 25.3%-26.3%. These changes are higher than the loss of
photosynthesis (Figures 2c and 2f), suggesting the decoupling of $O_3$ damages to leaf
photosynthesis and stomatal conductance as revealed by the L2013 scheme.

We further assessed the $O_3$ damage to GPP and transpiration (TR). For S2007

scheme, $O_3$ causes damages to GPP and TR approximately by 5.5% with low
sensitivity (Figures 4b and 4e) and 8.4% with high sensitivity (Figures 4a and 4d)
compared to the CTRL simulation. The model predicts high GPP damages over North
China Plain and moderate damages in the southeastern and northeastern regions. In
the northwest, GPP damage is very limited due to the low relative humidity (Figure 1b)
that constrains the stomatal uptake. For L2013 scheme, TR shows uniform reductions
exceeding -25% in most regions of China except for the northwest (Figure 4f), though
$O_3$ concentrations show distinct spatial gradient (Figure 1d). The changes of GPP are
similar to that of TR but with lower inhibitions (Figure 4c). On average, the GPP



reduction with the L2013 scheme is 2.5-3.9 times of that predicted with the S2007
scheme. The most significant differences are located in Tibetan Plateau with limited
damages in S2007 but strong inhibitions in L2013. Given the cold environment
(Figure 1a) that constrains stomatal uptake (Wilkinson et al., 2001), we consider the
low $O_3$ impacts in Tibetan Plateau predicted with S2007 scheme are more reasonable.

**3.3 The $O_3$-vegetation feedback to surface energy and meteorology**
The $O_3$ vegetation damage causes contrasting responses in surface sensible heat
(SH) and LH (Figure 5). For S2007 scheme, the SH fluxes on average increase by
3.17 W m$^{-2}$ (8.85%) with low sensitivity (Figure 5b) and 5.99 W m$^{-2}$ (16.22%) with
high sensitivity (Figure 5a). The maximum enhancement is located in southern China,
where the increased stomatal resistance (Figure 3a) reduces transpiration and the
consequent heat dissipation. Meanwhile, LH fluxes decrease by 3.26 W m$^{-2}$ (5.58%)
with low sensitivity (Figure 5e) and 6.43 W m$^{-2}$ (15.29%) with high sensitivity
(Figure 5d), following the reductions in transpiration (Figures 4d and 4e). We found
similar changes in surface energy by $O_3$-vegetation coupling between the S2007 and
L2013 schemes. The SH shows the same hotspots over southern China with national
average increase of 12.85% (Figure 5c), which is within the range of 8.85% to
16.22% predicted by the S2007 scheme. The LH largely decreases in central and
northern China with the mean reduction of 17.4% (Figure 5f), close to the magnitude
of 15.29% predicted with the S2007 scheme using the high $O_3$ sensitivity (Figure 5d).
Although the offline damages to GPP and TR are much larger with the L2013 than
S2007 (Figure 4), their feedback to surface energy shows consistent spatial pattern
and magnitude (Figure 5), likely because the $O_3$ inhibition in S2007 has the same
diurnal cycle with energy fluxes while the L2013 scheme shows almost constant
inhibitions through the day (Figure S1). The nighttime damages in L2013 have
limited contributions to the changes of surface energy, which usually peaks at the
daytime.
The $O_3$-induced damages to stomatal conductance weaken plant transpiration and



thus slow down the heat dissipation at the surface, leading to the higher temperature
but lower RH in China (Figure 6). On the national scale, temperature increases by
0.5 °C due to $O_3$ vegetation damage with the high sensitivity (Figure 6a) and 0.23 °C
with the low sensitivity (Figure 6b) predicted using the S2007 scheme. A similar
warming is predicted with the L2013 scheme except that temperature shows moderate
enhancement over Tibetan Plateau (Figure 6c). The average RH decreases by 3.68%
with the high $O_3$ sensitivity (Figure 6d) and 2.22% with the low sensitivity (Figure 6e)
in response to the suppressed plant transpiration. A stronger RH reduction of -3.85%
is achieved with the L2013 scheme, which predicts the maximum RH reductions in
the North (Figure 6f).

**3.4 The $O_3$-vegetation feedback to air quality**

The $O_3$-induced inhibition on stomatal resistance leads to a significant increase
in surface $O_3$ concentrations, particularly in eastern China (Figures 7a-7c). The main
cause of such feedback is the reduction in $O_3$ dry deposition, which exacerbates the $O_3$
pollution in China. For S2007 scheme, this positive feedback can reach up to 15 μg
m$^{-3}$ with high sensitivity (Figure 7a) and 8 μg m$^{-3}$ with low sensitivity (Figure 7b)
over North China Plain. On the national scale, surface $O_3$ enhances 3.31 μg m$^{-3}$ (1.25
μg m$^{-3}$) or 7.92 μg m$^{-3}$ (3.04%) with the high (low) $O_3$ sensitivity. For L2013 scheme,
the changes of $O_3$ concentration (Figure 7c) are comparable to that of the S2007
scheme with high sensitivity (Figure 7a), except that the $O_3$ enhancement is stronger
in the Southeast but weaker in the Northeast.
The $O_3$-vegetation coupling also increases surface isoprene emissions. For S2007
scheme, isoprene emissions increase by 6.13% with high sensitivity (Figure 7d) and
3.43% with low sensitivity (Figure 7e), with regional hotspots in North China Plain,
northeastern and southern regions. The predictions using L2013 scheme (Figure 7f)
show very similar patterns and magnitude of isoprene changes to the S2007 scheme
with high sensitivity. Such enhancement in isoprene emissions is related to the
additional surface warming by $O_3$-vegetation interactions (Figures 6a-6c). In turn, the



increased isoprene emissions contribute to the deterioration of $O_3$ pollution in China.

**4. Conclusions and discussion**

In this study, we explored the feedback of $O_3$-vegetation coupling to surface

meteorology and air quality in China using two $O_3$ damage schemes embedded in a
regional meteorology-chemistry coupled model. The two schemes predicted distinct
spatial patterns with much larger magnitude of GPP loss in the L2013 scheme than
that in the S2007 scheme. We further distinguished the leaf responses with different
illuminations. For the S2007 scheme, the damages to photosynthesis of sunlit leaves
are ~2.6 times of that to shaded leaves. However, for the L2013 scheme, limited
differences are found between the sunlit and shaded leaves. The damages to leaf
photosynthesis increase stomatal resistance, leading to the reductions of transpiration
but enhancement of sensible heat due to the less efficient heat dissipation. These
changes in surface energy and water fluxes feed back to increase surface temperature
but decrease relative humidity. Although the L2013 scheme predicts much stronger
offline damages, the feedback causes very similar pattern and magnitude in surface
warming as the S2007 scheme. Consequently, surface $O_3$ increases due to the stomatal
closure and isoprene emissions enhance due to the anomalous warming.

Our predicted $O_3$ damage to GPP was within the range of -4% to -40% as

estimated in previous studies using different models and/or parameterizations over
China (Ren et al., 2011; Lombardozzi et al., 2015; Yue et al., 2015; Sadiq et al., 2017;
Xie et al., 2019; Zhu et al., 2022; Jin et al., 2023). Such a wide span revealed the large
uncertainties in the estimate of $O_3$ impacts on ecosystem functions. In this study, we
employed two schemes and compared their differences. With the S2007 scheme, we
predicted GPP reductions of -5.5% to -8.5% in China, similar to the range of -4% to
-10% estimated by Yue et al. (2015) using the same $O_3$ damage scheme but lower than
the estimate of -12.1% predicted by Xie et al. (2019), likely due to the slight
overestimation of surface $O_3$ in the latter study. With the L2013 scheme, we predicted
much larger GPP reductions of -21.4%. However, such value was still lower than the



-28.9% in Jin et al. (2023) and -20% to -40% in Zhu et al. (2022) using the same
L2013 scheme embedded in WRF-Chem model, though all studies showed similar
spatial patterns in the GPP reductions. Such differences were likely attributed to the
varied model configuration as we ran the model from May while the other studies
started from the beginning of years. The longer time for the accumulation of $O_3$
stomatal uptake in other studies resulted in higher damages than our estimates with
the L2013 scheme.

The $O_3$-vegetation coupling caused strong feedback to surface meteorology and

air quality. Our simulations with either scheme revealed that surface SH increases by
2-28 W m$^{-2}$ and LH decreases by 4-32W m$^{-2}$ over eastern China, consistent with the
estimates of 5-30 W m$^{-2}$ by Zhu et al. (2022) using WRF-Chem model with the L2013
scheme. Consequently, surface air temperature on average increases by 0.23-0.51ºC
while relative humidity decreases by 2.2-3.8%, similar to the warming of 0.2-0.8ºC
and RH reduction of 3% as predicted by Zhu et al. (2022). However, these changes in
surface energy flux and meteorology are much higher than that in Jin et al. (2023),
likely because the latter focuses on the perturbations averaged throughout the year
instead of summer period as in this study and Zhu et al. (2022). We further predicted
that $O_3$ vegetation damage increased surface $O_3$ by 0.6-1.7 ppbv in China, similar to
the 1.2-2.1 ppbv estimated for eastern China using a global model (Gong et al., 2020).
Regionally, the $O_3$ enhancement reached as high as 4-7.5 ppbv in North China Plain,
consistent with the maximum value of 6 ppbv over the same domain predicted by Zhu
et al. (2022). However, limited feedback to surface $O_3$ was predicted in Jin et al.
(2023), mainly because the decreased dry deposition had comparable but opposite
effects to the decreased isoprene emissions due to the reductions of LAI. Such
discrepancy was likely caused by the stronger $O_3$ inhibition in Jin et al. (2023)
following the longer period of $O_3$ accumulation, consequently exacerbating the
negative impacts of LAI reductions on $O_3$ production.

There were some limitations in our parameterizations and simulations. The

WRF-Chem model slightly overestimated summer $O_3$ concentrations, which could



exacerbate the damages to stomatal conductance and the subsequent feedback. The
S2007 scheme employed the coupled responses in photosynthesis and stomatal
conductance to $O_3$ vegetation damage. However, some observations revealed that
stomatal response is slow under long-term $O_3$ exposure, resulting in loss of stomatal
function and decoupling from photosynthesis (Calatayud et al., 2007; Lombardozzi et
al., 2012). The L2013 scheme considered the decoupling between photosynthesis and
stomatal conductance. However, this scheme could not distinguish the responses of
sunlit and shaded leaves. In addition, the calculation of CUO heavily relied on the
ozone threshold and accumulation period, leading to varied responses among different
studies using the same scheme. Furthermore, the slopes of $O_3$ sensitivity in L2013
scheme were set to zero for some PFTs, leading to constant damages independent of
CUO. Finally, the current knowledge of the $O_3$ effects on stomatal conductance was
primarily derived from leaf-level measurements (Matyssek et al., 2008), which were
much fewer compared to that for photosynthesis. The limited data availability and
lack of inter-PFT responses constrain the development of empirical parameterizations.

Despite these limitations, our study provided the first comparison of different

parameterizations in simulating $O_3$-vegetation interactions. We found similar
feedbacks to surface energy and meteorology though the two schemes showed varied
magnitude and distribution in the offline responses of GPP and stomatal conductance
to surface $O_3$. The main cause of such inconsistency lied in the low feedback of
damages in L2013 with some unrealistic inhibitions of ecosystem functions at night
and over the regions with low $O_3$ level. Such similarity provides a solid foundation for
the exploration of $O_3$-vegetation coupling using different schemes. The positive
feedback of $O_3$ vegetation damage to surface air temperature and $O_3$ concentrations
posed emerging but ignored threats to both climate change and air quality in China.

**Data availability**

The observed hourly O3 concentrations were obtained from Chinese National

Environmental Monitoring Center (CNEMC, http://websearch.mep.gov.cn/). The



observed meteorological data were obtained from the National Meteorological
Information Center of China Meteorological Administration (CMA Meteorological
Data Centre, 2022, http://data.cma.cn/data/detail/dataCode/A.0012.0001.html). The
MEIC and MIX emission inventory are available at
(http://meicmodel.org.cn/?page_id=560 and http://meicmodel.org.cn/?page_id=89).

**Author contributions**
XY conceived the study. XY and JC designed the research and carried out the
simulations. JC completed data analysis and the first draft. MM provided useful
comments on the paper. XY reviewed and edited the manuscript.

**Competing interests**
The authors declare that they have no conflict of interest.

**Acknowledgments**
This study was jointly funded by the National Natural Science Foundation of
China (grant no. 42293323) and Jiangsu Funding Program for Excellent Postdoctoral
Talent (2023ZB737).

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



**Tables**

**Table 1.** Parameters used for S2007 $O_3$ damage scheme.

| PFTs [a] | $a_{PFT}$(nmol$^{-1}$ m$^2$ s) [b] | $y_{PFT}$(nmol m$^{-2}$ s$^{-1}$) |
|---|---|---|
| EBF | 0.075, 0.02 | 1.6 |
| NF | 0.075, 0.02 | 1.6 |
| DBF | 0.15, 0.04 | 1.6 |
| SHR | 0.1, 0.03 | 1.6 |
| GRA | 1.4, 0.25 | 5 |
| CRO | 1.4, 0.25 | 5 |

[a] The plant functional types (PFTs) include evergreen broadleaf forest (EBF), needleleaf forest (NF), deciduous broadleaf forest (DBF), shrubland (SHR), grassland (GRA), and cropland (CRO).

[b] The first number is for high sensitivity and the second is for low sensitivity.






**Table 2.** Slopes and intercepts used for L2013 $O_3$ damage scheme.

| PFTs | $a_p$ (mmol m$^{-2}$) | $b_p$ | $a_c$ (mmol m$^{-2}$) | $b_c$ |
|------|------|------|------|------|
| EBF | 0 | 0.8752 | 0 | 0.9125 |
| NF | 0 | 0.839 | 0.0048 | 0.7823 |
| DBF | 0 | 0.8752 | 0 | 0.9125 |
| SHR | 0 | 0.8752 | 0 | 0.9125 |
| GRA | -0.0009 | 0.8021 | 0 | 0.7511 |
| CRO | -0.0009 | 0.8021 | 0 | 0.7511 |








**Table 3.** Summary of simulation experiments

| Name | O$_3$ damage to vegetable | Scheme |
|---|---|---|
| CRTL | - | - |
| Offline_SH07 | High | Sitch et al. (2007) |
| Offline_SL07 | Low | Sitch et al. (2007) |
| Offline_L13 | - | Lombardozzi et al. (2013) |
| Online_SH07 | High | Sitch et al. (2007) |
| Online_SL07 | Low | Sitch et al. (2007) |
| Online_L13 | - | Lombardozzi et al. (2013) |





**Figure captions**

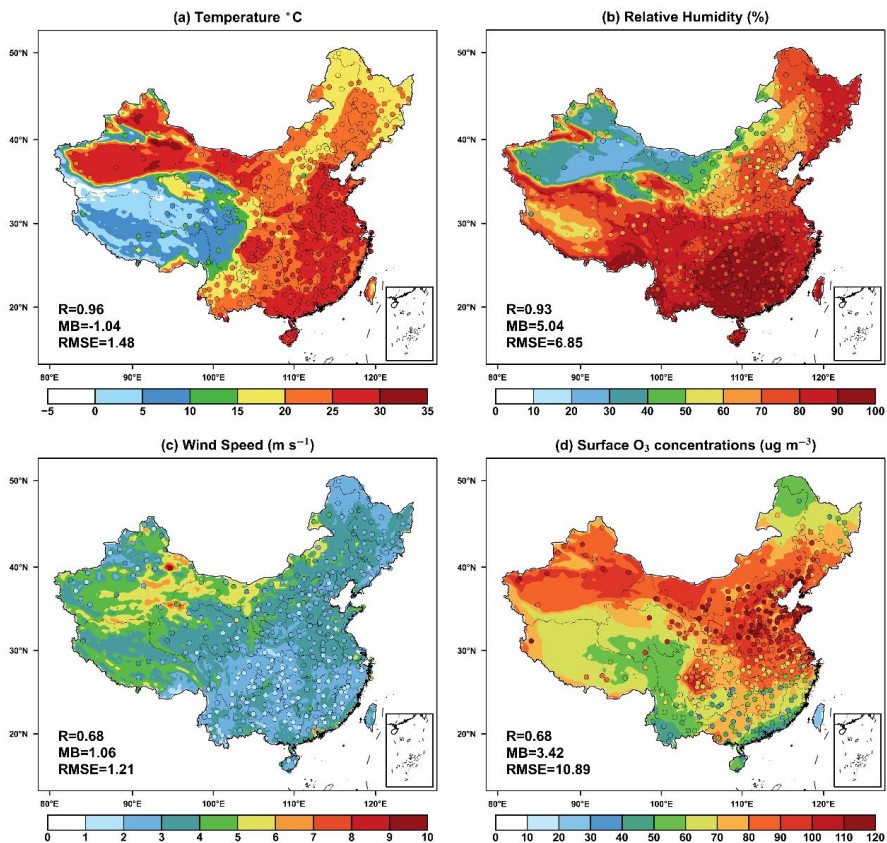

**Figure 1** Evaluations of simulated summer (June–August) daily (24-h average) (a) near-surface temperature, (b) relative humidity, (c) wind speed, and (d) surface $O_3$ concentrations in China. The dots represent the site-level observations. The correlation coefficients (R), mean biases (MB), and root-mean-square error (RMSE) for the comparisons are shown in the lower left corner of each panel.



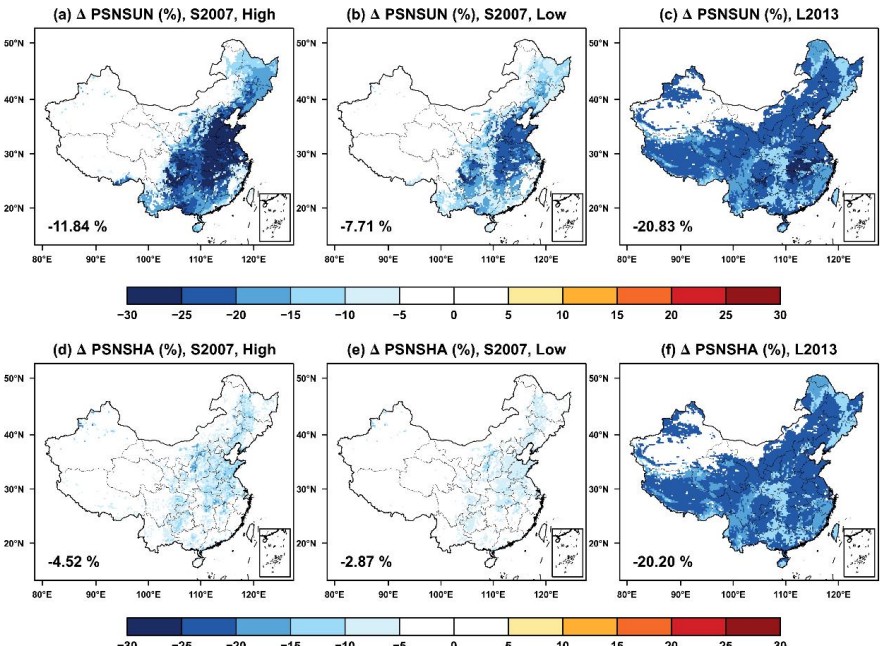

**Figure 2** Offline O$_3$ damage (%) to the summertime photosynthesis of (a-c) sunlit and (d-f) shaded leaves predicted by the S2007 scheme with (a, d) high and (b, e) low sensitivities or the (c, f) L2013 scheme. The area-weighted percentage changes are shown in the lower left corner.



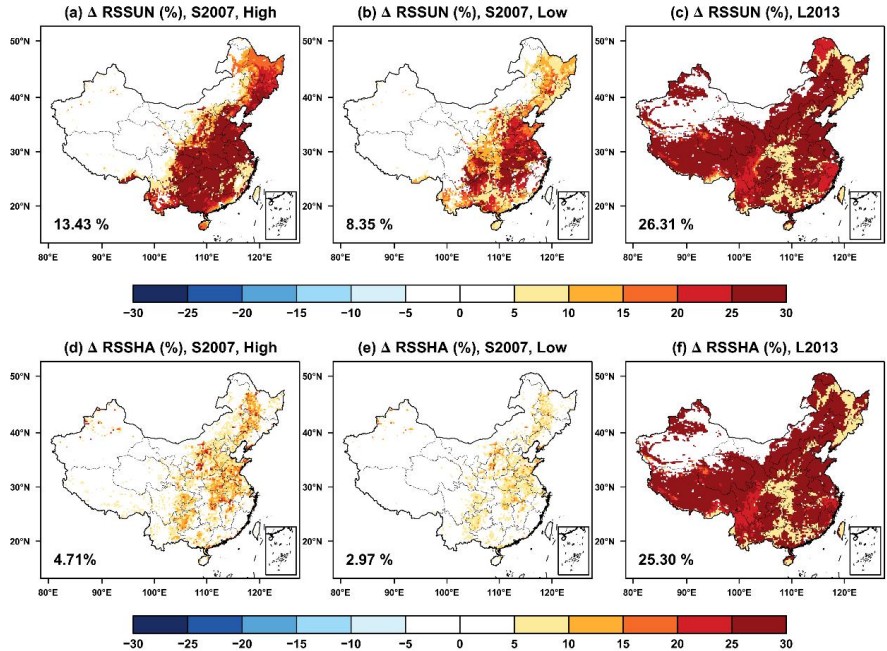




**Figure 3** The same as Figure 2 but for the changes in stomatal resistance.




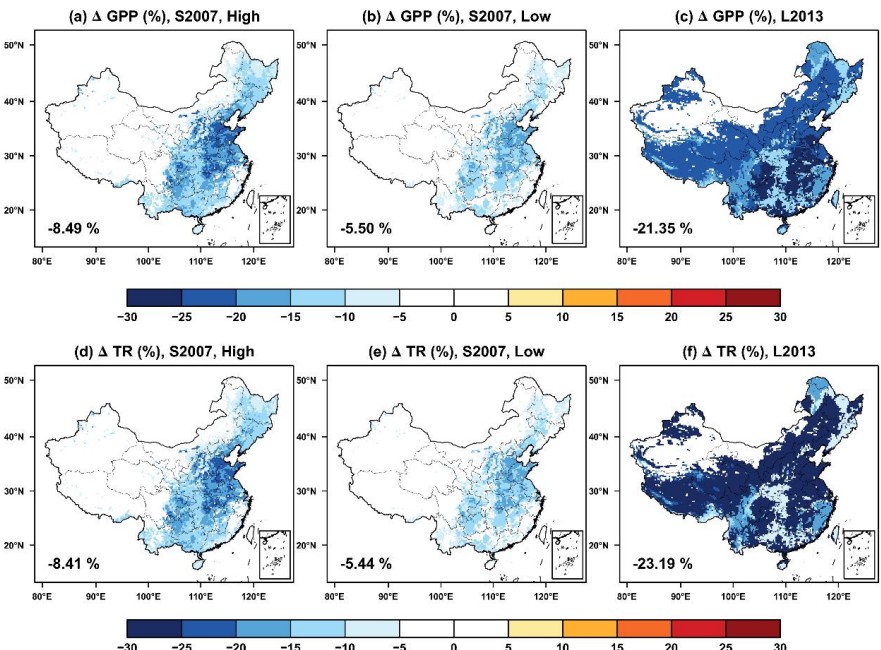

**Figure 4** Offline $O_3$ damage (%) to the (a-c) gross primary productivity (GPP) and
(d-f) transpiration rate (TR) predicted by the Sitch scheme with (a, d) high and (b,e)
low sensitivities or the (c, f) Lombardozzi scheme. The area-weighted percentage
changes are shown in the lower left corner.



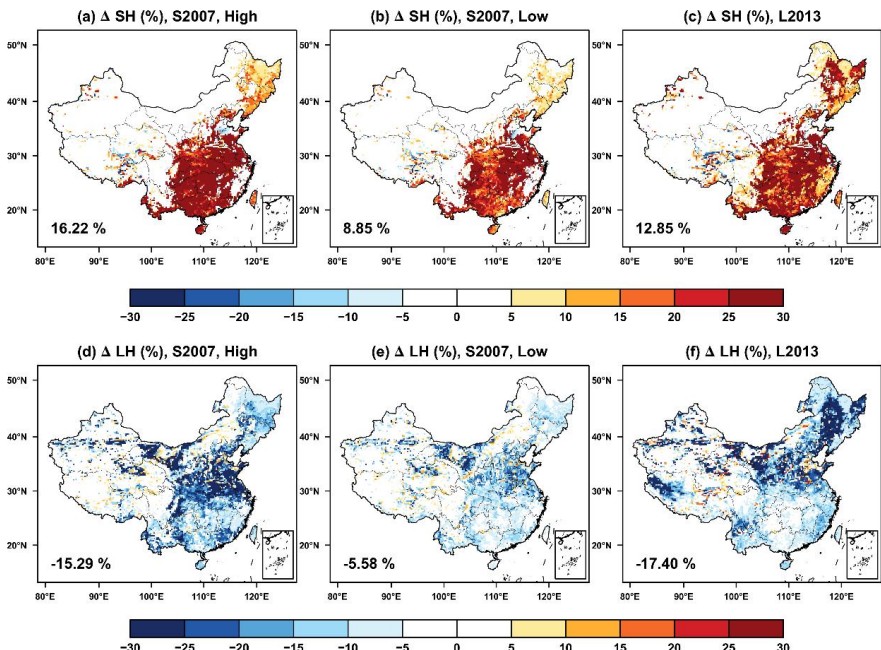

**Figure 5** The feedback of O$_3$-vegetation interaction to surface (a-c) sensible and (d-f) latent heat fluxes in the summer predicted by the S2007 scheme with (a, d) high and (b, e) low sensitivities or the (c, f) L2013 scheme. The relative changes are shown with area-weighted percentage changes indicated at the lower left corner.



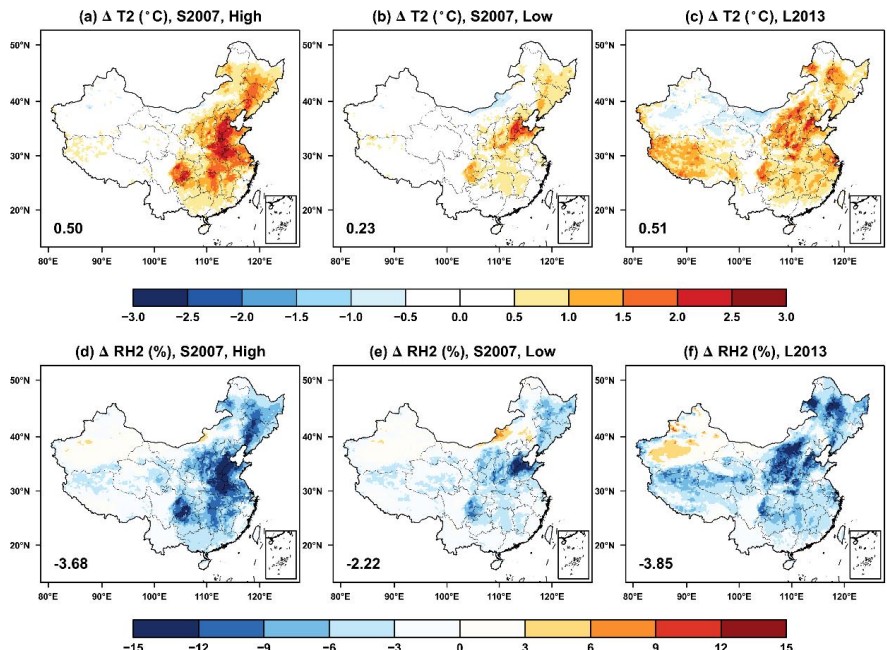

**Figure 6** The same as Figure 5 but for changes in (top) air temperature and (bottom) relative humidity at 2 meters.






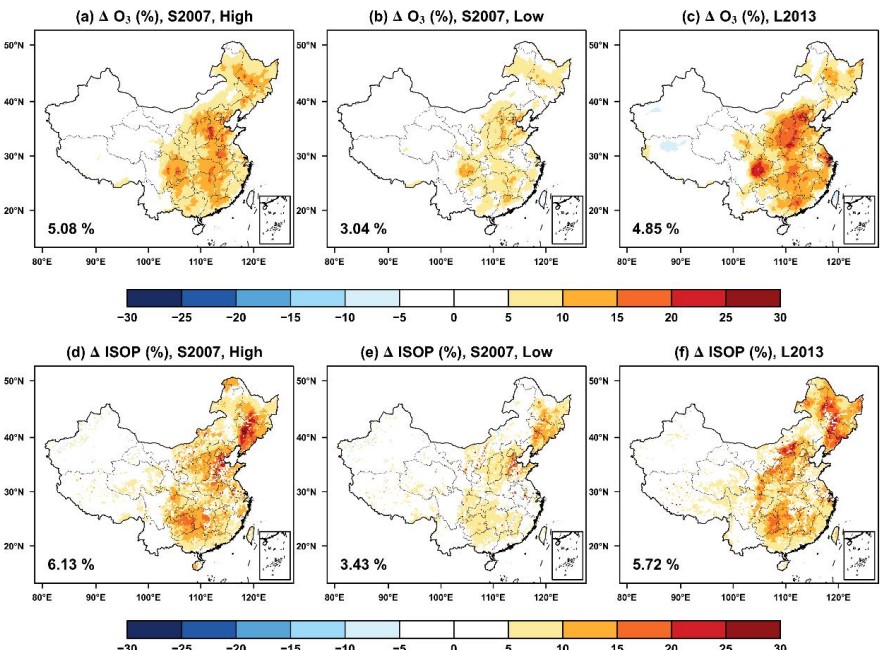


**Figure 7** The feedback of O$_3$-vegetation interaction to surface O$_3$ concentrations and isoprene emissions in the summer predicted by the S2007 scheme with (a, d) high and (b, e) low sensitivities or the (c, f) L2013 scheme. The area-weighted percentage changes are shown in the lower left corner.