# Peer review of "Simulation of ozone-vegetation coupling and feedback in"

_EGUsphere, 2023_

## Author Comment (AC1)

**Response to Comments of Reviewer #2**

Manuscript number: egusphere-2023-2149
Authors: Jiachen Cao, Xu Yue and Mingrui Ma
Title: Simulation of ozone-vegetation coupling and feedback in China using multiple ozone damage schemes

We are grateful to the referee for his/her time and energy in providing helpful comments and guidance that have improved the manuscript. In this document, we describe how we have addressed the reviewer's comments. Referee comments are shown in black and author responses are shown in blue text.

The manuscript firstly explores the different impact of the two commonly used O3 damage parametrizations which is an interesting comparison with relevant conclusions for the community. The authors additionally use measurements of O3 and meteorology to evaluate the model prediction which, however, could be more taken into account. In general, I feel more explanation and interpretation in the result section can imrove the manuscript, though it is overall well written and understandable. Please find my minor comments below:

Thank you for your positive evaluations. All the questions and concerns have been carefully answered.

l. 285: The terms 'warmings' and 'coolings' are not clear. This would more refer to model changes or even climate change experiments
*Response:* Thank you for your suggestions. We modified the sentence as follows: "The model reasonably reproduces the spatial pattern of higher near-surface temperature in Southeast and Northwest and lower temperature over the Tibetan Plateau (Figure 1a)". (Lines 279-281)

l. 288 "[...] but it shows a high correlation (R=0.96)"
*Response:* Corrected as suggested.

l. 292 For which model was it also reported ? Is it model-specific?
*Response:* WRF-CMAQ model was used in Hu et al. (2016) and Liu et al. (2020), and Zhu et al. (2022) used WRF-Chem model. We clarified as follows: "Such overestimation was also reported in other studies using WRF models …" (Lines 286-287)

l. 296 mention the reason for the overestimation of O3 (counteract the overestimation

of wind speed?)

*Response:* In the revised paper, we clarified as follows: "The model reasonably captures the hotspots over North China Plain though with some overestimations, potentially attributed to uncertain emissions and coarse model resolutions". (Lines 291-293)

l. 298 "reports" of "overestimated" (the model overestimates)

*Response:* Corrected as suggested.

l. 316 f: But the O3 damage not only depend on O3 concentration, right? How do you come to the conclusion that S2007 is more reasonable here?

*Response:* We agree with the referee's comments. In the revised paper, we removed the original statement on Lines 316-318 and clarified that S2007 reasonably captured the differences of $O_3$ damages to photosynthesis of sunlit and shaded leaves, which was supported by observations: "In contrast, the L2013 scheme depends on the accumulated $O_3$ flux and assumes constant damages for some PFTs (Table 2), resulting in reductions of photosynthesis even at low $O_3$ concentrations. Consequently, we found limited differences in the $O_3$ damages between sunlit (Figure 2c) and shaded (Figure 2f) leaves with L2013 scheme. Observations have reported that surface $O_3$ has limited impacts on the shaded leaves (Wan et al., 2014), consistent with the results simulated by the S2007 scheme. " (Lines 307-313)

l. 333: 5.5% is this an average over the model region?

*Response:* Yes, 5.5% is this an average over the model region. We clarified as follows: "For S2007 scheme, $O_3$ causes damages to national average GPP and TR approximately by 5.5% …" (Lines 327-328)

l. 344/345 Please explain the reason for the different changes by the two schemes
  You can be more concrete here.

*Response:* In the revised paper, we added explanations as follows: "The most significant differences are located in Tibetan Plateau with limited damages in S2007 but strong inhibitions of both GPP and TR in L2013. The low temperature (Figure 1a) and $O_3$ concentrations (Figure 1d) jointly constrain $O_3$ stomatal uptake (Figure S2), leading to low $O_3$ damages over Tibetan Plateau with the S2007 scheme. However, the L2013 scheme applies $b_p$=0.8021 for grassland (Table 2), suggesting strong baseline damages up to 20% even with CUO=0 over Tibetan Plateau where the grassland dominates (Figure S3)." (Lines 338-344)

l.366/367 Why is the L2013 O3 inhibition constant over day?

*Response:* In the revised paper, we clarified as follows: "the L2013 scheme shows

almost constant inhibitions throughout the day (Figure S1). The zero or near-zero slope parameters ($a_p$ and $a_c$) in the L2013 scheme (Table 2) lead to insensitive responses of photosynthesis and stomatal conductance to the variations of CUO. As a result, there were very limited diurnal variations in $O_3$ damage with the L2013 scheme." (Lines 364-368)

l. 388/389: The reffering of the different values is not clear. Perhaps, there is a bug with one unit or the brackets.

*Response:* In the revised paper, we corrected the numbers as follows: "On the national scale, surface $O_3$ enhances 4.40 µg m$^{-3}$ (5.08 %) with high $O_3$ sensitivity and 2.62 µg m$^{-3}$ (3.04%) with low $O_3$ sensitivity through the coupling to vegetation." (Lines 387-389)

l. 423 ff. please split the sentence in two or shorten it

*Response:* In the revised paper, we modified as follows: "With the S2007 scheme, we predicted GPP reductions of -5.5% to -8.5% in China. This is similar to the range of -4% to -10% estimated by Yue et al. (2015) using the same $O_3$ damage scheme. However, it is lower than the estimate of -12.1% predicted by Xie et al. (2019), likely due to the slight overestimation of surface $O_3$ in the latter study." (Lines 422-426)

l. 433-435: To my knowledge that shouldn't be the case? Didn't the other models consider leaf turnover?

*Response:* The other model studies did not mention whether their models took into account leaf turnover. Even if the models considered leaf turnover, they should have longer accumulation period of $O_3$ uptake than us, because they ran models from the beginning of the year while we ran the model from May. In the text, we added the word 'might' to suggest possible causes instead of making conclusions: "The longer time for the accumulation of $O_3$ stomatal uptake in other studies might result in higher damages than our estimates with the L2013 scheme" (Lines 432-433)

l. 446 f: Be consistent with the O3 unit.

*Response:* In the revised paper, we modified as follows: "We further predicted that $O_3$ vegetation damage increased surface $O_3$ by 1.0-3.33 µg m$^{-3}$ in China, similar to the 2.35-4.11 µg m$^{-3}$ estimated for eastern China using a global model (Gong et al., 2020). Regionally, the $O_3$ enhancement reached as high as 7.84-14.70 µg m$^{-3}$ in North China Plain, consistent with the maximum value of 11.76 µg m$^{-3}$ over the same domain predicted by Zhu et al. (2022)." (Lines 443-448)

l. 464/465: I would rephrase to "However, this scheme shows no significant different

changes for sunlit and shaded leaves"

*Response:*  Corrected as suggested.

---

## Author Comment (AC2)

**Response to Comments of Reviewer #1**

Manuscript number: egusphere-2023-2149
Authors: Jiachen Cao, Xu Yue and Mingrui Ma
Title: Simulation of ozone-vegetation coupling and feedback in China using multiple ozone damage schemes

We are grateful to the referee for his/her time and energy in providing helpful comments and guidance that have improved the manuscript. In this document, we describe how we have addressed the reviewer's comments. Referee comments are shown in black and author responses are shown in blue text.

The authors examined the meteorological and air quality feedback of O3 damage to vegetation by coupling WRF-Chem with two O3 damage schemes. This reviewer has a few questions.

First, S2007 seems to calculate instantaneous (for WRF-Chem's model integration time steps or hourly) values of the undamaged fraction F, whereas L2013 calculates the ozone damage ratio for the entire growing season. So, was one constant L2013-calculated, plant-specific, O3 damage ratio applied throughout the whole simulation period, whereas S2007-calculated O3 damage ratios were time-dependent, when the schemes were coupled with WRF-Chem?

*Response:* As mentioned by the referee, the ozone damage calculated by the S2007 scheme is related to instantaneous excessive ozone flux ($dFO_3$), while the ozone damage calculated by the L2013 scheme is related to the cumulative ozone uptake flux (CUO). As shown in Figure R1, both CUO and $dFO_3$ vary with time. The value of CUO increases month by month, reaching a maximum in August. In contrast, $dFO_3$ is affected by instantaneous $O_3$ concentration, which peaks in July, leading to highest $dFO_3$ in July.

[Figure]

**Figure R1** Monthly mean CUO and *$dFO_3$* calculated for L2013 and S2007 schemes,

respectively. Here $dFO_3 = max\{f_{O_3} - y_{PFT}, 0\}$ in equation (3) of main text.

Second, the way the manuscript was written did not show the distinction between sunlit and sunshade in S2007- and L2013-calculated O3 damage ratios, which leads to the question how the ratios were applied to NOAH-MP. This leads to the next question. Why were L2013-calculated sunlit and sunshade O3 damage values for both photosynthesis and stomatal conductance were almost the same, whereas S2007-calculated ones showed such a contrast?

*Response:* In supplementary material, we added Text S1 to explain how we distinguish O₃ damages to sunlit and shaded leaves:

"In NOAH-MP, stomatal resistance is calculated separately for sunlit and shaded leaves. Therefore, the undamaged fraction $F_{(sunlit/shaded)}$ in S2007 is dependent on the sensitivity parameter $a_{PFT}$ and excessive area-based stomatal O₃ flux, which is calculated as the difference between $f_{O3(sunlit/shaded)}$ and threshold $y_{PFT}$:

$$F = 1 - a_{PFT} \times max\{f_{O_{3(sunlit/shaded)}} - y_{PFT}, 0\} \tag{1}$$

The stomatal O₃ flux $f_{O_{3(sunlit/shaded)}}$ is calculated as:

$$f_{O_{3(sunlit/shaded)}} = \frac{[O_3]}{r_a + k_{O_3} \cdot r_{s(sunlit/shaded)}} \tag{2}$$

where $r_{s(sunlit/shaded)}$ represents stomatal resistance (s m⁻¹) for sunlit/shaded leaves.

For the L2013 scheme, the leaf-level CUO for sunlit and sunshade (mmol m⁻²) over the growing season is calculated as follows:

$$CUO_{(sunlit/shaded)} = \sum(k_{O_3}/r_{s(sunlit/shaded)} + 1/r_a) \times [O_3] \tag{3}$$

$$F_{PO3(sunlit/shaded)} = a_p \times CUO_{(sunlit/shaded)} + b_p \tag{4}$$

$$F_{cO3(sunlit/shaded)} = a_c \times CUO_{(sunlit/shaded)} + b_c \tag{5}$$

where $F_{pO3(sunlit/shaded)}$ and $F_{cO3(sunlit/shaded)}$ are the damage ratios of photosynthesis and stomatal conductance for sunlit/shaded leaves, respectively."

The main reason why in the L2013 scheme, the sunlit and shaded leaves showed very similar damages for photosynthesis and stomatal conductance is that the L2013 scheme employed $a_p$=0 or $a_c$=0 for many PFTs (Table 2). In this case, the damages are independent of CUO which is different between sunlit and shaded leaves. Even for PFTs with non-zero sensitivities, such as grassland and cropland, the values of $a_p$ and $a_c$ are too low that the damaging ratio is mainly determined by $b_p$ or $b_c$. In the revised paper, we clarified as follows: "In contrast, the L2013 scheme depends on the accumulated O₃ flux and assumes constant damages for some PFTs (Table 2), resulting in reductions of

photosynthesis even at low $O_3$ concentrations. Consequently, we found limited differences in the $O_3$ damages between sunlit (Figure 2c) and shaded (Figure 2f) leaves with L2013 scheme." (Lines 307-311)

Third, isn't Eq. 5 supposed to be the integration of Eq. 4 according to its definition?

*Response:* By theory the accumulative flux (Eq. 5) should be the integration of instantaneous flux (Eq. 4). In practice, Eq 4 was used in the S2007 scheme while Eq. 5 was used in L2013 scheme with some differences. We maintained such differences because $O_3$ sensitivity parameters were derived based on the corresponding $O_3$ stomatal fluxes.

---

## Author Comment (AC3)

**Response to Comments of Reviewer #3**

Manuscript number: egusphere-2023-2149
Authors: Jiachen Cao, Xu Yue and Mingrui Ma
Title: Simulation of ozone-vegetation coupling and feedback in China using multiple ozone damage schemes

We are grateful to the referee for his/her time and energy in providing helpful comments and guidance that have improved the manuscript. In this document, we describe how we have addressed the reviewer's comments. Referee comments are shown in black and author responses are shown in blue text.

This paper use the established methods of chemistry-meteorology-ecosystem modeling to simulate ozone damage on plants over China, and the associated impacts on surface energy balance, carbon sink, meteorology and air quality. The manuscript is well-organized. Compared to earlier papers in this topic, the authors focus on comparing several established methods of calculating ozone damage (S2007 vs L2013), which is an important and new contribution. Minor revision is recommended to address several linguistic and conceptual problems:

Thank you for your positive evaluations. All the questions and concerns have been carefully answered.

L48: Rewrite as "…adverse effects on ecosystem functions, global warming and O3 pollution through…"
*Response:* Rewrite as suggested.

L60: rewrite as "…growth, suppressing ecosystem carbon uptake."
*Response:* Rewrite as suggested.

L104: "surface energy balance"
*Response:* Rewrite as suggested.

L107: "but" -> "and"
*Response:* Corrected as suggested.

L311: what is "instant O3 concentration"?
*Response:* In revised paper, we modified inappropriate description as follows: "at low $O_3$ concentrations." (Line 309)

L 310 – 313: Clearer explanation is required. L2013 (Table 2) has a lot of PFTs with 0 slopes. That means when stomatal O3 flux is above 0.8 nmol m-2 s-1, the response of photosynthesis and stomatal conductance remain constant. I believe this causes the same phenomenon described in L 366 – 367, especially during ozone season. A few CUO and PFT plots could help explain/verify this.

*Response:* In the revised paper, we clarified as follows: "In contrast, the L2013 scheme depends on the accumulated $O_3$ flux and assumes constant damages for some PFTs (Table 2), resulting in reductions of photosynthesis even at low $O_3$ concentrations." (Lines 307-309) We also added Figures S2 and S3 to show the CUO and PFT over China.

L316 – 318: There is no direct observation suggesting plants in southwest receive less ozone damage. This is not a valid conclusion and not necessary for the paper. Remove this statement or provide more direct evidence. On the other hand it is fair to point out L2013 lacks distinction between sunlit and shaded leaves since direct evidence were given by the authors.

*Response:* We agree with the referee's comments. In the revised paper, we removed the original statement on Lines 316-318 and clarified that S2007 reasonably captured the differences of $O_3$ damages to photosynthesis of sunlit and shaded leaves, which was supported by observations: "In contrast, the L2013 scheme depends on the accumulated $O_3$ flux and assumes constant damages for some PFTs (Table 2), resulting in reductions of photosynthesis even at low $O_3$ concentrations. Consequently, we found limited differences in the $O_3$ damages between sunlit (Figure 2c) and shaded (Figure 2f) leaves with L2013 scheme. Observations have reported that surface $O_3$ has limited impacts on the shaded leaves (Wan et al., 2014), consistent with the results simulated by the S2007 scheme. " (Lines 307-313)

L 343 – 346: Like I explained above: for a lot of PFTs L2013 has constant response after stomatal O3 flux is higher than a threshold, while S2007 depends on instantaneous stomatal O3 flux. It's more appropriate to highlight the difference in model structure/assumptions that leads to different result between S2007 and L2013 than judge which scheme is better without comparing with direct empirical evidence (e.g. plant trait and EC measurements).

*Response:* We agree with the referee's comments. In the revised paper, we removed the original judgement on Lines 344-346 and explained the differences between schemes as follows: "The most significant differences are located in Tibetan Plateau with limited damages in S2007 but strong inhibitions of both GPP and TR in L2013. The low temperature (Figure 1a) and $O_3$ concentrations (Figure 1d) jointly constrain $O_3$ stomatal uptake (Figure S2), leading to low $O_3$ damages over Tibetan Plateau with the S2007 scheme. However, the L2013 scheme applies $b_p$=0.8021 for grassland (Table 2),

suggesting strong baseline damages up to 20% even with CUO=0 over Tibetan Plateau where the grassland dominates (Figure S3)." (Lines 338-344)

L 393: This paper suggests that O3 damage increase isoprene emission because of increased leaf temperature, which is in line with previous studies (Sadiq et al., 2017). However, isoprene production is coupled to photosynthesis. There are empirical evidence, that high O3 exposure actually reduces isoprene emission when O3 exposure is prolonged enough to suppress photosynthesis (Bellucci et al., 2023). As an empirical parameterization, MEGAN does not include this effect. While this does not completely invalidate the O3 feedback result, this possible artifact in isoprene emission and its potential impact on the result have to be discussed thoroughly.

*Response:* In revised paper, we added following discussion as suggested: "First, we predicted increases of isoprene emissions in eastern China mainly due to the increased leaf temperature, which is in line with previous studies (Sadiq et al., 2017; Zhu et al., 2022). However, isoprene production is coupled to photosynthesis. There are empirical evidences showing that high dose of $O_3$ exposure reduces isoprene emissions when $O_3$ exposure is prolonged enough to suppress photosynthesis (Bellucci et al., 2023). Inclusion of such negative feedback might alleviate the $O_3$-induced enhancement in isoprene emissions. " (Lines 454-461)

Reference:

Bellucci, M., Locato, V., Sharkey, T. D., De Gara, L., and Loreto, F.: Isoprene emission by plants in polluted environments, Journal of Plant Interactions, 18, 2266463, https://doi.org/10.1080/17429145.2023.2266463, 2023.

Sadiq, M., Tai, A. P. K., Lombardozzi, D., and Val Martin, M.: Effects of ozone-vegetation coupling on surface ozone air quality via biogeochemical and meteorological feedbacks, Atmospheric Chemistry and Physics, 17, 3055–3066, https://doi.org/10.5194/acp-17-3055-2017, 2017.

---

## Referee Report (RR1)

First, S2007 seems to calculate instantaneous (for WRF-Chem's model integration time steps or hourly) values of the undamaged fraction F, whereas L2013 calculates the ozone damage ratio for the entire growing season. So, was one constant L2013-calculated, plant-specific, O3 damage ratio applied throughout the whole simulation period, whereas S2007-calculated O3 damage ratios were time-dependent, when the schemes were coupled with WRF-Chem?

*Response:* As mentioned by the referee, the ozone damage calculated by the S2007 scheme is related to instantaneous excessive ozone flux ($dFO_3$), while the ozone damage calculated by the L2013 scheme is related to the cumulative ozone uptake flux (CUO). As shown in Figure R1, both CUO and $dFO_3$ vary with time. The value of CUO increases month by month, reaching a maximum in August. In contrast, $dFO_3$ is affected by instantaneous $O_3$ concentration, which peaks in July, leading to highest $dFO_3$ in July.

[Figure]

**Figure R1** Monthly mean CUO and $dFO_3$ calculated for L2013 and S2007 schemes.

The authors did not address my question. I originally asked how the F values calculated using S2007 and L2013 were applied in their simulations. Specifically, S2007 computed instantaneous F values, which could technically be included in every time step to quantify ozone damage to vegetation. L2013-calculated F values, however, depended on CUO obtained from integration "over the growing season" (L231) using Eqs. 5, 6, & 7, meaning that there'd be only one pair of $F_{PO3}$ and $F_{CO3}$ for their simulation period May – August 2017. So actually, two questions involving L2013: 1. How did they obtain CUO of the growing season for their F value calculations? 2. Was one pair of constant, time-independent $F_{PO3}$ and $F_{CO3}$ values applied to every time step throughout the simulation period? It was not apparent to me how L2013 was coupled with the land surface model and WRF-Chem all together.

Third, isn't Eq. 5 supposed to be the integration of Eq. 4 according to its definition?
*Response:* By theory the accumulative flux (Eq. 5) should be the integration of instantaneous flux (Eq. 4). In practice, Eq 4 was used in the S2007 scheme while Eq. 5 was used in L2013 scheme with some differences. We maintained such differences because $O_3$ sensitivity parameters were derived based on the corresponding $O_3$ stomatal fluxes.

What I meant was that in the manuscript, Eq. 5 was not the integration form of Eq. 4 as so intended.

$$f_{O_3} = \frac{[O_3]}{r_a + k_{O_3} \cdot r_s} \tag{4}$$

$$CUO = \sum (k_{O_3}/r_s + 1/r_a) \times [O_3] \tag{5}$$

If they used Eq. 5 to calculate CUO, their L2013-calculated results and subsequently a big hunk of their analysis would be questionable. Also, what were those "some differences"?

---

## Editor Decision (ED1)

**In this reviewer's opinion, the manuscript remains in need of much clarification.**

The authors did not address my question. I originally asked how the F values calculated using S2007 and L2013 were applied in their simulations. Specifically, S2007 computed instantaneous F values, which could technically be included in every time step to quantify ozone damage to vegetation. L2013-calculated F values, however, depended on CUO obtained from integration "over the growing season" (L231) using Eqs. 5, 6, & 7, meaning that there'd be only one pair of FPO3 and FCO3 for their simulation period May – August 2017. So actually, two questions involving L2013: 1. How did they obtain CUO of the growing season for their F value calculations? 2. Was one pair of constant, time-independent FPO3 and FCO3 values applied to every time step throughout the simulation period? It was not apparent to me how L2013 was coupled with the land surface model and WRF-Chem all together.

*Response:* Sorry for the confusion. The CUO accumulates at each time step during the growing season. Both FPO3 and FCO3 are calculated based on the CUO by each time step instead of the whole growth season. Therefore, FPO3 and FCO3 are different day by day during the growing season. At the end of the growing season, the L2013-based damages are greater than that at the early stage, theoretically. However, the L2013 scheme applies $a_p=0$ for evergreen broadleaf forest, needleleaf forest, deciduous broadleaf forest, and shrubland, $a_c=0$ for evergreen broadleaf forest, deciduous broadleaf forest, shrubland, grassland, and cropland (Table 2), suggesting that these PFTs employ constant F values due to time-independent $O_3$ sensitivity even if the CUO is varying day to day. In this revision, we clarified that "The leaf-level CUO (mmol m$^{-2}$) is calculated by accumulating stomatal $O_3$ fluxes of Equation 4 from the start of the growing season to the specific time step." (Lines 227-229)

**Reviewer: The authors stated that they used Lombardozzi et al. (2013)'s parameterizations for their study (L209). I am confused from where in Lombardozzi et al. (2013) the authors obtained their $a_p$, $a_c$, $b_p$, and $b_c$ for the 6 vegetation types in their Table 2. In their results from "the exposed to charcoal-filtered air with medium or high confidence in cumulative O3 uptake (CUO) calculations", Lomdardozzi et al. (2013) showed no significance in the linearly regressed equations of photosynthesis in % of control vs. CUO for all plant types except crops and showed no significance in the linearly regressed equations of conductance in % control vs. CUS for all plant types except temperate evergreen trees (L2013's Tables 2&3). In their results from "ambient air" data, Lomdardozzi et al. (2013) showed no significance in the linearly regressed equations of photosynthesis in % of control vs. CUO and conductance in % control vs. CUO for all plant types except "temperature deciduous trees" (L2013's Tables B1&B2).**

**The values the authors used that I recognized, albeit not the ones intended for their purposes in this reviewer's opinion, were 2 orders of magnitude smaller than those in Lombardozzi et al. (2013). This reviewer was taken by surprise by the authors' statement that most of their plant types had "time-independent" sensitivity to CUO since $a_c$ and $a_p$**

values were zero. First, I did not see zero values for $a_c$ and $a_p$ in Lombardozzi et al. (2013); instead, L2013 showed no significance in regression for most plants as stated above. Second, if what the authors stated were true, it'd totally defeat the purpose of that epic study of Lombardozzi et al. (2013)'s. In short, it was very confusing how and where the authors got the values in their Table 2 from.

Further, Lombardozzi et al. (2013) emphasized "chronic ozone exposure" throughout their work, and thus they included the studies that used experimental periods longer than 7 days. That means that the parameterizations derived from L2013 would be only applicable for calculations over periods > 7 days. Hence, the question is: how could the authors' calculations for times shorter than that be valid?

Since S2007 calculated instantaneous effects while L2013 the effect of CUO, it is critical to know what exactly was presented in Figures 2 and 3. The author just stated "$O_3$ damage", but they had 3 months simulations. The two figures must be showing post processed values. So, what exactly was shown in those figures? This question points to the comparability of those two figures and consequently their main findings.

---

## Author Response (AR2)

We thank very much for the helpful comments and suggestions from the reviewer, which help us improve our manuscript. The comments were carefully considered and revisions have been made in response to suggestions. The second round of author responses to review comments are shown in red text.

First, S2007 seems to calculate instantaneous (for WRF-Chem's model integration time steps or hourly) values of the undamaged fraction F, whereas L2013 calculates the ozone damage ratio for the entire growing season. So, was one constant L2013-calculated, plant-specific, O3 damage ratio applied throughout the whole simulation period, whereas S2007-calculated O3 damage ratios were time-dependent, when the schemes were coupled with WRF-Chem?

*Response:* As mentioned by the referee, the ozone damage calculated by the S2007 scheme is related to instantaneous excessive ozone flux ($dFO_3$), while the ozone damage calculated by the L2013 scheme is related to the cumulative ozone uptake flux (CUO). As shown in Figure R1, both CUO and $dFO_3$ vary with time. The value of CUO increases month by month, reaching a maximum in August. In contrast, $dFO_3$ is affected by instantaneous $O_3$ concentration, which peaks in July, leading to highest $dFO_3$ in July.

[Figure]

Figure R1 Monthly mean CUO and dFO3 calculated for L2013 and S2007 schemes, respectively. Here $dFO_3 = max\{f_{O_3} - y_{PFT}, 0\}$ in equation (3) of main text.

The authors did not address my question. I originally asked how the F values calculated using S2007 and L2013 were applied in their simulations. Specifically, S2007 computed instantaneous F values, which could technically be included in every time step to quantify ozone damage to vegetation. L2013-calculated F values, however, depended on CUO obtained from integration "over the growing season" (L231) using Eqs. 5, 6, & 7, meaning that there'd be only one pair of FPO3 and FCO3 for their simulation period May – August 2017. So actually, two questions involving L2013: 1. How did they obtain CUO of the growing season for their F value calculations? 2. Was one pair of constant, time-independent FPO3 and FCO3 values applied to every time

step throughout the simulation period? It was not apparent to me how L2013 was coupled with the land surface model and WRF-Chem all together.

*Response:* Sorry for the confusion. The CUO accumulates at each time step during the growing season. Both FPO3 and FCO3 are calculated based on the CUO by each time step instead of the whole growth season. Therefore, FPO3 and FCO3 are different day by day during the growing season. At the end of the growing season, the L2013-based damages are greater than that at the early stage, theoretically. However, the L2013 scheme applies ap=0 for evergreen broadleaf forest, needleleaf forest, deciduous broadleaf forest, and shrubland, ac=0 for evergreen broadleaf forest, deciduous broadleaf forest, shrubland, grassland, and cropland (Table 2), suggesting that these PFTs employ constant F values due to time-independent $O_3$ sensitivity even if the CUO is varying day to day. In this revision, we clarified that "The leaf-level CUO (mmol m$^{-2}$) is calculated by accumulating stomatal $O_3$ fluxes of Equation 4 from the start of the growing season to the specific time step." (Lines 227-229)

Third, isn't Eq. 5 supposed to be the integration of Eq. 4 according to its definition?

*Response:* By theory the accumulative flux (Eq. 5) should be the integration of instantaneous flux (Eq. 4). In practice, Eq 4 was used in the S2007 scheme while Eq. 5 was used in L2013 scheme with some differences. We maintained such differences because $O_3$ sensitivity parameters were derived based on the corresponding $O_3$ stomatal fluxes.

What I meant was that in the manuscript, Eq. 5 was not the integration form of Eq. 4 as so intended.

$$f_{O_3} = \frac{[O_3]}{r_a + k_{O_3} \cdot r_s} \tag{4}$$

$$\text{CUO} = \sum (k_{O_3}/r_s + 1/r_a) \times [O_3] \tag{5}$$

If they used Eq. 5 to calculate CUO, their L2013-calculated results and subsequently a big hunk of their analysis would be questionable. Also, what were those "some differences"?

*Response:* We appreciate the reviewer's kind suggestion and rigorous check of the equations. We had a careful check of our codes and found that we used incorrect expression in the text by citing the same formula from Jin et al.'s (2023) without verification. In the code (Figure R2), we actually applied equation (4) to calculate instantaneous $O_3$ stomatal flux and estimated CUO by from the start of the growing season to the specific time step. We clarified in the text that: "The leaf-level CUO (mmol m$^{-2}$) is calculated by accumulating stomatal $O_3$ fluxes of Equation 4 from the start of the growing season to the specific time step." (Lines 227-229)

We sincerely apologized for this text error and assured that correct codes were applied in the model. As shown in the first author's Ph.D dissertation (Figure R3), which was

published on September 2022, the correct format has been used (text in Chinese). For this revision, we have attached the complete Ph.D dissertation for your reference (Page 61)

```fortran
if ((ddvel(i,j,p_o3)*chem(i,1,j,p_o3)*(1960*1000)/48 .gt. 0.8)& !O
        .and.(depuse_lai(landinuse) .gt. 0.4))then
cuo_add(i,j)=cuo_add(i,j)+ &
        (o3conc/(1.67*rsadd+rb_to_chem(i,j))*dtstep*1.0e-6)
cuo_sun(i,j)=cuo_sun(i,j)+ &
        (o3conc/(1.67*rssunxy(i,j)+rb_to_chem(i,j))*dtstep*1.0e-6)
cuo_sha(i,j)=cuo_sha(i,j)+ &
        (o3conc/(1.67*rsshaxy(i,j)+rb_to_chem(i,j))*dtstep*1.0e-6)
select case(landinuse)
   case(1,16,19,24)
     fpo3_lh(i,j)=1.
     fpo3_sun_lh(i,j)=1.
     fpo3_sha_lh(i,j)=1.
     fco3_lh(i,j)=1.
     fco3_sun_lh(i,j)=1.
     fco3_sha_lh(i,j)=1.
   case(13,15,8,9,20,21,22,23) !EBF,SHR
     fpo3_lh(i,j)=(-0.0018)*cuo_add(i,j)+0.99
     fpo3_sun_lh(i,j)=(-0.0018)*cuo_sun(i,j)+0.99
     fpo3_sha_lh(i,j)=(-0.0018)*cuo_sha(i,j)+0.99
     fco3_lh(i,j)=-0.0082*cuo_add(i,j)+0.5892
     fco3_sun_lh(i,j)=-0.0082*cuo_sun(i,j)+0.5892
     fco3_sha_lh(i,j)=-0.0082*cuo_sha(i,j)+0.5892
   case(11) !DBF
```

Figure R2 Codes of L2013

调节，该理论已经在不同植被类型上得到了验证[309-310]。CUO (mmol·m⁻³) 的计算公式为：

$$CUO = 10^{-6} \sum \frac{[O_3]}{kO_3 R_s + R_b} \Delta t \qquad (4.3)$$

其中，$[O_3]$ 为地表 $O_3$ 浓度 (nmol·m⁻³)；$kO_3$=1.67 是叶片对 $O_3$ 的阻抗与叶片对水的水的比值[149]；$R_s$ 为气孔阻抗 (s·m⁻¹)；$R_b$ 为边界层阻抗 (s·m⁻¹)；$\Delta t$ 为模式模拟时间步长。在生长季，植被最容易受到空气污染的影响，因此将生长季节定义为 LAI 大于 0.4。且当 $O_3$ 通量高于阈值 0.8 nmolO₃m⁻²s⁻¹ 时，CUO 才会累积，以考虑生长季植物对 $O_3$ 的解毒作用[23]。

Figure R3 The CUO formula in the first author's Ph.D dissertation

Reference:

Jin, Z., Yan, D., Zhang, Z., Li, M., Wang, T., Huang, X., et al. (2023). Effects of elevated ozone exposure on regional meteorology and air quality in China through ozone-vegetation coupling. Journal of Geophysical Research: Atmospheres, 128, e2022JD038119. https://doi.org/10.1029/2022JD038119

**暨南大学博士学位论文**

题名（中英对照）：

典型下垫面臭氧干沉降机制改进及生态环境效应评估

**Improvement of ozone dry deposition mechanism and assessment of ecological effects over typical terrestrial ecosystems**

作者姓名：曹嘉晨

指导教师姓名及学位、职称：王雪梅 博士 教授

常鸣 博士 副教授

学科、专业名称：理学 生态学

学位类型：学术型博士学位

论文提交时间：2022年7月

论文答辩时间：2022年9月

答辩委员会主席：

论文评阅人：匿名评审

学位授予单位日期：暨南大学 2022年10月

**暨南大学博士学位论文**

**独 创 性 声 明**

本人声明所呈交的学位论文是本人在导师指导下进行的研究工作及取得的研究成果。除了文中特别加以标注和致谢的地方外，论文中不包含其他人已经发表或撰写过的研究成果，也不包含为获得 ___暨南大学___ 或其他教育机构的学位或证书而使用过的材料。与我一同工作的同志对本研究所做的任何贡献均已在论文中作了明确的说明并表示谢意。

学位论文作者签名： 曹嘉晨    签字日期：2022年 9月 25日

**学位论文版权使用授权书**

本学位论文作者完全了解 ___暨南大学___ 有关保留、使用学位论文的规定，有权保留并向国家有关部门或机构送交论文的复印件和磁盘，允许论文被查阅和借阅。本人授权 ___暨南大学___ 可以将学位论文的全部或部分内容编入有关数据库进行检索，可以采用影印、缩印或扫描等复制手段保存、汇编学位论文。

（保密的学位论文在解密后适用本授权书）

学位论文作者签名： 曹嘉晨    导师签名： 王季槐

签字日期：2022年9月25日    签字日期：2022年 9月 25日

学位论文作者毕业后去向：

工作单位：          电话：

通讯地址：          邮编：

**摘  要**

  对流层臭氧 ($O_3$) 是影响我国空气质量最重要的污染物之一，预计未来 $O_3$ 污染还会进一步加剧。大气中的 $O_3$ 主要通过干沉降过程作用于陆地生态系统，造成一系列负面的生态环境效应。一方面，高浓度的地表 $O_3$ 会导致气孔关闭，减少植被的光合作用，从而抑制其正常生长，导致作物产量降低；另一方面，$O_3$ 与植被的相互反馈不仅会影响植被的生理生态，还会对区域气象产生影响。相较于国外，我国对臭氧干沉降过程的数值模拟研究起步较晚。与此同时，当前主流大气化学传输模式 WRF-Chem 对 $O_3$ 干沉降过程的表达并未考虑 $O_3$ 与植被之间的相互反馈机制，严重阻碍了我们对中国地区臭氧干沉降过程变化的理解和预测。

  本文首先基于单点干沉降机理诊断模式 Noah-MP-WDDM，结合站点观测数据诊断和识别了影响森林和农田下垫面 $O_3$ 干沉降的关键过程，并评估不同气孔导度机制在不同下垫面的适用性，识别出最优气孔导度机制。随后在区域模式 WRF-Chem 中建立了最优气孔导度机制与 $O_3$ 化学沉降模块的程序接口，使得 WRF-Chem 中的 Wesely 干沉降机制可灵活调用此气孔导度机制进行计算。同时，构建了 $O_3$ 损伤植被的参数化方案，将臭氧累积吸收通量对植被光合速率和叶片气孔导度的影响过程考虑进区域数值模式中，并与 WRF-Chem 中的干沉降机制进行耦合，实现了在 WRF-Chem 中 $O_3$ 与植被的双向反馈和 $O_3$ 累积作用的表征。在此基础上，运用所改进的模式分析了我国典型下垫面 $O_3$ 干沉降过程及未来气候变化背景下潜在的生态环境效应。研究得到的主要结论如下：

  典型亚热带森林下垫面冠层外和冠层内观测的日间臭氧干沉降速率 ($V_d(O_3)$) 分别为 $0.75\,\mathrm{cm \cdot s^{-1}}$ 和 $0.30\,\mathrm{cm \cdot s^{-1}}$。农田下垫面观测的日间 $V_d(O_3)$ 为 $0.45\,\mathrm{cm \cdot s^{-1}}$。结合观测数据发现，不同气孔导度机制是短期内影响 $V_d(O_3)$ 日变化过程的关键因素，其中 Ball-Berry 气孔导度机制对典型亚热带森林和农田下垫面的 $V_d(O_3)$ 模拟效果均优于 Jarvis 气孔导度机制。此外，冠层阻抗 ($R_c$) 是影响白天 $V_d(O_3)$ 变化的主导因子，夜间则是由空气动力学阻抗 ($R_a$) 引起的湍流混合作用主导。

  基于改进后的模式对我国 $O_3$ 干沉降开展模拟，结果发现，改进后的模式对典型亚热带森林和农田下垫面上 $O_3$ 浓度的模拟精度分别提高了 10.0% 和 8.8%；对 $V_d(O_3)$ 模拟精度分别提高了 51.9% 和 28.6%。对于臭氧干沉降通量 ($O_3ddep$)，其在农田下垫面的改进效果不明显，仅提升了 0.3%；而对典型亚热带森林下垫面，模式改进后 $O_3ddep$ 模拟精度显著提高了 16.8%。此外，模式改进还引起区域植被生理过程和气象要素的一系列改变，其中使

气孔阻抗平均增加了 5.6%，光合速率降低了 6.4%。气孔阻抗和光合速率的变化进一步导致叶面积指数和总初级生产力平均降低了 4.1% 和 9.5%。蒸腾速率的降低导致潜热通量平均下降了 6.3%，显热通量增加了 8.5%，进而使温度升高，并造成了异戊二烯排放的增加。植被生理过程和气象要素的变化促使区域 $O_3$ 浓度平均上升了 3.2 $\mu g \cdot m^{-3}$，$V_d(O_3)$ 平均下降了 0.06 $cm \cdot s^{-1}$，$O_3ddep$ 平均下降了 0.9 $kg \cdot ha^{-1} \cdot month^{-1}$。

我国农田、森林、草地和城市下垫面的 $O_3ddep$ 分别为 596.3 $kg \cdot km^{-2} \cdot month^{-1}$、555.7 $kg \cdot km^{-2} \cdot month^{-1}$、528.9 $kg \cdot km^{-2} \cdot month^{-1}$ 和 323.9 $kg \cdot km^{-2} \cdot month^{-1}$。$O_3$ 浓度和 $V_d(O_3)$ 对 $O_3ddep$ 的相对贡献分别为 34.4% 和 63.8%。其中，影响 $O_3$ 浓度日内变化的关键大气物理化学过程存在显著的昼夜差异，白天主要由垂直混合、化学和干沉降过程主导，贡献率分别为 33.6%、29.7% 和 19.5%，而夜间 $O_3$ 浓度的变化则是由化学过程主导，贡献率达到 45.6%。对于 $V_d(O_3)$，$R_a$ 和 $R_c$ 主导了其日内变化，相对贡献分别为 53.2% 和 43.4%。

通过当前与未来气候变化情景下 (RCP6.0 BAU，RCP4.5 ECP 和 RCP4.5 BHE) 的 $O_3$ 干沉降过程及其对作物产量、经济损失影响的模拟研究发现，不同气候变化情景下 $O_3$ 浓度分别下降 3.7、5.8% 和 11.6%，$V_d(O_3)$ 分别上升了 4.3%，2.0% 和 2.3%。$O_3$ 浓度和 $V_d(O_3)$ 的变化进一步使 $O_3ddep$ 分别下降了 2.9%、6.5% 和 8.4%。同时，不同气候变化情景下 AOT40 的变化导致全国双季早稻的减产量分别为 169.3 万吨、70.5 万吨和 47.6 万吨；相应的经济损失分别为 689.1 百万美元、288.2 百万美元和 195.2 百万美元。未来如果继续执行当前的污染控制政策 (RCP6.0 BAU)，$O_3$ 污染会对作物造成更高的伤害和经济损失；执行最佳污染控制政策 (RCP4.5 BHE)，作物的减产量会减少一半以上，经济损失也会大幅度减少。

**关键词**：臭氧；干沉降；下垫面；WRF-Chem；Noah-MP

**ABSTRACT**

Tropospheric ozone ($O_3$) is one of the most critical pollutants affecting air quality in China, and $O_3$ pollution is expected to increase further. Atmospheric $O_3$ acts on terrestrial ecosystems through dry deposition, causing adverse ecological effects. On the one hand, high surface $O_3$ concentration can cause stomatal closure and reduce the photosynthesis of vegetation, thus inhibiting its average growth and resulting in lower crop yields. Compared with foreign countries, the numerical simulation study of the $O_3$ dry deposition process in China started late. Meanwhile, the current mainstream atmospheric chemical transport model WRF-Chem does not consider the mutual feedback mechanism between $O_3$ and vegetation, which seriously hinders our understanding and prediction of the changes in the $O_3$ dry deposition process in China.

Based on the single-point dry deposition mechanism diagnostic model Noah-MP-WDDM, we first diagnosed and identified the key processes affecting $O_3$ dry deposition over forests and agriculture in this study. Then we assessed the applicability of different stomatal conductance mechanisms on various underlying surfaces and identified the optimal one. Subsequently, a programmatic interface between the optimal stomatal conductance mechanism and the $O_3$ chemical deposition module was established in the WRF-Chem, so that the Wesely dry deposition scheme in the WRF-Chem could flexibly invoke this stomatal conductance mechanism for calculations. At the same time, a parametric scheme for $O_3$ damage to vegetation was constructed. The cumulative $O_3$ absorption flux on vegetation photosynthetic rate and leaf stomatal conductance was taken into account in the regional numerical model and coupled with the dry deposition scheme in the WRF-Chem to realize the two-way feedback between $O_3$ and vegetation and the characterization of cumulative $O_3$ effect in WRF-Chem. On this basis, the improved model was used to analyze the $O_3$ dry deposition process in the typical substratum of China and the potential ecological and environmental effects in the context of future climate change. The main conclusions of the study were as follows:

The daytime ozone dry deposition velocity ($V_d(O_3)$) observed outside and inside the canopy of a typical subtropical forest were 0.75 cm·s$^{-1}$ and 0.30 cm·s$^{-1}$, respectively. The daytime $V_d(O_3)$ was 0.45 cm·s$^{-1}$ in an agricultural understory. Combining the observed data, different stomatal conductance mechanisms were found to be the key factors affecting the daily variation of the $V_d(O_3)$

process in the short term. The Ball-Berry stomatal conductance mechanism outperforms the Jarvis stomatal conductance mechanism for simulating $V_d(O_3)$ in a typical subtropical forest and agriculture. In addition, canopy resistance (Rc) was the dominant factor affecting the daytime $V_d(O_3)$ variation, while the nighttime was dominated by turbulent mixing due to aerodynamic resistance (Ra).

Based on the improved model, the simulation of $O_3$ dry deposition in China was launched. It was found that the simulation accuracy of the enhanced model for the $O_3$ concentration over the typical subtropical forest and agriculture was increased by 10.9% and 13.2%, respectively. The accuracy was improved by 51.9% and 28.6%, respectively. For the ozone dry deposition flux ($O_3$ddep), the improvement effect on agriculture was not noticeable and only increased by 0.3%. In contrast, for the typical subtropical forest, the simulation accuracy of $O_3$ddep was significantly improved by 16.8% after the model improvement. In addition, the model improvement also caused a series of changes in regional vegetation physiological processes and meteorological elements, which increased the stomatal resistance by 5.6% on average and decreased the photosynthetic rate by 6.4%. Changes in stomatal resistance and photosynthetic rate further resulted in an average reduction of 4.1% and 9.5% in leaf area index and total gross productivity. The decline in transpiration rate resulted in an average 6.3% decrease in latent heat flux and an 8.5% increase in sensible heat flux, increasing temperature and increasing isoprene emissions. Changes in vegetation physiological processes and meteorological elements led to an average increase of $O_3$ concentration by 3.2 $\mu$g·m$^{-3}$, an average decrease of $V_d(O_3)$ by 0.06 cm·s$^{-1}$, and an average decrease of $O_3$ddep by 0.9 kg·km$^{-2}$·month$^{-1}$.

The simulation results of the $O_3$ dry deposition process showed that the $O_3$ddep over agriculture, forest, grassland, and urban areas were 596.3 kg·km$^{-2}$·month$^{-1}$, 555.7 kg·km$^{-2}$·month$^{-1}$, 528.9 kg·km$^{-2}$·month$^{-1}$, and 323.9 kg·km$^{-2}$·month$^{-1}$, respectively. The relative contributions of $O_3$ concentration and $V_d(O_3)$ to $O_3$ddep were 34.4% and 63.8%, respectively. Among them, there were significant diurnal differences in the critical atmospheric physicochemical processes affecting the intra-day variation of $O_3$ concentrations, which were dominated by vertical mixing, chemical and dry deposition processes during daytime with contributions of 33.6%, 29.7%, and 19.5%, respectively, and chemical processes were the main processes causing the variation of $O_3$ concentrations at night with contributions of 45.6%. For $V_d(O_3)$, Ra and Rc dominated its intra-day interpretation with relative contributions of 53.2% and 43.4%, respectively.

Through the simulation study of the $O_3$ dry deposition process and its impact on crop yield

and economic loss under the current and future climate change scenarios, it is found that under different climate change scenarios, the $O_3$ concentration was decreased by 3.7%, 5.8%, and 11.6%, respectively, and $V_d(O_3)$ was increased by 4.3%, 2.0%, and 2.3%, respectively. Changes in $O_3$ concentration and $V_d(O_3)$ further decreased the $O_3$ddep by 2.9%, 6.5%, and 8.4%, respectively. At the same time, the changes in AOT40 under different climate change scenarios led to the reduction of the national double-early rice yield of 169.3 million tons, 70.5 million tons, and 47.6 million tons, respectively; the corresponding economic losses were 689.1million USD, 288.2 million USD, and 195.2 million USD, respectively. In the future, if the current pollution control policy is continued (RCP6.0 BAU), $O_3$ pollution will cause more severe damage and economic losses to crops; if the optimal pollution control policy (RCP4.5 BHE) is implemented, crop yield reduction will be reduced by more than half, and economic losses will also be significantly reduced.

The simulation results of $O_3$ dry deposition process showed that the $O_3$ddep over agriculture, forest, grassland, and urban areas were 596.3 kg·km$^{-2}$·month$^{-1}$, 555.7 kg·km$^{-2}$·month$^{-1}$, 528.9 kg·km$^{-2}$·month$^{-1}$, and 323.9 kg·km$^{-2}$·month$^{-1}$, respectively. The relative contributions of $O_3$ concentration and $V_d(O_3)$ to $O_3$ddep were 34.4% and 63.8%, respectively. Among them, there were significant diurnal differences in the critical atmospheric physicochemical processes affecting the intra-day variation of $O_3$ concentrations, which were dominated by vertical mixing, chemical and dry deposition processes during daytime with contributions of 33.6%, 29.7%, and 19.5%, respectively, and chemical processes were the main processes causing the variation of $O_3$ concentrations at night with contributions of 45.6%. For $V_d(O_3)$, $R_a$ and $R_c$ dominated its intra-day variation with relative contributions of 53.2% and 43.4%, respectively.

**Key Words:** $O_3$; Dry deposition; Underlying surface; WRF-Chem; Noah-MP

**目  录**

**插图**

**表格**

**第一章 绪论**

**1.1 研究背景**

  臭氧 ($O_3$) 是大气中的一种痕量气体，占大气体积百分比不到 0.0012%，但却是一种非常重要的大气成分。对流层 $O_3$ 在气候、大气化学、生物圈和空气质量等方面起着重要作用。从 1950 年到 2000 年，由于 $O_3$ 前体物的人为排放不断增加，使得在中纬度地区的对流层 $O_3$ 浓度大约增加了一倍[1]。到下世纪末，$O_3$ 浓度仍然会持续上升，2100 年全球 $O_3$ 浓度水平较 2015 年预计增加 40~60%[2]。干沉降作为对流层 $O_3$ 清除的最关键过程，它通过调节 $O_3$ 从大气中的去除速度，对地表 $O_3$ 的生命周期和浓度至关重要[3]。图1.1总结了陆地 $O_3$ 干沉降的相关过程，以及对对流层大气化学、空气质量、生态系统和气候的影响[4]。

  随着近几十年来工业化和城市化进程的加快，我国空气质量问题日趋严重，对国民经济和健康造成了严峻的危害。2013 年我国颁布了《大气污染防治行动计划》，开始了以 $PM_{2.5}$ 为重点的大气污染防治工作 (www.gov.cn/zwgk/2013-09/12/content_2486773.htm)，颗粒物质量浓度显著下降，空气质量得到了改善，但是 $O_3$ 污染却日益严重。相比于 2015 年，2019 年 337 个城市 $SO_2$、$PM_{2.5}$、$CO$、$PM_{10}$ 和 $NO_2$ 年评价浓度平均值分别下降了 52.0%、22.0%、28.6%、19.5% 和 3.3%，而 $O_3$ 年评价浓度的平均值上升了 20.1%；$O_3$ 浓度年评价值的范围为 89~229 μg·m$^{-3}$，平均值为 161 μg·m$^{-3}$，有 161 个城市 (占 47.8%) $O_3$ 浓度年评价值高于国家二级标准限值 (GB 3905-2012，160 μg·m$^{-3}$)，未达标城市比 2018 年增加了 46 个[5]。日益严峻的 $O_3$ 污染已经成为了制约我国空气质量持续改善的关键性指标。

  对流层 $O_3$ 浓度受到光化学过程、$O_3$ 干沉降和大气扩散作用的影响[6-7]。过去几十年中对 $O_3$ 的研究主要集中于对 $O_3$ 前体物 (人为源和自然来源)，对流层的传输以及平流层输入对对流层 $O_3$ 分布等方面[1,8-10]。$O_3$ 作为一种空气污染物，许多国家出台了相应的管控政策，因此对 $O_3$ 的源汇变化和趋势研究具有政策相关性。一些研究中强调了源对于 $O_3$ 浓度的影响，但是忽略了 $O_3$ 干沉降对 $O_3$ 浓度的重要性。干沉降是 $O_3$ 从对流层大气中移除的重要途径，根据大气化学模式的估计，每年全球对流层 $O_3$ 损失的 20% 来自于向地表的干沉降[11-13]，表明干沉降是控制 $O_3$ 污染的重要手段之一。

  $O_3$ 的干沉降主要通过气孔途径和非气孔途径进行沉降。气孔沉降指植被通过气孔吸收 $O_3$ 通量，植被又可以通过气孔影响 $O_3$ 的源和汇来调节地表 $O_3$ 浓度。定量气孔 $O_3$ 吸收量不仅有利于准确估算 $O_3$ 的去除量，并且对进一步了解植物对 $O_3$ 的响应关系也至关重要。气

孔对 $O_3$ 的吸收会产生活性氧，导致细胞死亡和损伤，从而加速植物衰老[14-15]，损害光合酶活性，增强呼吸作用，干扰碳分配[14-15]，改变植物的叶面积和生物量累积[16-19]，因此气孔 $O_3$ 沉降通量被认为是对植物进行 $O_3$ 风险评估最有效的方法[20]。其次，气孔通过控制水分的排出和进入叶片内部的碳，从而影响陆气之间的水循环和碳循环[21-27]，边界层气象[25,28-30]和气候[31-32]。当植被暴露于高浓度的 $O_3$ 中时，会导致光合速率下降，从而改变 $CO_2$ 同化。$O_3$ 暴露还会导致气孔导度降低，从而减少 $O_3$ 的汇和增加地表 $O_3$ 浓度[25,29-30,33]。尽管 $O_3$ 和植被之间的相互作用对大气环境至关重要，但在大多数大气化学传输模式 (CTMs，Chemical Transport Models) 中，仍然缺乏对 $O_3$ 和植被相互反馈的描述，其主要原因是由于陆面过程和生物圈模式在高分辨率模式下的耦合能力不足，还有部分原因是观测所得的 $O_3$ 损害方案较难在区域中广泛适用。另外一个重要的 $O_3$ 干沉降途径是非气孔沉降，它包括叶片角质层和土壤等。非气孔沉降可以占到总沉降的 45% 左右[34]。试验表明，潮湿的植物叶片可以增强叶角质层上的 $O_3$ 损失，其强度与叶片释放出的化合物有关[35]。$O_3$ 沉降到土壤中可能是由于 $O_3$ 扩散到土壤孔隙中并与土壤中有机物的不饱和双碳键发生反应的结果[36]。当土壤比较潮湿时，随着可用于反应的表面积减少，沉降到土壤中的 $O_3$ 量变会降低[37]。在许多模式对非气孔沉降参数化的考虑，主要是通过查表的方式确定，但是该方法并不会考虑气象变化对参数的影响，这可能会错过许多影响非气孔沉降的重要过程[38-41]。

由于直接测量干沉降过程难度较高并且测量设备比较昂贵[42-43]，因此在区域尺度上多用 CTMs 来进行干沉降过程的研究。当前大多数 CTMs 通常使用 Wesely (1989) 方案计算气体干沉降速率。该方案主要是根据短期观测结果对白天与夜间以及不同土地覆盖类型和季节的 $O_3$ 干沉降进行调整，并成功应用在评估特定土地覆盖类型的 $O_3$ 干沉降速率的观测和模拟上[45-47]。一些模式根据 Zhang (2003) 的方案增加了由观测到得到的气象要素和叶片表面湿润度对 $O_3$ 干沉降速率的影响，一些模式增加了更多对气孔导度过程的表征[49-52]。而对流层 $O_3$ 的数值模拟，包括高浓度的 $O_3$ 污染事件和背景 $O_3$ 浓度的水平，对 $O_3$ 干沉降的参数化方案很敏感[9,46,53-65]。然而，许多广泛使用的 $O_3$ 干沉降参数化方案并不能准确表征出 $O_3$ 干沉降过程或者不能捕获到观测到的时空变化[45,50,66-68]，进而会影响对 $O_3$ 浓度的模拟。在不同的 CTMs 中，由于使用的干沉降参数化方案不同，对 $O_3$ 干沉降月或年均值的估算差异可达两到三倍[63,69-72]。总的来说，当前对 $O_3$ 的气孔和非气孔沉降仍缺乏了解。在 CTMs 中，若不能准确模拟出 $O_3$ 干沉降速率的变化[38,73-74]，可能会影响对 $O_3$ 浓度的模拟，并错误地反映其他过程对 $O_3$ 浓度的影响。因此，如何准确评估气孔和非气孔沉降是提高 $O_3$ 干沉降速率模拟效果的一个难题。

[Figure]

图 1.1: 陆地 $O_3$ 干沉降的相关过程及其对生态环境的影响[34]

**1.2 国内外研究现状**

**1.2.1 臭氧干沉降的观测方法**

$O_3$ 干沉降的测量方法主要包括直接测量和间接测量两种。直接测量是指垂直 $O_3$ 通量直接在近地面测量。而间接测量则是指 $O_3$ 沉降物质的平均浓度或者平均浓度的垂直梯度被测量，$O_3$ 通量数据则通过推演计算间接获得。直接测量一般采用箱式法和涡动相关法，间接测量主要是梯度法。

**1.2.1.1 箱式法**

箱式法可以对叶片、土壤、水或者其他表面的 $O_3$ 吸收进行直接测量[75-81]。但是，以往的研究主要集中于土壤 NO 的排放[82-84]或植物对 $O_3$ 的响应[75]，而不是 $O_3$ 干沉降过程。

箱式法主要是由通量箱、鼓风机、$O_3$ 发生器、$O_3$ 分析仪和布气管道等几部分组成。用通量箱覆盖于植物上方，可以直接测量植物整个冠层水平上的 $O_3$ 吸收通量[20]。该方法一般采用开放式测量系统，当箱内气体达到稳定状态时，气体质量平衡方程可表达为[75]：

$$FC_{in} - FC_{out} - F\Delta C_{ww} - K_{chamber}C_{out}V = 0 \tag{1.1}$$

其中，F 为通量箱内气体流速 $(m^3 \cdot s^{-1})$，$C_{in}$ 和 $C_{out}$ 分别为通量箱进、出气口气体浓度 $(mol \cdot m^{-3})$，$\Delta C_{ww}$ 为由箱内的植物体引起的气体浓度变化 $(mol \cdot m^{-3})$，$K_{chamber}$ 为箱壁气体吸收速率常数 $(s^{-1})$，V 为箱体体积 $(m^3)$。实验交替测量出空箱和植物箱内的气体交换，算出 $\Delta C_{ww}$ 后带入下式计算 $O_3$ 通量：

$$F_{O_3} = \frac{F\Delta C_{ww}P}{ART_a} \tag{1.2}$$

其中，$F_{O_3}$ 为 $O_3$ 沉降通量 $(nmol \cdot m^{-2} \cdot s^{-1})$；P 为气压 (Pa)；A 为静态箱所覆盖的土地面积 $(m^2)$；R 为普适气体常数 $(J \cdot mol \cdot K^{-1})$；$T_a$ 为绝对温度 (K)。

**1.2.1.2 涡动相关法**

涡动相关法 (EC，Eddy-covariance technique) 是一种更直接测量植被与大气间 $O_3$ 通量的方法。在最近几十年得到了快速的发展，被认为是现今最好的测量地气交换的手段。该方法最早应用于水汽 $(H_2O)$ 通量测量，后来拓展到二氧化碳 $(CO_2)$ 通量的研究中[85-88]。这一方法需要比较精密的仪器。其基本设备主要包括一个三维超声风速仪 (CSAT3) 和一个高速响应红外气体分析仪 (IRGA)。该方法应用于 $O_3$ 通量的测量时需要在原基础设备中增加一台快速和精确的化学发光的 $O_3$ 传感器 (FOS，Fast ozone sensor)[20]。该探测器通常是利用乙烯基或芳香物等与 $O_3$ 发生化学反应并产生蓝光的原理来实现对 $O_3$ 浓度变化的快速测量[44]。EC 法通过计算垂直风速 (w) 与气体浓度 (c) 之间的协方差来确定湍流通量 (F)：

$$F = \overline{w'c'} \tag{1.3}$$

其中，上划线 (—) 表示时间平均；撇号 (') 表示瞬时值与平均值的偏差 $(x'=x(t)-\overline{x}$，$\overline{x}=mean)$；正通量向上为排放；负通量向下为沉降。

涡动尺度的峰值功率谱取决于测量的高度，会随高度而增大，同时涡动会随植被/地表粗糙度和风速的增大而增大[20]。在慢速紫外 $O_3$ 分析仪观测的 $O_3$ 浓度校准的基础上，$O_3$ 通量可以由具备快速 $O_3$ 传感器的涡动相关法直接测量。

**1.2.1.3 通量梯度法**

通量梯度法通常使用慢速 $O_3$ 分析仪就可以监测 $O_3$ 通量，因此此方法比 EC 法更便宜和简单。该方法假设湍流输送类似与分子扩散[89]，认为由湍流所引起的局地的 $O_3$ 通量与局地 $O_3$ 浓度梯度成正比，通量的方向与梯度方向相反。该方法需要测量两个高度以上的 $O_3$ 浓度、风速和温度梯度，表达式为：

$$F_{O_3} = -K_c(z)\frac{dC}{dz} \tag{1.4}$$

其中，$K_c$ 为 $O_3$ 湍流交换系数；$dC/dz$ 为 $O_3$ 的垂直浓度梯度；$F_{O_3}$ 为 $O_3$ 沉降通量。其中，计算 $K_c$ 有两种比较常用的方法，分别是空气动力学法 (AGM，Aerodynamic gradient method) 和修改波文比法 (MBR，Modified Bowen ratio method)。

1. 空气动力学法

空气学动力法假设热量和动量以相同的方式在植被表面传输[90]，$K_c$ 与空气动力学阻抗 ($R_a$) 有关，表达式为：

$$R_a(z_1 : z_2) = \int_{z_2}^{z_1} dz/K_c(z) \tag{1.5}$$

其中，$z_1$ 和 $z_2$ 是冠层以上的两个相邻层的高度 ($z_1 > z_2$)。

使用公式1.4和1.5，沉降通量 (F) 表达式为：

$$F = -\frac{\Delta C}{R_a(z1 : z2)} = -\frac{C_1 - C_2}{R_a(z_1 : z_2)} \tag{1.6}$$

其中，$C_1$ 和 $C_2$ 分别表示 $z_1$ 和 $z_2$ 高度处的气体浓度。

$R_a$ 表达式为：

$$R_a(z1 : z2) = (ku^*)^{-1}[\ln\frac{z_1 - d}{z_2 - d} + \psi h(\frac{z_1 - d}{L}) - \psi h(\frac{z_2 - d}{L})] \tag{1.7}$$

其中，k 为冯卡门常数 (0.4)；$u^*$ 为基准高度的摩擦风速；d 为零位移高度；L 为莫宁-奥布霍夫长度；$\psi h$ 为稳定度函数。

2. 修改波文比法

修改波文比法也是基于通量梯度理论，但是 $K_c$ 值是由另一个标量 (如感热，$CO_2$，$H_2O$) 的通量推导计算的[91-92]。以 $CO_2$ 为例，$CO_2$ 的通量和梯度测量与 $O_3$ 相同的高度，因此 $O_3$ 的 $K_c$ 值可以根据由 $K_{CO_2}$ 的测量值推到计算：

$$K_c = K_{CO_2} = -\frac{F_{CO_2}\Delta Z}{\Delta C(CO_2)} \tag{1.8}$$

其中，$K_{CO_2}$ 为 $CO_2$ 的湍流交换系数，$F_{CO_2}$ 为 $CO_2$ 的涡动相关通量，$\Delta C(CO_2)$ 为不同高度观测的 $CO_2$ 浓度的梯度，$\Delta Z$ 为观测 $CO_2$ 浓度的高度间隔。

使用公式1.4和1.8，沉降通量 (F) 表达式为：

$$F = F_{CO_2}\Delta C(O_3)/\Delta C(CO_2) \tag{1.9}$$

**1.2.1.4 同位素法**

在实验室和野外进行的同位素实验，可以确定 $O_3$ 表面反应的主要地点，从而提高对 $O_3$ 沉降途径的理解[93-94]。Subke 等[94]提出了一种将 $^{18}O$ 加入放电 $O_3$ 发生器，并使用硅胶

将 $^{18}O\,O_3$ 从 $^{18}O\,O_2$ 分离出来的方法。但是，从 $O_3$ 中产生的 $^{18}O$ 会导致 $^{18}O$ 中富含水汽和其他气体，这些气体不一定会停留在表面，从而使得对 $O_3$ 沉降的估计变得复杂[93]。

**1.2.2 现有臭氧干沉降速率观测水平**

自 20 世纪 50 年代以来，就已有学者们利用梯度法对 $O_3$ 干沉降进行短期的观测实验。到了 20 世纪 70 年代，涡动相关法逐渐成为测量通量的首选方法[95]。由于干沉降速率 ($V_d$) 会消除 $O_3$ 浓度对通量的影响，所以通常使用 $V_d(O_3)$ 来研究 $O_3$ 在不同生态系统上的干沉降规律。$V_d(O_3)$ 的变化在很大程度上受到植被的生理活动和气象要素的影响，因此不同下垫面的 $V_d(O_3)$ 存在很大的差异。一方面，不同植被的覆盖状况使 $V_d(O_3)$ 存在明显不同；另一方面，有无植被的覆盖也会使 $V_d(O_3)$ 在量级上产生差别。同时，$V_d(O_3)$ 还会受到局地环境条件等影响。

目前，对 $O_3$ 干沉降的观测实验已经可以覆盖主要的陆地生态系统，时间分辨率从小时到月均值不等。由于特定的冠层特征和浓度测量的不确定性，不同方法测量 $V_d(O_3)$ 的不确定度可大于 50%[96]。Wu 等[91] 的研究表明，不同的测量方法在同一个哈佛森林站点，$V_d(O_3)$ 的差异可到达 2 倍。此外，由于我国对 $O_3$ 干沉降的研究相较于北美和欧洲国家起步较晚，因此观测数据较少，并且这些观测大多是短期的，通常只集中在几个地点 (如南京和郑州，图1.2)。

通过总结过去近 30 年来发表的 35 篇文献中的 $V_d(O_3)$ (表1.1) 可以看出，每个植被类型的 $V_d(O_3)$ 范围区间都比较大，最小值在 0 cm·s$^{-1}$ 附近，最大值可以达到 1.80 cm·s$^{-1}$ 附近。裸地的平均 $V_d(O_3)$ 约为 0.36 cm·s$^{-1}$，而在热带雨林平均 $V_d(O_3)$ 约为 0.80 cm·s$^{-1}$。在巴西亚马逊地区的一些研究表明，热带雨林的 $V_d(O_3)$ 能够超过 1.80 cm·s$^{-1}$[97-98]。此外，Rummel 等[98] 在亚马逊热带雨林发现，雨季的 $V_d(O_3)$ 明显高于旱季。其他森林生态系统中，针叶林 $V_d(O_3)$ 约为 0.54 cm·s$^{-1}$，落叶林 $V_d(O_3)$ 约为 0.63 cm·s$^{-1}$。

由于影响森林生态系统 $O_3$ 气孔吸收的因素很多，不同的森林生态系统 $O_3$ 气孔吸收所占的比例也大相径庭，这些实验所得的数据集并不能提供足够的细节来进一步分解区域或特定物种对 $O_3$ 的吸收。例如在意大利橡树林，$V_d(O_3)$ 全年都较高，干燥且温度较高的气象条件会显著影响 $V_d(O_3)$ 的日变化。此地区以非气孔吸收占主导作用[99]。美国加利福利亚州的橙子园和松林同样是非气孔吸收占主导[39,100]。在地中海的常绿阔叶林地区，则是以气孔吸收起主导作用[101]。在美国的科罗拉多亚高山带森林地区，Turnipseed 等[102]的观测发现，白天气孔吸收占 $O_3$ 通量的 81%。

在农田生态系统中，$V_d(O_3)$ 受作物不同物候阶段的影响[103]，其大小、日变化和季节变化会呈现不同的特点。观测到的最大 $V_d(O_3)$ 在马铃薯地，均值大约为 0.76 cm·s$^{-1}$[8]，最小

的 $V_d(O_3)$ 在玉米地观测到，均值大约为 0.29 cm·s$^{-1}$[104]。在意大利的麦田，正午光合作用最强烈，$V_d(O_3)$ 可达到 0.70~0.90 cm·s$^{-1}$[101]。从作物物候来看，$V_d(O_3)$ 在开花到灌浆期间最大，因为此时期作物光合作用最强，气孔打开进行光合作用[105]，$O_3$ 也更容易进入到作物的气孔。随着作物叶片的衰老，$V_d(O_3)$ 也逐渐减小。在意大利麦田和洋葱的生长旺盛期，气孔吸收比例不高于 50~60%，并随着作物的衰老此占比会逐渐减小[99,106]。然而在马铃薯的生长旺盛期，气孔吸收比例大约占 85% 左右，到了成熟期气孔吸收比例会减少到 20% 左右[8]。利用大叶模型，Lamaud 等[107]区分了玉米地上 $O_3$ 的气孔吸收和非气孔吸收。研究发现，$O_3$ 与 NO 的化学反应对 $O_3$ 的分解起到了重要的作用[105]，气孔吸收的占比会随着气象条件和生育期的变化出现很大的差异。朱治林 等[104]在鲁西北平原的玉米地对 $O_3$ 通量的观测推断白天玉米地上 $O_3$ 的汇主要是气孔吸收。

根据表1.1，目前针对 $V_d(O_3)$ 的观测更偏向于生长季节的月份 (春、夏、秋) 。因此，这制约了评估冬季参数化改进的有效性。但是考虑到干沉降对作物的影响多发生在生长季，这种制约也可能不太重要。如图1.2所示，可以看出，大多观测站点集中于中纬度地区，现有的数据集中多集中于在森林生态系统的测量 (n=23) ，农田生态系统有少量的数据集 (n=15) 。考虑到不同作物在植物物候方面存在巨大差异，有必要通过对不同作物的长期观测来丰富农业生态系统的数据，为干沉降模型提供更多的数据支撑。

[Figure]

图 1.2: 臭氧干沉降速率观测点分布

**1.2.3 臭氧干沉降的模型模拟**

在没有降水的情况下，$O_3$ 从大气中沉降到表面，单位表面积的 $O_3$ 沉降量被定义为垂直干沉降通量 ($F_{O_3}$)。对于干沉降过程的参数化，一般是将植被看作一个叶子并在模型中考

表 1.1: 臭氧干沉降速率观测数据

| 站点 | 纬度 | 经度 | 下垫面类型 | LAI (m$^{-2}$·m$^{-2}$) | 观测时间 | 观测方法 | $V_d(O_3)$(cm·s$^{-1}$) | 参考文献 |
|---|---|---|---|---|---|---|---|---|
| **森林** | | | | | | | | |
| Mea Moh, Thailand | 18.28°N | 99.72°E | 热带雨林 | - | 2002.1-2002.4 | 梯度法 | 0.32(旱季) | Matsuda 等[108] |
| Mea Moh, Thailand | 18.28°N | 99.72°E | 热带雨林 | - | 2004.1-2002.4 | 梯度法 | 0.38(旱季), 0.64(雨季) | Matsuda 等[109] |
| Datum, Borneo | 4.98°N | 117.85°E | 热带雨林 | 6 | 2008.7 | 梯度法 | 0.90(雨季) | Fowler 等[110] |
| Central Amazon, Brazil | 3.0°S | 59.9°W | 热带雨林 | 7 | 1987.4.22-1987.5.8 | 涡动相关法 | 1.80(雨季) | Fan 等[97] |
| Southwest Amazon, Brazil | 10.1°S | 61.9°W | 热带雨林 | 5.6 | 1999.9-1999.10 | 涡动相关法 | 0.50(旱季) | Rummel 等[98] |
| Southwest Amazon, Brazil | 10.1°S | 61.9°W | 热带雨林 | 5.6 | 1999.5.4-1999.5.22 | 涡动相关法 | 1.10(雨季) | Rummel 等[98] |
| Blodgett Forest, California | 38.90°N | 120.63°W | 松树园 | 3.6 | 1999.6-2020.6 | 涡动相关法 | 0.50 | Kurpius 等[111] |
| Blodgett Forest, California | 38.88°N | 120.62°W | 松树园 | 1.2-2.9 | 2001-2007 | 涡动相关法 | 0.92 | Fares 等[112] |
| Citrus Orchard, California | 36.35°N | 119.09°W | 柑橘园 | 3.0 | 2009.10-2010.11 | 涡动相关法 | 0.37 | Fares 等[100] |
| Manitou Forest, Colorado | 39.10°N | 105.10°W | 松树园 | | 2010.8.7-31 | 梯度法 | 0.50 | Park 等[113] |
| Niwot Ridge, Colorado | 40.03°N | 105.55°W | 松树林 | 4.2 | 2002-2005.5-8 | 涡动相关法 | 0.39 | Turnipseed 等[102] |
| Ontario, Canada | 44.32°N | 80.93°W | 落叶林 | 5.0 | 1988 年夏季 | 梯度法 | 1.00 | Padro[114] |
| Ontario, Canada | 44.19°N | 79.56°W | 落叶林 | 4.6 | 2008-2013 | 梯度法 | 0.91 | Wu 等[115] |
| Sand Flats, New York | 43.57°N | 75.24°W | 落叶林 | 5.0-6.0 | 1998.5-10 | 涡动相关法 | 0.82 | Finkelstein 等[116] |
| Harvard Forest, MA | 42.53°N | 72.18°W | 落叶林 | 3.4 | 1990-2000 | 涡动相关法 | 0.70 | Munger 等[117] |
| Kane, Pennsylvania | 41.60°N | 78.77°W | 落叶林 | 5.0-7.0 | 1997.4-10 | 涡动相关法 | 0.83 | Finkelstein 等[116] |
| Duke Forest, North Carolina | 35.97°N | 79.13°W | 落叶林 | - | 1996.4-5 | 涡动相关法 | 0.80 | Finkelstein 等[116] |
| Hyytiala, Finland | 61.85°N | 24.28°E | 松树林 | 6.0-8.0 | 2001-2010 | 涡动相关法 | 0.54 | Rannik 等[41] |
| Ulborg, Denmark | 56.28°N | 8.42°E | 针叶林 | 8 | 1996-2000 | 梯度法 | 0.92 | Mikkelsen 等[118] |
| Bily Kriz, Czech Republic | 49.55°N | 18.53°E | 针叶林 | 9-9.5 | 2008.7-8 | 涡动相关法 | 0.50 | Zapletal 等[19] |
| Alice Holt, UK | 51.17°N | 0.84°W | 落叶林 | | 2005.7.16-8.18 | 涡动相关法 | 0.85 | Fowler 等[38] |
| Les Landes Forest, France | 44.20°N | 0.70°W | 落叶林 | 2.1 | 1994.6 | 涡动相关法 | 0.62 | Lamaud 等[120] |
| Castelporziano, Italy | 41.70°N | 12.35°E | 落叶林 | 3.7 | 2012-2013 | 涡动相关法 | 0.29(旱季), 0.40(雨季) | Fares 等[121] |
| Marmirolo, Italy | 45.12°N | 10.44°E | 混交林 | 2.28 | 2012.6.12-2012.7.11 | 涡动相关法 | 1.92(冠层外), 0.42(冠层内) | Finco 等[122] |
| **农田** | | | | | | | | |
| Sacramento, CA | 36.51°N | 120.6°W | 葡萄园 | 3.4 | 1991.7.8-8.6 | 涡动相关法 | 0.50 | Padro[114] |
| Sacramento, CA | 36.48°N | 120.4°W | 棉花田 | 3 | 1991.7.8-8.6 | 涡动相关法 | 0.75 | Padro[114] |
| Gilchriston, Scotland | 55.90°N | 2.80°W | 土豆田 | | 2006.7-8 | 涡动相关法 | 0.76 | Coyle 等[8] |
| Comun Nuovo, Italy | 45.37°N | 9.30°E | 小麦田 | 2-5.5 | 2002.4-7 | 涡动相关法 | 0.56 | Gerosa 等[101] |
| Grignon, France | 48.85°N | 1.97°E | 玉米田 | 5.2 | 2008.6-8 | 涡动相关法 | 0.63 | Stella 等[123] |
| La CapeSud, France | 44.30°N | 0.63°W | 玉米田 | 5.1 | 2008.6-2008.8 | 涡动相关法 | 0.50 | Stella 等[123] |
| Lamasquere, France | 43.82°N | 1.38°E | 玉米田 | 3.2 | 2008.6-2008.8 | 涡动相关法 | 0.38 | Stella 等[123] |
| Bondville, Illinois | 40.05°N | 88.37°W | 大豆田 | 2.4-3.0 | 1994.8.18-1994.10.1 | 涡动相关法 | 0.60 | Meyers 等[124] |
| Nashville, Tennese | 36.65°N | 87.03°W | 玉米田 | 1.0-6.0 | 1995.6.22-1994.10.11 | 涡动相关法 | 0.70 | Meyers 等[124] |
| Zhengzhou, China | 34.44°N | 113.40°E | 玉米田 | 2.98-4.0 | 2018.7.16-2018.9.14 | 涡动相关法 | 0.54 | 魏莉[125] |
| Zhengzhou, China | 34.44°N | 113.40°E | 水稻田 | 2.95-8.5 | 2018.3.1-2018.5.31 | 涡动相关法 | 0.46 | 魏莉[125] |
| Nanjing, China | 32.19°N | 118.71°E | 冬小麦 | 1.1-3.5 | 2016.3.16-5.30 | 涡动相关法 | 0.45 | 徐静馨 等[126] |
| Nanjing, China | 32.21°N | 118.69°E | 水稻田 | 6 | 2016.8.17-10.10 | 涡动相关法 | 0.48 | 刘俊 等[127] |
| Yucheng, China | 36.50°N | 116.34°E | 玉米田 | 3.62 | 2011.8.9-9.28 | 涡动相关法 | 0.29 | 朱治林 等[104] |
| Nanjing, China | 32.14°N | 118.42°E | 冬小麦 | 1.09-1.55 | 2013.4.1-5.30 | 涡动相关法 | 0.71 | 李颀 等[128] |
| **草地** | | | | | | | | |
| Beijing, China | 40.05°N | 116.43°E | 低矮植被 | - | 2000 夏末 | 梯度法 | 0.26 | Sorimachi 等[129] |
| Beijing, China | 40.05°N | 116.43°E | 低矮植被 | - | 2000 初冬 | 梯度法 | 0.46 | Sorimachi 等[129] |
| Beijing, China | 40.55°N | 116.16°E | 草地 | 0.9 | 2007.9.23-10.12 | 梯度法 | 0.65 | 潘小乐 等[130] |
| Sacramento, CA. | 37.02°N | 119.48°W | 草地 | | 1991.7.8-8.6 | 涡动相关法 | 0.15 | Padro[114] |
| **其他** | | | | | | | | |
| Nanjing, China | 32.14°N | 118.40°E | 裸土 | - | 2015.12.2-12.22 | 涡动相关法 | 0.21 | 袁月[82] |
| Nanjing, China | 32.14°N | 118.40°E | 裸土 | - | 2016.6.6-6.18 | 涡动相关法 | 0.25 | 袁月[82] |
| La Crau plain, France | 43.34°N | 4.49°E | 裸土 | | 2001.4.20-5.31 | 涡动相关法 | 0.26 | Stella 等[131] |
| Lamasquère,France | 43.49°N | 1.23°E | 裸土 | | 2008.4.24-5.26 | 涡动相关法 | 0.35 | Stella 等[131] |
| La Cape Sud, France | 44.24°N | 0.38°W | 裸土 | | 2007.10.19-2008.3.4 | 涡动相关法 | 0.31 | Stella 等[131] |
| Lusignan, France | 46.24°N | 0.07°E | 裸土 | | 2011.3.17-5.5 | 涡动相关法 | 0.51 | Stella 等[131] |
| Turro, Italy | 44.59°N | 9.42°E | 裸土 | | 2014.3.24-4.14 | 涡动相关法 | 0.69 | Stella 等[131] |
| Auchencorth Moss, Scotland | 55.78°N | 3.23°W | 荒原 | 2 | 1995-1998 | 梯度法 | 0.65 | Fowler 等[36] |
| Nanjing, China | 32.12°N | 118.96°E | 城郊结合 | | 2016.9.1-2017.9.1 | 梯度法 | 0.55 | 赵雄飞 等[132] |

虑植被冠层结构的垂直变化，采用类比于欧姆定律的阻抗–速率 (Resistance-Velocity) 的方法来计算地表和大气之间各物种的干沉降速率[133]。该方法简单且适用于不同尺度的模型，并可以表示每个过程。

**1.2.3.1 大叶模型 (Big-leaf Model)**

大叶模型分为单面模型和双面模型。臭氧干沉降速率 ($V_d(O_3)$) 通常用垂直速度表示，由参考高度 (h) 的 $O_3$ 垂直通量 ($F_{O_3}$) 和 $O_3$ 浓度 ($C_{O_3}$) 之比得到，负号表示方向向下：

$$V_d(O_3) = -\frac{F_{O_3}^h}{C_{O_3}^h} \tag{1.10}$$

单层大叶模型 (图 1.3(a)) 是最简单的阻抗模型，它假设所有气体交换均发生在一个单一的表面，植物气孔和叶片位于冠层顶部。它由空气动力学阻抗 $R_a$ (Aerodynamic Resistance)，粘性副层阻抗 $R_b$ (Quasi-laminar layer Resistance) 和冠层阻抗 $R_c$ (Canopy Resistance) 三部分组成。

$$V_d = (R_a + R_b + R_c)^{-1} \tag{1.11}$$

其中，由于在模拟 $O_3$ 对植物的影响时需要将气孔沉降和非气孔沉降分离开来，单面大叶模型中的 $R_c$ 通常采用公式 (1.12) 方法进行计算，假设气孔表面和大部分非气孔表面在冠层中处于相同的高度：

$$R_c = (\frac{1}{R_{stom} + R_{meso}} + \frac{1}{R_{ns}})^{-1} \tag{1.12}$$

其中，$R_{stom}$ (Stomata Resistance) 是对 $O_3$ 通过扩散进入气孔吸收的阻抗；$R_{meso}$ (Leaf Mesophyll Resistance) 是叶片内部对 $O_3$ 反应的阻抗；$R_{ns}$ (Nonstomata Resistance) 是所有非气孔沉降途径的阻抗。通常 $R_{ns}$ 是使用 $O_3$ 通量和互补的微气象测量和单面的大叶模型推断的。

双面模型考虑了干沉降的两个表面 (图1.3(b))，即叶面和土壤，所有的叶片被认为是在一个高度。

$$V_d = (R_a + (\frac{1}{R_{b,leaf} + R_{stom} + R_{meso}} + \frac{1}{R_{b,leaf} + R_{cut}} + \frac{1}{R_{ac} + R_{b,soil} + R_{soil}})^{-1})^{-1} \tag{1.13}$$

其中，$R_{b,leaf}$ 为叶片准层状边界层阻抗；$R_{ac}$ (Aerodynamic Resistance from Canopy Top to Ground) 为冠层内空气动力阻抗；$R_{cut}$ 为表面阻抗；$R_{b,soil}$ 为土壤准层状边界层阻抗；$R_{soil}$ 为土壤阻抗；$R_{stom}$ 和 $R_{meso}$ 和单面大叶模型中定义的相同。

不同的干沉降方案，大叶模型的结构也不同。例如，Wesely 等[44]认为土壤和叶片是同一粘性副层阻抗，并将其与 $R_a$ 串联。而 Massman[37]则认为土壤和叶片有不同的粘性副层阻抗。

**1.2.3.2 多层冠层模型 (Multi-layer Canopy Model)**

大叶模型虽然应用广泛，但该模型没有考虑到叶片特型和功能的垂直变化 (例如响应太阳辐射的冠层衰减)。多层冠层模型 (图1.3(c)) 将植被冠层细分为多层，计算在冠层高度 (hc) 以下的每一层冠层 (z) 的表面阻抗 ($R_{surf}$)。它将冠层辐射、植物生理生化理论与湍流传输结合在一起，且可以预测和独立验证冠层内 $CO_2$、$H_2O$ 和 $O_3$ 等标量气体在多层叶片和大气之间进行交换的气孔通量[134]。

$$R_{surf}(z) = \left(\frac{1}{R_{b,leaf}(z) + R_{stom}(z) + R_{meso}(z)} + \frac{1}{R_{b,leaf}(z) + R_{cut}(z)}\right)^{-1} \quad if z \leqslant h_c \quad (1.14)$$

$$R_{surf}(Z) = \left(\frac{1}{R_{b,soil} + R_{soil}}\right)^{-1} \quad if z = 0 \quad (1.15)$$

用多层冠层模型计算 $V_d$，需要将上述 $R_{surf}$ 参数化嵌入到一个考虑冠层之间 $O_3$ 湍流传输的模型中。

(a) 单层模型  (b) 双层模型  (c) 多层模型

图 1.3: 不同模型臭氧干沉降阻抗结构图[34]

**1.2.3.3 不同臭氧干沉降模型的适用性**

基于以上介绍的大叶和多层的臭氧干沉降模型，目前一些区域和全球化学、空气质量模式也将臭氧干沉降模型作为模块和机制耦合嵌入自身模式之中，为区域和全球提供 $O_3$ 浓度和沉降的数据，以便评估空气质量对生态系统的影响。

在区域和全球模式的臭氧干沉降模块中大多使用大叶模型计算的。例如 EMEP，GEOS-CHEM 和 WRF-Chem 等模式。该模型一般忽略对陆地生态系统地面和下层植被 $O_3$ 通量以及冠层内平均 $O_3$ 浓度的垂直变化的考虑[135]，同时也忽略了 $O_3$ 和生物挥发性有机物 (BVOCs) 以及 NO 等气体的气相化学反应。多层冠层模型目前多集中应用于森林陆地生态系统[124,135-141]，但大多在单点模式中应用，极少用于区域或全球的 $O_3$ 干沉降模拟中。多层冠层模型能够克服大叶模型中的一些弊端，可以在每个高度计算 $O_3$，因此可以明确冠层上的气相化学反应[138,142]或湍流[143-144]的影响。但由于模型本身的复杂性，仍有很多假设和提议存在不确定性，例如这些模型的阻抗参数并不相同，尤其是非气孔沉降阻抗参数的使用并不一致[107,126,145-147]，其中土壤和表面沉降过程可能存在相互补偿的作用且它们各自在非气孔途径的分配比并不明确，亟待以后的研究和验证。

**1.2.3.4 影响臭氧干沉降的不同途径**

由于 $O_3$ 的高化学反应性，它会迅速沉降在干燥的表面，包括植被、土壤和建筑物等，而理论上 $O_3$ 较难溶于水，在潮湿的表面上沉降速度会较慢[3]。如图1.4所示，$O_3$ 的干沉降发生在不同的途径上，如通过植物叶片的气孔，或通过非气孔途径[38]。典型的非气孔沉降途径包括：$O_3$ 与植物外表面或叶片角质层以及植被冠层下的土壤发生反应[38]，以及与 BVOCs 在冠层表面的化学反应[39,112]。

[Figure]

图 1.4: 臭氧吸收和控制大气-生物圈交换的过程[38]

**1.2.3.5 气孔沉降途径**

气孔是植物叶片上调节大气与植被间气体交换的门户。准确估算气孔臭氧通量 ($F_{stom}$) 是解释臭氧通量观测和评估臭氧植物损害的关键。$F_{stom}$ 的计算公式如下：

$$F_{stom} = \frac{-O_{3leaf}}{R_{b,leaf} + R_{stom} + R_{meso}} \tag{1.16}$$

其中，$R_{b,leaf}$ 为叶片与外界空气之间的粘性副层阻抗；$R_{stom}$ 为气孔阻抗；$R_{meso}$ 为叶肉阻抗。$R_{stom}$ 的倒数为气孔导度 ($g_s$)。

$O_3$ 的 $R_{stom}$ 的观测方法和预测模型通常利用水汽在冠层的气孔导度进行推导。这种方法是假设 $O_3$ 通过气孔向内扩散和水汽向外扩散之间成比例关系。空气中水汽的扩散率与 $O_3$ 在空气中的扩散率之比约为 0.61[148]。这一假设有一个局限性，$O_3$ 和水汽分子之间的碰撞可能会导致估计的 $F_{stom}$ 误差在 4~10%[149]。

气孔 $O_3$ 吸收通过短期和长期反应改变气孔导度。在短期内，气孔 $O_3$ 吸收通过改变保卫细胞和信号转导途径降低 $g_s$[150-152]。从长期来看，在植物生理学研究中气孔对 $O_3$ 吸收的平均 $g_s$ 响应呈下降趋势[153]。然而，在研究中 $g_s$ 有可能增加或减少。例如，气孔对 $O_3$ 的吸收会导致光合效率降低，从而导致内部二氧化碳增加和气孔关闭[151,154-156]。另一方面，气孔对 $O_3$ 的吸收通过降低对脱落酸的敏感性导致 $g_s$ 的长期增加[157]，从而改变气孔细胞的离子交换[151,158]以及保卫细胞周围的表皮细胞的脱落[150]，导致气孔对外界刺激反应迟缓[151,156,159-160]，从而使 $g_s$ 的长期增加。气孔对 $O_3$ 的吸收也可能导致生长发育迟缓、衰老过程加速、叶面积减小[15]，从而使 $g_s$ 降低[7,38,161]。

目前有两种类型的参数化方案用于模拟量化气孔 $O_3$ 吸收对植被的影响。第一种是 $O_3$ 对单个植被生理过程的影响[21,31]。例如，气孔 $O_3$ 吸收对植物生物量或作物产量的影响，可以在模型中等同于 $O_3$ 对光合速率的影响并进行相应地参数化[31,162]。第二种模型考虑 $O_3$ 对同一植被生理过程的影响[23,153,163]。例如，Lombardozzi 等[153]通过 meta 分析，研究了累积气孔 $O_3$ 吸收对 $g_s$ 和光合作用的影响。此外，模型间气孔 $O_3$ 吸收的参数化方案的差异也与方案中气孔 $O_3$ 吸收是瞬时还是累积 $O_3$ 造成的有关。通常认为，植物的损伤与累积气孔吸收的关系更密切[164]。

**1.2.3.6 非气孔沉降途径**

通常认为，在植物生长过程中，$O_3$ 干沉降主要是气孔沉降。然而，许多研究表明，非气孔沉降也是总沉降的重要组成部分。Ganzeveld 等[165]强调，$O_3$ 非气孔沉降与气孔沉降量相当。大多数对植被冠层 $O_3$ 干沉降过程的观测都是短期[166]，可能受到周围环境化学的影响。

除了估计可用于 $O_3$ 干沉降的数量和反应性的挑战之外，这些因素还阻碍了对个别沉降途径的相对重要性以及相对重要性在时间和空间上变化的深入理解。此外 Ganzeveld 等[165]还指出，量化 $O_3$ 干沉降总通量对于研究碳固存和 $O_3$ 对植物功能的影响至关重要，因为它们取决于气孔和非气孔 $O_3$ 沉降的准确划分。Fowler 等[38]指出，约 40~60% 的总沉降量由气孔吸收构成，而非气孔沉降量是可变的。此外，Fowler 等[38]还指出，温度、太阳辐射、水分和风速是影响非气孔沉降的关键因素，并确定了导致 $O_3$ 沉降增加的一些关键机制。

1. 臭氧与水溶液在叶片表面的相互作用

非均相化学是控制 $O_3$ 沉降到叶片角质层的主要机制。叶片外部表面覆盖着表皮蜡和其他相关化合物，一些化合物如盐类离子 (如 Na、K、$NH_4^+$、$NO_3^-$ 和 $SO_4^{2-}$ 等) 、无机气体和 BVOCs 等可以通过沉积到叶片上，另一些化合物如也抗坏血酸盐等可以由植物内部分泌[35]。Altimir 等[167]在气室内测量的 $O_3$ 和 $CO_2$ 的叶片吸收，证明角质层 $O_3$ 吸收依赖于相对湿度。

根据 Altimir 等[167]的研究，叶片表面的水溶液反应，增加了 $O_3$ 的溶解度，并且较高的 pH 值有利于 $O_3$ 的溶解。实验证据表明，叶面上的化学溶液的成分可能会随着时间的变化而变化[80,167]。气溶胶在叶片表面的沉降会带来盐类和有机酸，可能是潜在溶液成分变化的原因。Coyle 等[168]提出，苏格兰草原上空 $SO_2$ 和 $NH_3$ 协同效应可以解释观测到的高水平的 $O_3$ 非气孔沉降，这是由于 $NH_3$ 增加了溶液的 pH 值，形成了大量 $O_3$ 的汇。Potier 等[139]通过实验室实验和机理模型表明，潮湿叶片气孔中抗坏血酸的流失可能与 $O_3$ 在某些植被物种上的化学反应和植被的物候状态有关。

由于缺乏可用的实验室和外场观测数据，$O_3$ 角质层沉降的模型主要是经验性的。许多模型只包括叶面积指数 (LAI) 和一个调节因子[37]。一些模型区分了湿角质层和干角质层的沉降，但不同模型在模拟响应方向上存在差异。例如，Wesely 等[44]认为当叶片是湿的时候叶片角质层沉降较低，而 Zhang 等[145]则认为当叶片是湿的时候叶片角质层沉降较高。

2. 冠层内的臭氧化学反应

植物冠层外表面和内表面周围的空气成分由植被排放的 BVOCs、粘性边界层和细胞间空间决定，并受到小气候条件的影响[167]。根据冠层尺度的观测，植物排放的 BVOCs，如一些倍半萜烯和单萜烯，与 $O_3$ 在冠层内发生气相反应[169]，可导致植被冠层内形成大量的 (非气孔) $O_3$ 的汇[39,170]。这些 BVOCs 与 $O_3$ 的反应可能占到 $O_3$ 总通量的一个不可忽略的部分[100,138,170-173]。然而，由于缺乏全面的可分辨 BVOCs 的异构体的测量技术以及与 $O_3$ 反应速率的不确定性，量化化学反应的相对贡献受到了限制[169,173-174]。

在美国内达华山脉的 Blodgett 森林观测到，臭氧通量对空气温度的依赖与植被单萜烯

的排放一致[112,170]，可导致树冠内显著的 $O_3$ 损失；在森林疏伐后，臭氧通量和单萜烯类化合物的浓度也有类似的提高[173]。Kurpius 等[170]对内华达山脉松树的总臭氧沉降通量进行了分析，研究表明树冠内的 $O_3$ 化学汇是白天 $O_3$ 损失的主要过程 (45~55%)，而夏季气孔吸收和非气孔沉降分别占 25~35% 和 20%。

在亚马逊森林中观测到大量土壤排放的高活性倍半萜烯[175]。虽然需要在其他地点进行测量以了解这种现象是否发生在其他地方，但在解释臭氧通量时忽视土壤排放的高活性 BVOCs 时可能会导致过度强调其他过程的作用。

在冠层中，环境化学不仅影响 $O_3$，还影响其他气体。$O_3$ 和高活性 BVOCs 之间的反应可能是冠层氧化的有机物排放的原因[176-177]，以及随后的二次有机气溶胶形成[178-180]，羟基自由基的生产[170]和活性氮氧化物排放[181]。氧化的有机和无机化合物的通量可以为化学影响的臭氧通量的机制提供观测约束。

3. 土壤沉降

$O_3$ 沉降到土壤的主要途径是与土壤有机质中不饱和碳键的反应[182]，撒哈拉沙漠短期野外观测到日间的平均 $V_d$ 大约为 0.1 cm·s$^{-1}$，这表明 $O_3$ 与土壤有机物质的反应并不是土壤沉降的唯一途径[183]。通过与 NO 或 BVOCs 的反应，$O_3$ 有可能在土壤表面发生热分解或在土壤孔隙中发生气相损失。

一些外场试验[40,121,184-185]和实验室实验[93,182]结果表明，土壤水分通过限制土壤孔隙大小抑制 $O_3$ 吸收。随着土壤水分的增加，土壤 $O_3$ 吸收的减少表明水分减少了可与 $O_3$ 反应的表面积，超过了水分促进异质化学的任何影响。事实上，使用同位素方法来限制土壤对土壤孔隙水的 $O_3$ 吸收，Toet 等[93]研究结果表明，在土壤水分为 60% 时，$O_3$ 沉降到土壤孔隙水中占土壤 $O_3$ 吸收的很大一部分，但土壤水分为 30% 的情况下，$O_3$ 吸收率要低得多 (<10%)，因为土壤水分只占总吸收率的一小部分。

用 EC 法对欧洲裸露的农业土壤和半干旱平原上进行短期的 $O_3$ 干沉降观测显示，至少在其中一个地点，土壤 $O_3$ 吸收与近地表相对湿度的关系大于与土壤含水量的关系[186]，并且呈指数级下降[131]。Stella 等[186]假设认为相较于较浅深度的土壤含水量，是地面上的水分子阻止了 $O_3$ 进入土壤。Stella 等[131]在 6 个地点进行实验认为，土壤 $O_3$ 吸收和近地表相对湿度之间的关系变化是由土壤粘粒含量引起的。进一步回归分析表明，土壤吸收量随粘粒含量的增加而增加，但粘粒含量越高的土壤，土壤吸收量随表面相对湿度的增加而减少得越快[131]。

在区域或全球模式中，$O_3$ 对土壤的阻抗值通常是恒定的，随下垫面类型和季节的变化而变化[44]。大多数 $O_3$ 土壤干沉降的建模研究使用的都是短期数据。Massman[37]首次提

出了 $O_3$ 干沉降到土壤中的建模方法，指出土壤干/湿程度会导致土壤阻抗变小/变大，从而对沉降产生影响。早期的研究结果[37,183]表明，土壤有机含量、土壤孔隙度、土壤湿度和土壤阻抗之间可能存在一定的关系。由于对上述土壤阻抗与土壤性质之间的关系缺乏了解，Massman[37]从观测中推断出的土壤 ($R_{soil}$) 对 $O_3$ 干沉降的两个恒定阻抗值，建议干土为 100 s·m$^{-1}$，湿土为 500 s·m$^{-1}$。相比之下，Ganzeveld 等[165]实验测量结果表明，土壤层面相对湿度的增加是控制 $O_3$ 向裸土沉降的一个因素[131,165,186]。

4. 冠层内湍流的输送和扩散

大气湍流是由切变力或浮力产生的，在复杂下垫面下，气团和下垫面接触，湍流会更强烈的。湍流使空气移动，并将富含 $O_3$ 的空气包裹带向地表，是 $O_3$ 沉降的基础。摩擦速度与 $V_d(O_3)$ 或 $F_{O_3}$ 之间的相关性[120-121,187-188]表明，湍流输送是 $O_3$ 干沉降一个重要的驱动因素。

痕量气体的输送通常是用 Monin-Obukhov 相似理论 (MOST，Monin-Obukhov similarity theory) 计算，一种基于量纲分析的经验形式，它考虑到了大气稳定性对近地表湍流的影响，并保持在惯性亚层中。不同的 MOST 经验公式可能导致在稳定条件下空气质量模式中模拟的 $V_d$ 的存在差异[96,189]。利用 MOST 模拟冠层大气交换的方法存在一定的局限性。亚马逊高塔天文台的观测表明，森林上方的粗糙亚层直接融合到混合层中，甚至没有形成惯性亚层，而 MOST 在惯性亚层中是有效的[190]。另一个问题是许多森林冠层位于丘陵或山区地形，但是 MOST 适用于下垫面水平均匀的条件。

多层冠层模型通常模拟冠层之间的垂直交换，但模型中模拟的湍流会影响 $O_3$ 的冠层分布，从而影响干沉降和环境化学。大多数多层冠层模型采用 K 理论[136,138,140,142,179,191]，不能真实模拟复杂冠层的湍流。

在大叶阻抗干沉降模型中,冠层湍流的表示也受到限制。例如,在 Wesely 方案[44]中,有两种对冠层湍流的阻抗，一种是对较低冠层沉降的阻抗，另一种是对土壤吸收的阻抗。Wesely 等[44]为前者定义了一个基于太阳辐射和地形坡度的简单模型，为后者规定了随不同下垫面类型和季节而变化的常数。后来的大叶模型不包括向较低冠层的沉降[48,50,90,145,192-193]，而是采用一个基于 6 天玉米田下午观测的 $F_{O_3}$，LAI，摩擦风速和冠层高度对湍流传输到地面的模型进行修改[185]。一些研究确实证明摩擦速度是 $O_3$ 干沉降变化的驱动因素[120-121,187]。然而，这样的特定地点的模型[185]可能无法捕捉到跨越不同下垫面和边界层条件的冠层湍流阻抗的变异性。

在目前的建模方法中，$O_3$ 在冠层内的湍流传输被部分捕获，通过准层流表面的传输是通过分子扩散的[165]。Ganzeveld 等[165]也指出，与冠层湍流机制相关的两种潜在机制尚未被

模拟：冠层下部部分或完全脱钩，这取决于风速、植被冠层密度和热分层；在冠层上方形成一致的湍流结构，可导致冠层空气的恢复，并随之产生热量和质量的输送。此外，几乎所有的参数都是稳态模型，其中 $O_3$ 在冠层内的储存及其与 NO 和 VOCs 等其他化合物的化学相互作用无法及时跟踪，导致额外的不确定性。Ganzeveld 等[136]研究表明，冠层植被的两层 (冠层和冠层-土壤层) 表征能够更好地模拟大气-生物圈 $O_3$ 和氮氧化物交换的主要特征，指出该方法较好地解决了干沉降和冠层内 $O_3$ 湍流交换的问题。

**1.2.4 臭氧干沉降模拟的不确定性来源**

在区域和全球模式中，$O_3$ 干沉降方案大多采用大叶模型[44]。该方案通过查表的方式确定不同阻抗值，以及其随季节和下垫面类型的变化[44,145,194]。虽然环境因素 (例如温度或表面湿度) 有时会改变查找值，但在干沉降方案中，很少有机制表示这一过程。干沉降的不确定性部分来自于亚网格级别的一些过程没有被很好地捕获，并且该过程在模式中被严重参数化[38,47]，以及缺乏长期测量，土地利用和植被特征的准确性不足等方面[34]。这种不确定性会影响模式在模拟 $O_3$ 干沉降对空气污染、生态系统和气候影响时的准确性。

Wesely 等[44] (简称 W89) 干沉降方案是目前许多全球或区域尺度模式中使用的干沉降方案。W89 方案通过初始查找表值和物理依赖关系来计算每个陆地类型和季节的各沉降阻抗。在 CTMs 中，传统的 W89 方案中的植被生理活动是通过气象参数 (如太阳辐射、气温、季节和下垫面类型) 计算确定的[195]，而不是直接模拟植被的生理活动。气象方法大幅度减少了计算时间，但是相对于直接模拟植被的生理活动，在光合作用低的时期，这种方法无法很好的捕捉到干沉降的大小，并且此方案也没有考虑到 $O_3$ 对植被的损伤[195]。W89 方案最初用于计算中北纬度地区的土地类型的干沉降，后来被进行了一系列的修改[46,48,196-197]。针对 W89 方案的许多修改大多来自于某单一研究地区的观测数据[198]，虽然类似于 W89 的参数化已经在一定程度上根据不同地区的观测结果进行了评估[48]，但很少有区域性的评估。Hardacre 等[72]第一个对全球尺度的 $O_3$ 干沉降进行了评估，将 15 个大气污染半球传输组 (HTAP, Task Force on Hemispheric Transport of Air Pollution) 中不同模式模拟月均 $V_d(O_3)$ 与各种观测结果进行了比较。他们的研究发现不同模式之间的模拟偏差能够高达 2 倍。不同的干沉降方案之间模拟的 $V_d(O_3)$ 也存在很大的差异[63,96,199]。例如，在加拿大 Borden 森林使用了相同的强迫数据驱动了 5 个不同的干沉降方案，$V_d(O_3)$ 差异能够到达 2~3 倍[96]。在 GEOS-Chem 中实施了四个干沉降方案，$V_d(O_3)$ 在夏季的年际变化率和 30 年趋势上也存在很大差异[63]。深入比较各方案后[63,199]，发现各种物理过程和参数导致了方案间的差异。Schwede 等[199]利用几种干沉降方案对统一地点的 $O_3$ 干沉降进行了模拟，结果发现对于 $V_d(O_3)$，$R_c$ 的模拟是不同模式之间差异最大的原因。

气孔阻抗是 $R_c$ 主要组成部分之一[47]。气孔导度 (气孔阻抗的倒数) 对区域模式中量化

地表与大气之间的能量、水汽和碳交换也至关重要[200-201]。在 CTMs 中，最常用于计算气孔导度的是基于 Jarvis 连乘型的经验模型[44,192]。在 Jarvis 模型中，是将最大的气孔导度参数值与环境因子相乘，每个因子都是特定环境条件的函数，可以是气象条件 (如光合有效辐射、温度和湿度等)，也可以是生物的物理条件 (如土壤湿度和叶龄)。Jarvis 模型是第一个相对完整的气孔导度模型，为系统地研究气孔随外界环境变化的机理创造了条件[202]。但是此模型也具有一定的局限性，模型中的参数大多没有生物学上的意义[203]，这给机理性的研究增加了困难。同时模型只针对于各环境因子对气孔导度的单独作用，因此会忽略各环境因子对气孔的协同作用[204]。Jarvis 模型还需要根据所代表的生态系统和环境条件对参数进行调整，受到许多生态系统 (例如热带森林) 数据的缺乏的限制。DO₃SE (The Deposition of O₃ for Stomatal Exchange) 模型使用物种特定参数的 Jarvis 模型计算 O₃ 气孔沉降，并预测 O₃ 污染造成的欧洲相关树木和作物的损伤[192,205-207]。然而，DO3SE 模型是针对欧洲北部和温带地区的物种开发的，普适性不足[205]，无法对其他地区 $V_d(O_3)$ 进行准确模拟[208]。随后发展了一种基于光合作用的 Ball-Berry 气孔导度的模型。该模型是一种半经验模型，气孔导度表示为光合速率的函数关系[209]，与相对湿度正相关，与 $CO_2$ 浓度负相关，并且受到叶片水平数据的限制[210-211]。相较于 Jarvis 模型，Ball-Berry 模型具有一定的优势，它把复杂的环境因子影响综合到湿度、$CO_2$ 和光合作用 3 个要素中[212]，用较少的经验参数表示植物气孔对环境变化的响应以及植被的生理活动[213-214]。基于 Ball-Berry 模型计算的气孔导度方案已独立或耦合进地球系统模型或者陆面模型中，但很少应用在 CTMs 中计算干沉降速率，只有少数在模拟气候-化学相互作用的耦合气候化学模型中试图将化学模块中的干沉降与其耦合起来[46,215]。

一些研究评估比较了 Jarvis 模型和 Ball-Berry 模型计算的气孔导度，Misson 等[216]和 Niyogi 等[217]结果表明，基于 Ball-Berry 模型计算的气孔导度要优于 Jarvis 模型，然而在 Büker 等[218]和 Uddling 等[219]研究中却没有。很少有研究比较和评估在完全一致的方法框架和一致的模型输入参数下不同的气孔导度模型的模拟效果。深入研究不同类型的气孔导度模型，不仅有助于统一 CTMs 内部气孔行为的表征，而且有助于更好地理解与大气化学相关的植物生理过程。

由于技术和方法上的困难，气孔沉降难以连续观测，非气孔沉降目前还无法直接观测[20]。Flechard 等[69]强调，O₃ 的非气孔沉降的参数化在不同的 CTMs 中存在差异。EMEP 中非气孔沉降考虑了增加表面湿度对 O₃ 沉降的影响[38]。在 Surfatm-O₃ 模式中，实施了更为完善的非气孔沉降方案[186]，其考虑到了生长期和衰老期间相对湿度 (RH) 对叶片表皮沉降的影响，以及土壤表面 RH 会土壤沉降的影响。在 MuSICA 模式中，O₃ 非气孔沉积采用了一种更机械的方法，模拟 O₃ 沉降在干燥和湿润叶片生长和衰老期间的不同[139]。Zhang

等[48]开发了更为动态的 $O_3$ 非气孔方案,考虑了关键的环境因素,如 RH、LAI 和摩擦风速。根据 Zhang 的方案[48],一些模式增加了气象和地面湿度对 $V_d(O_3)$ 的观测效应[220],以及气孔导度过程的表征[46,49,51-52,221]。空气质量模式 CAMx 目前实现应用了 Zhang 的方案[48]在欧洲进行 $O_3$ 模拟的研究方法[222]。

由于 $O_3$ 干沉降不仅与环境因子、植物生理状况密切相关,而且还与植物和土壤排放的气体相关,且不同的环境因子之间也存在交互影响的关系,模拟起来较为复杂,因此很多参数和物理方案仍存在很多的不确定性,亟待进一步的研究。此外,Hogrefe 等[65]研究结果表明地表 $O_3$ 浓度对 $V_d(O_3)$ 敏感,这意味着如果在大气化学传输模式中不能准确模拟 $V_d(O_3)$ 的变化[8,38,73-74,107,223],可能会影响对 $O_3$ 浓度的模拟,并误报其他过程对 $O_3$ 浓度的影响。

**1.2.5 臭氧的生态环境效应**

地表 $O_3$ 具有强氧化性,是地表大气光化学烟雾以及温室气体的主要成分,会严重危害植物和农作物的生长[224]。地表 $O_3$ 除了少量来自平流层的输送外,绝大部分是由氮氧化物 (NOx)、挥发性有机物 (VOCs)、甲烷 ($CH_4$)、一氧化碳 (CO) 和许多其他污染物之间在太阳光照射下发生光化学反应的产物[225-226]。在 NOx 和 VOCs 都存在的情况下,$O_3$ 形成的关键化学过程如公式1.17-1.22所示[7,227-228]:

$$VOC + OH \xrightarrow{O_2} RO_2 + H_2O \tag{1.17}$$

$$CO + OH \xrightarrow{O_2} HO_2 + CO_2 \tag{1.18}$$

$$RO_2 + NO \xrightarrow{O_2} SVOC + HO_2 + NO_2 \tag{1.19}$$

$$HO_2 + NO \longrightarrow OH + NO_2 \tag{1.20}$$

$$NO_2 + hv \longrightarrow NO + O \tag{1.21}$$

$$O + O_2 + M \longrightarrow O_3 + M \tag{1.22}$$

VOCs 被 OH 自由基、$O_3$ 或硝酸盐自由基 ($NO_3$) 氧化产生过氧自由基 ($HO_2$),进而氧化成 NO 生成 $NO_2$,这些新生成的 $NO_2$ 再次参与到光化学循环中,导致 $O_3$ 在大气中积累[226,228]。

地表 $O_3$ 的生成受到排放源、气象条件 (如光照、温度、湿度及风速等) 和光化学反应等影响。其光化学形成、区域传输、还原氧化及沉降分解等过程共同决定着某一地区的地表 $O_3$ 浓度状况。根据 EKMA (Empirical kinetics modeling approach) 模型和观测数据的研究表明,$O_3$ 浓度与 NOx 和 VOCs 呈显著的非线性响应关系[229]。城市中心的 $O_3$ 产生是 VOCs 敏感区,而农村地区的 $O_3$ 产生是 NOx 敏感区,超大城市集群区域处于过渡状态[230]。地表

$O_3$ 形成后在大气中的寿命大约为 22 (±2) 天[13]，一方面它会随大气环流进行跨区域、长距离的传输，另一方面与大气污染物的氧化产物 OH 自由基及 $HO_2$ 自由基等发生反应还原为 $O_2$，或直接向地表沉降并最终分解[229]。

随着我国城镇化和工业化进程快速发展，$O_3$ 前体物 (NOx 和 VOCs) 排放量的增加，导致我国尤其是人口密集经济发达的京津冀、长三角和珠三角地区 $O_3$ 浓度显著升高，$O_3$ 污染日益严重。目前，我国大部分地区的地表 $O_3$ 浓度水平与美国 20 世纪 90 年代的水平相当，甚至更高[231]。根据环境空气质量站点的的 $O_3$ 监测数据显示，2015 年到 2019 年我国 337 个城市臭氧浓度年评价值从 134 ug·m$^{-3}$ 增加到 161 ug·m$^{-3}$，平均上升了 20.1% (图1.6)。2015 年，337 个城市中臭氧年评价浓度超过国家二级标准限值 (GB 3905-2012) 的城市数为 55 个 (占比 16.3%)，到了 2019 年增加到 161 个 (占比 47.8%) (图1.5)。高浓度的地表 $O_3$ 已经成为中国大多数城市夏季最主要的空气污染物。另外，我国 $O_3$ 污染还呈现明显的季节和地域特征[5]：京津冀和汾渭平原 $O_3$ 浓度高值主要出现在 7 月份；长三角 $O_3$ 浓度高值出现在 5 月份；珠三角 $O_3$ 浓度高值主要一般出现在 10 月份；川渝地区 $O_3$ 浓度高值出现在 8 月份。如此高浓度的 $O_3$ 不可避免地会对我国作物的生长和产量造成严重的危害。

[Figure]

图 1.5: 2015-2019 年 337 个城市臭氧浓度年评价值空间分布

[Figure]

图 1.6: 2015-2019 年 337 个城市污染物浓度年际变化

**1.2.5.1 臭氧对生态系统的影响**

$O_3$ 是一种植物毒素，其对植物的破坏会影响生态系统。$O_3$ 主要通过气孔途径进入到植物内部，气孔对 $O_3$ 吸收的变化有短期和长期的响应。

大量实验结果表明：小麦，玉米、棉花、水稻、大豆、土豆等农作物[157,232-235]，萝卜、生菜、菠菜、洋葱、荠菜、番茄、西兰花等蔬菜[232-233]，还有水果中的柑橘、黑樱桃、葡萄，西瓜[157,236-237]，苜蓿、风信子、毛茛草等草类[157,233]，树木中的山毛榉、云杉、白桦树、圣栎树、火炬松等[238-240]对 $O_3$ 较为敏感，$O_3$ 浓度升高能够对这些植物产生较为显著的伤害，进而会威胁到植物所提供的生态系统服务，如：粮食安全、碳固定、木材生产、植被针对土壤侵蚀作用的保护作用、雪崩和洪水等[7]。

相比于国外，我国针对 $O_3$ 污染造成的农业风险研究开展较晚，但是发展迅速。基于 AOT40 指标，研究结果表明 $O_3$ 污染造成 2015 年我国水稻和小麦分别减产 8.0% 和 6.0%，导致的经济损失高达 1304 亿元[241]，Hu 等[242]以县级尺度，进一步针对 $O_3$ 高污染地区的华北平原 2014~2017 的小麦产量进行了评估，研究发现 $O_3$ 导致小麦分别减产 18.5%、22.7%、26.2% 和 30.8%，造成总经济损失约为 2599.2 亿元。姚芳芳 等[243]对长三角地区主要粮食作物小麦和水稻和油菜的产量损失进行了综合估算，指出 $O_3$ 污染造成的农作物分别减产了 17.1% 和 3.0%，总的经济损失约为 13.4 亿元；此外，$O_3$ 污染还造成该地区油菜减产 5.9%，产量损失 11 万吨，经济损失 2.6 亿元。基于通量指标，Feng 等[244]的研究表明 $O_3$ 造成我国 2015~2016 年的冬小麦平均损失了 10.4%。Zhang 等[245]的研究结果显示 $O_3$ 污染可能会对中国东北地区的大豆产量减少 23.4~30.2%。Avnery 等[246]利用大气化学输运模式 (MOZART-2) 对全球的主要粮食作物在未来 $O_3$ 浓度不断升高的情境下的产量损失进行了预测。结果

表明，到了 2030 年不断升高的 $O_3$ 浓度将导致全球小麦、大豆和玉米减产 4~17%、9.5~15% 和 2.5~6.0%，造成的经济损失约为 12~21 亿美元。Van Dingenen 等[247]利用大气化学输运模式 (TM5) 对全球的主要粮食作物在未来情景下的产量进行了预测。预计到 2030 年在相对乐观的 "当前立法" 情景下小麦、水稻、玉米和大豆将分别额外减产 2~6%、1~2% 和 2%。在区域范围内，最显著的额外损失主要发生在欧洲和中国，预估将导致 14~26 亿美元的经济损失。

**1.2.5.2 臭氧与气候变化的相互影响**

在对流层中，$O_3$、$CH_4$ 和二次气溶胶 (硫酸盐、硝酸盐和二次有机气溶) 这样的化合物，由于其辐射特性，可以对地球气候产生强烈的影响[248]。对流层 $O_3$ 被认为是导致气候变化的第三大温室气体，其产生的辐射强迫为 0.35 $W·m^{-2}$ (0.25~0.65 $W·m^{-2}$) ，高于 $CO_2$、$CH_4$ 和 $N_2O$ 等温室气体的辐射强迫作用，占全球净辐射强迫的 25%[2]。IPCC 第五次评估报告[249]指出 1750~2010 年对流层 $O_3$ 的变化引起的全球平均辐射强迫约为 +0.4 $W·m^{-2}$ (+0.2~0.6 $W·m^{-2}$) 。

其次，对流层 $O_3$ 变化除了产生辐射强迫影响气候变化之外，还能影响温度和降水的变化。Xie 等[250]模拟了 1850~2013 年全球对流层 $O_3$ 变化引起的气候变化，发现其变化产生的有效辐射强迫为 0.46 $W·m^{-2}$，并且导致了全球年平均地表温度增加 +0.43 ℃，导致降水增加 +0.02 $mm·d^{-1}$。其中，地表温度的增加在南北半球高纬度更加显著，在西伯利亚最大升温为 1.4 ℃；赤道地区的降水有相反的变化：夏威夷降水升高了 0.5 $mm·d^{-1}$，相反印度洋中部地区降水降低了 0.6 $mm·d^{-1}$。此外，Chang 等[251]模拟了 1951~2000 年中国东部地区的对流层 $O_3$ 变化产生的辐射强迫引起的气候变化，发现对流层 $O_3$ 变化导致了中国东部地区 1951~2000 地面空气温度增加了 0.43 ℃，同时还造成降水减少了 0.08 $mm·d^{-1}$ 的变化。与 Chang 等[251]相似，Hansen 等[252]研究结果显示中国东部地区 1900~2003 对流层 $O_3$ 的变化导致了温度的较大变化 (0.5 K) 。Chen 等[253]使用 IPCC 的 A2 排放场景模拟了 2000~2100 年全球对流层 $O_3$ 变化引起的气候变化，结果表明其变化将导致全球年平均地表温度 0.32 K 的升温，其中温度变化最大的区域为北半球高纬度地区。此外，周彦丽[254]研究结果分别揭示了对流层 $O_3$ 和黑炭气溶胶变化对北半球热带地区扩张的影响以及东亚的对流层 $O_3$ 变化对该区域夏季温度的影响。

最后，对流层 $O_3$ 也会对生态系统的总初级生产力 (GPP) 产生影响，降低生态系统的碳同化能力，从而间接影响气候变化。对流层高浓度的 $O_3$ 浓度会导致植被的生产能力有数十亿美元的损失。$O_3$ 通过进入植被的气孔进入植被叶片，产生活性氧和氧化压力，这反过来降低植物的光合作用，减缓植物的生长，危害植物生物量的累积。$O_3$ 通过这些复杂的反应过程影响生态系统的 GPP，威胁碳封存，进而对气候变化产生影响[15]。Fares 等[255]通过

对美国内华达山脉森林站点，意大利罗马附近海滨森林站点，以及美国加州中央山谷甜橙果园站点的长期野外观测实验，结果显示除意大利由于较低 $O_3$ 浓度导致该站点森林 GPP 没有降低以外，美国的森林和果园站点的黄松和甜橙树的碳同化能力降低了 12~19%。Ren 等[162]通过使用对流层 $O_3$ 历史数据以及动态陆地生态模式模拟 1961~2000 年中国陆地生态系统对 $O_3$ 浓度升高的响应，发现 $O_3$ 浓度升高导致陆地生态系统净初级生产力 (NPP) 平均降低了 4.5%，全国范围总碳储量降低了 0.9%。与 Ren 等[162]研究相似，Sitch 等[31]通过使用气候-碳循环耦合模式模拟计算了 1900~2100 年全球陆地生态系统在控制 $CO_2$ 排放量不变，对流层 $O_3$ 浓度升高情景下的响应，发现陆地生态系统 GPP 降低了 $16.1{\sim}26.4\,\mathrm{Pg{\cdot}Cyr^{-1}}$。

气候变化对 $O_3$ 的影响，一方面通过局地的气象场来影响 $O_3$ 浓度，另一方面，气候变化也可以通过影响 $O_3$ 形成的化学环境来影响地表 $O_3$ 的浓度[256]。Monks 等[7]和 Fiore 等[257]对关键的气候变化-空气质量途径进行了总结，并指出如陆地上空温度的升高、大气湿度的变化以及与反气旋条件变化相关的停滞事件发生率的增加是重点的途径。其中，对 $O_3$ 影响最大的是温度。气候变暖可以对 $O_3$ 前体物的排放造成很大的影响，同时地表 $O_3$ 的生成与 NOx 和 VOCs 呈显著的非线性关系[257]。Jacob 等[258]通过 GCM-CTM 研究发现，在未来几十年中气候变化会导致污染地区的夏季地表 $O_3$ 浓度增加。Xie 等[259]通过 WRF-CALGRID 模式模拟长三角地区的当前和未来的地表 $O_3$ 浓度，对比目前和 IPCC SRES A1B 情景，发现气候的变化可以显著改变长三角地区地表 $O_3$ 的空间分布，南部减少 5-15 ppbv，北部增加 5-15 ppbv。虽然大多数模式都预测气候变化会导致 $O_3$ 浓度的变化，但气象因素的变化如何导致 $O_3$ 干沉降的变化，仍存在很大的不确定性[7]。

**1.3  本文的主要研究内容**

综上所述，目前国内对臭氧干沉降的研究开展较晚，并且研究也相对较少，尤其对全国区域臭氧干沉降过程的模拟研究更为不足。目前对干沉降机制的评估多集中于单一干沉降机制计算的臭氧干沉降速率在单一下垫面上的表现；同时在区域模式上，对臭氧干沉降的模拟没有考虑与植被的反馈。这种对干沉降及其影响过程处理的不足，严重限制了我们对中国地区臭氧干沉降过程变化的理解和预测。

因此，为了深入探讨区域臭氧干沉降的时空分布特征，全面认识不同干沉降机制在典型下垫面对臭氧干沉降速率模拟的影响，以及臭氧与植被的相互反馈对区域臭氧干沉降过程的影响。本文以中国地区为研究区域，基于数值模拟的方法，结合典型亚热带森林和农田下垫面的观测资料，综合利用干沉降机理诊断模式 Noah-MP-WDDM 和区域空气质量模式 WRF-Chem，探讨了干沉降机制改进对区域植被生理过程，气象要素以及臭氧干沉降过程的影响，以提升区域尺度 $O_3$ 干沉降的模拟效果。

主要的研究内容包括：

1. 利用单点干沉降机理模式，识别影响臭氧干沉降速率的关键过程

基于典型亚热带森林和农田下垫面的观测数据，分析不同下垫面臭氧干沉降速率的日变化特征。利用陆面过程模式 Noah‑MP 与区域大气化学模式 WRF-Chem 的干沉降模块 (WDDM) 耦合形成的单点干沉降机制模式 Noah-MP-WDDM，评估不同气孔导度机制对臭氧干沉降速率的影响，以及影响典型下垫面臭氧干沉降速率的关键过程。

2. 改进区域大气化学模式的臭氧干沉降机制，提升对区域臭氧干沉降过程的模拟能力

基于 Noah-MP-WDDM 模式对典型下垫面上不同干沉降机制的评估，改进区域大气化学模式中干沉降机制的气孔导度方案，并将臭氧胁迫导致的植被破坏的半经验方案[23]耦合进模式中，进一步完善臭氧干沉降机制的改进。探究模式改进后，$O_3$ 与植被耦合对区域植被生理过程、气象以及对 $O_3$ 干沉降过程的影响，分析典型下垫面臭氧干沉降过程的差异，量化影响臭氧干沉降通量，臭氧浓度和臭氧干沉降速率的关键过程。

3. 评估臭氧干沉降过程对气候变化的响应，量化臭氧污染对作物产量及经济的损失

探讨在碳中和背景下，探究臭氧干沉降过程对气候变化的两个因子 (气象条件和人为排放源) 的响应，分析在气候变化情景下臭氧污染对未来作物产量和经济损失的影响。

根据上述研究内容，设计了本研究的技术路线，如图1.7所示。具体章节安排为：第二章介绍 WRF-Chem 和 Noah-MP 模式，以及模式中干沉降机制的计算过程；第三章分析了森林和农田下垫面臭氧干沉降速率的日变化趋势，利用单点干沉降机制模式 Noah-MP-WDDM 对森林和农田下垫面干沉降机制中的气孔导度方案进行评估，识别最优的气孔导度方案；第四章针对评估出的气孔导度方案，对区域模式 WRF-Chem 中的干沉降机制进行改进，并将臭氧胁迫导致的植被破坏的半经验方案耦合进干沉降机制中；第五章将改进后的 WRF-Chem 模式应用到中国地区臭氧干沉降过程的估算中，分析典型下垫面臭氧干沉降过程的差异，揭示影响臭氧干沉降过程的关键因素；第六章探讨气候变化情景下臭氧干沉降过程对气候变化的响应，以及当前和未来臭氧水平下导致的作物产量损失和经济损的影响；第七章对本文的主要工作进行总结，并讨论本文研究的不足之处和未来可以进一步研究的方向。

[Figure]

图 1.7: 技术路线

本研究的科学问题：如何改进干沉降机制以提升对臭氧生态风险的评估能力？

**第二章 数值模式及评估方法介绍**

本研究采用新一代大气化学传输模式 WRF‑Chem 和陆面过程模式 Noah‑MP，研究干沉降机制对我国典型下垫面臭氧干沉降过程的影响。其中第一部分详细介绍了 WRF-Chem 模式中以及与本研究相关的主要物理和化学方案，第二部分介绍了 Noah-MP 模式的主要结构和参数化方案。

**2.1 WRF-Chem 模式**

中尺度气象数值模拟与预报模式 Weather Research Forecast (WRF) 是在美国国家大气研究中心 (NCAR，National Center of Atmospheric Research) 、美国国家海洋和大气管理局 (NOAA，National Oceanic and Atmospheric Administration) 中的环境预报中心 (NCEP，National Centers for Environmental Prediction) 联合俄克拉荷马大学 (Oklahoma University) 、美国空军气象局 (AWFA，Air Force Weather Agency) 、美国海军研究实验室 (NRL，United States Naval Research Laboratory) 和美国联邦航空管理局 (FAA，Federal Aviation Administration) 等机构在 1997 年联合发起的新一代高分辨率中尺度气象模式，重点针对水平分辨率 1-10km 左右、时效为 60 小时的优先区域的天气预报和模拟问题[260-261]。WRF 模式被誉为新一代中尺度天气预报模式和同化系统，包含能够适应不同的地形、地貌边界和物理过程的参数化方案，可以较好模拟和预报对各种天气过程，为研究大气科学问题提供了一个好的平台和工具。WRF 为开源模式，可通过其官网 (http://www2.mmm.ucar.edu/wrf/users/) 进行获取，同时 WRF 高度模块化和分层设计，便于模式的发展、维护和管理。

目前 WRF 包含两个动力核心框架分别为 Advanced Research WRF (ARW) 和 Nonhydrostatic Mesoscale Model (NMM) 。本文所用的 WRF-ARW 是为研究理想和真实大气数值模拟所常用，是在 NCAR 的 MM5 模式 (The Fifth‑Generation NCAR/Penn State Mesoscale Model) 基础上发展而来，并由 NCAR 中小尺度气象部门负责维护管理并持续更新。

WRF-ARW 模式由前处理 WPS (WRF Preprocessing System) ，主模式 WRF 和后处理 (Post-Processing Programs) 三部分组成。WPS 用于对实时资料的处理，包括定义模拟区域，地形资料的插值，气象数据的垂直和水平插值等，提供 WRF 所需要的气象和地形数据。而主程序 WRF 通过各种物理过程的积分计算，输出特定模拟区域的气象结果。WRF 的计算框架如图2.1所示。

[Figure]

图 2.1: WRF 模式流程图 (摘自 WRF Version 3.9.1 User's Guide)

  WRF-Chem 是 NOAA 和 NCAR 在现有的 WRF-ARW 框架下加入了化学模块 (气溶胶化学、气相化学、气溶胶化学和光化学等模块)，以模拟大气中的多相态的化学物质的生成及其相互转换。WRF-Chem 模拟的化学场和气象模块使用了相同的时间积分步长，并且它们拥有相同的网格，垂直和坐标系设置，因此避免了时间或者空间插值产生的误差[262-263]。此外网格尺度上的物质传输由动力过程决定，次网格传输与物理参数化方案部分保持一致。化学模块和物理参数化方案的主要的相互作用如图2.2所示，化学模块主要与微物理、辐射以及边界层方案相互传递信息。这样在数值模拟的计算过程中，气象条件会驱动大气化学物质的传输和分布，并影响大气化学过程，部分化学过程也会对气象场造成反馈影响，从而实现了气象场和化学场在时空上的完全耦合，能够比较真实的反应大气物理化学过程[260]。WRF-Chem 的化学模块由 NOAA 地球系统研究实验室负责，并联合其他研究机构共同开

发，管理和维护，并与 WRF 同步更新，可通过官网 (https://ruc.noaa.gov/wrf/WG11/) 获取相关源代码和教程。

[Figure]

图 2.2: WRF-Chem 中物理和化学参数化过程之间的主要相互作用

**2.1.1 物理方案**

大气当中的多种物理过程在数值模式中由参数化方案的形式呈现。它们包括，生成网格尺度云水降水过程的微物理方案、生成次网格尺度的对流性降雨的积云方案、计算大气边界层中热力和动力过程的边界层方案、计算长波辐射和短波辐射的辐射方案以及提供地表下垫面属性的陆面过程。3.9.1 版本包含的主要物理参数化方案如表2.1所示。

表 2.1: 目前版本 WRF 包含的物理参数化方案

| 物理过程 | 参数化选项 |
| --- | --- |
| 云微物理 | Kessler，Lin，WSM3，WSM5，WSM6，WDM5,WDM6,Eta,Goddard,new Thompson,Milbrandt-Yau Double-Moment,Morrison double-moment,Y. Lin,NSSL 2-moment,CAM,Thompson Aerosol-Aware，HUJI，P3 |
| 长波辐射 | RRTM，GFDL，CAM，RRTMG，New Goddard，Fu-Liou-Gu |
| 短波辐射 | Dudhia，Goddard，GFDL，CAM，RRTMG,New Goddard,Fu-Liou-Gu，Held–Suarez relaxation |
| 近地面层 | MM5，Eta，Pleim-Xiu，QNSE，MYNN,TEMF，Revised MM5，Chen-Zhang |
| 陆面过程 | 5-layer thermal diffusion，Noah,RUC,Pleim-Xiu,Noah-MP,SSiB，Fractional sea-ice，CLM4 |
| 城市冠层 | UCM，BEP，BEM |
| 行星边界层 | YSU，MYJ，MRF，ACM2，QNSE，MYNN2,MYNN3,BouLac,UW,TEMF,LES,Grenier-Bretherton-McCaa，Shin-Hong |
| 积云对流 | KF，BMJ，GD，SAS，G3D，Tiedtke，Zhang-McFarlane,NSAS,GF,(old)KF,Multi-KF,New Tiedtke,Kain-Fritsch-Cumulus Potential |
| 其他物理过程 | Lake Physics，Ocean Physics，Gravity Wave Drag 等 |

常用的物理参数化方案如下：

1. 行星边界层方案

大气边界层对下垫面和自由大气之间的热量、动量和物质交换以及辐射的收支平衡有着非常重要的作用[264]，影响着各种变量在边界层内时间和空间的分布变化。在中尺度数值模式中湍流是一种次网格尺度的大气运动，需要利用参数化来模拟次网格过程及其与大尺度运动或平均流的相互作用[265]WRF 模式中的行星边界层方案主要包括：MRF、MYJ 和 YSU 方案。YSU 边界层方案[266]是一阶非局地闭合的梯度输送理论方案，由 MRF 参数化方案改进而来，相比 MRF 方案对夹卷过程有了更精确的描述，增加了由热力驱动的自由对流混合强度，减少了机械动力强迫性对流的混合强度[267]。改进后 YSU 方案中非局地通量项的量级小于 MRF 方案，逆梯度项的值减小，使大气层结接近中性，解决了 MRF 方案中逆梯度项过大而导致的大气层结过于稳定的问题[265]。

2. 云微物理方案

云微物理过程在中尺度数值模式中起着关键作用。云微物理过程描述云中水汽、云水、雨水、冰晶、雪、霰和雹七类水物质的形成与演变的过程。云微物理参数化方案主要包括：

Kessler 方案[268]、WSM3 方案[269]、WSM5 方案[266]和 Lin 方案[270]。其中，Lin 方案来自 Purdue 云模式，是云微物理过程描述较为复杂的方案，适合高分辨率模拟和理论研究。该方案中考虑的物质包括云冰、雪、云水、雨水、水汽和霰，其参数化方案主要来自 Lin 等[270]和 Rutledge 等[271]等。

**3. 积云对流方案**

积云对流参数化过程主要处理次网格降水。WRF 模式中的积云对流参数化方案主要包括：Kain-Fritsch 方案[272]，Betts-Miller-Janjic 方案[273]和 Grell-Devenyi 方案[274]。其中，Kain-Fritsch 方案利用一个伴有水汽上升下沉的简单云模式，考虑了云中上升气流卷入和下曳气流卷出及相对粗糙的微物理过程的影响。KF 方案在边缘不稳定、干燥的环境场中考虑了最小卷入率，以抑制大范围的对流，而对于没有达到最小降水云厚度的上升气流，考虑浅对流，最小降水云厚度随云底温度变化。

**4. 辐射过程方案**

WRF 模式中的辐射过程主要分为长波辐射和短波辐射两大部分。主要的长波辐射参数化方案包括：RRTM 方案[275]、ETA-GFDL 方案[276]、CAM 方案和 RRTMG 方案[275]。RRTM 方案源于 MM5 模式，该方案利用分段波谱法以及 K 分布来计算长波辐射对于温度的影响，利用一个预先处理的辐射参数表来计算水汽、$O_3$、$CO_2$ 及其他气体，以及云与辐射之间的相互作用。主要的短波辐射参数化方案包括：Dudhia 方案[277]，Goddard 方案[278]，ETA-GFDL 方案[276]和 RRTMG 方案[275]。RRTMG 为基于 MCICA 方法的一种新的短波辐射参数化方案。

**2.1.2 化学方案**

WRF-Chem 模式作为新一代三维欧拉空气质量模型，其污染物浓度的变化如以下的连续性方程2.1所示：

$$\frac{\partial n}{\partial t} = -\frac{\partial F_x}{\partial x} - \frac{\partial F_y}{\partial y} - \frac{\partial F_z}{\partial z} + P - L = -\nabla \cdot (nU) + P - L \tag{2.1}$$

其中，n 为某个网格中的污染物的浓度，P 为生成项，包含排放、化学生成，L 代表损耗项，包括干湿沉降、化学损耗，U=(u, v, w)。

针对各个过程，WRF-Chem 提供了多个化学参数化方案 (如表2.2所示)，能够较为全面的描述大气污染物的排放、化学反应、损耗等过程，主要考虑的过程包括气相化学反应，云水液相化学反应，光解过程，气溶胶化学，干湿沉降过程，自然源和人为源排放过程，野火抬升过程等。其每一个过程和所包含的机制都高度模块化从而方便维护和发展，各方案的具体内容可参考 WRF-Chem 网站上的相关论文 (http://ruc.noaa.gov/wrf/WG11/References/

WRF-Chem.references.htm)。

表 2.2: WRF-Chem 模式包含的化学过程参数化方案

| 化学过程机制 | 参数化选项 |
| --- | --- |
| 气相化学 | RADM2，RACM，RACM-MIM，RACM-ESRL,CBMZ,CB4,CB05,MOZART，SAPRC99，NMHC9，CRIMECH |
| 气溶胶化学 | MADE/SORGAM，VBS，MOSAIC (4 bins 或 8 bins)，GOCART，MAM |
| 云水液相化学 | VSRM |
| 光解 | TUV，Fast-J，Madronich F-TUV |
| 生物源排放 | Gunther95，BEIS3.14，MEGAN |
| 温室气体排放 | VPRM |
| 沙尘排放 | 适用于 GOCART 的沙尘机制，适用于 GOCART 并经过 AFWA 修改的沙尘机制，适用于 MOSAIC 和 MADE/SORGAM 的沙尘机制 |
| 海盐排放 | 适用于 GOCART 的海盐机制，适用于 MOSAIC 或 MADE/SORGAM 的海盐机制 |
| 气体干沉降 | Wesely |
| 气溶胶干沉降 | MADE 沉降机制，MOSAIC 沉降机制，Zhang 机制 |
| 湿沉降 | Grell 简单湿沉降方案，Easter 方案 |

常用化学参数化方案如下：

1. 光解方案

WRF-Chem 中主要包含三种光解方案：TUV 方案[279]、FAST-J 方案[280]和 Madronich F-TUV 方案[279]。Fast-J 方案主要考虑云、气体和气溶胶对光解率的影响，在对流层化学的模拟中利用每个粒径段粒子的数浓度、折射指数和湿半径，根据 Mie 散射理论分别计算了在 300 nm、400 nm、600 nm 和 999 nm 波长下的光学厚度、单次散射反照率和不对称因子，然后计算气体的光解速率。

2. 气相化学机制

气相化学机制是区域空气模型中一个重要的组成部分，气相转化率、排放、传输和沉降等过程决定了气体的种类和浓度。在对流层中，有机物、硫酸和氮氧化物等的排放以及臭氧和酸沉降过程与区域空气污染的关系受到气相化学机制的影响，同时气象化学也决定了液相化学的种类以及反应速度。当前 WRF-Chem 模式中可供选择的气相化学机制多达 9

种以上，并且所有的气相化学反应机制的气相化学动力学问题都采用动力学预处理器 KPP (Kinetic PreProcessor) 来处理，通过读取二进制输入格式的气相化学反应速率常数，并且自动生成用 Rosenbrok 求解的集成气相化学的代码进行后续计算。目前 KPP 只适用于气相化学反应机制，还没有应用到气溶胶化学机制。SAPRC99 (Statewide Air Pollution Research Center) 机制是光化学模拟中最常用的机制之一，是在 SAPRC90 的基础上发展形成。与 CB4 和 CB05 这类碳键机制相比，使用 SAPRC99 机制模拟的 $O_3$ 浓度通常更高[281]。SAPRC99 机制一共包含 80 个化学种类和 235 个化学反应，采用结构集总的方法处理有机化学反应。

**2.2 陆面过程模式**

Noah‑MP (The Community Noah Land Surface Model With Multi‑Parameterization Options) 陆面过程模式是基于 Noah-LSM 陆面过程模式发展而来的，用于研究陆气之间的相互作用，并对关键的陆气相互作用过程使用多种选项[282]。Noah-MP 包含一个单独的植被冠层，由冠层顶部和底部、冠层半径以及具有规定的尺寸、方向、密度和辐射特性的叶片定义。冠层采用了双流辐射传输方法，以及实现适当地表能量和水分传输过程所必需的遮阳效果[283]。此外，Noah-MP 能够区分 C3 和 C4 作物的光合作用途径，并为植物光合作用和呼吸作用定义特定的植被参数。

Noah-MP 可用于预测植被的生长，它结合了 Ball-Berry 基于光合作用的气孔阻抗机制，将碳分配到植被的各个部分 (如根、茎、叶和木) 和土壤碳库[210,284]，然后通过光合作用预测初级生产力、叶面积指数和冠层高度等。Noah-MP 还分别考虑了阳光下和遮荫下叶片两面的光合作用，阳光下的叶片更受 $CO_2$ 浓度的限制，而遮荫的叶片则更受日照的限制，因此可能对 $O_3$ 的损害有不同的反应程度。动态的叶面积指数和冠层高度的计算将进一步影响地表能量平衡。Noah-MP 陆面过程模式的结构图 (https://www.jsg.utexas.edu/noah-mp/) 如图2.3所示。

[Figure]

图 2.3: Noah-MP 陆面过程模式的结构图

**2.3 干沉降过程计算框架**

数值模式中，大气污染物的干沉降通量 $F_d$ 是通过污染物浓度 (C) 和污染物的干沉降速率 ($V_d$(z)) 的乘积计算得到的：

$$F_d = -C(z) \times V_d(z) \tag{2.2}$$

其中，z 为参考高度，一般为模式靠近地面的第一层，负号表示向下的通量。干沉降参数化方案是计算每种污染物在每个网格点每个时间步长下的干沉降速率。目前大部分模式采用类比于欧姆定律的阻抗方法来计算干沉降速率，该方法认为 $V_d$(z) 为大气污染沉降到地表过程的总阻抗 ($R_t$) 的倒数。

WRF-Chem 模式目前采用 Wesely 方案[44]对气体污染物的干沉降过程进行计算,如图2.4所示。Wesely 方案是基于大叶阻抗干沉降理论发展而来的，它对冠层阻抗的计算充分考虑了下垫面植被的生长结构和分布特征，并根据每个部分的实际特征再进行细分，得到了与冠层阻抗有关的 7 个阻抗因子。污染物从底层大气到近地表下垫面的迁移经历了三个过程，首先是污染物通过湍流输送作用，从大气边界层迁移到粘性副层之上，这一过程表示为空气动力学阻抗 ($R_a$，Aerodynamic Resistance)；然后通过分子扩散作用，从粘性副层迁移至下垫面附近，这一过程表示为用粘性副层阻抗 ($R_b$，Quasi-laminar layer Resistance)；最终下垫面如土壤、植被、水面等对污染物进行吸附和捕捉，被固定在地表导致干沉降过程的最

终发生，用冠层阻抗 ($R_c$，Canopy Resistance) 表示。阻抗模型为大气污染物的干沉降速度与三个阻抗之和呈反比，计算公式如下：

$$V_d = R_t^{-1} = \frac{1}{R_a(z) + R_b + R_c} \tag{2.3}$$

[Figure]

图 2.4: Wesely 干沉降方案的阻抗结构图

其中，$R_a$ 与摩擦速度 ($u^*$)、莫宁-奥布霍夫长度 (L，Monin–Obukhov length) 和大气稳定度 ($z/L$) 等大气湍流特征量有关。Wesely 方案采用 MOST 相似理论[285]进行计算，该方法将大气稳定度分为了稳定、中性和不稳定三种情况分别进行计算：

$$R_a = \begin{cases} 0.74(\kappa u^*)^{-1}[ln(\frac{z}{z_0}) + 4.7\frac{z - z_0}{L}] & \text{稳定条件} \\[2mm] 0.74(\kappa u^*)^{-1}ln(\frac{z}{z_0}) & \text{中性条件} \\[2mm] 0.74(\kappa u^*)^{-1}\{ln[\frac{(1 - 9\frac{z}{L})^{0.5} - 1}{(1 - 9\frac{z}{L})^{0.5} + 1}] - ln[\frac{(1 - 9\frac{z_0}{L})^{0.5} - 1}{(1 - 9\frac{z_0}{L})^{0.5} + 1}]\} & \text{不稳定条件} \end{cases} \tag{2.4}$$

其中，$z_0$ 为动量粗糙长度；$\kappa$ 为卡门常数 (0.4)；L 由以下计算公式得到：

$$L = -\frac{\rho c_p (u^*)^3 \theta}{\kappa g(H + L_w E/14)} \tag{2.5}$$

其中，$\rho$ 为空气密度；$c_p$ 为常压下的空气比热；$\theta$ 为绝对位温；g 为重力加速度；H 为显热通量；$L_w E$ 为潜热通量。

$R_b$ 与粘性副层的厚度、气态污染物分子半径、温度和 $u^*$ 等因素有关，采用 Wesely 等[286]中的方法进行计算：

$$R_b = 2(\kappa u^*)^{-1}(S_c/P_r)^{2/3} \tag{2.6}$$

其中，$S_c$ 为施密特数，与气态污染物分子半径和温度有关；$P_r$ 为空气中的普朗特数 (0.72)。

$R_c$ 不仅与下垫面的物理、化学性质有关还与植被的生理过程以及冠层内的气象条件 (温度，湿度和辐射等) 有关，因此 $R_c$ 的计算也是最为复杂的。如图2.4所示，Wesely 方案认为气态污染物迁移至地表，可以沉降至植被冠层高层覆盖的部分，冠层低层的部分或地表土壤，这三部分叠加为最终总的地表上的沉降。这三部分共用 7 个串行或并行的阻抗分项来表示，计算式如下：

$$R_c = (\frac{1}{R_s + R_m} + \frac{1}{R_{lu}} + \frac{1}{R_{dc} + R_{cl}} + \frac{1}{R_{ac} + R_{gs}})^{-1} \tag{2.7}$$

其中，$R_s$ 为叶片气孔阻抗 (Leaf Stomata Resistance)，$R_m$ 为叶肉阻抗 (Leaf Mesophyll Resistance)，$R_{lu}$ 为叶片角质层阻抗 (Leaf Cuticles Resistance)，$R_{dc}$ 为植被冠层内的大气浮力阻抗 (Buoyant Convection Resistance)，$R_{cl}$ 为树枝、树皮等的表面阻抗 (Resistance for Lower Canopy Exposed Surface)，$R_{ac}$ 为冠层顶部至地表面的空气动力学阻抗 (Aerodynamic Resistance from Canopy Top to Ground)，$R_{gs}$ 为土壤、枯枝落叶层等覆盖物的表面阻抗 (Resistance at the Ground Surface)。

下面分别介绍 $R_c$ 的每个分项：

冠层气孔阻抗 ($R_s$) 主要取决于叶片气孔张开孔径的大小，叶片气孔的张开或闭合主要取决于植物的光合作用需要。当有太阳辐射时，气孔打开，吸收 $CO_2$ 进行光合作用，阻抗变小，在夜间气孔一般闭合，阻抗增大。此外阻抗的变化也与外界的温度有关，Wesely 方案采用了一个简单的经验公式对 $R_s$ 进行计算：

$$R_s = R_i\{1 + \frac{1}{[200(G + 0.1)]^2}\}\frac{400}{T_s(40 - T_s)}\frac{D_{H_2O}}{D_x} \tag{2.8}$$

其中，$R_i$ 为在理想状况下的最小冠层气孔阻抗 (Minimum Canopy Stomatal Resistance)，G 为地表净太阳辐射 (Net Solar Irradiation at Ground)，$T_s$ 为表面温度 (Surface Temperature)，$D_x$ 为气态污染物分子的扩散速率，$D_{H_2O}$ 为水汽分子的分子扩散速率，当 $T_s$ 小于 0°C 或大于 40°C 时，$R_s$ 设置为 9999。

$R_c$ 的其他阻抗分项如 $R_m$、$R_{lu}$、$R_{cl}$ 和 $R_{gs}$ 大多与气态污染物的溶解性和氧化性相关。水溶性大、氧化性强的气体较容易沉降至地表面。Wesely 方案中，这些阻抗值是基于有效亨利常数 ($H^*$) 和反应活性因子 ($f_0$) 来计算的。对于不同的气态污染物，这两个参数的取值如表2.3所示。

表 2.3: Wesely 干沉降方案中气态污染物的部分属性参数

| 气态污染物 | $D_{H_2O}/D_x$ | $H^*$ (M atm$^{-1}$) | $f_0$ |
|---|---|---|---|
| 二氧化硫 ($SO_2$) | 1.9 | $1 \times 10^5$ | 0 |
| 臭氧 ($O_3$) | 1.6 | 0.01 | 1 |
| 二氧化氮 ($NO_2$) | 1.6 | 0.01 | 0.1 |
| 一氧化氮 ($NO$) | 1.3 | $2 \times 10^{-3}$ | 0 |
| 硝酸 ($HNO_3$) | 1.9 | $1 \times 10^{14}$ | 0 |
| 过氧化氢 ($H_2O_2$) | 1.4 | $1 \times 10^5$ | 1 |
| 乙醛 (ALD) | 1.6 | 15 | 0 |
| 甲醛 (HCHO) | 1.3 | $6 \times 10^3$ | 0 |
| 甲基过氧化氢 (OP) | 1.6 | 240 | 0.1 |
| 过氧乙酸 (PAA) | 2.0 | 540 | 0.1 |
| 甲酸 (ORA) | 1.6 | $4 \times 10^6$ | 0 |
| 氨气 ($NH_3$) | 1.0 | $2 \times 10^4$ | 0 |
| 过氧酰基硝酸酯 (PAN) | 2.6 | 3.6 | 0.1 |
| 亚硝酸 ($HNO_2$) | 1.6 | $1 \times 10^5$ | 0.1 |

$R_m$ 的计算公式为:

$$R_m = (\frac{H^*}{3000} + 100 f_0)^{-1} \tag{2.9}$$

$R_{lu}$ 对植被冠层表面分为干燥和潮湿两种状况作分别进行计算。对于干燥的冠层表面：

$$R_{lu,dry} = R_{lu}(10^{-5} H^* + f_0)^{-1} \tag{2.10}$$

其中，$R_{lu}$ 可查表获得，见表2.4。

对于潮湿的冠层表面，分为由于露水导致的和由于雨水导致的。其中，由于露水导致的植被冠层表面潮湿，Wesely 方案中将 $SO_2$ 的 $R_{lu}$ 设置为 50 s·m$^{-1}$ (城市下垫面) 和 100 s·m$^{-1}$ (其他下垫面)。$O_3$ 的 $R_{lu}$ 则由下式计算：

$$R_{lu,O_3,dew} = (\frac{1}{3000} + \frac{1}{3R_{lu}})^{-1} \tag{2.11}$$

其他污染物的 $R_{lu}$ 则由 $R_{lu,O_3,dew}$ 的进行推导估计：

$$R_{lu,dew} = (\frac{1}{3R_{lu,dry}} + 10^{-7}H^* + \frac{f_0}{R_{lu,O_3,dew}})^{-1}$$ (2.12)

由于雨水导致的植被冠层表面潮湿，将城市下垫面的 $SO_2$ 的 $R_{lu}$ 统一设置为 50 s·m$^{-1}$，其他下垫面的计算式为：

$$R_{lu,SO_2,rain} = (\frac{1}{5000} + \frac{1}{3R_{lu}})^{-1}$$ (2.13)

$O_3$ 的 $R_{lu}$ 的计算则变为：

$$R_{lu,O_3,rain} = (\frac{1}{1000} + \frac{1}{3R_{lu}})^{-1}$$ (2.14)

其他污染物在这一状况下的 $R_{lu}$ 仍由 $R_{lu,O_3,dew}$ 的进行推导估计：

$$R_{lu,rain} = (\frac{1}{3R_{lu,dry}} + 10^{-7}H^* + \frac{f_0}{R_{lu,O_3,rain}})^{-1}$$ (2.15)

$R_dc$ 主要取决于地形坡度和太阳辐射对地表的加热状况决定，其计算式为：

$$R_{dc} = \frac{100(1 + \frac{1000}{G + 10})}{1 + 1000\theta}$$ (2.16)

其中，G 为太阳辐射，$\theta$ 为地形坡度。

对于 $SO_2$ 和 $O_3$ 的 $R_{cl}$ 都由查表获得。其他气态污染物则根据自身的水溶性、氧化性以及 $R_{cl,SO_2}$ 和 $R_{cl,O_3}$ 进行推导估计：

$$R_{cl} = (\frac{10^{-5}H^*}{R_{cl,SO_2}} + \frac{f_0}{R_{cl,O_3}})^{-1}$$ (2.17)

$R_{gs}$ 与 $R_{cl}$ 类似，由 $R_{gs,SO_2}$ 和 $R_{gs,O_3}$ 推导估计：

$$R_{gs} = (\frac{10^{-5}H^*}{R_{gs,SO_2}} + \frac{f_0}{R_{gs,O_3}})^{-1}$$ (2.18)

$R_{ac}$ 值可以查表获得，见表2.4。

表 2.4: Wesely 干沉降方案中气态污染物的输入参数

| 阻抗 | 土地利用类型 | | | | | | | | | | |
|------|------|------|------|------|------|------|------|------|------|------|------|
| 分项 | 1 | 2 | 3 | 4 | 5 | 6 | 7 | 8 | 9 | 10 | 11 |
| 季节类型 1：有茂盛植被的夏季 | | | | | | | | | | | |
| ri | 9999 | 60 | 120 | 70 | 130 | 100 | 9999 | 9999 | 80 | 100 | 15 |
| rlu | 9999 | 2000 | 2000 | 2000 | 2000 | 2000 | 9999 | 9999 | 2500 | 2000 | 4000 |
| rac | 100 | 200 | 100 | 2000 | 2000 | 2000 | 0 | 0 | 300 | 150 | 200 |
| rgss | 400 | 150 | 350 | 500 | 500 | 100 | 0 | 1000 | 0 | 220 | 400 |
| rgso | 300 | 150 | 200 | 200 | 200 | 300 | 2000 | 400 | 1000 | 180 | 200 |
| rcls | 9999 | 2000 | 2000 | 2000 | 2000 | 2000 | 9999 | 9999 | 2500 | 2000 | 4000 |
| rclo | 9999 | 1000 | 1000 | 1000 | 1000 | 1000 | 9999 | 9999 | 1000 | 1000 | 1000 |
| 季节类型 2: 农作物未被收割的秋季 | | | | | | | | | | | |
| ri | 9999 | 9999 | 9999 | 9999 | 250 | 500 | 9999 | 9999 | 9999 | 9999 | 9999 |
| rlu | 9999 | 9000 | 9000 | 9000 | 4000 | 8000 | 9999 | 9999 | 9000 | 9000 | 9000 |
| rac | 100 | 150 | 100 | 1500 | 2000 | 1700 | 0 | 0 | 200 | 120 | 140 |
| rgss | 400 | 200 | 350 | 500 | 500 | 100 | 0 | 1000 | 0 | 300 | 400 |
| rgso | 300 | 150 | 200 | 200 | 200 | 300 | 2000 | 400 | 800 | 180 | 200 |
| rcls | 9999 | 9000 | 9000 | 9000 | 2000 | 4000 | 9999 | 9999 | 9000 | 9000 | 9000 |
| rclo | 9999 | 400 | 400 | 400 | 1000 | 600 | 9999 | 9999 | 400 | 400 | 400 |
| 季节类型 3: 霜冻后的晚秋 (无雪覆盖) | | | | | | | | | | | |
| ri | 9999 | 9999 | 9999 | 9999 | 250 | 500 | 9999 | 9999 | 9999 | 9999 | 9999 |
| rlu | 9999 | 9999 | 9000 | 9000 | 4000 | 8000 | 9999 | 9999 | 9000 | 9000 | 9000 |
| rac | 100 | 10 | 100 | 1000 | 2000 | 1500 | 0 | 0 | 100 | 50 | 120 |
| rgss | 400 | 150 | 350 | 500 | 500 | 200 | 0 | 1000 | 0 | 200 | 400 |
| rgso | 300 | 150 | 200 | 200 | 200 | 300 | 2000 | 400 | 1000 | 180 | 200 |
| rcls | 9999 | 9999 | 9000 | 9000 | 3000 | 6000 | 9999 | 9999 | 9000 | 9000 | 9000 |
| rclo | 9999 | 1000 | 400 | 400 | 1000 | 600 | 9999 | 9999 | 800 | 600 | 600 |
| 季节类型 4: 零度以下有雪覆盖的冬季 | | | | | | | | | | | |
| ri | 9999 | 9999 | 9999 | 9999 | 400 | 800 | 9999 | 9999 | 9999 | 9999 | 9999 |
| rlu | 9999 | 9999 | 9999 | 9999 | 6000 | 9000 | 9999 | 9999 | 9000 | 9000 | 9000 |
| rac | 100 | 10 | 10 | 1000 | 2000 | 1500 | 0 | 0 | 50 | 10 | 50 |

续表

| 阻抗 | 土地利用类型 | | | | | | | | | | |
|---|---|---|---|---|---|---|---|---|---|---|---|
| 分项 | 1 | 2 | 3 | 4 | 5 | 6 | 7 | 8 | 9 | 10 | 11 |
| rgss | 100 | 100 | 100 | 100 | 100 | 100 | 0 | 1000 | 100 | 100 | 50 |
| rgso | 600 | 3500 | 3500 | 3500 | 3500 | 3500 | 2000 | 400 | 3500 | 3500 | 3500 |
| rcls | 9999 | 9999 | 9999 | 9000 | 200 | 400 | 9999 | 9999 | 9000 | 9999 | 9000 |
| rclo | 9999 | 1000 | 1000 | 400 | 1500 | 600 | 9999 | 9999 | 800 | 1000 | 800 |
| 季节类型 5: 有部分矮小一年生绿色植物的春季 | | | | | | | | | | | |
| ri | 9999 | 120 | 240 | 140 | 250 | 190 | 9999 | 9999 | 160 | 200 | 300 |
| rlu | 9999 | 4000 | 4000 | 4000 | 2000 | 3000 | 9999 | 9999 | 4000 | 4000 | 8000 |
| rac | 100 | 50 | 80 | 1200 | 2000 | 1500 | 0 | 0 | 200 | 60 | 120 |
| rgss | 500 | 150 | 350 | 500 | 500 | 200 | 0 | 1000 | 0 | 250 | 400 |
| rgso | 300 | 150 | 200 | 200 | 200 | 300 | 2000 | 400 | 1000 | 180 | 200 |
| rcls | 9999 | 4000 | 4000 | 4000 | 2000 | 3000 | 9999 | 9999 | 4000 | 4000 | 8000 |
| rclo | 9999 | 1000 | 500 | 500 | 1500 | 700 | 9999 | 9999 | 600 | 800 | 800 |

**2.4 生态效应评估方法**

**2.4.1 AOT40 指标的计算**

AOT40 是目前最常用的 $O_3$ 评估指标之一，表示为每小时 $O_3$ 浓度超过 40 $nmol \cdot mol^{-1}$ 部分的累积值 (单位为 $nmol \cdot mol^{-1} \cdot h^{-1}$)[287]。AOT40 是一种剂量指标，它考虑了 $O_3$ 浓度与累积暴露时间，能较好地反映 $O_3$ 对作物长时间的胁迫效应[288]。其计算公式如下：

$$AOT40 = \sum_{i=1}^{n} max(C_{O_3 i} - 40, 0) \tag{2.19}$$

其中，$C_{O_3}$ 表示大气中的 $O_3$ 浓度。
* * *
[1]9999 表示阻抗极大，该沉降途径无沉降发生；

[2]土地利用类型：(1) 城市用地，(2) 农业用地，(3) 牧场，(4) 落叶林，(5) 针叶林，(6) 包含湿地的混交林，(7) 水体，(8) 沙漠等荒漠地区，(9) 无森林覆盖的湿地，(10) 农业与牧场混合用地，(11) 长有矮小灌木的岩石地带。

**2.4.2 作物产量和经济损失的估算**

**2.4.2.1 全国各省市水稻产量**

2019 年全国双季稻种植区 (不包含港澳台地区) 水稻产量数据来源于中国农业统计年鉴 (http://www.stats.gov.cn/)，如表2.5所示。在我国，水稻的种植分布十分广泛，可分为双季早稻、双季晚稻和单季水稻三类。在本研究中主要讨论双季早稻，其种植区主要分布于东南沿海的浙江、福建、广东、广西和海南，长江中下游的安徽、江西、湖南和湖北，以及云南地区。

表 2.5: 全国各省市双季早稻产量情况

| 省份 | 浙江 | 安徽 | 福建 | 江西 | 湖北 | 湖南 | 广东 | 广西 | 海南 | 云南 |
|------|------|------|------|------|------|------|------|------|------|------|
| 产量 (万吨) | 50.3 | 101.0 | 61.6 | 626.2 | 84.3 | 661.4 | 488.3 | 452.6 | 69.2 | 21.7 |

**2.4.2.2 基于 AOT40 指标的响应方程**

研究表明，$O_3$ 对作物的损伤主要是由 $O_3$ 长期累积效应引起[157]。当 $O_3$ 浓度超过某一临界值时，$O_3$ 才会影响作物的生长和产量。近几十年来，我国研究人员利用 OTC/FACE 系统开展了大量实验探究 $O_3$ 浓度升高对不同作物的影响，并在此基础上建立了不同作物相对产量 (RY，Relative yield) 与 AOT40 指标之间的线性函数关系。本研究使用表2.6中的剂量响应方程评估地表 $O_3$ 浓度对水稻产量的影响。

其中，AOT40 指标表示作物生长期内的白天时段，大于 40 $nmol \cdot mol^{-1}$ 的小时平均 $O_3$ 浓度与 40 $nmol \cdot mol^{-1}$ 差值的累积。根据 Avnery 等[246]的研究方法，在计算 AOT40 指标时将白天时段定义为每天的 08:00-19:59。

表 2.6: 水稻相对产量与 AOT40 指标的响应方程

| 作物 | 实验方式 | 剂量响应方程 | 文献 |
|------|---------|-------------|------|
| 双季早稻 | OTC | $RY = -0.0053 \times AOT40 + 1$ | Feng 等[289] |
| | OTC | $RY = -0.0095 \times AOT40 + 1$ | Wang 等[290] |
| | OTC | $RY = -0.010 \times AOT40 + 1$ | 耿春梅 等[291] |
| | FACE | $RY = -0.022 \times AOT40 + 0.969$ | 张继双 等[292] |

**2.4.2.3 水稻产量和经济损失的估算**

根据 Avnery 等[246]所用的作物产量和经济损失的计算公式，将作物生长期内 AOT40 的值代入到上述方程中，计算出作物相对产量及作物产量损失率，再结合各省市收获的当年

作物实际产量，即可得到因 $O_3$ 污染造成的作物减产量，具体的计算公式如下：

$$
\begin{aligned}
RYL &= 1 - RY \\
CPL &= CP \times \frac{RYL}{1 - RYL}
\end{aligned}
$$
(2.20)

其中，RYL 为作物产量损失率 (RYL，Relative yield loss) 定义为在不受 $O_3$ 损伤的情况下，以理论产量为基础的作物产量的损失；RY 为作物相对产量 (Relative yield)；CPL 为作物减产量 (Crop production loss)；CP 为作物实际产量 (Crop production)。

同时，因 $O_3$ 损伤所造成的作物经济损失的计算方法如下如下：

$$
ECL = CPL \times MPP
$$
(2.21)

其中,ECL 为经济损失 (Economic cost loss) ,MPP 为作物的最低市场收购价格 (Minimum purchase price)，本研究使用的是联合国粮农组织统计数据库提供的价格作为国内市场价格 (FAOSTAT，2019，http://faostat.fao.org/)。

**第三章 干沉降参数化方案评估**

本章采用 Chang 等[293]发开的单点干沉降机理诊断耦合模式。该模式是将 WRF‑Chem 干沉降模块 (WDDM) 从 WRF‑Chem 区域模式中剥离出来，与 Noah-MP 模式进行耦合，形成单点的干沉降机理诊断模式 (Noah-MP-WDDM)，便于在单点上首先进行关键参数和过程的识别。Noah-MP-WDDM 模式中，首先由气象数据、土壤和植被参数驱动 Noah-MP，计算该时刻内的地表物候、湍流特征、能量和水分的分配，并由此计算得到影响各沉降阻抗部分的冠层、地表、土壤变量输入至 WDDM 模块计算得到各沉降阻抗，而后得到各大气成分的干沉降速率，计算碳组分的积累量从而进行植被的动态模拟，得到更新了的下一时刻地表的植被、湍流参数，再进入下一时间步长的迭代。并将每个部分的能量、水分变量输出，以判断和评估其对该部分沉降阻抗乃至最终沉降速率的影响。其中，干沉降部分的计算流程，已在2.3节详细介绍。

本章基于 Noah-MP-WDDM 模式，评估干沉降方案中的两种气孔机制，并对影响气孔机制的主要参数和过程进行了改进和敏感性实验，评估在典型下垫面不同干沉降机制对臭氧干沉降速率的模拟效果。

**3.1 观测站点介绍**

本研究其中一个观测站点位于广东省珠江三角洲西北边的肇庆市鼎湖山自然生态保护区内的森林生态系统交换通量观测塔。鼎湖山自然保护区北回归线附近，属典型的南亚热带季风湿润型气候。通量观测塔 (112°32' E，23°10' N，以下简称鼎湖山站) 的观测平台位于鼎湖山自然生态保护区的核心区域，是一片针阔混交林样地。通量观测塔高 37 m，其所位于的区域森林冠层高度约为 17 m。两套涡动相关测量系统分别安装在塔的两个高度 (30 米和 7 米)，代表在冠层外和冠层内的测量结果。该系统主要由两个三维超声风速仪 (CSAT3，Campbell Scientific，USA)，两个快速化学发光 $O_3$ 分析仪 (FOS，Sextant，New Zealand)，$H_2O/CO_2$ 浓度红外分析仪 (Li-750，LI-COR，USA)，一个慢速紫外 $O_3$ 分析仪 (型号 49i, Thermo Fisher，USA) 和 CR6 数据采集器组成。其中闭路的 $H_2O/CO_2$ 浓度红外分析仪用来测量 $H_2O$ 和 $CO_2$ 密度；三维超声风速仪用来测量三维风速和超声虚温；$O_3$ 浓度由慢速紫外 $O_3$ 分析仪进行观测；$O_3$ 的快速响应则利用快速 $O_3$ 分析仪监测，该仪器的主要工作原理为香豆素涂层片被 $O_3$ 氧化消耗发出蓝光，被光电倍增管探测到后转化成为电压输出，获得的数据需要利用慢速紫外 $O_3$ 分析仪获得的 30 min 平均 $O_3$ 浓度进行时时校准。系统昼夜

连续自动采集数据，原始采样频率是 10 Hz，由 CR6 数据采集器自动存储并在线计算每 30 min 平均 $O_3$ 通量。鼎湖山的观测时间为 2019 年 8 月 21 日至 2019 年 11 月 21 日。此站点作为森林生态系统的代表。

本研究的另一个观测站点位于江苏省南京市浦口区南京信息工程盘城镇永丰生态试验地 (32°19' N，118°71' E，以下简称永丰站)，属于北亚热带季风性气候。观测时间为 2017 年 3 月 25 日至 2017 年 5 月 25 日，观测期间试验田所种作物为冬小麦，此站点作为农田生态系统的代表。试验观测仪器同样基于涡度相关系统，与鼎湖山的使用的仪器相同，数据采集器为 CR3000。

**3.2 不同干沉降参数化方案对比实验设计**

**3.2.1 模式输入设置**

Noah-MP-WDDM 模式需要大气驱动数据、植被和土壤等多项地表参数做为输入数据。其中大气驱动数据包括研究时段内的温度、风速、风向、湿度、短波辐射、长波辐射、气压及降水。Wu 等[96]研究表明，模拟气象数据驱动的 $V_d$ 值比观测气象数据驱动的 $V_d$ 值的高 30%，且相关性更差，因此为了减少气象要素模拟偏差对 $V_d(O_3)$ 模拟的影响，本研究中两个站点均采用观测得到的 30 min 气象数据作为输入数据。

依据鼎湖山站和永丰站的输入数据结果，修改对应的输入参数，主要包括有：模拟的开始及结束的时间、模拟循环次数、模拟站点的坐标、时间步长、土地覆被类型、土壤类型、温度和风速的测量高度等。具体设置值如表3.1所示。

表 3.1: 模式输入参数设置

| 参数名称 | 鼎湖山站 | 永丰站 | 备注 |
|---|---|---|---|
| Startdate | 201908211600 | 201703250000 | 采用 UTC 时间 |
| Enddate | 201911211600 | 201705270000 | 采用 UTC 时间 |
| Loop for a while | 20 | 20 | 模式 spin-up |
| Latitude (°) | 23.17 | 32.19 | 站点纬度 |
| Longitude (°) | 112.53 | 118.71 | 站点经度 |
| Forcing timestep (s) | 1800 | 1800 | 驱动数据时间步长 |
| Noahlsm timestep (s) | 900 | 900 | 模拟时间步长 |
| Sea ice point | False | False | 海冰点 |
| Soil layer thickness (m) | 0.1,0.3，0.6，1.0 | 0.1，0.3，0.6，1.0 | 土壤层厚度 |
| Soil temperature (K) | 266.09,274.04,276.90,279.92 | 266.09，274.04，276.90,279.92 | 各层土壤平均温度 |
| Soil moisture (m$^3$·m$^{-3}$) | 0.298,0.294,0.271,0.307 | 0.298，0.294，0.271，0.307 | 各层土壤平均湿度 |
| Skin temperature (K) | 304.29 | 292.24 | 地表平均温度 |
| Deep soil T (K) | 297.40 | 292.78 | 1m 深度平均温度 |
| Vegetation type | Mixed Forest | Dryland Cropland and Pasture | USGS 分类 |
| Soil type | Loamy Sand | Sandy Clay Loam | STAS 土壤类型分类 |
| Air temperature level (m) | 30.0(冠层外)，7.0(冠层内) | 7.0 | 气温驱动数据高度 |
| Wind level (m) | 30.0(冠层外)，7.0(冠层内) | 7.0 | 风速驱动数据高度 |

  本研究所使用的过程方案与默认的 Noah‑MP 模式一致，其中由于鼎湖山站和永丰站两个站点都处于亚热带地区，与冻土中的过冷液态水 (FRZ)、冻土渗透率 (INF)、积雪反照率 (ALB)、雨雪分离过程 (SNF)、土壤底部温度 (TBOT) 和雪/土壤温度时间方案 (STC) 因素相关的物理过程对模式的结果基本没有影响，因此这些过程都选择默认选项。而动态植被模型 (DVEG)、土壤水势因子 (BTR)、地表交换系数 (SFC)、径流与地下水 (RUN)、冠层辐射传输 (RAD) 和冠层气孔阻抗 (CRS) 对结果影响较大，因此本研究根据 Chang 等[294]的研究，这些过程参数化方案选择如表3.2所示。

表 3.2: Noah-MP 模式参数化方案

| 参数化方案 | 方案选择 | 内容 |
| --- | --- | --- |
| 动态植被模型 | 2 | Dynamic |
| 冠层气孔阻抗 | 1; 2 | Ball-Berry; Jarvis |
| 土壤水势因子 | 3 | SSiB Scheme |
| 径流与地下水 | 3 | Schaake96 Scheme |
| 地表交换系数 | 2 | Chen97 Scheme |
| 冻土中的过冷液态水 | 1 | NY06 Scheme |
| 冻土渗透性 | 1 | NY06 Scheme |
| 冠层辐射传输 | 2 | Vegetation gap = 0 |
| 积雪反照率 | 1 | BATS Scheme |
| 雨雪分离过程 | 1 | Jardan91 Scheme |
| 土壤底部温度 | 1 | Zero-flux Scheme |
| 雪/土壤温度时间方案 | 1 | Fully implicit |

**3.2.2 不同干沉降参数化方案**

在区域和全球的大气化学传输模式中，干沉降速率的计算通常使用的是大叶模型。根据对干沉降方案的文献调研，$R_s$ ($R_s$=1/$g_s$) 通常使用以下的方法进行计算：

a. 简单的 Jarvis 气孔导度机制，只考虑了温度、辐射影响[44]；

b. 使用 Jarivs 气孔导度机制的方法，考虑了太阳辐射、温度、湿度和土壤水分胁迫[48,50,124,295]；

c. Ball-Berry 气孔导度机制，该机制利用了光合作用 ($A_n$) 和气孔导度 ($g_s$) 之间的函数关系[210,296]。

为了探究不同干沉降参数化方案对 $V_d(O_3)$ 模拟的影响，本研究将上述描述的三个气孔导度机制与 Noah-MP-WDDM 中的干沉降方案重新修改耦合编译，具体方案如下：

1. Wesely 等[44]方案 (以下简称为 W89)，CTMs 中应用最广泛的气态干沉降方案[72]，气孔导度机制使用方法a；

2. Zhang 等[48]方案 (以下简称为 Z03)，常用于北美的空气质量模型[297]和加拿大空气和降水监测网络 (CAPMoN，Canadian Air and Precipitation Monitoring Network)[298]，气孔导度机制使用方法b；

3. Z03 方案但是气孔导度机制使用方法c (以下简称为 Z03&BB) ；

4. W89 方案但是气孔导度机制使用方法c (以下简称为 W89&BB) 。

    其中 W89 为 Noah-MP-WDDM 中的默认干沉降方案。

    四种干沉降方案的具体描述如表3.3所示。

表 3.3: 四种干沉降参数化方案比较

| | Z03 | W89 | Z03&BB | W89&BB |
|---|---|---|---|---|
| $R_a$ | 稳定条件，$$R_a = \frac{0.74}{\kappa u^*}[ln(\frac{z}{z_0}) + 4.7\frac{z}{L}]$$ 不稳定条件，$$R_a = \frac{0.74}{\kappa u^*}[ln(\frac{z}{z_0}) - 2ln(\frac{\sqrt{(1-9\frac{z}{L}+1)}}{2})]$$ | 稳定条件，$$R_a = \frac{0.74}{\kappa u^*}[ln(\frac{z}{z_0}) + 4.7\frac{z-z_0}{L}]$$ 不稳定条件，$$R_a = \frac{0.74}{\kappa u^*}[ln(\frac{\sqrt{(1-9\frac{z}{L}-1)}}{\sqrt{(1-9\frac{z}{L}+1)}}) - ln(\frac{\sqrt{(1-9\frac{z_0}{L}-1)}}{\sqrt{(1-9\frac{z_0}{L}+1)}})]$$ | 与 Z03 一致 | 与 W89 一致 |
| $R_b$ | $$R_b = \frac{2}{\kappa u^*}(D_\theta/D_c)^{2/3}$$ | $$R_b = \frac{2}{\kappa u^*}(S_c/P_r)^{2/3}$$ | | |
| $R_s$ | $$R_s = \frac{1}{G_s(PAR)f(T)f(D)f(\Psi)}\frac{D_{H_2O}}{D_{O_3}}$$ | $$R_s = r_{s,min}(1 + \frac{1}{[200(R_G+0.1)]^2}\frac{400}{T_s(40-T_s)})\frac{D_{H_2O}}{D_{O_3}}$$ | $$g_s = g_0 + m\frac{A_n}{C_s}\frac{h_s}{}$$ $$R_s = \frac{1}{(1-W_{st})}\frac{D_{H_2O}}{D_{O_3}}$$ | $$g_s = g_0 + m\frac{A_n}{C_s}\frac{h_s}{C_s}$$ $$R_s = \frac{1}{g_s}\frac{D_{H_2O}}{D_{O_3}}$$ |
| $R_{cut}$ | 干燥表面，$$R_{cut} = \frac{R_{cutd0}}{e^{0.03RH}LAI^{0.25}u^*}$$ 潮湿表面，$$R_{cut} = \frac{R_{cutw0}}{LAI^{0.5}u^*}$$ | $$R_{cut} = \frac{R_{cut0}}{LAI}$$ | 与 Z03 一致 | 与 W89 一致 |
| $R_{ac}$ | $$R_{ac} = R_{ac0}\frac{LAI^{0.25}}{u^*}$$ | 查表值 | 与 Z03 一致 | 与 W89 一致 |
| $R_g$ | | 查表值 | | |

其中：z 为参考高度，$z_0$ 为粗糙度长度，$\kappa$ 为卡门常数，$u^*$ 为摩擦风速，L 为莫宁-奥布霍夫长度，$D_x$ 为物种扩散系数，$D_\theta$ 为扩散率，$D_c$ 为扩散系数，$S_c$ Schmidt 数，$P_r$ 为 Prandtl 数，$W_{st}$ 为气孔阻塞率，$R_G$ 为太阳辐射，m 为气孔导度斜率，$A_n$ 为光合作用速率，$g_0$ 为最小气孔导度，$C_s$ 为叶片表面 $CO_2$ 浓度，$h_s$ 为叶片表面相对湿度，$T_s$ 地表温度，RH 为相对湿度，LAI 为叶面积指数，$f(T)$ 温度函数，$f(D)$ 水汽压差函数，$f(\Psi)$ 为叶片水势函数。

**3.3 典型下垫面臭氧干沉降速率模拟特征分析**

**3.3.1 臭氧干沉降速率日变化特征**

观测和模拟的 $V_d(O_3)$ 昼夜变化比较如图3.1所示。结果表明,在鼎湖山站,冠层外变化范围在 $0.7\sim5.2$ cm·s$^{-1}$,平均值为 $0.34$ cm·s$^{-1}$;冠层内变化范围在 $-1.9\sim3.8$ cm·s$^{-1}$,平均值为 $0.1$ cm·s$^{-1}$。与观测值相比,冠层外四种方案模拟的日变化在 $0.05$ cm·s$^{-1}\sim0.9$ cm·s$^{-1}$,冠层内四种方案模拟的日变化在 $0.004$ cm·s$^{-1}\sim0.4$ cm·s$^{-1}$。永丰站观测值范围为 $0.008\sim2.0$ cm·s$^{-1}$,平均值为 $0.32$ cm·s$^{-1}$,模拟值范围为 $0.005\sim1.1$ cm·s$^{-1}$。

值得注意的是,在鼎湖山站的冠层内,日出前和日落后出现了比较明显的的反梯度输送现象。然而,没有一种方案可以模拟出这个观测结果,这导致了模拟中在日出前和日落后的 $V_d(O_3)$ 被高估。相比之下,夜间冠层外的模拟的结果较好,而白天的模拟结果有较大的差异。这可能是由于方案中冠层内外所使用的植被形态参数都是一致的,这并不能反映真实的森林生态系统结构。Wu 等[70]认为冠层的结构在模拟结果中起着重要的作用,这可能进一步解释了这种差距。

如图3.1所示,在鼎湖山站,气孔导度机制主导了 $V_d(O_3)$ 的日变化趋势,基于 Jarvis 气孔导度机制的方案 $V_d(O_3)$ 呈现双峰型变化趋势,在 8:00-9:00 和 16:00-17:00 左右达到峰值;基于 Ball-Berry 气孔导度机制的方案的 $V_d(O_3)$ 中午趋于稳定,16:00 以后逐渐下降。在永丰站中,观测的 $V_d(O_3)$ 在 9:00 达到峰值 ($0.68$ cm·s$^{-1}$),随后快速下降直到 18:00。而在此时段,模拟的 $V_d(O_3)$ 变化较为平缓,在 17:00 左右迅速下降。四种方案的模拟结果虽然有一定的差异,但日变化情况非常相似,两种气孔导度机制没有呈现明显的不同。

在鼎湖山和永丰站中,白天 Z03 和 Z03&BB 方案对 $V_d(O_3)$ 的模拟差异分别达到 7.8% 和 31.9%,W89 和 W89&BB 方案模拟差异分别为 18.6% 和 56.2%。所有方案均能模拟日出后 $V_d(O_3)$ 的快速上升,但无法模拟出峰值后 $V_d(O_3)$ 的快速下降。这可能是因为 $O_3$ 浓度快速上升导致 $O_3$ 消解效率降低[299]。而在现有的干沉降方案中,并没有考虑到 $V_d(O_3)$ 与 $O_3$ 浓度的相互影响,因此导致了 $V_d(O_3)$ 在此时段模拟的误差。夜间,Z03 型方案的模拟 $V_d(O_3)$ 比 W89 型方案高 2 倍以上,在永丰站表现尤为明显。这是由于夜间气孔关闭后非气孔阻抗占主导地位,这一差异反映了两种方案非气孔机制的差异[145]。

在鼎湖山站中,冠层外四个方案的模拟结果比观测值低 $18\sim70\%$,冠层内模拟结果与观测值的偏差在 $-18\sim64\%$ 之间。对于永丰站,与观测值相比,所有方案产生的偏差在 $-56\sim42\%$ 之间。总体来看,在永丰站和鼎湖山站的的冠层外,W89&BB 方案的模拟效果都更好。当只考虑白天的模拟效果时,鼎湖山站冠层内模拟效果最好的也是 W89&BB 方案。所有方案

与观测值之间的相关性均在 0.38 至 0.56 之间，其中 W89&BB 方案在两个生态系统中相关性都最好的。

[Figure]

| (a) 鼎湖山站，冠层外 | (b) 鼎湖山站，冠层内 | (c) 永丰站 |

图 3.1: 观测和模拟的平均 $V_d(O_3)$ 昼夜变化过程比较

表 3.4: 不同干沉降方案下 $V_d(O_3)$ 观测和模拟的统计特征量

| | 鼎湖山站 (冠层外) | | | | 鼎湖山站 (冠层内) | | | | 永丰站 | | | |
|---|---|---|---|---|---|---|---|---|---|---|---|---|
| | Mean | Median | Bias | R | Mean | Median | Bias | R | Mean | Median | Bias | R |
| 观测值 | 0.34 | 0.14 | - | - | 0.10 | -0.004 | - | - | 0.33 | 0.28 | - | - |
| Z03 | 0.10 | 0.06 | -0.25 | 0.50 | 0.08 | 0.04 | -0.02 | 0.46 | 0.14 | 0.12 | -0.19 | 0.41 |
| Z03&BB | 0.11 | 0.10 | -0.24 | 0.40 | 0.07 | 0.06 | -0.02 | 0.35 | 0.21 | 0.20 | -0.11 | 0.45 |
| W89 | 0.12 | 0.08 | -0.23 | 0.41 | 0.12 | 0.08 | 0.02 | 0.38 | 0.47 | 0.40 | 0.14 | 0.43 |
| W89&BB | 0.15 | 0.09 | -0.20 | 0.57 | 0.14 | 0.07 | 0.04 | 0.56 | 0.35 | 0.36 | 0.02 | 0.45 |
| W89&$rs$ | 0.29 | 0.09 | -0.06 | 0.47 | 0.15 | 0.08 | 0.06 | 0.40 | 0.37 | 0.36 | 0.04 | 0.44 |
| W89&BN | 0.13 | 0.09 | -0.22 | 0.56 | 0.13 | 0.08 | 0.03 | 0.54 | 0.36 | 0.37 | 0.03 | 0.45 |

其中，W89&$rs$ 与 W89 相同，只是改变了 $R_{s,min}$ 的值；W89&BN 是在 W89&BB 的基础上采用了改进的气孔阻抗机制和光合作用过程中的氮限制方案。

**3.4 典型下垫面影响臭氧干沉降速率主控因子识别**

为了进一步探究不同方案模拟 $V_d(O_3)$ 差异的原因，本研究构建的干沉降速率的过程诊断模块，对 $V_d(O_3)$ 的日变化经行了过程分解 (图3.2，(a)-(d) 为鼎湖山站，冠层外；(e)-(h) 为鼎湖山站，冠层内；(i)-(l) 为永丰站)。Saylor 等[300]的研究表明 $R_b$ 对 $V_d$ 没有显著的影响。因此，本研究只考虑 $R_a$、$R_c$ 和残差对 $V_d(O_3)$ 的影响。具体的公式如式3.1所示。

$$\Delta V_d = V_{d(t1)} - V_{d(t0)} = \frac{1}{R_{a(t1)} + R_{b(t1)} + R_{c(t1)}} - \frac{1}{R_{a(t0)} + R_{b(t0)} + R_{c(t0)}} \tag{3.1}$$

$$= \frac{1}{R_{a(t1)} + R_{b(t0)} + R_{c(t0)}} - \frac{1}{R_{a(t0)} + R_{b(t0)} + R_{c(t0)}} \tag{3.2}$$

$$+ \frac{1}{R_{a(t0)} + R_{b(t0)} + R_{c(t1)}} - \frac{1}{R_{a(t0)} + R_{b(t0)} + R_{c(t0)}} \tag{3.3}$$

$$+ \frac{\Delta R_a \cdot \Delta R_c{}^2 + \Delta R_a{}^2 \cdot \Delta R_c + 2 \cdot (R_{a(t0)} + R_{c(t0)}) \cdot \Delta R_a \cdot \Delta R_c + 2 \cdot (R_{b(t0)}) \cdot \Delta R_a \cdot \Delta R_c}{(R_{a(t1)} + R_{b(t1)} + R_{c(t1)}) \cdot (R_{a(t0)} + R_{b(t0)} + R_{c(t0)}) \cdot (R_{a(t0)} + R_{b(t1)} + R_{c(t1)}) \cdot (R_{a(t1)} + R_{b(t1)} + R_{c(t0)})}$$
$$\tag{3.4}$$

其中，t1 为下一个时刻，t0 为上一个时刻，式(3.2)为 $R_a$ 变化产生的贡献，式(3.3)为 $R_c$ 变化产生的贡献，式(3.4)为残差产生的贡献。

在鼎湖山站，冠层内外四种方案各阻抗贡献的变化趋势基本一致。$R_c$ 是鼎湖山站中 $V_d(O_3)$ 变化的主要因素。具体来说就是气孔导度机制决定了 $R_c$ 对 $V_d(O_3)$ 是正贡献还是负贡献。使用 Jarvis 气孔导度机制的方案，在日出和日落时刻呈现明显的正贡献，而在其余的时刻为负贡献，在第 8-9 和第 17-18 时刻之前 $R_c$ 变化范围最大。Jarvis 气孔导度机制受到太阳和辐射两个气象因素的影响，在这两个时刻气象因素变化较大，因此造成了 $R_c$ 的变化。而 Ball-Berry 气孔导度机制的方案，$R_c$ 在中午之后都为负贡献。此外，日出或日落前后大气温度差异增大 (图3.4)，导致湍流强度增加，使得 $R_a$ 和残差的贡献变多。残差几乎为正贡献，$R_a$ 夜间为负贡献，白天为正贡献。在永丰站，Z03 型方案的各阻抗变化趋势与鼎湖山站基本一致，但是 W89 型方案中，除了午间由 $R_c$ 占主导外，其余时刻基本由 $R_a$ 占主导。如表3.5所示，在两个陆地生态系统上，Z03&BB 方案中 $R_c$ 对 $V_d(O_3)$ 的贡献都是最高的，W89&BB 方案中 $R_c$ 对 $V_d(O_3)$ 的贡献都是最低的。不同方案中各阻抗对干沉降速率的贡献的差异，主要是由于陆地生态系统的不同造成的。

(a) Z03   (b) Z03&BB   (c) W89   (d) W89&BB

(e) Z03   (f) Z03&BB   (g) W89   (h) W89&BB

(i) Z03   (j) Z03&BB   (k) W89   (l) W89&BB

图 3.2: $V_d(O_3)$ 日内变化过程分解

大气阻抗 ($V_{d,max}=1/(R_a+R_b)$) 反映了臭氧干沉降在大气输送过程中的总阻抗。图3.3显示了 Z03 和 W89 型的 $V_{d,max}(O_3)$ 的平均日变化趋势。W89 和 Z03 这两种类型方案采用了 Monin-Obukhov 相似理论，因此他们的日变化趋势都是相似的。大气阻抗与大气湍流状况密切相关，白天地表热对流强烈，且风速较大 (图3.4)，因此冠层上方的阻抗大于冠层下方的阻抗。另一方面，由于大气层结区域稳定，夜间阻抗显著减小。在较低的冠层 (图3.3(a)和 (c))，W89 型的 $V_{d,max}$ 较高，而在较高的冠层 Z03 型模拟出的 $V_{d,max}$ 较高的。在永丰站，两种类型方案的差异约为 11.6%，而对于鼎湖山站，两种方案的差异分别为 5% (冠层外) 和 10.3% (冠层内) 。

在 Z03 型方案中，$V_{d,max}$ 对总阻抗的贡献约为 10.0% (鼎湖山站) 和 12.3% (永丰站) 。它对模式模拟的 $V_d(O_3)$ 没有显著的影响。本研究结果与 Wu 等[96] 的结果相似，他的研究结果表明 $V_{d,max}$ 对总阻抗的贡献约为 15%。然而，在 W89 型方案中，$V_{d,max}$ 对总阻抗的贡献变多，在鼎湖山站约为 22.0%，永丰站中约为 42.0%。值得注意的是，对于一些快速沉降的化学物质，如 $HNO_3$，其 $R_c$ 值接近于 0[96]，那么两种方案之间的 $V_{d,max}$ 的差异就会变得重要很多。

表 3.5: 干沉降速率变化贡献的统计结果

| | 鼎湖山 (冠层外) | | | 鼎湖山 (冠层内) | | | 永丰站 | | |
|---|---|---|---|---|---|---|---|---|---|
| | $\Delta R_a$ | $\Delta$Residual | $\Delta R_c$ | $\Delta R_a$ | $\Delta$Residual | $\Delta R_c$ | $\Delta R_a$ | $\Delta$Residual | $\Delta R_c$ |
| Z03 | 13.30 (13.63,13.07) | 22.53 (16.45,26.86) | 64.17 (69.91,60.07) | 8.21 (10.94,6.27) | 7.00 (9.27,5.38) | 84.79 (79.79,88.36) | 24.43 (38.25,14.63) | 16.89 (13.27,19.48) | 58.68 (48.58,65.89) |
| Z03&BB | 14.61 (16.46,13.29) | 20.73 (17.46,23.07) | 64.66 (66.08,63.64) | 15.41 (26.75,7.31) | 9.05 (7.02,10.51) | 75.54 (66.23,82.19) | 15.87 (19.33,13.40) | 15.83 (13.96,17.16) | 68.30 (66.70,69.44) |
| W89 | 43.26 (11.11,66.23) | 5.84 (8.35,4.05) | 50.90 (80.55,29.72) | 39.34 (15.54,56.34) | 5.37 (6.13,4.83) | 55.29 (78.33,38.83) | 68.15 (46.35,83.72) | 6.51 (8.37,5.18) | 25.34 (45.29,11.09) |
| W89&BB | 55.86 (21.17,80.63) | 4.69 (8.66,1.85) | 39.45 (70.16,17.52) | 53.71 (22.63,78.88) | 6.28 (10.21,3.02) | 40.02 (67.16,18.10) | 78.84 (61.98,92.40) | 2.56 (2.48,2.44) | 18.60 (35.54,5.16) |

括号中右边的值是白天 (9:00-16:00) 的平均值，左边的值是夜间 (17:00-8:00) 的平均值。

(a) 鼎湖山站，冠层外     (b) 鼎湖山站，冠层内     (c) 永丰站

图 3.3: 大气阻抗平均昼夜变化比较

[Figure]

图 3.4: 鼎湖山站气象剖面图

  由于 $R_c$ 主要是影响 $V_d(O_3)$ 的主控因子，因此本研究又进一步探究了 $V_d(O_3)$、冠层导度 ($g_c=1/R_c$)、气孔导度 ($g_s=1/R_s$) 和非气孔导度 ($g_{ns}$) 之间的关系。在鼎湖山站，不同方案在冠层内外的表现是一致的，但是幅度略有差异。冠层内和冠层外的平均气孔对 $R_c$ 的贡献率分别为 42~73% 和 32~61%。Ball-Berry 气孔导度机制计算的气孔对 $R_c$ 的贡献率都大于 50%。影响 $O_3$ 气孔导度贡献率的因素很多，不同森林生态系统中气孔导度对 $R_c$ 的贡献率也不同。温带混交林夏季气孔对 $R_c$ 的贡献率为 28~50%[187,301]，落叶林大约为 55%[301]。在永丰站，Lin 等[64]研究指出，农田下垫面气孔对 $R_c$ 的贡献率高达 40%。Clifton 等[34]梳理了以往文献中由观测测量的气孔对 $R_c$ 的贡献率。结果表明，气孔贡献率大约为 45%。本研究结果表明，永丰站平均气孔对 $R_c$ 的贡献率为 30~72%。与鼎湖山站不同，Jarvis 气孔导度机制估算出的气孔对 $R_c$ 的贡献率更高 (>55%)。

(a) 鼎湖山站，冠层外     (b) 鼎湖山站，冠层内     (c) 永丰站

图 3.5: 冠层导度、气孔导度、非气孔导度以及气孔对冠层导度的贡献率

  图3.6为四种方案模拟的气孔导度的平均日变化趋势 ($g_s$=1/$R_s$)。由图可以看出，两个站点 $g_s$ 的平均日变化趋势与 $V_d(O_3)$ 的平均日变化趋势相似。这表明 $g_s$ 是可能短期内影响 $V_d(O_3)$ 的关键决定因素。在鼎湖山站，四种方案计算的平均 $g_s$ 在 $0.10\sim0.27$ cm·s$^{-1}$ 之间，冠层内外的幅度基本一致；在永丰站中，$g_s$ 在区间为 $0.15\sim0.80$ cm·s$^{-1}$ 之间。此外，Jarvis 气孔导度型机制在鼎湖山站和永丰站表现出两种不同的日变化趋势。在鼎湖山站中，Jarvis 气孔导度机制估算的 $g_s$ 呈双峰型变化，在 8:00 和 17:00 到达峰值。Jarvis 机制与温度高度相关，因此在中午温度升高，气孔关闭，$g_s$ 显著下降。而 Ball-Berry 机制估计的 $g_s$ 呈单峰趋势，大约在 11:00 到达峰值。Ball-Berry 气孔导度机制估算的 $g_s$ 大约是 Jarvis 机制的两倍。在永丰站中，$g_s$ 在 $0.15\sim0.70$ cm·s$^{-1}$ 之间，两种机制的估算的 $g_s$ 日变化趋势基本一致。Jarvis 机制没有出现明显的双峰型变化，可能与局地温度不高有关。Jarvis 机制估算的 $g_s$ 约为 Ball-Berry 机制的 3 倍。为了比较四种方案对 $g_s$ 的模拟效果，我们使用彭曼公式[302]对永丰站的 $g_s$ 进行估算。P-M 法估算的 $g_s$ 平均值为 0.23 cm·s$^{-1}$。四种方案基本上可以捕捉到 $g_s$ 的日变化趋势。然而，Jarvis 机制与彭曼公式估算的相差 2 倍以上。Z03&BB 模拟效果最好，与彭曼公式的偏差为 0.04 cm·s$^{-1}$。

(a) 鼎湖山站，冠层外     (b) 鼎湖山站，冠层内     (c) 永丰站

图 3.6: 气孔导度的平均昼夜变化比较

  Jarvis 气孔导度机制是早期且较为完整的计算冠层气孔导度的机制之一。但是 Jarvis 机制中大多数的参数没有生物学上的意义，缺乏对气象和对植被生理的考虑，以及它的线性

关系依赖于最小气孔导度，忽略了各因子之间的协同或相互作用 ($R_{s,min}$)[70]。此外，Jarvis 机制中的参数主要来自野外测量的经验拟合，具有很大的不确定性。在 Noah-MP 中，$R_{s,min}$ 值通常取决于下垫面类型，而下垫面类型的分类是根据特定植物的叶水平测量得出的。本研究中，初始 $R_{s,min}$ 值森林下垫面为 125 s·m$^{-1}$，农田下垫面为 40 s·m$^{-1}$。为了探究 $R_{s,min}$ 对模式 $V_d(O_3)$ 的影响，我们采用 W89 方案，通过调整 $R_{s,min}$ 值 (以下简称 W89&rs)，进行了一系列敏感性实验。

如图3.8所示，在鼎湖山站中，将冠层内 $R_{s,min}$ 的值调整为 70 s·m$^{-1}$，冠层外调整为 25 s·m$^{-1}$ 后，与观测结果更为匹配，使冠层内和冠层外日间的 $V_d(O_3)$ 的平均偏差分别减少了 51.4% 和 27.1%。在永丰站中，由于对 W89 方案估算的 $V_d(O_3)$ 明显高估，当 $R_{s,min}$ 由 40 s·m$^{-1}$ 改变为 80 s·m$^{-1}$ 时，W89&rs 可以与观测值相匹配，使日间模拟的 $V_d(O_3)$ 的精度提高 52.8%。Wu 等[96]的研究也得出了相似的结论。他们发现当 $R_{s,min}$ 降低 25% 时，C5DRY (CMAQ 模式的干沉降模块) 模拟的夏季 $V_d(SO_2)$ 和 $V_d(O_3)$ 分别增加 14% 和 16%。

与 Jarvis 气孔导度机制相比，Ball-Berry 的气孔导度机制[210]是一种更为精确的估算冠层气孔导度的机制，它根据冠层的光合速率、$CO_2$ 浓度和叶面相对湿度计算气孔导度。但是，Noah-MP 中将 Ball-Berry 机制的气孔导度斜率 (m) 设为常数，这从机理上并不合适，会造成较大的模拟偏差。此外，Noah-MP 中 DVEG (Dynamic vegetation Model) 机制的原有方案将叶片含氮量因子参数 FOLNMX 设为常数 1.5 (使得叶片含氮量因子 $f(N)$ 为 0.67)，这是为了方便在区域上进行陆面模拟而做的简化。Chang 等[294]通过结合改进的气孔阻抗机制[303]和氮限制光合过程方案[304]，使模拟的 $V_d(NO_2)$ 与观测的平均偏差降低了 50.1%。因此，我们使用这种方法在 W89&BB 的基础上对 Ball-Berry 机制进行改进 (以下简称 W89&BN) 来探讨改进这两种机制对 $V_d(O_3)$ 的影响 (具体公式见 Table3.6)。

表 3.6: 改进的气孔导度机制和光合作用机制方程

| | 默认 | 改进 | 参考文献 |
|---|---|---|---|
| 气孔导度 | $g_s = g_0 + m\dfrac{A_n}{C_s}h_s$ | $g_s = g_0 + a \cdot A_n \dfrac{RH}{C_s - \Gamma^*}$ | Leuning[303] |
| 叶片氮 | $V_{cmax} = V_{cmax25} \cdot f(T_v) \cdot f(W_s) \cdot f(N)$ | $V_{cmax} = V_{cmax25} \cdot f(T_v) \cdot f(W_s) \cdot f(N)$ | Lin 等[304] |
| 限制因子 | $f(N) \in [0,1]$ | $f(N) = \dfrac{f(GPP) \cdot k}{1 - f(GPP)}$ | |

由图3.8可以看出，W89&BN 方案模拟结果并没有表现出明显的改进，与 W89&BB 方案相比，在鼎湖山冠层内和冠层外，$V_d(O_3)$ 白天的平均偏差增加了 12.6% 和 6.9%。尽管如此，W89&BN 方案的模拟结果仍然比 Z03 和 W89 方案的模拟结果有所改善。在永丰站，W89&rs 方案计算的 $g_s$ 比 W89&BB 方案提高了 7.1%，精度提高了 4.3% (图3.7)。随后 $V_d(O_3)$ 的改善与 W89&BB 方案相差不大，但优于 Z03 和 W89 方案，与 DHS 方案相同。与 Chang

等[294]的结果不同，该方案对 $V_d(O_3)$ 模拟效果的提升不如对 $V_d(NO_2)$ 模拟效果提升的显著。这可能是由于该方案的改进与 $NO_2$ 直接相关，而对 $O_3$ 的影响则还涉及其他的物理和生物过程，更为复杂。

(a) 永丰站

图 3.7: 改进后 $g_s$ 平均昼夜变化比较

(a) 鼎湖山站，冠层外     (b) 鼎湖山站，冠层内     (c) 永丰站

图 3.8: 改进后 $V_d(O_3)$ 平均昼夜变化比较

**3.5 本章小结**

本章中，我们基于观测试验分析了不同下垫面 $V_d(O_3)$ 的日变化特征，随后利用观测得到气象数据驱动单点干沉降机理诊断模式 Noah-MP-WDDM，评估不同干沉降方案对森林和农田下垫面上 $V_d(O_3)$ 的模拟效果以及不同气孔机制对 $V_d(O_3)$ 的影响，并进一步分析了影响典型下垫面 $V_d(O_3)$ 的关键过程。

在观测方面，鼎湖山站冠层外和冠层内的白天 $V_d(O_3)$ 分别为 $0.75\,\mathrm{cm\cdot s^{-1}}$ 和 $0.30\,\mathrm{cm\cdot s^{-1}}$，永丰站的白天 $V_d(O_3)$ 为 $0.45\,\mathrm{cm\cdot s^{-1}}$。与其他文献报道相比，鼎湖山站的白天 $V_d(O_3)$ 比其

他混合林略低。这是可能由于不同森林下垫面的森林结构和局地 NOx 和 $O_3$ 反应不同造成的。而在永丰站，观测期间所种植的作物为冬小麦，不同作物的物候和胁迫响应以及光合途径存在潜在差异，因此 $V_d(O_3)$ 大小存在差异。冬小麦下垫面的 $V_d(O_3)$ 与其他农作物相比，处于中等水平。目前对臭氧干沉降过程的观测多偏向于生长季，且由于我国臭氧干沉降的观测开展较晚，大多都是进行的短期观测，这会限制验证对干沉降参数化方案改进的有效性，因此我们需要对不同的下垫面，不同的作物进行长期的观测试验，为干沉降参数化方案提供更多的依据及理论。

在模式模拟方面，本章对四种干沉降方案模拟的 $V_d(O_3)$ 进行了评估。四种方案在永丰站的模拟效果相似，相关系数在 0.41~0.45；在鼎湖山站，Ball-Berry 气孔导度机制的模拟效果要优于 Jarvis 气孔导度机制，$V_d(O_3)$ 与观测值的相关系数在 0.38~0.56。此外，在鼎湖山站，相较于冠层外的模拟结果，四种方案在冠层内的模拟效果均更好。这可能是由于模式中森林结构参数无法体现实际森林下垫面冠层内外的差异导致。总体而言，W89&BB 方案相较于其他三个方案更适用于农田和森林下垫面。

为了进一步识别典型下垫面影响 $V_d(O_3)$ 的主控因子，首先分析了不同方案在森林和农田下垫面对 $V_{d,max}$ 的模拟。结果表明，在较低冠层，W89 型模拟的 $V_{d,max}$ 较高，而在较高冠层 Z03 型模拟出的 $V_{d,max}$ 较高。在 Z03 型方案中，$V_{d,max}$ 对总阻抗的贡献约为 10.0% (鼎湖山站) 和 12.3% (永丰站)。在 W89 型方案中，$V_{d,max}$ 对总阻抗的贡献变多，鼎湖山站约为 22.0%，永丰站中约为 42.0%。值得注意的是，对于一些快速沉降的化学物质，如 $HNO_3$，其 $R_c$ 值接近于 0[96]，那么不同方案之间 $V_{d,max}$ 的差异就会变得更为重要。

在 W89 型方案中，$R_c$ 是引起鼎湖山站白天 $V_d(O_3)$ 变化的主导因素，夜间是由 $R_a$ 引起的湍流混合主导；在永丰站，$R_c$ 仅在中午时段占主导地位，其余时段 $R_a$ 的影响更大。在 Z03 型方案中，鼎湖山站和永丰站 $V_d(O_3)$ 变化的都受到 $R_c$ 的影响。进一步分析 $V_d(O_3)$、$R_c$、$g_s$ 和 $g_{ns}$ 之间的关系，研究发现这两个站点 $g_s$ 的平均日变化趋势与 $V_d(O_3)$ 的平均日变化趋势一致。这表明 $g_s$ 可能是短期内影响 $V_d(O_3)$ 的关键因素。在鼎湖山站中，Jarvis 机制估算的 $g_s$ 呈双峰型变化，而 Ball-Berry 机制呈单峰趋势。冠层内和冠层外的平均气孔率分别为 42~73% 和 32~61%，Ball-Berry 机制计算的气孔率均大于 50%。在永丰站中，两种机制的估算的 $g_s$ 日变化趋势基本一致。Jarvis 机制估算出更高的气孔率，平均气孔率为 30~72%。

对 Jarvis 气孔导度机制，其不确定性很大程度上来自于 $R_{s,min}$ 的取值。在鼎湖山站，$R_{s,min}$ 在冠层外和冠层内设置的值相同。但是，$R_{s,min}$ 通常是对树冠顶部受阳光照射的叶片经行测量，而冠层内的叶子通常与树冠顶部的叶子形状和角度不同。在鼎湖山站，Jarvis 型的方案在冠层内模拟效果更好，这可能是因为模式默认的 $R_{s,min}$ 值更适用于冠层内的树冠结构。本研究在修改 $R_{s,min}$ 值后，鼎湖山站白天平均偏差降低了 39.3%，永丰站白天平均

偏差提高了 60.9%。对于 Ball-Berry 气孔导度机制，与模式参数的选择相比 $g_s$ 直接受物种特异响应参数变化的影响更大。本研究在默认的 Ball-Berry 方案上改进了气孔导度机制和光合作用机制方程。结果表明，改进后的方案相较于原始方案模拟的 $V_d(O_3)$ 在两个站点均有所改善。

**第四章  WRF-Chem 数值模式改进及评估**

对流层 $O_3$ 目前是最受关注的空气污染物之一。研究表明，对流层 $O_3$ 不仅损害人类健康，还会损害植被和作物生长。植被也可以通过影响 $O_3$ 源汇来调节地表 $O_3$ 浓度。通过气孔吸收，$O_3$ 在植被上的干沉降是 $O_3$ 重要的汇。当植被暴露在高浓度的 $O_3$ 水平下时，会导致光合作用速率下降，气孔导度降低，从而减少 $O_3$ 的干沉降的汇[25,33]。此外，$O_3$ 胁迫会导致气孔对环境条件的变化反应更加缓慢，对气孔的开闭和干沉降产生复杂的影响。植被也影响 $O_3$ 的源，植被释放出的异戊二烯在受污染的高 NOx 地区，是 $O_3$ 形成的主要前体。异戊二烯的产生与光合作用密切相关，会受到气孔导度的影响。因此，$O_3$ 与植被的相互作用可以通过影响植物生理过程对气象产生影响，并通过一系列反馈机制进一步影响 $O_3$ 浓度本身。然而，这种 $O_3$ 引起的植被破坏以及对 $O_3$ 浓度本身的影响，并未在当前区域大气化学传输模式中形成反馈。

本章基于 WRF-Chem 数值模式，将 $O_3$ 胁迫导致的植被破坏的半经验方案[23]耦合进模式中，进一步完善臭氧干沉降机制的改进，研究 $O_3$ 与植被耦合引起的区域植被生理过程、气象以及对 $O_3$ 干沉降过程的影响。

**4.1 数值模式的干沉降机制的参数化方案改进**

为了更好地对气象、化学过程以及大气干沉降过程进行模拟，本研究在公开发布的 WRF-Chem v3.9.1 版本的基础上进行了一些模式的改进与调整，主要包括以下几方面：

1. WRF-Chem v3.9.1 目前仅支持在 MOZART 机制下干沉降速率的输出且对计算干沉降速率的各阻抗并不进行输出，因此本研究增加了在 SAPRC99/MOSAIC 机制下气体和气溶胶粒子的干沉降速率 ($V_d$)，空气动力学阻抗 ($R_a$)，粘性副层阻抗 ($R_b$) 以及冠层阻抗 ($R_c$) 输出的程序接口；

2. 在气孔阻抗方面，由第三章对干沉降方案的评估可知，基于 Ball-Berry 气孔导度方案计算干沉降速率在典型亚热带森林和农田生态系统中的模拟效果均要优于 Jarvis 气孔导度方案计算的结果。然而，在 WRF-Chem 中，陆面过程与干沉降过程分别属于物理、化学两个不同的模块。Noah‑MP 作为物理模块中的一种陆面过程参数化方案选择 (module_sf_noahmplsm.F)，在其中使用 Ball-Berry 方案对气孔导度进行计算，虽然会对气象场产生影响，但其与化学模块是整体不相接的，化学模块中的干沉降模块 (module_dep_simple.F)

对气孔导度的计算依然使用的是 Jarivs 方案。陆面过程参数化选择的改变，只会导致气象场的变化从而影响 Jarvis 方案中计算所需的辐射和温度，进而对干沉降速率的模拟产生影响。但本质上在干沉降模块中并没有使用 Ball-Berry 方案对气孔进行计算。因此在本研究中，建立了 Ball-Berry 气孔导度方案由物理模块中陆面模式 Noah-MP 输出进化学模块的程序接口，使得 Wesely 干沉降机制可灵活调用 Ball-Berry 方案进行气孔导度计算，并使用 LAI 来对向阳面和遮荫面的叶片在冠层尺度上进行缩放。此外，在无植被覆盖的下垫面，气孔导度方案任然使用的是 Jarvis 气孔导度方案。

此外，研究表明当植被受到 $O_3$ 胁迫时，气孔导度会降低，Feng 等[305]通过 meta 分析结果显示 $O_3$ 会引起小麦的气孔导度降低约 31%。同时，植被暴露于高浓度的 $O_3$ 中时，细胞和组织损伤会导致光合速率下降[25]。因此，本研究将 WRF-Chem 化学模块模拟的每一个时刻的 $O_3$ 浓度，动态传递到 Noah-MP 中，修正因为 $O_3$ 胁迫引起的光合速率和气孔导度变化。Noah-MP 模拟的陆面变量也被动态的传递回大气，从而允许气象场、$O_3$ 和其他大气化学成分产生即时的双向反馈效应。这样，在 WRF-Chem 模式中的陆面过程、大气动力学和大气化学就完全耦合起来了。具体模式耦合图，如图4.1所示。

[Figure]

图 4.1: 改进机制在 WRF-Chem 中的耦合

具体改进为，在 Noah-MP 中，光合速率 A 的计算采用了 Farquhar 的方案[284]，此方案考虑了植被光合作用的生物、化学过程对于环境变化的响应，并且比较详细的考虑了 C3 和 C4 植物的气孔导度对光合速率的响应过程，并且对光照和遮荫面叶片分别进行解算，其叶片的光合速率受到三个因子之一的限制，即：

$$A = \min(W_c, W_j, W_e)I_{gs} \tag{4.1}$$

其中，$W_c$ 为受到叶绿素的光合酶 (RuBisCO) 浓度所限制的羧化效率；$W_j$ 为受光通量

所限制的羧化效率；$W_e$ 为受有机磷限制的羧化效率；Igs 指生长季节指数，取值范围从 0 到 1。

冠层气孔导度的计算采用了 Ball-Berry 的方案[210]，此方案基于光合速率，叶片表面的 $CO_2$ 浓度和湿度计算[210,306]，即：

$$g_s = \frac{1}{R_s} = m \times \frac{A}{C_{air}} \times \frac{e_{air}}{e_{sat}(T_v)} \times P_{air} + b \tag{4.2}$$

其中，$g_s$ 为叶片气孔导度 $(m^2 \cdot s^{-1})$；$R_s$ 为叶片气孔阻抗 $(s \cdot m^1)$；m 为关气孔导度和光合速率的经验参数，取值范围为 5～9，模式中取值为 9；A 为光合速率 (由4.1计算)；$C_{air}$ 为叶片表面 $CO_2$ 分压 (Pa)；$e_{air}$ 为叶片表面水汽压 (Pa)；$e_{sat}(T_v)$ 为冠层温度下的叶片饱和水汽压 (Pa)；$P_{air}$ 为地表气压 (Pa)；b 为 A=0 时的最小气孔导度。

目前认为，基于臭氧累计吸收通量 (CUO，Cumulative uptake of ozone) 与 $O_3$ 植被响应关系更适合应用于全球或区域模式中[307]。因此在本研究中，我们使用 Lombardozzi 等[163]开发的参数化方案，将 CUO 对植物光合速率和叶片气孔导度影响的过程添加到 WRF-Chem v3.9.1 版本的 Noah-MP 中。根据 Reich[308]的理论，分别对植物的光合和叶片气孔导度进行调节，该理论已经在不同植被类型上得到了验证[309-310]。CUO $(mmol \cdot m^{-3})$ 的计算公式为：

$$CUO = 10^{-6} \sum \frac{[O_3]}{kO_3 R_s + R_b} \Delta t \tag{4.3}$$

其中，$[O_3]$ 为地表 $O_3$ 浓度 $(nmol \cdot m^{-3})$；$kO_3$=1.67 是叶片对 $O_3$ 的阻抗与叶片对水的水的比值[149]；$R_s$ 为气孔阻抗 $(s \cdot m^{-1})$；$R_b$ 为边界层阻抗 $(s \cdot m^{-1})$；$\Delta t$ 为模式模拟时间步长。在生长季，植被最容易受到空气污染的影响，因此将生长季节定义为 LAI 大于 0.4。且当 $O_3$ 通量高于阈值 $0.8\ nmolO_3 m^{-2} s^{-1}$ 时，CUO 才会累积，以考虑生长季植物对 $O_3$ 的解毒作用[23]。

$O_3$ 损伤因子是基于 CUO 计算的，计算公式如4.4所示。利用两组 $O_3$ 损伤因子 $F_{pO_3}$ 和 $F_{cO_3}$ 分别对光合速率和气孔导度进行单独修正，然后分别乘以 Farquhar 方案和 Ball-Berry 气孔导度方案计算出的 A 和 $g_s$，得到 $O_3$ 胁迫下的光合速率和气孔导度的变化。

$$F_{pO_3} = a_p \cdot CUO + b_p \tag{4.4}$$

$$F_{cO_3} = a_C \cdot CUO + b_c \tag{4.5}$$

其中，$F_{pO_3}$ 为 $O_3$ 对光合速率的影响因子；$F_{cO_3}$ 为 $O_3$ 对气孔导度的影响因子；$a_p$、$b_p$、$a_c$ 和 $b_c$ 是根据 Lombardozzi 等[23]研究得出的经验斜率和截距，这些参数依赖于植物的类型，具体数值如表4.1所示。

表 4.1: 公式中臭氧损伤因子的斜率

| | 光合速率 | | 气孔导度 | |
|---|---|---|---|---|
| | 斜率 ($a_p$) | 截距 ($b_p$) | 斜率 ($a_c$) | 截距 ($b_c$) |
| 阔叶林 | 0.0000 | 0.8752 | 0.0000 | 0.9125 |
| 针叶林 | 0.0000 | 0.8390 | 0.0048 | 0.7823 |
| 农田和草地 | -0.0009 | 0.8021 | 0.0000 | 0.7511 |

3. 在非气孔阻抗方面，模式默认计算潮湿状况下的 $R_{lu}$ 使用的并不是模式所介绍的 Wesely 方案，因此本研究首先将潮湿状况下的 $R_{lu}$ 修正为 Wesely 的方案。其次，观测实验显示，$R_{lu}$ 在湿润条件下阻抗降低，从而会加快臭氧的干沉降速率，而在默认的干沉降方案忽略了此现象，因此本研究考虑了 RH 的变化对 $R_{lu}$ 的影响，并使用 LAI 对其在冠层上尺度上进行缩放，具体改进公式如4.2所示。

表 4.2: 叶片角质层阻抗的修正与改进

| | 默认方案 | 本研究方案 |
|---|---|---|
| For dry condition | $R_{lu} = \dfrac{R_{luO}}{10^{-5} \times H \times f_0}$ | $R_{lu} = \dfrac{R_{luO}}{LAI \times exp(RH) \times (10^{-5} \times H + f_0)}$ |
| For dew condition | $R_{lu} = \dfrac{1}{\dfrac{1}{3000} + \dfrac{3}{R_{luO}1}}$ | $R_{lu} = \dfrac{1}{\dfrac{1}{3000} + \dfrac{1}{3R_{luO}1}}$ |
| For rain condition | $R_{lu} = \dfrac{1}{\dfrac{1}{1000} + \dfrac{3}{R_{luO}2}}$ | $R_{lu} = \dfrac{1}{\dfrac{1}{1000} + \dfrac{1}{3R_{luO}2}}$ |

**4.2 模拟方案设置**

**4.2.1 模式设置**

本研究选取 2019 年 1、4、7、10 四个月作为典型代表月份。图4.2给出了本研究的模拟区域，采用一重嵌套网格进行模拟，模拟区域为中国地区其范围为 65.46°E–140.53°E，15.14°N-55.65°N，其水平网格数为187×158，水平网格分辨率为 27×27 km，垂直层数设为 35 层，模式顶气压设置在 50 hPa。模式使用的物理方案如表4.3所示。

[Figure]

图 4.2: 模拟区域设置与模式评估的观测站点

表 4.3: 模式物理化学方案设置

| 过程 | 方案 |
| --- | --- |
| 云微物理方案 | Lin Scheme[270] |
| 长波辐射方案 | RRTM Longwave[275] |
| 短波辐射方案 | RRTMG Shortwave[275] |
| 近地面层方案 | Monin-Obukhov[273] |
| 陆面过程方案 | Noah-MP LSM[282] |
| 行星边界层方案 | YSU[266] |
| 积云参数化方案 | Kain-Fritsch[272] |
| 光解方案 | Fast-J[280] |
| 气象化学机制 | SAPRC99[311] |
| 气溶胶机制 | MOSAIC 4Bins[312] |

**4.2.2 模式输入资料**

1. 初始和边界条件

2017 年模式气象驱动数据来自 NCEP 再分析资料 FNL (http://dss.ucar.edu/datasets/ds083.2/)，水平分辨率为 1°×1°，时间间隔为 6 个小时，垂直分层为 26 层，气压高度从 1000hPa 到 10hPa。化学的初始和边界条件采用 NCAR 提供的 CAM-Chem 全球大气化学模拟场 (https://www.acom.ucar.edu/cam-chem/cam-chem.shtml)，其水平分辨率为 1.9°×2.5°，时间分辨率为 6 个小时，垂直方向上有 56 层[313]。

2. 人为源排放清单

人为源排放清单采用清华大学提供的 2017 年中国污染源排放清单，该清单由中国多尺度排放清单模型 (Multi-resolution Emission Inventory for China,MEIC) (http://meicmodel.org/) 构建，时间分辨率为月，水平分辨率为 27×27km，涵盖了 10 种主要的大气污染物和二氧化碳 ($SO_2$、NOx、CO、NMVOC、$NH_3$、$PM_{2.5}$、$PM_{10}$、BC、OC 和 $CO_2$)，其中 VOCs 物种分配机制包括 CBIV，CB05，SAPRC99，SAPRC07，RADM2 五种，便于在模式中针对不同的化学机制使用相应的物种分配机制。

2017 年中国地区人为源排放空间分布如图4.3所示，$SO_2$、NOx、CO、VOCs 和 $PM_{2.5}$ 排放主要集中在城市地区，如华北平原、四川盆地、长三角和珠三角地区等。而 $NH_3$ 主要集中在河北和四川盆地。各排放部门对污染物的排放贡献率如图4.4所示，$SO_2$ 主要来源于工业排放，占比率达到 57%，其次是民用源和电力源，两者一共占比约 39%；NOx 主要来源于工业源和交通源，分别占比 42% 和 35%，其次是电力源，占比 19%；CO 主要来源于民用源和工业源，分别占比 42% 和 36%，其次是交通源，占比 18%；农业源是 $NH_3$ 的绝对主要来源，占比高达 94%；工业源是 $PM_{2.5}$ 的主要来源，占比率约 46%，其次是民用源，约占 40%，电力源和交通源占比相当，都约为 7%；VOCs 主要来源于工业源，占比达到 65%，民用源和交通源占比相当，都约为 17%。人为源各月排放量变化如图4.5所示，除 $NH_3$ 外，其余的污染物均是在 1~2 月和 11~12 月排放量最大，在 4~9 月排放量低；而 $NH_3$ 排放趋势正好相反，在夏季 5~8 月排放量较高，而 1~3 月和 10~12 月排放量较低。

[Figure]

图 4.3: 中国地区人为源排放空间分布

[Figure]

图 4.4: 中国地区各部门人为源排放占比

[Figure]

图 4.5: 中国地区人为源各月排放量变化

**3. 土地利用分布**

本研究使用欧盟联合研究中心 (JRC，JOINT RESEARCH CENTRE) 发布的 2019 年土地利用资料 (GLC2019)，根据 USGS24 类分类体系，重新反演的土地利用资料驱动 WPS[314]。中国地区土地覆被类型分布如图4.6所示。

[Figure]

图 4.6: 中国地区土地覆被类型分布

**4.3 模式模拟性能评估**

本小节利用站点观测的数据对 WRF-Chem 模拟气象和污染物浓度进行验证。图4.2同时展示了本研究气象和污染物浓度的观测站点，模拟区域范围内气象观测站点 699 个，空气质量站点 1600 个。

**4.3.1 模式验证计算方法**

为了定量描述模式模拟结果的准确性，本研究主要采用以下统计指标量化验证模式的结果。平均偏差 (MB)、绝对误差 (MAE)、归一化平均偏差 (NMB)、均方根误差 (RMSE)、相关系数 (r) 和拟合指数 (IOA)，其中 MB 反映的是模拟值与实测值的平均偏离程度，MAE 反映了平均绝对误差，RMSE 反映了模拟值与实测值的偏离程度，NMB 反映了模拟值与实测值的平均偏离程度，以上四个都是有量纲的统计量，它们越接近 0，表明模拟的效果越好。同时，使用 r 和 IOA 来表征模拟值与观测值之间变化趋势的吻合程度，r 越接近 1，表明模拟效果越好；而 IOA 为无量纲统计量，一般可按三级划分：|IOA|<0.4 为低拟合度，0.4<|IOA|<0.7 为显著拟合，0.7<|IOA|<1.0 为高度拟合。这些指标的计算公式如下，其中 $M_i$ 与 $O_i$ 分别为模拟值和观测值，$\bar{M}$ 与 $\bar{O}$ 分别为模拟值和观测值的平均，n 为样本数，计算公式如下：

$$MB = \frac{1}{n}\sum_{i=1}^{n}(M_i - O_i) \tag{4.6}$$

$$MAE = \frac{1}{n}\sum_{i=1}^{n}|M_i - O_i| \tag{4.7}$$

$$NMB = \frac{\sum_{i=1}^{n}(M_i - O_i)}{\sum_{i=1}^{n}(O_i)} * 100\% \tag{4.8}$$

$$RMSE = \sqrt{\frac{1}{n}\sum_{i=1}^{n}(M_i - O_i)^2} \tag{4.9}$$

$$r = \frac{\sum_{i=1}^{n}(M_i - \bar{M})(O_i - \bar{O})}{\sqrt{\sum_{i=1}^{n}(M_i - \bar{M})^2 \sum_{i=1}^{n}(O_i - \bar{O})^2}} \tag{4.10}$$

$$IOA = 1 - [\frac{n \cdot RMSE^2}{\sum_{i=1}^{n}(|M_i| - |O_i|)^2}] \tag{4.11}$$

**4.3.2 气象要素评估**

本研究对 2019 年全国 699 个气象站点 1、4、7、10 四个月的日平均 2 米温度 (T2)、相对湿度 (RH)、10 米风速 (WS) 和大气压 (Press) 的模拟结果进行评估。气象数据来自中国

气象数据网 (http://data.cma.cn/) ，时间分辨率为日。表4.4和图4.7~4.10分别给出了 T2、RH、WS10 和 Press 各月观测和模拟的统计结果、日均观测与模拟时间序列和全国范围内各站点的模拟偏差分布图。

如表4.4所示，总体而言，模拟的 2 米温度在四个月低估约 0.5~1.6 °C，模拟和观测的温度相关性都达到了 0.8 以上。从各站点偏差分布可以看出，西部地区温度模拟普遍低估，南部沿海地区除了 7 月模拟偏高以外，其余三个月模拟偏低。相对湿度在 1 月和 10 月模拟偏差较低，分别为-1.2% 和 2.3%，模拟和观测的相对湿度相关系数大于 0.5；与 2 米温度不同的是，相对湿度在西部地区模拟呈现偏高的趋势。全年模拟风速高估 3 m·s$^{-1}$ 左右，绝大部分站点模拟的风速均高于观测值，这可能是由于模式对城市热动力过程描述不够准确，因此当前模式普遍高估风速[315]；模拟和观测的风速相关系数大约在 0.4 左右。模拟与观测的大气压相关性最好，均高于 0.9，偏差在-14.0~-5.0 hPa 之间，绝大部分站点模拟大气压表现为低估，西部地区大气压低估最为明显。总体而言，模式可以较好抓住观测的气象参数趋势，量级与观测吻合较好，模拟结果可靠。

表 4.4: 模拟和观测的日均气象要素的比较结果

| 时期 | 气象变量 | 平均值 | | MB | MAE | RMSE | r |
| | | 观测 | 模拟 | | | | |
|---|---|---|---|---|---|---|---|
| 1 月 | T2 (°C) | -0.5 | -1.0 | -0.5 | 3.1 | 3.6 | 0.8 |
| | RH (%) | 64.7 | 65.9 | -1.2 | 12.5 | 15.0 | 0.6 |
| | WS10 (m·s$^{-1}$) | 2.1 | 3.5 | 1.4 | 1.6 | 1.9 | 0.5 |
| | Press (hPa) | 936.8 | 922.8 | -14.0 | 19.5 | 19.7 | 0.9 |
| 4 月 | T2 (°C) | 14.7 | 13.7 | -1.0 | 2.6 | 3.1 | 0.9 |
| | RH (%) | 61.2 | 69.7 | 8.5 | 13.6 | 16.1 | 0.5 |
| | WS10 (m·s$^{-1}$) | 2.5 | 3.7 | 1.5 | 1.8 | 2.2 | 0.4 |
| | Press (hPa) | 926.9 | 913.9 | -13.0 | 18.5 | 18.7 | 0.9 |
| 7 月 | T2 (°C) | 24.2 | 23.6 | -0.6 | 2.6 | 2.9 | 0.8 |
| | RH (%) | 73.8 | 79.5 | 5.7 | 11.9 | 13.9 | 0.4 |
| | WS10 (m·s$^{-1}$) | 2.1 | 3.1 | 1.0 | 1.4 | 1.6 | 0.3 |
| | Press (hPa) | 920.8 | 905.8 | -5.0 | 19.1 | 19.2 | 0.9 |
| 10 月 | T2 (°C) | 13.7 | 12.1 | -1.6 | 2.6 | 3.0 | 0.9 |

续表4.4

| 时期 | 气象变量 | 平均值 | | MB | MAE | RMSE | r |
|---|---|---|---|---|---|---|---|
| | | 观测 | 模拟 | | | | |
| | RH (%) | 68.8 | 71.1 | 2.3 | 13.0 | 15.5 | 0.5 |
| | WS10 (m·s$^{-1}$) | 2.1 | 3.3 | 1.2 | 1.5 | 1.7 | 0.5 |
| | Press (hPa) | 933.1 | 919.5 | -13.6 | 18.6 | 18.8 | 0.9 |

[Figure]

图 4.7: 2019 年 1 月观测与模拟气象参数时间序列及各气象站点偏差分布

[Figure]

图 4.8: 2019 年 4 月观测与模拟气象参数时间序列及各气象站点偏差分布

[Figure]

图 4.9: 2019 年 7 月观测与模拟气象参数时间序列及各气象站点偏差分布

[Figure]

图 4.10: 2019 年 10 月观测与模拟气象参数时间序列及各气象站点偏差分布

**4.3.3 大气污染物浓度评估**

全国 2017 年 1、4、7、10 四个月的污染物质量浓度数据来自全国空气污染观测网络，本文对 $NO_2$，$O_3$ 和 $PM_{2.5}$ 进行评估。

图4.11展现了四个月污染物模拟和观测的平均昼夜变化趋势，可以看出，不同月份三种污染物的模拟昼夜变化趋势与观测都较为一致，其中 $NO_2$ 和 $PM_{2.5}$ 呈现白天时段模拟低估，夜晚时段模拟高估的现象。$O_3$ 除了在 1 月清晨到中午时段呈现明显高度现象，其余月份与观测都较为吻合，但是中午过后和夜间 $O_3$ 都明显低估。中国地区各空气质量站点的污染物模拟与观测月均浓度对比如图4.12所示，其中圆点表示各站点观测的污染物月均浓度。总体而言，模拟的污染物和站点观测浓度空间吻合较好。$NO_2$ 的高值中心出现在华北地区；$PM_{2.5}$ 的高值区同样也在华北地区，其中 1 月的华中和西南部分地区模拟存在高估现象。4月 $O_3$ 模拟的空间吻合度较差，大体上高值区模拟偏低，低值区模拟偏高，其余月份基本吻合。

表4.5展示了全国范围内不同月份污染物的统计结果。可以看出，全国平均而言，$NO_2$ 全年模拟呈现高估，平均高估在 6.3～9.5 $\mu g \cdot m^{-3}$，$PM_{2.5}$ 除 10 月高估 8.3 $\mu g \cdot m^{-3}$ 以外，其余月份低估 1.3～2.3 $\mu g \cdot m^{-3}$。而 $O_3$ 全年模拟呈现低估，平均低估在 6.4 ～20.5 $\mu g \cdot m^{-3}$。平均而言，模拟和观测的 IOA 均高于 0.8，为高度吻合。

总体而言，在量级和趋势上，WRF-Chem 都能较为合理地模拟出中国地区主要大气污染物浓度的变化特征，可用于后续中国地区污染物沉降过程的研究。

表 4.5: 模拟和观测的污染物浓度的比较结果

| 时期 | 污染物 | 平均值 | | MB | MAE | RMSE | IOA |
|---|---|---|---|---|---|---|---|
| | | 观测 | 模拟 | | | | |
| 1 月 | NO$_2$ | 41.3 | 52.7 | 11.4 | 30.1 | 39.2 | 0.8 |
| | O$_3$ | 34.6 | 31.5 | -3.1 | 22.8 | 29.2 | 0.9 |
| | PM$_{2.5}$ | 70.4 | 72.5 | 1.1 | 34.5 | 49.3 | 0.7 |
| 4 月 | NO$_2$ | 27.2 | 37.8 | 10.6 | 25.6 | 34.1 | 0.9 |
| | O$_3$ | 70.6 | 53.9 | -3.5 | 35.0 | 44.0 | 0.9 |
| | PM$_{2.5}$ | 34.7 | 37.6 | 2.9 | 22.6 | 31.1 | 0.8 |
| 7 月 | NO$_2$ | 19.7 | 26.5 | 6.7 | 18.6 | 24.4 | 0.9 |
| | O$_3$ | 78.1 | 65.2 | 12.9 | 35.8 | 44.6 | 0.9 |
| | PM$_{2.5}$ | 22.1 | 28.4 | 6.3 | 16.0 | 22.1 | 0.8 |
| 10 月 | NO$_2$ | 31.0 | 39.9 | 8.9 | 22.8 | 30.3 | 0.9 |
| | O$_3$ | 55.1 | 43.5 | -11.6 | 29.4 | 38.2 | 0.9 |
| | PM$_{2.5}$ | 34.2 | 44.6 | 10.4 | 27.4 | 38.0 | 0.8 |

[Figure]

图 4.11: 2019 年污染物观测与模拟昼夜变化时间序列

[Figure]

图 4.12: 2019 年污染物观测与模拟月均浓度

**4.4 臭氧干沉降模拟性能评估**

为了进一步验证干沉降机制改进后模式对臭氧干沉降速率的模拟效果，本研究对 WRF-Chem 模拟的 $O_3$、$V_d(O_3)$ 和臭氧干沉降通量 ($O_3$ddep) 与鼎湖山站 (10 月) 和永丰站 (4 月) 的观测数据进行了比较验证。由于网格框内的异质性，来自特定位置的 $V_d(O_3)$ 观测结果与 WRF-Chem 模式输出的模拟值 (27×27km 分辨率) 的比较可能存在一定的偏差。表4.6是两

种干沉降方案下观测站点臭氧干沉降的模式验证统计特征量。

表 4.6: 两种干沉降方案下的臭氧干沉降的模式验证统计特征量

| 变量 | 鼎湖山站 (冠层外) | | | | | | 永丰站 | | | | | |
| | 平均值 | | r | MB | RMSE | NMB | 平均值 | | r | MB | RMSE | NMB |
| | 观测 | 模拟 | | | | | 观测 | 模拟 | | | | |
| **原始方案** | | | | | | | | | | | | |
| O$_3$ | 57.0 | 50.6 | 0.5 | -6.5 | 29.0 | -11.3 | 71.3 | 36.9 | 0.6 | -34.4 | 51.8 | -48.3 |
| V$_d$(O$_3$) | 0.4 | 0.6 | 0.5 | 0.2 | 0.5 | 54.8 | 0.3 | 0.5 | 0.7 | 0.2 | 0.3 | 59.3 |
| O$_3$ddep | -4.2 | -6.7 | 0.6 | -2.8 | 5.2 | 67.1 | -6.1 | -6.6 | 0.5 | -0.5 | 7.6 | 8.0 |
| **改进方案** | | | | | | | | | | | | |
| O$_3$ | 57.0 | 58.4 | 0.6 | 1.4 | 29.0 | 2.5 | 71.3 | 46.4 | 0.7 | -24.9 | 48.5 | -37.3 |
| V$_d$(O$_3$) | 0.4 | 0.5 | 0.6 | 0.1 | 0.4 | 26.2 | 0.3 | 0.4 | 0.8 | 0.1 | 0.2 | 7.7 |
| O$_3$ddep | -4.2 | -6.1 | 0.7 | -2.1 | 4.4 | 50.3 | -6.1 | -5.6 | 0.6 | 0.5 | 6.0 | -7.4 |

**4.4.1 臭氧干沉降速率模拟验证**

从图4.13可以看出，在永丰站原始方案 (Default) 模拟的 V$_d$(O$_3$) 明显大于观测值，改进方案 (Improved) 模拟的 V$_d$(O$_3$) 比原始方案的模拟值小的多，模式能客观反映出 V$_d$(O$_3$) 的日变化过程。总体而言，改进方案的模拟 V$_d$(O$_3$) 比较接近观测值，但是模拟 V$_d$(O$_3$) 峰值出现的时间和观测值存在不一致性，改进模拟的 V$_d$(O$_3$) 在下午 13:00 左右出现峰值，而观测值一般在早上 09:00 左右出现峰值。其中整个模拟时段原始方案和改进方案的平均 V$_d$(O$_3$) 分别为 0.5 和 0.4 cm·s$^{-1}$，最大值分别为 1.3 和 0.8 cm·s$^{-1}$；最小值分别为 0.006 和 0.003 cm·s$^{-1}$。由表4.6可以看出，改进方案的模拟 V$_d$(O$_3$) 效果明显优于默认方案，原始方案和改进方案的平均偏差 (NMB) 分别为 59.3% 和 7.4%，表明原始方案和改进方案模拟的 V$_d$(O$_3$) 均偏高，而改进方案模拟的 V$_d$(O$_3$) 偏高幅度较小；r 分别为 0.7 和 0.8。在鼎湖山站，同样改进方案模拟的 V$_d$(O$_3$) 比原始方案的模拟值要小，但是两个方案都无法模拟出观测的峰值。对于整个模拟时段原始方案和改进方案的平均 V$_d$(O$_3$) 分别为 0.6 和 0.5 cm·s$^{-1}$，最大值分别为 1.3 和 0.6 cm·s$^{-1}$；最小值分别为 0.003 和 0.007 cm·s$^{-1}$。原始方案和改进方案的 NMB 分别为 54.8% 和 26.2%，r 分别为 0.5 和 0.6。综合比较，改进方案模拟的 V$_d$(O$_3$) 效果较好。

(a) 鼎湖山站，冠层外      (b) 永丰站

图 4.13: 改进前后 $V_d(O_3)$ 的模拟和观测值的昼夜变化过程

**4.4.2 臭氧浓度模拟验证**

图4.14是观测站点在改进前后两种干沉降方案下 $O_3$ 浓度的观测值和模拟值的昼夜变化过程。在永丰站，两种方案都大致能模拟出 $O_3$ 浓度的日变化趋势，上午 $O_3$ 浓度随着太阳辐射的增加而逐渐增加，下午随着太阳辐射的减弱逐渐下降，在凌晨左右达到最低值。虽然两种方案的模拟结果与观测值在日变化趋势上基本一致，但模拟结果明显低于观测值；默认方案和改进方案下的 $O_3$ 浓度在夜间差别很小，改进方案的模拟值白天高于默认方案。其中整个模拟时段默认方案和改进方案的平均 $O_3$ 浓度分别为 36.9 $\mu g \cdot m^{-3}$ 和 46.4 $\mu g \cdot m^{-3}$，最大值分别为 145.3 $\mu g \cdot m^{-3}$ 和 164.0 $\mu g \cdot m^{-3}$；最小值均为 0 $\mu g \cdot m^{-3}$。从表4.6可以看出，该观测站点默认方案和改进方案的 r 分别为 0.6 和 0.7，MB 和 NMB 的负值显示了两个方案的模拟结果相比观测值均偏低，其中改进方案的模拟效果比默认方案略好。在鼎湖山站，两种方案同样都大致能模拟出 $O_3$ 浓度的日变化趋势。但是，与永丰站不同的是，两种方案模拟值在夜间都低于观测值，白天高于观测值。其中整个模拟时段默认方案和改进方案的平均 $O_3$ 浓度分别为 49.1 $\mu g \cdot m^{-3}$ 和 56.7 $\mu g \cdot m^{-3}$，最大值分别为 162.3 $\mu g \cdot m^{-3}$ 和 222.5 $\mu g \cdot m^{-3}$；最小值均为 0 $\mu g \cdot m^{-3}$。从表4.6可以看出，该观测站点默认方案和改进方案的 r 分别为 0.52 和 0.56，略低于永丰站的 r，MB 和 NMB 的同样均为负值，改进方案的模拟效果要好于比默认方案。总体而言，改进方案模拟的 $O_3$ 浓度效果较好。

(a) 鼎湖山站，冠层外           (b) 永丰站

图 4.14: 改进前后 $O_3$ 浓度的模拟和观测值的昼夜变化过程

  图4.15显示了两种干沉降方案下的 1、4、7、10 四个月各观测站点地表 $O_3$ 浓度模拟值与观测值的 1：1 图。从图可以看出，改进方案对 $O_3$ 浓度模拟增加是显著的。在原始方案下四个月对 $O_3$ 浓度的模拟都是低估的。干沉降方案改进后，与观测相比模拟结果仍然是低估，但是对植被较为茂盛或 $O_3$ 浓度本身就较为高的季节 4 月和 7 月，改进方案显著改善了对 $O_3$ 浓度的模拟效果，更加接近观测值。其中，4 月的 NMB 由-19.3% 降低到-10.3%，7 月由-21.8% 降低到-13.4%。Val Martin 等[46]的研究结果也表明，地表 $O_3$ 浓度的模拟受植被物候的影响较大，因此干沉降方案的改变对于植被贫瘠的地区和时期几乎没有影响。干沉降机制的改进效果要小于人为排放导致的潜在地表 $O_3$ 浓度的变化，但是它更能反映化学-气象之间的相互作用[316]。

[Figure]

图 4.15: 改进前后 $O_3$ 浓度观测站点的模拟值和观测值比较

**4.4.3 臭氧干沉降通量模拟验证**

图4.16为两个观测站点在两种干沉降方案下 $O_3$ddep 的观测值和模拟值的昼夜变化过程。在永丰站，夜间默认方案和改进方案下的 $O_3$ddep 差别很小，但都高于观测值，白天默认方案的模拟值低于改进方案。两种方案都大致能模拟出 $O_3$ddep 的日变化趋势。从表4.6可以看出，改进方案要略好于默认方案，r 分别为0.6 和0.5。其中整个模拟时段默认方案和改进方案的平均 $O_3$ddep 分别为-6.6 nmol·m$^{-2}$·s$^{-2}$ 和-5.6 nmol·m$^{-2}$·s$^{-2}$，最小值为-35.0 nmol·m$^{-2}$·s$^{-2}$ 和-32.8 nmol·m$^{-2}$·s$^{-2}$，最大值均为 0 nmol·m$^{-2}$·s$^{-2}$，改进方案于默认方案的差别不大。在鼎湖山站，整个模拟时段默认方案和改进方案的平均 $O_3$ddep 分别为-6.7 nmol·m$^{-2}$·s$^{-2}$ 和-6.1 nmol·m$^{-2}$·s$^{-2}$，最小值分别为-22.8 nmol·m$^{-2}$·s$^{-2}$ 和-17.9 nmol·m$^{-2}$·s$^{-2}$，最大值均为 0 nmol·m$^{-2}$·s$^{-2}$。从表4.6可以看出，MB 和 NMB 的负值显示了两个方案的模拟结果相比观测值均偏低，改进方案的模拟效果要略好于默认方案，模拟精度提高了 16.8%。总体而言，改进方案模拟的 $O_3$ddep 要优于于默认方案。

(a) 鼎湖山站，冠层外        (b) 永丰站

图 4.16: 改进前后 $O_3$ddep 的模拟和观测值的日变化过程

**4.5 本章小结**

本章基于 WRF-Chem 模式，建立了 Ball-Berry 导度机制与 $O_3$ 化学沉降模块的程序接口，使模式中的 Wesely 干沉降机制可灵活调用此气孔导度机制进行计算，并将原始模式中对 $R_{lu}$ 的计算修正为 Wesely 干沉降机制的计算方法。其次，构建了 $O_3$ 损伤植被的参数化方案，将臭氧累积吸收通量对植被光合速率和叶片气孔导度的影响过程考虑进区域数值模式中，并与 WRF-Chem 模式中的干沉降机制进行耦合，实现了在 WRF-Chem 中 $O_3$ 与植被的双向反馈和 $O_3$ 累积作用的表征。基于改进的 WRF-Chem 模式，取 2019 年 1、4、7、10 四个月作为典型代表月份，验证和评估了模式的改进效果。

根据 699 个气象观测站点和 1600 个空气质量站点的观测数据，综合评估了模式的气象场和化学场的模拟效果。综合而言，模式模拟的气象参数趋势和观测趋势匹配，量级上吻合度较好，模拟结果可靠。大气污染物方面，不同月份污染物的模拟昼夜变化趋势与观测都较为一致，模拟的污染物和站点观测浓度空间吻合较好。总体而言，在量级和趋势上，WRF-Chem 都能较为合理地模拟出中国地区主要大气污染物浓度的变化特征，可用于后续中国地区臭氧干沉降过程的研究。

模式改进后，在永丰站原始方案模拟的 $V_d(O_3)$ 明显大于观测值，改进方案模拟的 $V_d(O_3)$ 比原始方案的模拟值小的多，模式能客观反映出 $V_d(O_3)$ 的日变化过程。总体而言，改进方案模拟的 $V_d(O_3)$ 更接近于观测值，但是模拟 $V_d(O_3)$ 峰值出现的时间和观测值存在不一致性，改进方案模拟的 $V_d(O_3)$ 在下午 13:00 左右出现峰值，而观测值一般在早上 09:00 左右出现峰值。其中整个模拟时段原始方案和改进方案的平均 $V_d$ 分别为 0.5 和 0.4 cm·s$^{-1}$，改进方案相较于原始方案对 $V_d(O_3)$ 模拟精度提高了 51.9%。在鼎湖山站，同样改进方案模拟

的 $V_d(O_3)$ 比原始方案的模拟值要小，但是两个方案都无法模拟出观测的峰值。对于整个模拟时段原始方案和改进方案的平均 $V_d(O_3)$ 分别为 0.6 cm·s$^{-1}$ 和 0.5 cm·s$^{-1}$，相关系数分别为 0.5 和 0.6，改进方案相较于默认方案对 $V_d(O_3)$ 模拟精度提高了 28.6%。

对于 $O_3$ 浓度，从观测站点看，两个站点模拟值都低于观测值，永丰站和鼎湖山站改进方案相较于默认方案对 $O_3$ 浓度的模拟精度分别提高了 10.0% 和 8.8%。从全国站点地表 $O_3$ 浓度的模拟值看，改进方案显著改善了对 $O_3$ 浓度的模拟，更加接近观测值。其中，4 月的 NMB 由-19.3% 降低到-10.3%，7 月由-21.8% 降低到-13.4%。Val Martin 等等[46]的研究结果也表明，地表 $O_3$ 浓度的模拟受植被物候的影响较大，因此干沉降方案的改变对于植被贫瘠的地区和时期几乎没有影响。干沉降机制的改进效果要小于人为排放导致的潜在 $O_3$ 的变化，但是它更能反映化学-气象之间的相互作用[316]。

对于 $O_3$ddep，永丰站改进方案和默认方案模拟的 $O_3$ddep 差别不大，模拟精度仅提高了 0.3%；鼎湖山站，改进方案相较于和默认方案模拟进度提高了 16.8%。总体而言，改进方案的模拟效果更好。

**第五章 典型下垫面臭氧干沉降过程的数值模拟研究**

干沉降是臭氧影响生态系统的主要途径，由于我国臭氧干沉降通量和干沉降速率的研究起步较晚，目前尚缺乏对不同下垫面臭氧干沉降过程的系统评估。干沉降通量由污染物浓度和干沉降速率共同决定，厘清影响干沉降通量的关键过程有助于改善空气质量以及理解潜在的生态环境风险。本章基于改进后的 WRF-Chem 模式，选取 1、4、7、10 四个月作为代表月份，评估模式改进后对植被生理生态过程，气象要素以及臭氧干沉降过程的影响。利用 WRF-Chem 中已有对臭氧浓度生成的过程诊断模块，第三章构建的干沉降速率过程诊断模块，结合本章构建的污染物干沉降通量的过程诊断模块，分析和评估了典型下垫面影响臭氧干沉降通量，臭氧浓度以及臭氧干沉降速率的关键过程及其差异。

**5.1 模式改进对区域臭氧-植被空间格局的影响**

模式改进对区域植被生理过程、气象要素和臭氧干沉降过程的相关参数以及相对于原始模式的差值与相对变化统计如表5.1所示。本研究模拟了 2019 年 1、4、7、10 四个月，其中 1 月由于植被生理生态活动低，因此下文将主要对 4、7、10 三个月这些不同的参数进行详细讨论。

**5.1.1 模式改进对臭氧干沉降格局的影响**

图5.1显示了臭氧干沉降速率 ($V_d(O_3)$) 的空间分布以及模式改进引起的 $V_d(O_3)$ 的绝对变化和相对变化。空间分布上，东部地区 $V_d(O_3)$ 的下降高于西都地区，减幅程度也更大；季节上，4 月和 7 月 $V_d(O_3)$ 的变化区域分布更为广泛，而 10 月 $V_d(O_3)$ 的变化主要集中在南方地区的森林下垫面。模式改进后 $V_d(O_3)$ 的变化分别为-0.18~0.02 cm·s$^{-1}$ (4 月)，-0.26~0.03 cm·s$^{-1}$ (7 月) 和-0.20~0.01 cm·s$^{-1}$ (10 月)，相对变化分别为-35.2~2.8%，-50.9~4.5% 和-28.9~2.3%，平均下降 25.0%。本研究的结果相较于 Sadiq 等[25]的研究结果略高，他们的研究结果表明考虑 $O_3$ 与植被的相互作用，使得中国，欧洲和北美的 $V_d(O_3)$ 下降了约 20%。从不同下垫面看 (表5.1)，农田下垫面平均下降了 19.2% (4 月)、19.0% (7 月) 和 1.2% (10 月)；森林下垫面平均下降了 19.8% (4 月)、25.8% (7 月) 和 12.4% (10 月)；草地下垫面平均下降了 7.8% (4 月)、15.7% (7 月) 和 5.8% (10 月)；城市下垫面平均下降了 0.3% (4 月)、0.5% (7 月) 和 0.9% (10 月)。

地表 $O_3$ 浓度的空间分布以及由模式改进引起的 $O_3$ 浓度的绝对变化和相对变化如

表 5.1: 模式改进后中国地区各参数统计及各改进模式相对于原始模式的差值与相对变化 (%)

| 参数 | | 全国 1月 | 全国 4月 | 全国 7月 | 全国 10月 | 农田 1月 | 农田 4月 | 农田 7月 | 农田 10月 | 森林 1月 | 森林 4月 | 森林 7月 | 森林 10月 | 草地 1月 | 草地 4月 | 草地 7月 | 草地 10月 | 城市 1月 | 城市 4月 | 城市 7月 | 城市 10月 |
|---|---|---|---|---|---|---|---|---|---|---|---|---|---|---|---|---|---|---|---|---|---|
| $R_s$ | 改进 | 0.19 | 1.06 | 3.71 | 0.98 | 0.38 | 1.48 | 5.23 | 2.62 | 0.62 | 3.13 | 9.88 | 2.89 | 0.19 | 1.33 | 4.39 | 1.24 | 0.01 | 0.11 | 1.34 | 0.12 |
| | MB | 0.00 | 0.04 | 0.22 | 0.05 | 0.00 | 0.06 | 0.32 | 0.20 | 0.06 | 0.11 | 0.48 | 0.13 | 0.01 | 0.03 | 0.32 | 0.19 | 0.00 | 0.00 | 0.08 | 0.02 |
| | NMB | 1.17% | 3.77% | 6.94% | 5.67% | 1.21% | 2.96% | 8.26% | 6.90% | 1.10% | 3.30% | 7.39% | 7.33% | 0.98% | 3.78% | 5.52% | 4.71% | 0.96% | 2.96% | 3.39% | 2.43% |
| PSN | 改进 | 61.81 | 69.17 | 75.25 | 58.67 | 49.10 | 63.42 | 65.91 | 47.66 | 57.29 | 74.26 | 70.94 | 60.54 | 43.44 | 65.60 | 68.25 | 52.63 | 28.54 | 57.63 | 54.60 | 50.14 |
| | MB | -0.71 | -3.69 | -4.78 | -2.15 | -0.66 | -6.58 | -6.40 | -2.26 | -1.70 | -6.03 | -6.55 | -3.40 | -0.49 | -1.47 | -3.82 | -0.83 | -0.29 | -1.58 | -1.97 | -1.05 |
| | NMB | -1.36% | -6.11% | -7.66% | -4.66% | -1.58% | -12.03% | -11.05% | -4.98% | -3.29% | -9.03% | -11.55% | -5.95% | -0.68% | -2.20% | -5.36% | -1.18% | -0.91% | -2.48% | -3.29% | -3.09% |
| LAI | 改进 | 0.62 | 1.38 | 1.92 | 1.45 | 0.94 | 2.53 | 2.91 | 2.31 | 1.64 | 2.94 | 3.69 | 3.24 | 0.08 | 0.21 | 1.09 | 0.46 | 0.15 | 0.17 | 0.17 | 0.10 |
| | MB | 0.00 | -0.01 | -0.07 | -0.03 | -0.01 | -0.03 | -0.17 | -0.06 | -0.01 | -0.06 | -0.12 | -0.06 | 0.00 | -0.01 | -0.06 | -0.01 | -0.01 | -0.02 | -0.08 | -0.03 |
| | NMB | -0.40% | -2.11% | -7.00% | -3.27% | -0.85% | -2.34% | -11.15% | -4.44% | -0.35% | -1.87% | -6.36% | -1.66% | -0.06% | -2.74% | -5.91% | -3.89% | -0.41% | -1.02% | -1.62% | -0.91% |
| GPP | 改进 | 1.05 | 3.89 | 5.30 | 2.91 | 1.83 | 8.40 | 9.54 | 5.83 | 2.59 | 6.97 | 9.00 | 5.60 | 0.01 | 0.33 | 2.36 | 0.45 | 0.03 | 0.86 | 1.85 | 0.61 |
| | MB | 0.00 | -0.09 | -0.49 | -0.11 | -0.01 | -0.23 | -1.21 | -0.28 | -0.01 | -0.42 | -0.77 | -0.31 | 0.00 | -0.01 | -0.22 | -0.02 | 0.00 | -0.02 | -0.04 | -0.03 |
| | NMB | -1.35% | -5.94% | -13.49% | -9.05% | -1.12% | -7.82% | -19.23% | -12.33% | -0.53% | -6.01% | -9.59% | -4.98% | -4.28% | -7.22% | -12.07% | -8.85% | -0.30% | -3.09% | -1.67% | -1.87% |
| TR | 改进 | 0.13 | 0.66 | 1.01 | 0.41 | 0.22 | 1.39 | 1.68 | 0.78 | 0.33 | 1.21 | 1.65 | 0.80 | 0.00 | 0.07 | 0.57 | 0.08 | 0.05 | 0.16 | 0.17 | 0.12 |
| | MB | 0.00 | -0.03 | -0.12 | -0.03 | 0.00 | -0.06 | -0.26 | -0.06 | 0.00 | -0.04 | -0.18 | -0.05 | 0.00 | 0.00 | -0.07 | 0.00 | 0.00 | -0.06 | -0.03 | -0.05 |
| | NMB | -1.15% | -5.98% | -14.07% | -8.59% | -0.77% | -6.99% | -18.87% | -10.98% | -0.50% | -3.30% | -12.24% | -4.93% | -4.01% | -7.14% | -13.76% | -8.72% | -0.38% | -4.41% | -6.80% | -5.32% |
| ISOP | 改进 | 0.19 | 1.06 | 3.71 | 0.98 | 0.38 | 1.48 | 5.23 | 2.62 | 0.62 | 3.13 | 9.88 | 2.89 | 0.19 | 1.33 | 4.39 | 1.24 | 0.01 | 0.11 | 1.34 | 0.12 |
| | MB | 0.00 | 0.02 | 0.19 | 0.05 | 0.00 | 0.03 | 0.22 | 0.06 | 0.01 | 0.07 | 0.48 | 0.13 | 0.00 | 0.03 | 0.13 | 0.10 | 0.00 | 0.00 | 0.03 | 0.00 |
| | NMB | 1.17% | 2.28% | 5.05% | 3.67% | 1.21% | 1.96% | 5.69% | 3.77% | 1.10% | 2.30% | 4.69% | 4.04% | 0.98% | 2.78% | 3.52% | 3.11% | 0.96% | 1.96% | 2.09% | 1.43% |
| T2 | 改进 | 264.14 | 281.73 | 292.73 | 280.00 | 270.23 | 287.67 | 298.13 | 286.43 | 269.23 | 284.51 | 293.79 | 283.32 | 255.94 | 273.00 | 284.71 | 271.95 | 278.70 | 291.97 | 300.92 | 293.06 |
| | MB | 0.04 | 0.08 | 0.23 | 0.11 | 0.06 | 0.09 | 0.30 | 0.14 | 0.04 | 0.08 | 0.13 | 0.09 | 0.03 | 0.08 | 0.25 | 0.12 | 0.04 | 0.09 | 0.25 | 0.19 |
| | NMB | 0.02% | 0.03% | 0.08% | 0.04% | 0.02% | 0.03% | 0.10% | 0.05% | 0.01% | 0.03% | 0.04% | 0.03% | 0.01% | 0.03% | 0.09% | 0.05% | 0.01% | 0.03% | 0.08% | 0.07% |
| RH | Base | 62.36 | 63.21 | 70.89 | 64.04 | 68.08 | 72.84 | 83.32 | 74.45 | 70.53 | 75.74 | 88.35 | 82.00 | 59.80 | 60.49 | 69.76 | 59.81 | 52.85 | 65.84 | 77.18 | 62.12 |
| | MB | -0.18 | -0.46 | -1.43 | -0.68 | -0.18 | -0.65 | -1.94 | -0.98 | -0.13 | -0.49 | -0.85 | -0.66 | -0.24 | -0.43 | -1.61 | -0.61 | -0.09 | -0.71 | -1.42 | -1.48 |
| | NMB | -0.32% | -0.77% | -2.28% | -1.11% | -0.31% | -0.99% | -2.63% | -1.41% | -0.22% | -0.66% | -1.01% | -0.82% | -0.42% | -0.76% | -2.55% | -1.08% | -0.21% | -1.13% | -2.21% | -2.41% |
| LH | 改进 | 12.57 | 55.36 | 76.81 | 35.23 | 24.05 | 89.39 | 100.84 | 53.81 | 25.53 | 82.26 | 106.11 | 59.43 | 25.53 | 82.26 | 106.11 | 59.43 | 3.53 | 10.70 | 11.63 | 7.58 |
| | MB | -0.19 | -1.76 | -4.93 | -1.55 | -0.40 | -3.45 | -10.05 | -3.06 | -0.37 | -2.40 | -4.48 | -2.13 | 0.00 | -0.55 | -1.95 | -0.44 | -0.04 | -0.31 | -1.24 | -0.50 |
| | NMB | -1.37% | -4.11% | -8.22% | -6.61% | -1.99% | -4.84% | -13.57% | -9.26% | -1.72% | -2.99% | -4.30% | -3.72% | -0.52% | -2.94% | -3.16% | -3.32% | -1.63% | -3.67% | -7.23% | -5.03% |
| SH | 改进 | 10.62 | 51.09 | 48.06 | 28.97 | 14.71 | 43.62 | 25.64 | 20.54 | 34.26 | 66.83 | 35.47 | 34.91 | 2.36 | 43.45 | 58.73 | 30.40 | 35.38 | 82.05 | 63.90 | 48.04 |
| | MB | 0.07 | 1.22 | 3.05 | 1.03 | 0.11 | 1.99 | 5.78 | 1.82 | 0.10 | 1.80 | 2.56 | 1.37 | 0.02 | 0.44 | 1.28 | 0.33 | 0.12 | 1.94 | 3.40 | 2.70 |
| | NMB | 2.89% | 2.87% | 15.89% | 6.83% | 4.06% | 5.01% | 28.69% | 16.29% | 1.02% | 3.00% | 8.47% | 4.57% | 4.65% | 1.52% | 2.69% | 2.35% | 0.42% | 2.33% | 5.23% | 5.75% |
| $O_3$ | 改进 | 61.81 | 69.17 | 75.25 | 58.67 | 49.10 | 63.42 | 65.91 | 47.66 | 57.29 | 74.26 | 70.94 | 60.54 | 43.44 | 65.60 | 68.25 | 52.63 | 28.54 | 57.63 | 54.60 | 50.14 |
| | MB | 0.71 | 3.69 | 4.78 | 1.15 | 0.66 | 6.58 | 6.40 | 1.26 | 1.70 | 6.03 | 6.55 | 2.40 | 0.49 | 1.47 | 3.82 | 0.83 | 0.29 | 3.58 | 2.97 | 3.05 |
| | NMB | 1.36% | 6.11% | 7.66% | 1.92% | 1.58% | 12.03% | 11.05% | 2.33% | 3.29% | 9.03% | 11.55% | 4.01% | 0.68% | 2.20% | 5.36% | 1.18% | 0.91% | 7.48% | 6.29% | 6.39% |
| $V_d(O_3)$ | 改进 | 0.12 | 0.35 | 0.37 | 0.29 | 0.19 | 0.42 | 0.43 | 0.34 | 0.20 | 0.40 | 0.45 | 0.34 | 0.04 | 0.37 | 0.40 | 0.31 | 0.19 | 0.26 | 0.26 | 0.24 |
| | MB | -0.01 | -0.07 | -0.08 | -0.02 | -0.01 | -0.11 | -0.10 | 0.00 | -0.04 | -0.11 | -0.15 | -0.06 | 0.00 | -0.03 | -0.08 | -0.02 | 0.00 | 0.00 | 0.00 | 0.00 |
| | NMB | -6.15% | -12.14% | -15.95% | -4.80% | -3.07% | -19.21% | -18.99% | -1.20% | 14.03% | -19.76% | -25.75% | -12.41% | -6.55% | -7.80% | -15.73% | -5.82% | 0.21% | -0.28% | -0.50% | -0.87% |
| $O_3$dep | 改进 | 1.64 | 6.87 | 7.74 | 4.76 | 2.19 | 8.10 | 8.48 | 4.93 | 2.94 | 8.11 | 7.49 | 5.94 | 0.71 | 6.76 | 8.69 | 5.04 | 1.74 | 3.86 | 3.69 | 3.27 |
| | MB | -0.17 | -1.10 | -1.23 | -0.29 | -0.09 | -1.82 | -1.38 | -0.09 | -0.63 | -1.89 | -1.90 | -1.03 | -0.03 | -0.59 | -1.38 | -0.12 | -0.02 | -0.02 | -0.07 | -0.04 |
| | NMB | -6.17% | -10.93% | -12.28% | -4.78% | -3.75% | -16.66% | -13.64% | -2.05% | -13.00% | -17.34% | -19.78% | -11.40% | -6.17% | -7.65% | -13.11% | -4.39% | -1.18% | -0.55% | -2.09% | -1.89% |

图5.2所示。从图可以看出，$O_3$ 浓度和 $V_d(O_3)$ (图5.1) 的空间变化呈相反趋势，地表 $O_3$ 浓度的显著增加是由于 $V_d(O_3)$ 的下降引起的，尤其是对 4 月和 7 月植被生理活动较为强烈以及地表 $O_3$ 浓度较高的季节。4 和 7 月，$O_3$ 浓度在东部地区显著增加，最大增幅分别为 13.7 $\mu g \cdot m^{-3}$ (26.3%) 和 23.3 $\mu g \cdot m^{-3}$ (46.3%)。10 月，$O_3$ 浓度在南方地区显著上升，最大增幅约为 13.5 $\mu g \cdot m^{-3}$ (32.0%)。从不同下垫面看 (表5.1)，农田和森林下垫面 $O_3$ 浓度增幅最大，平均增加了 8.5% 和 8.2%；其次是城市下垫面，平均增加了 6.7%。本研究中地表 $O_3$ 浓度的增幅度 Sadiq 等[25]和 Zhu 等[317]等人的研究结果相似。在 Sadiq 等[25]的研究中，模拟的欧洲、北美和中国的地表 $O_3$ 浓度可增加 7~12 $\mu g \cdot m^{-3}$。Zhou 等[33]在 GEOS-Chem 模式中对 $O_3$ 和 LAI 进行了动态耦合，发现随着 $O_3$ 浓度水平的增加，大多数植物功能类型的 LAI 显著下降。$O_3$ 对 LAI 的破坏可进一步导致 $O_3$ 浓度产生-1.8~3 ppb 的变化。Gong 等[318]利用 ModelE2-YIBs 模式评估 $O_3$ 与植被的相互作用。结果表明，$O_3$ 胁迫引起气孔导度降低，进而使地表 $O_3$ 浓度平均增了 2.1 ppb。在 Zhu 等[317]的研究中，河北，山西的 $O_3$ 浓度增加最高，大约为 12 $\mu g \cdot m^{-3}$。本研究结果表明，$O_3$ 与植被的相互作用通过减少 $V_d(O_3)$ (通过抑制气孔导度) 和增加化学形成 (通过减弱蒸腾作用和增加温度) 从而引起地表 $O_3$ 浓度的增加。

图5.3显示了臭氧干沉降通量 ($O_3$ddep) 的空间分布以及由模式改进引起的 $O_3$ddep 的绝对变化和相对变化。由图可以看出，模式改进后 $O_3$ddep 整体呈现下降的现象，其中 4 月平均下降了 1.1 $kg \cdot ha^{-1} \cdot month^{-1}$，7 月平均下降了 1.2 $kg \cdot ha^{-1} \cdot month^{-1}$，10 月平均下降了 0.3 $kg \cdot ha^{-1} \cdot month^{-1}$。从不同的下垫面看，森林下垫面降幅最多，平均下降了 16.2%；农田下垫面下降了 10.8%；森林下垫面下降了 8.4%，城市下垫面降幅最小，仅有 1.1%。此外，虽然 $O_3$ddep 是由 $O_3$ 浓度和 $V_d(O_3)$ 相乘所得，但是本研究中 $O_3$ddep 变化的空间分布与 $V_d(O_3)$ 变化的空间分布更为一致，这反映了可能 $V_d(O_3)$ 在 $O_3$ddep 变化中占主导地位。

(a) 1 月

(b) 4 月

(c) 7 月

(d) 10 月

图 5.1: 模式改进对臭氧干沉降速率空间分布的影响

(a) 1 月

(b) 4 月

(c) 7 月

(d) 10 月

图 5.2: 模式改进对臭氧浓度空间分布的影响

(a) 1 月

(b) 4 月

(c) 7 月

(d) 10 月

图 5.3: 模式改进对臭氧干沉降通量空间分布的影响

**5.1.2 模式改进对植被生理过程的影响**

$O_3$ 胁迫会危害对植被的光合速率和气孔导度，从而干扰植被的生长、生产力和蒸腾作用。为了解 $O_3$ 胁迫对植被生理的影响，本小节主要分许模式改进对本研究对光合速率 (PSN)、气孔阻抗 ($R_s$)、叶面积指数 (LAI)、总初级生产力 (GPP)、蒸腾速率 (TR) 和异戊二烯排放 (ISOP) 的影响。1、4、7、10 四个月进行了，其中 1 月由于植被生理活动低，因此主要分析 4、7、10 三个月模式改进对

图5.4为模式改进后 $R_s$ 的空间分布及变化。总体而言，$R_s$ 的空间分布呈现明显的东西分布，东部地区高于西部地区。模式改进后 $R_s$ 有所增加，东部地区的变化高于西部地区，最大为 $3.5 \times 10^3$ s·m$^{-1}$，相比于原始模式增加约了 30%。$R_s$ 增加是引起 $V_d(O_3)$ 下降的主要原因。从不同下垫面看 (表5.1)，7 月农田和森林下垫面 $R_s$ 较高，其相对变化幅度较大，分别为 5.7% 和 4.1% (7 月)。草地下垫面的 $R_s$ 季节变化和农田森林下垫面相似，相对变化在 7 月也能达到 5.5%。城市下垫面 $R_s$ 相较于其他下垫面较低，相对变化在 1%～2% 之间。

图5.5为模式改进后 PSN 的空间分布，以及模式改进引起的变化。从图可以看出，东部地区的 PSN 普遍高于西部，最大值可达到 16 $\mu$molCO$_2$·m$^{-2}$·s$^{-1}$ (4 月)，17 $\mu$molCO$_2$·m$^{-2}$·s$^{-1}$ (7 月) 和 12 $\mu$molCO$_2$·m$^{-2}$·s$^{-1}$ (10 月)。模式改进后，PSN 整体呈下降趋势，东部地区的变化高于西部地区，三个月的下降范围分别为 0～5.3 $\mu$molCO$_2$·m$^{-2}$·s$^{-1}$,0～10.2 $\mu$molCO$_2$·m$^{-2}$·s$^{-1}$ 和 0～3.8 $\mu$molCO$_2$·m$^{-2}$·s$^{-1}$。东部地区，PSN 最高下降幅度可达 60%；西部地区，主要为草地下垫面，PSN 值较小，下降幅度最高约为 ~35%。从不同下垫面看 (表5.1)，PSN 在森林下垫面最高，其次是农田和草地下垫面。由于 PSN 与植被的生理活动相关，因此在城市下垫面较低, 相对变化范围在 0.9%～3.3%。森林下垫面相对变化范围在 3.3%～11.6%，农田下垫面范围在 1.6%～12.0%，草地下垫面范围在 0.7%～5.4%。

LAI (图5.6) 和 GPP (图5.7) 的空间分布与 PSN 的分布相似。不同月的最大值份别达到 5.2 m²·m$^{-2}$ 和 16.8 gC·m$^{-2}$·day$^{-1}$ (4 月)，5.3 m²·m$^{-2}$ 和 17.8 gC·m$^{-2}$·day$^{-1}$ (7 月)，5.0 m²·m$^{-2}$ 和 13.3 gC·m$^{-2}$·day$^{-1}$(10 月)。随着 PSN 的下降，LAI 和 GPP 也随之下降。LAI 下降区域主要集中在东部地区，平均相对变化约 24%；西部地区，由于 LAI 本身较小，变化不明显，最高相对变化可达到 24%。GPP 的下降区域同样集中在东部地区，最高可超过 1.2 gC·m$^{-2}$·day$^{-1}$，对应的相对变化可达 40%；西部地区，相对变化大约为 35%。LAI 和 GPP 在农田和草地下垫面相对变化最大，变化范围分别为 0.9%～11.2% 和 0.1%～6.9% (LAI)，1.1%～19.2% 和 4.3%～12.1% (GPP)。Yue 等[21]的研究表明，$O_3$ 胁迫导致美国东海岸 GPP 降幅在 11～17%。Lombardozzi 等[23]使用 CLM 模式估计当前的 $O_3$ 胁迫会导致全球 GPP 减少 8～12%。Xie 等[319]使用区域气候模式 (RegCM-CHEM4) 和耶鲁动态植被模式 (YIBs)，估算了 $O_3$ 胁迫导致中国地区 GPP 显著下降 (12.1±4.4%)，夏季下降幅度高达 35%。Zhu

等[317]利用 WRF-Chem 模式，揭示了 $O_3$ 与植被的反馈会导致 GPP 减少 20～40%。本研究的结果与 Xie 等[319]和 Zhu 等[317]的研究结果相近，但是量级大于 Yue 等[21]和 Lombardozzi 等[23]的研究。不同的模式设置是产生差异的原因之一。Yue 等[21]和 Lombardozzi 等[23]使用的是离线模式，相较于 Xie 等[319]，Zhu 等[317]和本研究使用的是在线模式产生了更小的损伤。

图5.8为 TR 的空间分布以及模式改进引起的变化。东部地区由于植被覆盖度较大，TR 值较高。如图所示，东部地区的 TR 值变化范围在 0.3～1.2 mm·day$^{-1}$，相对变化减少了 20% 以上。西部地区虽然 TR 值没有东部地区高，但是相对变化与东部地区相似，草地下垫面的减幅范围在 7.1%～13.8%。TR 受到 $R_s$ 和 LAI 变化的影响。在 $O_3$ 胁迫下，$R_s$ 的增加和 LAI 的减小都会导致 TR 的下降。对比 $R_s$ (图5.4)、LAI (图5.6) 和 TR (图5.8)，本研究发现 TR 变化的空间分布与 $R_s$ 变化的空间分布更为一致，这反映了在 TR 变化中 $R_s$ 可能占主导地位。

(a) 1 月

(b) 4 月

(c) 7 月

(d) 10 月

图 5.4: 模式改进对气孔阻抗空间分布的影响

(a) 1 月

(b) 4 月

(c) 7 月

(d) 10 月

图 5.5: 模式改进对光合速率空间分布的影响

(a) 1 月

(b) 4 月

(c) 7 月

(d) 10 月

图 5.6: 模式改进对叶面积指数空间分布的影响

(a) 1 月

(b) 4 月

(c) 7 月

(d) 10 月

图 5.7: 模式改进对总初级生产力空间分布的影响

(a) 1 月

(b) 4 月

(c) 7 月

(d) 10 月

图 5.8: 模式改进对蒸腾速率空间分布的影响

图5.9显示了 ISOP 排放的空间分布及其在 $O_3$ 胁迫下的变化。模式改进后，ISOP 排放整体呈现增加的趋势，三个月分别平均增加了 $1.1\ mol\cdot km^{-2}\cdot h^{-1}$ (4 月)，$3.7\ mol\cdot km^{-2}\cdot h^{-1}$ (7 月) 和 $1.0\ mol\cdot km^{-2}\cdot h^{-1}$ (10 月)。东部地区增幅较为明显，尤其是森林下垫面，最高增加了 $1.9\ mol\cdot km^{-2}\cdot h^{-1}$；农田下垫面相对变化较高，变化范围为 2.0～5.7%。总体上看，4 月 ISOP 排放变化在 $0～1.0\ mol\cdot km^{-2}\cdot h^{-1}$(0.02～55.1%)，7 月在 $-0.2～3.8\ mol\cdot km^{-2}\cdot h^{-1}$(-0.7～67.6%)，10 月在 $-0.1～1.4\ mol\cdot km^{-2}\cdot h^{-1}$(-0.05～53.8%)。尽管 LAI 呈现下降趋势 (图5.6)，但是 T2 的增加 (图5.12)，TR (图5.8) 和 LH (图5.10) 的下降抵消了因 LAI 降低导致的 ISOP 排放的减少。

(a) 1 月

(b) 4 月

(c) 7 月

(d) 10 月

图 5.9: 模式改进对异戊二烯排放空间分布的影响

**5.1.3 模式改进对气象敏感要素的影响**

通过与植被的相互作用，$O_3$ 胁迫可能会对区域的地表热通量，温度和湿度等气象因素产生影响。

图5.10和图5.11显示了潜热通量 (LH) 和感热通量 (SH) 的空间分布，以及模式改进引起的 LH 和 SH 的变化。模式改进后，随着 TR 的降低，LH 平均下降了 1.8 W·m$^{-2}$ (4 月)，4.9 W·m$^{-2}$ (7 月) 和 1.6 W·m$^{-2}$ (10 月)，相对变化下降了 4.1%，8.2% 和 6.6%。中部和北部地区降幅最为显著，最高可达 30.1 W·m$^{-2}$ (4 月)，41.6 W·m$^{-2}$ (7 月) 和 26.9 W·m$^{-2}$ (10 月)。同时，SH 平均上升了 1.2 W·m$^{-2}$ (4 月)，3.1 W·m$^{-2}$ (7 月) 和 1.0 W·m$^{-2}$ (10 月)，相对变化增加了 2.9%，15.9% 和 6.8%。$O_3$ 与植被耦合后，地表热通量在农田下垫面变化最大，变化范围为 4.8~13.6% (LH) 和 5.0~45.8% (SH) (表5.1)。$O_3$ 胁迫使地表能量平衡发生转变，净辐射更多的被 SH 耗散而不是被 LH 耗散，并对 T2 产生影响。

图5.12和图5.13显示了 T2 和 RH 的空间分布及其变化。TR 的降低同时还会导致 RH 的降低。总体而言，北方地区的 RH 的降幅大于南方地区，最大降幅可达 5.0% (4 月)，16.2% (7 月) 和 10.1% (10 月)。LH 的降低驱动了 T2 的升高。如图5.12所示，T2 的变化分布与 RH 的变化分布相似。北方地区增幅较大，在 0.2~1.2 K 之间。本研究的结果与 Li 等[29]，Sadiq 等[25]和 Zhu 等[317]的研究结果相当。Li 等[29]对美国德克萨斯州和北部地区的模拟结果表明，$O_3$ 胁迫使 LH 降低了 10~27 W·m$^{-2}$，T2 升高了 0.6~2.0 ℃。Sadiq 等[25]对全球区域的模拟结果表明，$O_3$ 与植被的相互作用使 LH 变化在-15~5 W·m$^{-2}$，T2 升高 2 K 以上，其中中国区域 T2 升高 0.2~1.0 K。Zhu 等[317]对中国地区的研究结果表明，LH 降幅 5~30 W·m$^{-2}$，T2 升高 0.2~0.8 K。值得注意的是，Li 等[29]的研究中采用的 $O_3$ 阈值为 20 ppb，他们认为 $O_3$ 浓度超过 20 ppb 时，$O_3$ 就会对植物造成损伤。而本研究与 Sadiq 等[25]和 Zhu 等[317]均采用以往研究中普遍使用的 40 ppb 为阈值，这种差异可能会导致气象场有更大的变化。$O_3$ 胁迫通过减弱 TR 使 T2 升高，RH 和 LH 降低，形成了有利于 $O_3$ 产生的气象条件。

(a) 1 月

(b) 4 月

(c) 7 月

(d) 10 月

图 5.10: 模式改进对潜热通量空间分布的影响

(a) 1 月

(b) 4 月

(c) 7 月

(d) 10 月

图 5.11: 模式改进对感热通量空间分布的影响

(a) 1 月

(b) 4 月

(c) 7 月

(d) 10 月

图 5.12: 模式改进对 2m 温度空间分布的影响

(a) 1 月

(b) 4 月

(c) 7 月

(d) 10 月

图 5.13: 模式改进对相对湿度空间分布的影响

**5.2 典型下垫面臭氧干沉降日变化特征分析**

**5.2.1 典型下垫面臭氧干沉降日变化差异**

图5.14显示了 1、4、7、10 四个月典型下垫面 $O_3ddep$，$V_d(O_3)$ 和 $O_3$ 浓度的昼夜变化。由图可以看出，$O_3ddep$，$O_3$ 浓度和 $V_d(O_3)$ 都具有典型的日变化过程。受到太阳辐射、温度等气象要素的影响，$O_3$ 浓度和 $V_d(O_3)$ 白天高于夜间，呈单峰型变化趋势，$O_3ddep$ 由 $O_3$ 浓度和 $V_d(O_3)$ 共同影响，日变化与之相似，三者的最高值均出现在 12～15 点之间。

城市下垫面，$V_d(O_3)$ 的日变化较为平缓，$O_3ddep$ 和 $O_3$ 浓度的昼夜变化趋势一致。4 月臭氧沉降量最高，平均为 399.4 $kg \cdot km^{-1} \cdot month^{-1}$；其次是 7 月和 10 月，平均约为 376.6 $kg \cdot km^{-1} \cdot month^{-1}$ 和 336.3 $kg \cdot km^{-1} \cdot month^{-1}$；1 月由于 $O_3$ 浓度较低，导致臭氧沉降量在四个月里最低，为 183.5 $kg \cdot km^{-1} \cdot month^{-1}$。

草地下垫面，$O_3$ 浓度在 1 月未出现明显的日变化特征，$V_d(O_3)$ 受到叶面积指数，以及温度辐射等气象要素的影响 1 月明显低于其他三个月，但是日变化特征明显，$O_3ddep$ 受到 $V_d(O_3)$ 的影响臭氧沉降量较低，仅为 74.0 $kg \cdot km^{-1} \cdot month^{-1}$。其他三个月，$O_3ddep$，$V_d(O_3)$ 和 $O_3$ 浓度均呈现典型的日变化趋势，$O_3ddep$ 出现峰值的时间点基本在 $O_3$ 浓度和 $V_d(O_3)$ 出现峰值的中间。4、7、10 三个月的臭氧沉降量分别为 697.2 $kg \cdot km^{-1} \cdot month^{-1}$,845.6 $kg \cdot km^{-1} \cdot month^{-1}$ 和 498.9 $kg \cdot km^{-1} \cdot month^{-1}$。

农田下垫面的臭氧沉降量是四个典型下垫面中最高的，这意味着 $O_3$ 胁迫可能会对农田生态系统造成较高的损伤。其中，4 月和 7 月臭氧沉降量相当，分别达到 803.2 $kg \cdot km^{-1} \cdot month^{-1}$ 和 804.1 $kg \cdot km^{-1} \cdot month^{-1}$；10 月臭氧沉降量为 487.7 $kg \cdot km^{-1} \cdot month^{-1}$；同样 1 月的臭氧沉降量最低，仅为 290.2 $kg \cdot km^{-1} \cdot month^{-1}$。

在森林下垫面，4 月和 7 月臭氧沉降量最高，分别为 756.4 $kg \cdot km^{-1} \cdot month^{-1}$ 和 724.4 $kg \cdot km^{-1} \cdot month^{-1}$。这是可能是由于 4 月和 7 月森林的 BVOC 排放较高，从而导致较高 $O_3$ 浓度引起的。10 月次之，沉降量约为 495.1 $kg \cdot km^{-1} \cdot month^{-1}$，1 月沉降量最低，约为 290.2 $kg \cdot km^{-1} \cdot month^{-1}$。

从全国来看，中国地区的平均臭氧沉降量为 1850.7 $kg \cdot km^{-1} \cdot month^{-1}$。其中，1、4、7、10 四个月臭氧沉降量分别为 180.6 $kg \cdot km^{-1} \cdot month^{-1}$、610.8 $kg \cdot km^{-1} \cdot month^{-1}$、650.7 $kg \cdot km^{-1} \cdot month^{-1}$ 和 408.6 $kg \cdot km^{-1} \cdot month^{-1}$。

(a) 1 月

(b) 4 月

(c) 7 月

(d) 10 月

图 5.14: 典型下垫面臭氧干沉降通量，臭氧浓度和臭氧干沉降速率的日变化过程

**5.3 典型下垫面影响臭氧干沉降的主要过程分析**

**5.3.1 影响典型下垫面臭氧干沉降通量日变化的过程诊断分析**

为进一步解析影响 $O_3ddep$ 的大气过程，本小结基于 WRF-Chem 模式，构建了针对污染物干沉降量的过程诊断分析模块。$O_3ddep$ 的变化由地表污染物浓度和干沉降速率共同决定，计算方程如式5.1所示：

$$\Delta O_3ddep = O_3ddep_{(t1)} - O_3ddep_{(t1)} = C_{O3(t1)} * V_d(O_3)_{(t0)} - C_{O3(t0)} * V_d(O_3)_{(t1)} \qquad (5.1)$$

$$= (C_{O3(t1)} - C_{O3(t0)}) * V_d(O_3)_{(t0)} \qquad (5.2)$$

$$+ (V_d(O_3)_{(t1)} - V_d(O_3)_{(t0)}) * C_{O3(t0)} \qquad (5.3)$$

$$+ (C_{O3(t1)} - C_{O3(t0)}) * (V_d(O_3)_{(t1)} - V_d(O_3)_{(t0)}) \qquad (5.4)$$

其中，t1 为下一个时刻，t0 为上一个时刻，式(5.2)为 $O_3$ 浓度变化贡献量，式(5.3)为 $V_d(O_3)$ 变化贡献量，式(5.4)为两者变化的残差。

图5.15和5.16显示了 2019 年 1、4、7、10 四个月 $O_3ddep$ 日变化的过程分解。由图可以看出，不同下垫面不同季节的 $O_3ddep$ 日变化的趋势基本相同，在 8 时左右开始增加，$V_d(O_3)$ 和

$O_3$ 浓度此时对 $O_3$ddep 均为正贡献，在正午左右到达峰值然后开始下降，16 时左右 $V_d(O_3)$ 和 $O_3$ 浓度对 $O_3$ddep 开始变为负贡献，在 19 时左右到达最低值。此外，$O_3$ 浓度较低的季节，$O_3$ddep 的变化主要由 $V_d(O_3)$ 的变化主导 (除城市下垫面)。

城市下垫面，1、4、7、10 四个月 $O_3$ddep 的日内变化范围分别为-0.12～0.11 kg·km$^{-2}$·hr$^{-1}$,-0.15～0.12 kg·km$^{-2}$·hr$^{-1}$,-0.12～0.11 kg·km$^{-2}$·hr$^{-1}$ 和-0.19～0.12 kg·km$^{-2}$·hr$^{-1}$。由于城市下垫面 $V_d(O_3)$ 的日变化趋势非常平缓 (图5.14) ,$O_3$ddep 的变化主要由是 $O_3$ 浓度的变化引起的。1 月 $O_3$ 浓度相对较低，所以 $V_d(O_3)$ 变化贡献量较其他三个月略高，约为 27.4%。而其余三个月，$O_3$ 浓度变化的贡献量均在 80% 以上。

草地下垫面，1 月 $O_3$ddep 较低 (0.04～0.30kg·km$^{-2}$·hr$^{-1}$，图5.14) ，因此日内变化幅度较小。与城市下垫面不同的是，$O_3$ 浓度的日变化不明显 (图5.14) ，主导 $O_3$ddep 变化的主要过程是 $V_d(O_3)$，贡献量高达 91.5%，而 $O_3$ 浓度变化的贡献量仅为 8.0%。4 月和 7 月两个过程量对 $O_3$ddep 贡献的变化趋势基本相同，$V_d(O_3)$ 变化的平均贡献量分别为 58.9% 和 50.7%，$O_3$ 变化的平均贡献量分别为 39.1% 和 47.3%。此外，11～16 时基本由 $O_3$ 浓度变化主导的，贡献量分别为 64.5% (4 月) 和 75.4% (7 月) 。10 月，$V_d(O_3)$ 和 $O_3$ 浓度变化的平均贡献量分别为 69.1% 和 28.9%，除了 13～15 时 $O_3$ddep 的变化以 $O_3$ 浓度为主导外 ($\geq 50\%$) ，其余时刻均由 $V_d(O_3)$ 的变化主导。

农田下垫面，1、4、7、10 四个月 $O_3$ddep 的日内变化范围分别为-0.18～0.19 kg·km$^{-2}$·hr$^{-1}$,-0.35～0.34 kg·km$^{-2}$·hr$^{-1}$,-0.33～0.28 kg·km$^{-2}$·hr$^{-1}$ 和-0.26～0.23 kg·km$^{-2}$·hr$^{-1}$。$O_3$ 浓度和 $V_d(O_3)$ 的变化对 $O_3$ddep 的贡献率分别为 43.5% 和 53.5%(1 月)，55.2% 和 42.0% (4 月) ，60.4% 和 37.1% (7 月) 和 58.7% 和 38.2% (10 月) 。$V_d(O_3)$ 主要受到气象因素影响，日出和日落前后气象要素差异较大，导致 $V_d(O_3)$ 变化幅度较大 (图5.19) ，从而主导了 $O_3$depp 的变化。夜晚，$V_d(O_3)$ 变化趋于平缓，此时 $O_3$ddep 的变化主要由 $O_3$ 浓度的变化引起。午间，$O_3$ 浓度到达峰值，此刻对 $O_3$ddep 的贡献量超过 85%。

森林下垫面，1、4、7、10 四个月 $O_3$ddep 的日内变化范围分别为-0.16～0.13 kg·km$^{-2}$·hr$^{-1}$,-0.33～0.32 kg·km$^{-2}$·hr$^{-1}$,-0.26～0.23 kg·km$^{-2}$·hr$^{-1}$ 和-0.24～0.21 kg·km$^{-2}$·hr$^{-1}$。1 月 $O_3$ddep 的变化主要由 $V_d(O_3)$ 主导的，其贡献量超过 75%，其余三个月 $V_d(O_3)$ 和 $O_3$ 浓度对 $O_3$ddep 的贡献量相当。与农田下垫面相似，午间引起 $O_3$ddep 变化的主要过程是 $O_3$ 浓度。

从全国来看，1、4、7、10 四个月 $O_3$ddep 的变化范围分别为-0.10～0.09 kg·km$^{-2}$·hr$^{-1}$,-0.22～0.20 kg·km$^{-2}$·hr$^{-1}$,-0.18～0.20 kg·km$^{-2}$·hr$^{-1}$ 和-0.18～0.16 kg·km$^{-2}$·hr$^{-1}$。1 月，$V_d(O_3)$ 的变化是引起 $O_3$ddep 变化的主要过程，贡献率高达 80%；4 月和 10 月 $V_d(O_3)$ 贡献率变低，但仍是主要过程，约为 63%。7 月，$O_3$ 浓度和 $V_d(O_3)$ 对 $O_3$ddep 变化的贡献率相当，均约为 49%。1 月，$O_3$ 浓度和 $V_d(O_3)$ 的变化对 $O_3$ddep 的贡献率的基本没有日变化趋势，$O_3$ 浓

度变化的贡献率在 10.2～28.8%，$V_d(O_3)$ 变化的贡献率在 70.8～81.7%。其他三个月，两个过程量的日变化趋势基本相同，日出和日落前后 $V_d(O_3)$ 变化的贡献率最多 (≥90%)，午后 $O_3$ 浓度变化的贡献率最多 (≥90%)。

(a) 1 月

(b) 4 月

(c) 7 月

(d) 10 月

图 5.15: 典型下垫面臭氧浓度和臭氧干沉降速率对臭氧干沉降通量日内变化的绝对贡献

(a) 1 月

(b) 4 月

(c) 7 月

(d) 10 月

图 5.16: 典型下垫面臭氧浓度和臭氧干沉降速率对臭氧干沉降通量日内变化的相对贡献

**5.3.2 影响典型下垫面臭氧浓度日变化的过程诊断分析**

随后，我们进一步对影响 $O_3$ddep 中的 $O_3$ 浓度的变化进行了过程分析。对流层中 $O_3$ 浓度的变化受到各项物理化学过程的直接影响[320]。WRF-Chem 模式中的化学诊断模块 (chem-diag) 提供了过程诊断变量 (advz_o3，advh_o3，chem_o3，vmix_o3 和 conv_o3) 来显示主要过程对 $O_3$ 浓度的贡献，对于任意网格上的 $O_3$ 浓度，每一个时间步长的 $O_3$ 浓度变化 ($\Delta C_{O3}$) 等于：

$$\Delta C_{O3} = ADV + VMIX + CONV + CHEM \tag{5.5}$$

其中，ADV (平流贡献量) 反映平流输送作用对 $O_3$ 浓度的贡献，与风场和 $O_3$ 浓度有关，为 advh_o3 和 advz_o3 的和 (ADV=advh_o3+advz_o3)；VMIX (垂直混合贡献量)，体现垂直混合作用对 $O_3$ 浓度的贡献，与湍流和 $O_3$ 垂直梯度有关；CHEM (化学贡献量)，是光化学反应对 $O_3$ 浓度的净贡献量，包括生成与消耗；CONV (对流贡献量)，是对流运动对

$O_3$ 浓度的贡献[321]。

对于地表 $O_3$ 浓度，干沉降是其重要的汇，与地表 $O_3$ 浓度和干沉降速率高度相关，因此还应考虑干沉降过程 (DRY) 的贡献。在 WRF-Chem 模式中，污染物的干沉降过程在 dry_dep_driver 模块 (chem/dry_dep_driver.F) 的子程序 vertmx (chem/module_vertmx_wrf.F) 中作为垂直混合过程的一部分一起计算，且只发生在第一层的高度。因此，在第一层中，vmix_o3 是 VMIX 和 DRY 的贡献之和。在第一层之上，vmix_o3 等于 VMIX 的贡献。由此，$\Delta O_3$ 在任意网格和每个时间步长的质量平衡方程为：

$$\Delta C_{O3} = \begin{cases} ADV + VMIX + DRY + CONV + CHEM & layer = 1 \\ ADV + VMIX + CONV + CHEM & layer \geq 1 \end{cases} \tag{5.6}$$

在进行干沉降计算时，压力和温度是不变的。因此，每个时间步长 (dt) 的 DRY 对 $O_3$ 浓度 ($C_{O3}$) 的贡献为：

$$DRY = C_{O3} * V_d(O_3) * \frac{dt}{dz} \tag{5.7}$$

其中，$V_d(O_3)$ 为臭氧干沉降速率，dz 为栅格高度。VMIX 在第一层的贡献为：

$$VMIX = vmix\_o3 - DRY \tag{5.8}$$

2019 年 1、4、7、10 四个月典型下垫面 $O_3$ 浓度变化的过程分析结果如图5.17和5.18所示。由图可以看出，$O_3$ 日内变化呈现明显的日变化过程，白天主要为 $O_3$ 生成，夜间 $O_3$ 主要为消耗过程，这与其他过程分析的研究结果相似[322-325]。

在城市下垫面，地表 $O_3$ 浓度的日内变化没有明显的季节差异，1、4、7、10 四个月 $O_3$ 浓度的日内变化范围在-21.7~14.5 $\mu g \cdot m^{-3} \cdot hr^{-1}$, -17.0~11.7 $\mu g \cdot m^{-3} \cdot hr^{-1}$, -12.5~11.3 $\mu g \cdot m^{-3} \cdot hr^{-1}$ 和-22.9~14.0 $\mu g \cdot m^{-3} \cdot hr^{-1}$。四个月均是化学过程和垂直混合过程占主导地位。白天，地表 $O_3$ 的主要来源于光化学的产生，垂直混合过程倾向于消耗 $O_3$。夜间，则通过气相反应去除 $O_3$，垂直混合变为促进 $O_3$ 生成。总体而言，化学过程所占的比例约 50%，垂直混合过程约占 41%，干沉降过程占比不到 2%。

在草地下垫面，1 月由于 $O_3$ 浓度本身较低，因此 $O_3$ 浓度日内变化幅度也较小，在-4.3~1.3$\mu g \cdot m^{-3} \cdot hr^{-1}$。与城市下垫面不同，垂直混合对 $O_3$ 的生成主要是正贡献，占比约为 20.2%。化学过程，平流过程和干沉降过程对 $O_3$ 变化的贡献分别为 41.9%，30.0% 和 8.0%。干沉降作为地表 $O_3$ 重要的汇，对 $O_3$ 的贡献基本为负，白天湍流活动较为剧烈，干沉降对 $O_3$ 贡献的占比增加到 19.0%。4、7、10 三个月 $O_3$ 的日内变化范围分别为-17.2~7.1 $\mu g \cdot m^{-3} \cdot hr^{-1}$, -15.8~9.1 $\mu g \cdot m^{-3} \cdot hr^{-1}$ 和-15.2~5.0 $\mu g \cdot m^{-3} \cdot hr^{-1}$。垂直混合过程是引起白天地表 $O_3$ 浓度增

加的关键过程，贡献量分别为43.8%，44.9%和40.6%。干沉降过程均为负贡献，清晨温度和辐射开始增强，干沉降贡献变多，达到峰值后随之减少，傍晚由于气象要素的变化，导致干沉降贡献量逐步增多。干沉降对 $O_3$ 浓度变化的贡献率平均为14.4%，15.1%和13.5%。平流过程白天为负贡献，夜间为正贡献，对 $O_3$ 浓度变化的贡献率平均为26.2%，19.0%和27.4%，白天高于夜间。

在农田下垫面，1、4、7、10四个月 $O_3$ 的日内变化范围分别为-14.2~8.5 $\mu g \cdot m^{-3} \cdot hr^{-1}$,-24.0~11.9 $\mu g \cdot m^{-3} \cdot hr^{-1}$,-21.1~10.7 $\mu g \cdot m^{-3} \cdot hr^{-1}$ 和-21.9~11.3 $\mu g \cdot m^{-3} \cdot hr^{-1}$。相较于草地下垫面，农田下垫面中化学过程对 $O_3$ 浓度变化的贡献增多。白天，$O_3$ 浓度的来源主要是化学和垂直混合过程，汇是干沉降和平流过程。夜间化学和垂直混合过程变为汇，平流输出过程变为 $O_3$ 的主要来源。1、4、7、10四个月化学过程对 $O_3$ 浓度变化的贡献均超过了50%，分别为54.1%，52.4%，50.6%和53.0%。其次是垂直混合过程对 $O_3$ 浓度变化的贡献，分别为20.3%，18.8%，20.2%和21.6%。1月和10月平流过程对 $O_3$ 浓度变化的贡献比干沉降过程多，分别为17.8%和14.5%，而干沉降过程则为7.3%和10.0%。4月平流过程和干沉降过程占比相当，均约为13.8%。7月由于气象环境有利于 $O_3$ 的干沉降，因此干沉降过程占比较多，达到16.4%，而平流过程为10.3%。

森林下垫面与草地下垫面的变化情况相似，主要是垂直混合过程影响 $O_3$ 的生成与消散。1、4、7、10四个月 $O_3$ 浓度的日内变化范围分别为-9.8~3.8 $\mu g \cdot m^{-3} \cdot hr^{-1}$,-21.6~6.9 $\mu g \cdot m^{-3} \cdot hr^{-1}$,-16.7~7.4 $\mu g \cdot m^{-3} \cdot hr^{-1}$ 和-15.9~5.7 $\mu g \cdot m^{-3} \cdot hr^{-1}$。1月，白天和夜晚均为化学过程对 $O_3$ 变化的贡献最多，分别36.3%和43.2%。此外，平流过程在夜间对 $O_3$ 变化的贡献也占到了约40.0%，干沉降过程在夜间只占到3.4%。其余三个月，白天垂直混合过程对 $O_3$ 浓度变化的贡献约为46.1%，而干沉降过程和化学过程对 $O_3$ 浓度变化的贡献均占20.0%左右，平流过程占比不到15.0%。夜间，4月和10月平流过程占比最多，分别为29.5%和36.4%；7月则是垂直混合过程占比最高，达到36.7%。

从全国来看，$O_3$ 的日内变化一般在中午到达峰值，然后逐渐下降，在傍晚达到最低值，1、4、7、10四个月变化范围分别为-7.4~3.4 $\mu g \cdot m^{-3} \cdot hr^{-1}$,-14.7~6.7 $\mu g \cdot m^{-3} \cdot hr^{-1}$,-11.8~7.5 $\mu g \cdot m^{-3} \cdot hr^{-1}$ 和-12.6~6.0 $\mu g \cdot m^{-3} \cdot hr^{-1}$。1月，化学过程在白天和夜间的贡献都是最高的，分别为43.1%和53.0%，干沉降过程分别为15.0%和3.3%，垂直混合过程分别为27.1%和11.8%，平流过程分别为14.4%和31.2%。其他三个月，干沉降过程对白天 $O_3$ 变化的贡献率大约占到20%，垂直混合过程大约为38%，平流过程大约为12%。夜间化学过程贡献最高，超过32%。本研究的结果与Tao 等[323]研究结果相似，Tao 等[323]对华东地区2008~2012年7月的研究显示，白天地表 $O_3$ 的主要来源是光化学产生 (37%) 和垂直混合 (63%)，干沉降是其主要的汇。

(a) 1 月

(b) 4 月

(c) 7 月

(d) 10 月

图 5.17: 典型下垫面大气物理化学过程对臭氧浓度日内变化的绝对贡献

(a) 1 月

(b) 4 月

(c) 7 月

(d) 10 月

图 5.18: 典型下垫面大气物理化学过程对臭氧浓度日内变化的相对贡献

**5.3.3 影响典型下垫面臭氧干沉降速率日变化的过程诊断分析**

对于影响 $O_3ddep$ 日内变化的 $V_d(O_3)$ 过程，本小节根据公式3.1对其进行过程诊断分析。图5.19和5.20显示了 1、4、7、10 四个月 $V_d(O_3)$ 的日内变化特征。由图可以看出，不同下垫面 $V_d(O_3)$ 的日内变化趋势一致，清晨之后呈现净增加，并在中午左右达到峰值，午后开始减少，傍晚左右到达变化的最低值。

在城市下垫面，由于 $V_d(O_3)$ 本身没有呈现明显的昼夜差异，因此 $V_d(O_3)$ 日内变化差异很小，季节变化特征不明显，变化范围在-0.008~0.01 cm·s$^{-1}$。1 月影响 $V_d(O_3)$ 变化的主控因子是 $R_c$，占 96.0%。其他三个月，由 $R_a$ 和 $R_c$ 共同影响，其对 $V_d(O_3)$ 的贡献分别为41.6% 和 57.8% (4 月)，40.7% 和 58.8% (7 月) 和 54.8% 和 44.5% (10 月)。

草地下垫面，1 月，$R_a$ 和 $R_c$ 对 $V_d(O_3)$ 变化的贡献率分别为 54.0% 和 42.5%，白天时段受 $R_c$ 影响较多，夜间受 $R_a$ 影响较多。4、7 和 10 月 $V_d(O_3)$ 的变化范围分别为-0.15~0.17

cm·s$^{-1}$，-0.16～0.17 cm·s$^{-1}$ 和-0.13～0.15 cm·s$^{-1}$；R$_a$ 和 R$_c$ 对 V$_d$(O$_3$) 的贡献率在三个月相似，分别为 77.6% 和 20.0%、74.5% 和 23.4% 以及 71.8% 和 21.7%。不同的是，4 月和 7 月 V$_d$(O$_3$) 正午的变化由 R$_c$ 影响，而 10 月基本上都受到 R$_a$ 的影响。

农田下垫面，1、4、7、10 四个月 V$_d$(O$_3$) 的日内变化范围分别为-0.007～0.008 cm·s$^{-1}$，-0.14～0.21 cm·s$^{-1}$，-0.16～0.09 cm·s$^{-1}$ 和-0.12～0.14 cm·s$^{-1}$。四个月，R$_a$ 和 R$_c$ 对 V$_d$(O$_3$) 的贡献率在农田下垫面显示出一致性，白天主要受 R$_c$ 控制 (1 月 67.1%，4 月 58.5%，7 月 65.2%，10 月 59.1%)，夜晚主要受 R$_a$ 控制 (1 月 58.4%，4 月 70.4%，7 月 83.2%，10 月 64.1%)。

森林下垫面，1、4、7、10 四个月 V$_d$(O$_3$) 的日内变化范围分别为-0.001～0.002 cm·s$^{-1}$，-0.16～0.15 cm·s$^{-1}$，-0.11～0.08cm·s$^{-1}$ 和-0.12～0.11 cm·s$^{-1}$。7 月 V$_d$(O$_3$) 的变化呈现比较明显的昼夜差异，白天 R$_c$ 的影响逐步增加，并在第 15 时刻到达最大值 (88.8%) 之后影响逐步减小，变为受到 R$_a$ 的影响较多；R$_c$ 白天的贡献率为 62.3%，夜晚 R$_a$ 的贡献率为 82.7%。1、4 和 10 月影响 V$_d$(O$_3$) 变化的主控因子均为 R$_c$，其贡献率分别为 85.1%、75.7% 和 77.8%。

从全国来看，1、4、7、10 四个月 V$_d$(O$_3$) 的日内变化范围分别为-0.003～0.003 cm·s$^{-1}$，-0.04～0.03 cm·s$^{-1}$，-0.03～0.04 cm·s$^{-1}$ 和-0.04～0.04 cm·s$^{-1}$。正午 R$_c$ 是引起 V$_d$(O$_3$) 变化的主控因子，其余时段为 R$_a$。从日平均来看，R$_a$ 的贡献率更高，在四个月分别为 53.6%，55.5%，65.8% 和 58.2%；R$_c$ 的贡献率则分别为 43.9%，39.8%，32.8% 和 37.1%；残差的贡献率在 4 月和 10 月较高，大约有 5%。

(a) 1 月

(b) 4 月

(c) 7 月

(d) 10 月

图 5.19: 典型下垫面各阻抗对臭氧干沉降速率日内变化的绝对贡献

(a) 1 月

(b) 4 月

(c) 7 月

(d) 10 月

图 5.20: 典型下垫面各阻抗对臭氧干沉降速率日内变化的相对贡献

**5.4 本章小结**

植被通过气孔吸收 $O_3$，从而影响其光合作用，导致气孔关闭，叶面积指数发生变化。植被生理和结构的变化不仅会影响区域的碳循环、水循环和气候，还会影响 $O_3$ 浓度本身。本章基于改进后的 WRF-Chem 模式，研究了模式改进后对 $O_3$ 干沉降过程、气象要素和植被生理过程的影响及其相互作用。同时，分析了中国地区典型下垫面臭氧干沉降的差异，探讨影响 $O_3ddep$，$O_3$ 浓度以及 $V_d(O_3)$ 的关键过程。

研究结果表明，模式改进后 $R_s$ 最高增加了约 30%。对于 PSN，东部地区最高下降了约 60%；西部地区，最高下降了约 35%。随着 PSN 的减弱，LAI 下降区域同样集中在东部，下降幅度在 3.0~24.3%，GPP 的下降幅度约为 2.1~54.4%。随着 $R_s$ 的增加和 LAI 的降低，东部地区 TR 减少了 20% 以上。ISOP 排放整体呈现增加的趋势，变化范围在-0.05~67.6%。植被生理过程的变化，对农田和森林下垫面的影响最为显著。

模式改进对气象要素的影响主要表现为，LH 平均下降了 9.5%，SH 平均上升了了 8.5%。地表能量平衡发生转变，净辐射更多的被 SH 耗散而不是被 LH 耗散，因此对 T2 产生影响。LH 的降低驱动了 T2 的升高。其中，北方地区增幅较大，在 0.2~1.2 K 之间。TR 的降低进一步引起 RH 的降低，北方地区 RH 的降幅大于南方地区，降幅范围在 5.0~16.2%。

模式改进后引起植被生理和气象的变化，会进而影响臭氧干沉降过程。由于 $R_s$ 的增加导致 $V_d(O_3)$ 平均下降了 11.0%，进而引起地表 $O_3$ 浓度的增加，尤其在 4 月和 7 月，植被生理活动较为强烈以及 $O_3$ 浓度较高的季节，地表 $O_3$ 浓度平均上升了 6.1% 和 7.7%。$O_3$ddep 整体呈现下降的现象，平均下降了 9.3%。森林下垫面降幅最多，平均下降了 16.2%；农田下垫面次之，下降了 10.8%；城市下垫面降幅最小，仅有 1.1%。

基于改进后的模式模拟结果表明，我国臭氧月均沉降量约为 1850.7 $kg \cdot km^{-1} \cdot month^{-1}$。农田下垫面的臭氧沉降量最高，约为 596.3 $kg \cdot km^{-1} \cdot month^{-1}$，这意味着 $O_3$ 沉降可能会对农田生态系统的造成较高的损伤。其次是森林和草地下垫面，沉降量分别为 559.7 $kg \cdot km^{-1} \cdot month^{-1}$ 和 528.9 $kg \cdot km^{-1} \cdot month^{-1}$。城市下垫面，沉降量最低，为 323.9 $kg \cdot km^{-1} \cdot month^{-1}$。

$O_3$ddep 在不同下垫面不同季节的日内变化趋势基本相同，在 8 时左右开始增加，$V_d(O_3)$ 和 $O_3$ 浓度此时对 $O_3$ddep 均为正贡献，在正午左右到达峰值然后开始下降，16 时左右 $V_d(O_3)$ 和 $O_3$ 浓度对 $O_3$ddep 开始变为负贡献，并在在 19 时左右到达最低值。此外，在 $O_3$ 浓度较低的季节，$O_3$ddep 的变化主要由 $V_d(O_3)$ 的变化主导。1、4、7、10 四个月 $O_3$ 浓度和 $V_d(O_3)$ 对 $O_3$ddep 的贡献率分别为 18.9%、36.2%、48.4%、34.2 和 79.8%、61.8%、50.0%、63.7%。

$O_3$ 浓度日内变化呈现明显的昼夜差异，白天主要为 $O_3$ 生成，夜间 $O_3$ 主要为消耗过程。1 月，化学过程在白天和夜间的贡献都是最高的，分别为 43.1% 和 53.0%，干沉降过程分别为 15.0% 和 3.3%，垂直混合过程分别为 27.1% 和 11.8%，平流过程分别为 14.4% 和 31.2%。其他三个月，干沉降过程对白天 $O_3$ 变化的贡献率大约占到 20%，垂直混合过程大约为 38%，平流过程大约为 12%。夜间均为化学过程贡献最高，超过 32%。

不同下垫面 $V_d(O_3)$ 的日内变化趋势一致，清晨之后呈现净增加，并在中午左右达到峰值，午后开始减少，傍晚左右变化到达最低值。整体来看，1、4、7、10 四个月，$R_a$ 的贡献率分别为 53.6%，55.5%，65.8% 和 58.2%；$R_c$ 的贡献率则分别为 43.9%，39.8%，32.8% 和 37.1%。

**第六章 臭氧沉降对我国生态环境风险评估**

气候变化造成的大气环流的变化能够影响大气污染物的浓度和沉降，同时气象场的变化也能够影响植被和污染物的沉降[256]。目前的研究大多集中在气候变化对空气质量的影响上。针对气候变化是如何影响区域臭氧干沉降格局的研究相对较少，进一步考虑未来碳中和背景下臭氧前体物排放变化对臭氧暴露的研究更是缺乏。Wang 等[326]使用 GEOS-Chem 模式研究发现，2050 年气候变化导致中国东部地区 $O_3$ 浓度增加了 0.5~3 ppb 左右。Chuwah 等[327]利用 TM5 大气化学和 IMAGE 综合评估模式的结果表明，到了 2050 年，由于臭氧暴露全球作物损失可能高达 20%。如果实施严格的气候政策 (RCP2.6)，作物产量损失将得到显著的限制，所有地区的损失都不超过 10%。因此本研究探讨在碳中和背景下臭氧干沉降对气候变化的两个因子 (气象条件和人为排放源) 的响应，继而定量分析这种响应关系对作物产量和经济损失的影响，为未来区域 $O_3$ 风险评估提供科学依据。

**6.1 模式设置与数据介绍**

**6.1.1 气象要素数据**

WRF-Chem 模式运行所需的气象驱动数据由通用气候系统模式 (CCSM，Community Climate System Model) 提供。该数据来自美国 NCAR，可通过网站http://rda.ucar.edu/data/ds316.1/获取。CCSM 提供了 RCP4.5，RCP6.0 和 RCP8.5 三种路径下的气候资料，时间尺度从 2006 年到 2100 年，时间分辨率为 6 小时，水平分辨率为 0.9°×1.25°，垂直分层为 26 层，气压高度从 1000 hPa 到 10 hPa；气象参数包括相对湿度、温度、风速和风向等。本研究选取 2060 年 RCP4.5 和 RCP6.0 数据集驱动 WRF-Chem 模式，估算中国地区臭氧干沉降过程。

RCP4.5 和 RCP6.0 路径下降尺度到 WRF 模式模拟的气象要素空间分布如图6.1所示。相对于 2019 年 7 月，2060 年 RCP4.5 路径下，中国大部分地区的 T2 呈现上升趋势，华北、西北和珠三角部分地区呈现下降趋势；RH 的变化则呈现相反趋势；RCP6.0 路径下，T2 和 RH 变化相较于 RCP4.5 路径下更为明显，变化幅度分别达到 5 K 和 20%。

[Figure]

图 6.1: 2019 年 7 月以及 RCP4.5 和 RCP6.0 路径下温度和相对湿度的空间分布

**6.1.2 排放源数据**

本研究采用了清华大学提供的 2060 年中国人为排放源清单，该清单由中国未来排放动态评估模型 DPEC (Dynamic Projection model for Emissions in China) 构建 (http://meicmodel.org/?page_id=1917)，实现了本地排放情景与全球社会经济情景 (SSP) 气候目标约束 (RCP) 的衔接[328]。时间分辨率，水平分辨率，涵盖的大气污染物以及 VOCs 物种分配机制选取均与 MEIC 清单一致。该排放清单时间范围覆盖拓展至 2060 年，能够支持碳达峰碳中和背景下的中国未来大气成分变化及其影响研究。

为了探究碳达峰碳中和背景下的中国未来臭氧干沉降的变化，本研究选取了 DPEC 的三套排放情景，具体描述如表6.1所示。

表 6.1: 不同排放源场景描述

| 场景 | 气候政策 | 社会经济路径 | 排放场景说明 |
| --- | --- | --- | --- |
| SSP4-60-BAU | RCP6.0 | SSP4 | RCP6.0 气候目标下和当前污染控制政策 (BAU) 情景组合而成 |
| SSP2-45-ECP | RCP4.5 | SSP2 | 中国自主减排承诺气候目标 (RCP4.5) 与强化污染控制政策 (ECP) 情景组合而成 |
| SSP2-45-BHE | RCP4.5 | SSP2 | 中国自主减排承诺下的能源经济转型与最佳污染控制政策 (BHE) 组合而成 |

  SSP2-45-BHE、SSP2-45-ECP 和 SSP4-60-BAU 路径下 2060 年与当前按部门划分的中国地区各污染物排放源总量如图6.3所示。在 BAU 情景下，NOx 和 $SO_2$ 排放上升显著，NOx 的排放上升主要集中在交通 (85.2%)、电力 (53.2%) 和工业 (38.3%) 部门，而 $SO_2$ 的排放上升主要集中在工业 (90.0%) 和交通 (52.5%) 部门。在 ECP 和 BHE 情景下，除 $NH_3$ 以外，各污染物的减排主要发生在工业、交通和居民部门。SSP2-45-BHE、SSP2-45-ECP 和 SSP4-60-BAU 三种路径下中国地区各污染物排放总量差异如图6.3所示。总体而言，相对于 2017 年，SSP-2-45-BHE 和 SSP-2-45-ECP 两种路径下，污染物排放均呈现下降趋势，下降百分比分别在-30~78% 和-23~61%。SSP-4-60-BAU 路径下，$SO_2$、NOx、CO 和 NMVOCs 分别上升了 50%、55%、6% 和 5%，其余污染物排放呈现下降趋势，下降百分比分别在-5~23%。从空间差异上看 (图6.4)，SSP-2-45-BHE 和 SSP-2-45-ECP 两种路径下，各污染物排放的空间变化比较一致；而 SSP-4-60-BAU 路径下，NOx 全国都呈现上升的趋势，$SO_2$ 除四川、贵州、湖南和湖北略微下降，其他地区均上升，而 $PM_{2.5}$、BC、OC 和 CO 在北方地区大幅度上升。

[Figure]

图 6.2: 不同排放路径下 2060 年与当前按部门划分的中国地区各污染物排放源总量

[Figure]

图 6.3: 不同排放路径下 2060 年与当前中国地区各污染物排放源总量差异

[Figure]

图 6.4: 不同排放路径下 2060 年与当前各污染物排放源空间差异

**6.1.3 模式设置**

本研究选择 7 月作为模拟时段，模式设置和物理化学方案设置与第四章相同，实验设置如表6.2所示。其中 Base 实验使用 2019 年气象场 (MET)，2017 年 MEIC 人为源排放清单 (EI)；Case1 和 Case2 使用 RCP4.5 和 RCP6.0 路径下的 2060 年气象场数据，排放源清单保持不变，探究气象要素变化对中国臭氧干沉降的影响；Case3，Case4 和 Case5 分别使用 SSP2-45-BHE、SSP2-45-ECP 和 SSP4-60-BAU 路径下的 2060 年人为源排放清单，气象条件保持不变，探究人为源排放变化对中国臭氧干沉降的影响；Case6，Case7 和 Case8 分别使用 RCP4.5 和 RCP6.0 路径下的 2060 年气象数据，三组不同控制下人为排放源数据，探究气候变化下两个因子同时变化对中国臭氧干沉降格局的影响。

表 6.2: 模式实验设置

| 实验 | 气象场 (MET) | 排放源 (EI) | 说明 |
|---|---|---|---|
| Base | 2019 | MEIC 2017 | 基准 |
| Case1(MET4.5) | 2060(RCP4.5) | MEIC 2017 | RCP4.5 路径下气象要素变化影响 |
| Case2(MET6.0) | 2060(RCP6.0) | MEIC 2017 | RCP6.0 路径下气象要素变化影响 |
| Case3(EI BHE) | 2019 | SSP2-45-BHE | SSP2-45-BHE 路径下人为排放源变化影响 |
| Case4(EI ECP) | 2019 | SSP2-45-ECP | SSP2-45-ECP 路径下人为排放源变化影响 |
| Case5(EI BAU) | 2019 | SSP4-60-BAU | SSP4-60-BAU 路径下人为排放源变化影响 |
| Case6(RCP4.5BHE) | 2060(RCP4.5) | SSP2-45-BHE | RCP4.5 路径下气候变化影响 |
| Case7(RCP5.4ECP) | 2060(RCP4.5) | SSP2-45-ECP | RCP4.5 路径下气候变化影响 |
| Case8(RCP6.0BAU) | 2060(RCP6.0) | SSP4-60-BAU | RCP6.0 路径下气候变化影响 |

Base 方案下中国地区各参数以及各 Case 相对于 Base 方案的差值与相对变化统计如表6.3所示。相对于 Base 方案，不同方案对中国地区各参数的扰动如图6.5所示，下节将针对臭氧干沉降过程进行详细讨论。

表 6.3: Base 方案中国地区各参数统计及各 Case 相对于 Base 的差值与相对变化 (%)

| 类别 | 参数 | Base | MET | | EI | | | RCP | | |
|------|------|------|-----|-----|-----|-----|-----|-----|-----|-----|
| | | | 6.0 | 4.5 | BAU | ECP | BHE | 6.0BAU | 4.5ECP | 4.5BHE |
| 气象 | T2 | 21.4 | 0.9 | 0.3 | -0.08 | -0.02 | 0.01 | 0.8 | 0.4 | 0.7 |
| | ($^\circ$C) | | 4.6% | 1.5% | -0.4% | -0.1% | 0.1% | 4.5% | 2.0% | 3.8% |
| | RH | 70.9 | 5.3 | 3.4 | 0.5 | 0.3 | 0.4 | 5.5 | 3.5 | 3.0 |
| | (%) | | 7.5% | 4.9% | 0.7% | 0.4% | 0.6% | 7.7% | 4.9% | 4.7% |
| | SW | 242.4 | -8.2 | -7.4 | -1.7 | 2.3 | 3.0 | -9.3 | -3.6 | -0.5 |
| | (W·m$^{-2}$) | | -3.3% | -3.1% | -0.7% | 1.0% | 1.2% | -3.8% | -1.4% | -0.2% |
| | LH | 61.3 | 13.1 | 5.7 | 0.3 | 1.2 | 2.1 | 12.8 | 7.2 | 8.5 |
| | (W·m$^{-2}$) | | 21.2% | 9.2% | 0.4% | 2.0% | 3.5% | 20.7% | 11.7% | 13.8% |
| | SH | 48.1 | -6.6 | -3.2 | -1.3 | 0.6 | 0.4 | -6.7 | -1.9 | -1.5 |
| | (W·m$^{-2}$) | | -13.3% | -6.5% | -2.6% | 1.3% | 0.8% | -13.5% | -3.9% | -3.1% |
| 植被生理 | GPP | 5.3 | 1.1 | 0.6 | 0.1 | 0.1 | 0.2 | 1.1 | 0.7 | 0.8 |
| | (gC·m$^{-2}$·day$^{-1}$) | | 20.9% | 10.9% | 2.1% | 2.5% | 3.6% | 20.6% | 12.3% | 14.1% |
| | LAI | 1.9 | 0.2 | 0.1 | 0.02 | 0.01 | 0.03 | 0.2 | 0.1 | 0.1 |
| | (m$^2$·m$^{-2}$) | | 10.8% | 6.8% | 1.1% | 0.8% | 1.4% | 10.8% | 7.0% | 7.2% |
| | ISOP | 3.7 | 0.3 | -0.04 | 0.05 | 0.1 | 0.1 | 0.4 | 0.2 | 0.2 |
| | (mol·km$^{-2}$·h$^{-1}$) | | 10.0% | -1.6% | 0.8% | 1.7% | 6.8% | 11.3% | 4.7% | 4.2% |
| | TR | 1.0 | 0.2 | 0.2 | 0.01 | 0.03 | 0.05 | 0.2 | 0.2 | 0.2 |
| | (mm·day$^{-1}$) | | 22.2% | 15.6% | 1.3% | 2.8% | 4.9% | 22.0% | 19.3% | 21.7% |
| | PSN | 5.1 | 1.1 | 0.6 | 0.1 | 0.1 | 0.2 | 1.1 | 0.6 | 0.7 |
| | ($\mu$mol CO$_2$·m$^{-2}$·s$^{-1}$) | | 20.9% | 10.9% | 2.1% | 2.5% | 3.6% | 20.6% | 12.3% | 14.1% |
| | $R_s$ | 6.5 | -0.13 | 0.004 | -0.1 | -0.1 | -0.1 | -0.1 | 0.04 | 0.04 |
| | (*10$^3$ s·m$^{-1}$) | | -2.0% | 0.06% | -1.4% | -1.1% | -1.1% | -1.9% | 0.6% | 0.6% |
| 污染物 | NO$_2$ | 5.6 | -0.5 | -0.07 | 2.9 | -2.1 | -3.2 | 2.1 | -2.3 | -3.3 |
| | ($\mu$g·m$^{-3}$) | | -8.6% | -1.3% | 53.2% | -38.4% | -56.8% | 37.9% | -40.1% | -58.7% |
| | PM$_{2.5}$ | 14.4 | -2.1 | 1.7 | 3.3 | -4.6 | -6.1 | -0.04 | -3.1 | -5.3 |
| | ($\mu$g·m$^{-3}$) | | -14.9% | 12.0% | 23.1% | -32.6% | -43.1% | -0.3% | -21.9% | -36.9% |
| 臭氧干沉降 | O$_3$ | 75.2 | -0.8 | -1.3 | -2.4 | -4.0 | -7.8 | -2.8 | -4.6 | -8.7 |
| | ($\mu$g·m$^{-3}$) | | -0.5% | -2.6% | -3.0% | -5.2% | -10.0% | -2.1% | -4.5% | -12.0% |
| | V$_d$(O$_3$) | 0.4 | 0.02 | 0.0 | -0.0 | 0.0 | 0.0 | 0.01 | 0.0 | 0.01 |
| | (cm·s$^{-1}$) | | 4.1% | 1.2% | -0.3% | 0.1% | 0.1% | 4.0% | 2.0% | 2.3% |
| | O$_3$ddep | 7.8 | 0.07 | -0.03 | -0.2 | -0.5 | -0.5 | -0.2 | -0.4 | -0.6 |
| | (kg·ha$^{-1}$·month$^{-1}$) | | 0.6% | -0.4% | -2.8% | -4.8% | -6.9% | -2.9% | -6.5% | -8.4% |

[Figure]

图 6.5: 不同方案对中国地区各参数的扰动贡献 (%)

**6.2 气候变化对臭氧干沉降过程格局的影响**

**6.2.1 气候变化对臭氧浓度时空分布的影响**

不同方案下中国地区的 $O_3$ 浓度与 Base 方案的差值的空间分布以及不同下垫面的差值统计如图6.6和6.7 所示。排放与气象因素的改变对 $O_3$ 的影响不尽相同。气象因素的改变，导致中国大部分地区的 $O_3$ 浓度有轻微的下降，其中 MET4.5 方案下城市和农田下垫面降幅最大，分别下降了 29.0% 和 10.0%；在 MET6.0 方案下城市和森林下垫面 $O_3$ 浓度呈现下降趋势，而农田和森林呈现上升趋势。在 BAU 和 ECP 两种排放情景下，$O_3$ 浓度分别平均下降了 2.4 $\mu g \cdot m^{-3}$ (3.0%) 和 4.0 $\mu g \cdot m^{-3}$ (5.2%)，农田、森林和草地下垫面呈现下降趋势，但是城市下垫面上升了 5.3 $\mu g \cdot m^{-3}$ (14.8%) 和 6.4 $\mu g \cdot m^{-3}$ (19.1%)；BHE 方案下，农田、森林、草地和城市下垫面 $O_3$ 浓度均呈现下降趋势，平均下降了 10%。

整体而言，气象要素和排放源两个因子导致中国绝大部分地区的 $O_3$ 浓度呈现下降趋势，三种路径下中国地区 $O_3$ 平均浓度变化分别为-2.8 $\mu g \cdot m^{-3}$ (-2.1%)、4.6 $\mu g \cdot m^{-3}$ (-4.5%) 和 8.7 $\mu g \cdot m^{-3}$ (12.0%)。其中，城市下垫面在 RCP6.0 BAU 和 RCP4.5 ECP 两种方案下呈现上升趋势，这与在固定气象的情景下变化相同；在 RCP4.5 BHE 方案下城市下垫面变化幅度最大，其次是农田下垫面。如图6.5所示，排放源对中国地区 $O_3$ 浓度的影响大于气象要素。

[Figure]

图 6.6: 不同方案下臭氧浓度及其与 Base 方案差值的空间分布

[Figure]

图 6.7: 不同方案下各下垫面上臭氧浓度与 Base 方案差值

不同方案下臭氧 MDA8 的累积分布如图6.8所示。总体而言，与 Base 方案相比，Case 方案的累积分布向较低值移动，表明在不同的 Case 下臭氧浓度降低，但是在不同下垫面下不同 Case 表现不同。在农田下垫面，MET 类方案在 $0\sim100$ $\mu g\cdot m^{-3}$ 向右偏移，在高于 100 $\mu g\cdot m^{-3}$ 浓度区向左偏移，表明该方案下低浓度臭氧区呈上升趋势，高浓度臭氧区呈下降趋势；EI 方案和 RCP 方案 (除 RCP6.0 BAU) 都是向左偏移，表明在这两类方案下，臭氧浓度呈下降趋势。在森林下垫面，MET 类方案的变化与农田下垫面相似；RCP6.0 BAU 和 RCP4.5 ECP 下的累积分布大体上向右偏移；RCP4.5 BHE 与 EI ECP 的累积分布整体向左偏移；而 EI BAU 与 Base 变化大体一致。在草地下垫面，除 EI BAU 方案，其余方案的累积分布基本都向左偏移，表明草地下垫面臭氧浓度整体下降。而城市下垫面的 RCP 和 EI 方案大体是呈现下降的趋势；MET6.0 在低浓度臭氧区增加最为显著。三种路径下气候变化两个因子共同作用对 $O_3$ 浓度的贡献分别为-3.7%、-5.8% 和-11.6%。不同下垫面臭氧变化方案的差异可能与不同下垫面臭氧前体物的排放有关[329]。

除臭氧 MDA8 累积分布外，臭氧 MDA8 超过 100 $\mu g\cdot m^{-3}$ (一级标准) 和 160 $\mu g\cdot m^{-3}$ (二级标准) 的百分如图6.8所示，图例左列表明臭氧 MDA8 超过 $100\mu g\cdot m^{-3}$ 的百分比，右列表明臭氧 MDA8 超过 $160\mu g\cdot m^{-3}$ 的百分比，不同颜色表明各方案相对于 Base 的变化。就全国而言，MET 6.0、EI BAU 和 RCP6.0 BAU 与 Base 相比，在未来臭氧超过 100 $\mu g\cdot m^{-3}$ 的趋势有所增加，其余方案都显示有所下降，其中 RCP4.5 BHE 方案下最为显著；MET6.0、MET4.5 以及 RCP6.0 BAU 和 RCP4.5 BHE 在未来超过 160 $\mu g\cdot m^{-3}$ 的趋势有所下降。农田下垫面，各方案在超过 100 $\mu g\cdot m^{-3}$ 趋势的变化与全国的变化基本一致，而超过 160 $\mu g\cdot m^{-3}$ 标准下的各方案变化幅度均超过 1%。森林下垫面，RCP4.5 BHE 方案下降最为显著，而 RCP6.0 BAU 则呈现上升趋势。草地下垫面，$O_3$ 浓度基本没有超过 160 $\mu g\cdot m^{-3}$ 的情况，下降最为显著的方案是 EI BHE 和 RCP4.5 ECP。在城市下垫面，各方案的变化幅度相对较大，其中 MET6.0 方案显示在未来超过一级标准的情况增幅明显，但是超过二级标准却有所减少，MET4.5 方案情况相反；在 RCP 和 EI 两类方案下，未来臭氧超标的情况都有所减少。

[Figure]

图 6.8: 不同方案下臭氧 MDA8 的累积分布

**6.2.2 气候变化对臭氧干沉降速率时空分布的影响**

不同方案下中国地区的 $V_d(O_3)$ 与 Base 方案的差值的空间分布以及不同下垫面的差值统计如图6.9和6.10 所示。气象因素的改变，导致中国地区 $V_d(O_3)$ 呈现上升趋势，其中农田下垫面升幅最大，两种路径下分别上升了 7.0% 和 3.0%，其次是森林和草地下垫面，对于城市下垫面影响最小。在 ECP 和 BHE 两种排放情景下，$V_d(O_3)$ 轻微上升；而在 BAU 排放源路径下，$V_d(O_3)$ 小幅减小，其中农田下垫面变化幅度最大。

整体而言，气象和排放两个因素共同导致中国绝大部分地区 $V_d(O_3)$ 呈现上升趋势，三种路径下中国地区 $V_d(O_3)$ 平均上升了 4.3%，2.0% 和 2.3%，其中农田下垫面升幅最大，平均上升了 $0.013 \ \text{cm·s}^{-1}$（3%),$0.011 \ \text{cm·s}^{-1}$ (2.6%) 和 0.03 (7.0%) ，其次是森林和草地，城市下垫面基本没有变化。总体而言，与 $O_3$ 浓度不同，气象因素是影响对中国地区 $V_d(O_3)$ 的主要因素 (图6.5)。

[Figure]

图 6.9: 不同方案下臭氧干沉降速率及其与 Base 方案差值的空间分布

[Figure]

图 6.10: 不同方案下各下垫面上臭氧干沉降速率与 Base 方案差值

**6.2.3 气候变化对臭氧干沉降通量时空分布的影响**

不同气候变化因子对 $O_3$ddep 的影响如图6.11和6.12所示。$O_3$ddep 的变化与 $O_3$ 浓度的变化类似。气象因素的变化使中国地区 $O_3$ddep 呈现下降的趋势，并且对 $O_3$ddep 的影响在空间分布上与对 $O_3$ 浓度的影响类似。气象因素导致中部地区和东北地区 $O_3$ 浓度下降，进而导致该地区的 $O_3$ddep 下降超过约 $1.6\,\mathrm{kg·ha^{-1}·month^{-1}}$。在 EI BAU 方案下，排放源的改变对南方地区 $O_3$ddep 的影响显著，臭氧干沉降通量上升超过 $1.6\,\mathrm{kg·ha^{-1}·month^{-1}}$；EI ECP 方案下，大部分地区的 $O_3$ddep 呈现下降趋势，而在 EI BHE 方案下，华北地区 $O_3$ddep 下降显著，超过 $2.8\,\mathrm{kg·ha^{-1}·month^{-1}}$；三种 EI 路径下，全国 $O_3$ddep 平均下降 2.8%、6.5% 和 6.9%。同时考虑气象和排放源两个因素的影响，$O_3$ddep 的变化幅度超过单个变化因素的影响，全国 $O_3$ddep 下降分别为 2.6% (RCP6.0 BAU)、4.8% (RCP4.5 ECP) 和 8.4% (RCP4.5 BHE)。总体而言，排放源对中国地区 $O_3$ddep 的影响大于气象因素 (图6.5)。

[Figure]

图 6.11: 不同方案下臭氧干沉降通量及其与 Base 方案差值的空间分布

[Figure]

图 6.12: 不同方案下各下垫面上臭氧干沉降通量与 Base 方案差值

**6.3 气候变化情景下臭氧对水稻产量和经济损失的影响**

**6.3.1 气候变化情景下对 AOT40 时空分布的影响**

高浓度 $O_3$ 会破坏植物的新陈代谢[15]，促使叶片衰老[330]，降低作物的叶绿素含量和光合速率[246]，改变碳分配[331]，导致作物减产，尤其是对 $O_3$ 敏感的作物[157,290]。本小节以对 $O_3$ 敏感的农作物双季早稻为例，讨论未来气候变化下 $O_3$ 污染对双季早稻产量与经济损失的影响。

基于模式的模拟时段，本小节只探讨气候变化情景下对双季早稻产量的影响。图6.13为气候变化下 AOT40 的时空分布。Base 方案下双季早稻生长季 AOT40 的平均值为 3.36 ppm·h，其中湖北 (9.7 ppm·h)、安徽 (5.9 ppm·h)、江西 (4.7 ppm·h) 和湖南 (3.8 ppm·h) 省份的 AOT40 值较高。在不同 MET 方案下，气象变化导致了 AOT40 分别减少了 3.0% (MET6.0) 和 17.9% (MET4.5)。不同省份之间的差异较大，比如浙江、福建和广西在 MET4.5 下 AOT40 呈现增加的现象，而在 MET6.0 下则呈现下降的现象。不同 EI 方案下，排放源变化导致了 AOT40 分别变化了 52.9% (BAU)、-25.5% (ECP) 和-43.7% (BHE) (表6.4)。在 BAU 方案下，NOx、CO 和 NMVOCs 排放增加 (图6.3)，而在 ECP 和 BHE 方案下，NOx、CO 和 NMVOCs 排放下降明显。在光照及温度较高的季节，光化学反应对 $O_3$ 的生成中起着重要的作用，因此 $O_3$ 前体物排放的降低，会减少通过光化反应生成的 $O_3$[332-333]，进而导致了 AOT40 的降低。

整体而言，在气象和排放源共同影响下，到了 2060 年 AOT40 分别为 4.2 ppm·h (RCP6.0 BAU)、2.1 ppm·h (RCP4.5 ECP) 和 1.2 ppm·h (RCP4.5 BHE)，相对变化为 24.1%、-40.0%

和-65.7% (表6.4)。这意味着相较于 Base 方案，到了 2060 年，在 RCP4.5 ECP 和 RCP4.5 BHE 路径下，未来 $O_3$ 对作物的损伤将会减小，而在 RCP6.0 BAU 方案下，$O_3$ 对作物的损伤将会增加。

[Figure]

图 6.13: 气候变化下 AOT40 的空间分布

表 6.4: Base 方案下各省份 AOT40 统计及各 Case 相对于 Base 的差值与相对变化 (%)

| 省份 | Base | MET | | EI | | | RCP | | |
|------|------|-----|-----|-----|-----|-----|--------|--------|--------|
| | | 6.0 | 4.5 | BAU | ECP | BHE | 6.0BAU | 4.5ECP | 4.5BHE |
| 浙江 | 2.8 | 1.3 | 2.4 | 3.3 | 0.4 | -1.3 | 1.5 | 1.0 | -2.1 |
| | (%) | 46.4% | 87.5% | 117.8% | 15.6% | -47.9% | 55.1% | 36.1% | -74.7% |
| 安徽 | 5.9 | 2.1 | -0.2 | 6.7 | 2.0 | -2.3 | -0.5 | -1.2 | -2.8 |
| | (%) | 35.5% | -2.6% | 114.2% | 34.0% | -39.3% | -8.6% | -20.4% | -48.4% |
| 福建 | 1.9 | -0.5 | 0.4 | 1.0 | -1.0 | -1.1 | 0.9 | -0.4 | -1.6 |
| | (%) | -25.5% | 20.4% | 54.7% | -52.6% | -55.9% | 46.2% | -20.0% | -85.2% |
| 江西 | 4.7 | -1.9 | -1.5 | 0.9 | -2.8 | -2.4 | 2.2 | -2.8 | -3.6 |
| | (%) | -40.5% | -32.5% | 20.2% | -60.0% | -51.6% | 47.7% | -60.6% | -76.1% |
| 湖北 | 9.7 | -0.8 | -3.3 | 3.2 | -3.4 | -4.0 | 0.2 | -4.5 | -6.4 |
| | (%) | -8.3% | -33.9% | 32.9% | -34.9% | -41.4% | 2.5% | -46.7% | -66.2% |
| 湖南 | 3.8 | -0.5 | -0.7 | 1.9 | -2.2 | -2.3 | 1.5 | -1.8 | -3.2 |
| | (%) | -12.6% | -17.5% | 49.8% | -57.9% | -58.9% | 38.7% | -47.4% | -83.3% |
| 广东 | 2.0 | -0.8 | -1.5 | -0.3 | -1.5 | -0.4 | 0.4 | -1.4 | -1.2 |
| | (%) | -37.5% | -71.2% | -15.9% | -71.3% | -18.4% | 17.3% | -69.3% | -56.3% |
| 广西 | 0.6 | -0.1 | 0.1 | 0.6 | -0.2 | -0.5 | 1.2 | -0.3 | -0.5 |
| | (%) | -20.0% | 22.4% | 106.1% | -36.0% | -79.1% | 206.2% | -47.9% | -88.7% |
| 海南 | 1.1 | -0.3 | -1.0 | -0.2 | -0.3 | -0.4 | 0.6 | -1.1 | -0.4 |
| | (%) | -27.2% | -88.9% | -14.5% | -25.9% | -34.2% | 54.7% | -96.3% | -40.8% |
| 云南 | 1.2 | 0.7 | -0.9 | 0.6 | 0.4 | -0.1 | 0.1 | -1.0 | -0.3 |
| | (%) | 41.2% | -80.5% | 51.6% | 30.7% | -7.6% | 6.5% | -83.8% | -22.9% |
| 平均 | 3.4 | -0.1 | -0.6 | 1.8 | -0.9 | -1.5 | 0.8 | -1.3 | -2.2 |
| | (%) | -3.0% | -17.9% | 52.9% | -25.5% | -43.7% | 24.1% | -40.0% | -65.7% |

**6.3.2 气候变化情景下臭氧对双季早稻相对产量损失的影响**

根据表2.6中基于 AOT40 指标的响应方程，得到不同响应方程下气候变化引起的全国双季早稻相对产量损失的平均值，如图6.14所示。由图可以看出，基于 Wang 等[290]和耿春梅 等[291]的 AOT40 响应方程预测的水稻相对产量损失结果差别不大，但是略高于 Feng 等[289]的 AOT40 响应方程预测的结果，基于张继双 等[292]的 AOT40 响应方程预测的水稻相对产量损失结果最高。Feng 等[289]和张继双 等[292]计算出的水稻相对产量损失差异可达到 4 倍。由此可见，使用不同的 AOT40 响应方程预测作物的相对产量损失会存在一定的不确定性。

模式改进后 (Base)，相较于默认模式 (Org)，对双季早稻相对产量损失的预测提高了 19.5%。Base 方案下通过平均不同 AOT40 响应方程下的水稻相对产量损失可以看出，双季早稻的平均相对产量损失为 4.7%。

气候变化下使用不同响应方程预测的双季早稻相对产量损失的变化与 Base 方案一致。基于 Feng 等[289]的 AOT40 响应方程预测的结果最低，基于张继双 等[292]的 AOT40 相应方程预测的双季早稻相对产量损失结果最高。通过平均不同 AOT40 响应方程下的双季早稻相对产量损失可以看出，在 MET 方案下，全国双季早稻平均相对产量损失都略有下降，分别下降了 0.3% (MET6.0) 和 0.7% (MET4.5)；在 EI 方案下，BAU 路径全国双季早稻平均相对产量损失最高，约为 6.8%，相较于 Base 方案增加了 1.1%，而在 ECP 和 BHE 路径，全国双季早稻平均相对产量损失都有所下降，分别下降了为 1.0% 和 1.7%。

整体而言，到了 2060 年在气候变化情景下基于当前污染控制政策 (RCP6.0 BAU)，全国双季早稻平均相对产量损失为 5.7%，相较于 Base 方案增加了 0.9%；强化污染控制政策 (RCP4.5 ECP) 下，全国双季早稻平均相对产量损失为 3.1%，相较于 Base 方案减少了 1.6%；最佳污染控制政策 (RCP4.5 BHE) 下，全国双季早稻平均相对产量损失为 2.2%，相较于 Base 方案减少了 2.6%。

[Figure]

图 6.14: 基于不同响应方程的未来双季早稻相对产量损失

图6.15为全国各省份双季早稻相对产量损失的平均值。由图可以看出，Base 方案下湖北 (12.1%)、安徽 (7.6%) 和江西 (6.2%) 相对产量损失最高。广西 (1.9%) 和海南 (2.1%) 相对损失产量较低在 MET 方案下，浙江双季早稻相对损失产量相较于 Base 方案，分别增加了 37.5% (MET4.5) 和 70.7% (MET6.0)。安徽和云南在 MET6.0 路径下双季早稻相对损失产量分别增加了 31.8% 和 26.1%。福建和广西则是在 MET4.5 方案下相对损失产量增加，分别

达到 15.1% 和 10.6%。其余省份均呈现下降的趋势，下降幅度最高的省份为海南 (MET4.5，-55.3%)。相较于 Base 方案，绝大部分省份在 EI BAU 路径下双季早稻相对损失产量都增幅明显，其中安徽增幅达到了 102.6%。而广东和海南则略有减少，分别为-12.0% 和-9.0%。在 EI ECP 路径下，除浙江 (12.6%)、安徽 (30.5%) 和云南 (19.4%) 双季早稻相对损失产量增加，其余省份均呈现下降趋势，其中广东下降最多 (-53.8%)。在 EI BHE 路径下，各省份双季早稻相对损失产量均有不同程度的下降，其中湖南下降最多 (-50.2%)，云南下降最少 (-4.8%)。

总体而言，到了 2060 年在 RCP6.0 BAU 路径下，除安徽略有下降 (-7.7%) 以外，其余省份双季早稻相对损失产量均有所增加，其中广西增幅达到 98.0%。浙江在 RCP4.5 ECP 路径下双季早稻相对损失产量仍然上升，这意味着只有执行最严的的排放政策 (RCP4.5 BHE) 双季早稻相对损失产量才会下降 (-60.4%)。除浙江以外的其余省份在 RCP4.5 ECP 路径下，相对损失产量都开始下降，海南下降最多为-59.9%。在 RCP4.5 BHE 路径下，所有种植区双季早稻的相对损失产量都有所下降 (-71.0～-14.5%)。

[Figure]

图 6.15: 各省份未来双季早稻相对产量损失的平均值

表 6.5: Base 方案下各省份双季早稻相对产量损失统计及各 Case 相对于 Base 的差值与相对变化 (%)

| 省份 | Base | MET | | EI | | | RCP | | |
|---|---|---|---|---|---|---|---|---|---|
| | | 6.0 | 4.5 | BAU | ECP | BHE | 6.0BAU | 4.5ECP | 4.5BHE |
| 浙江 | 4.0 | 1.5 | 2.9 | 3.8 | 0.5 | -1.6 | 1.8 | 1.2 | -2.4 |
| | (%) | 37.5% | 70.7% | 95.2% | 12.6% | -38.7% | 44.5% | 29.2% | -60.4% |
| 安徽 | 7.6 | 2.4 | -0.2 | 7.8 | 2.3 | -2.7 | -0.6 | -1.4 | -3.3 |
| | (%) | 31.8% | -2.4% | 102.6% | 30.5% | -35.3% | -7.7% | -18.3% | -43.5% |
| 福建 | 3.0 | -0.6 | 0.5 | 1.2 | -1.2 | -1.2 | 1.0 | -0.4 | -1.9 |
| | (%) | -18.8% | 15.1% | 40.4% | -38.9% | -41.3% | 34.1% | -14.8% | -63.0% |
| 江西 | 6.2 | -2.2 | -1.8 | 1.1 | -3.3 | -2.8 | 2.6 | -3.3 | -4.2 |
| | (%) | -35.5% | -28.4% | 17.7% | -52.6% | -45.1% | 41.7% | -53.0% | -66.6% |
| 湖北 | 12.1 | -0.9 | -3.9 | 3.7 | -4.0 | -4.7 | 0.3 | -5.3 | -7.5 |
| | (%) | -7.8% | -31.8% | 30.8% | -32.6% | -38.8% | 2.4% | -43.7% | -61.9% |
| 湖南 | 5.2 | -0.6 | -0.8 | 2.2 | -2.6 | -2.6 | 1.7 | -2.1 | -3.7 |
| | (%) | -10.7% | -14.9% | 42.5% | -49.3% | -50.2% | 33.0% | -40.4% | -71.0% |
| 广东 | 3.2 | -0.9 | -1.7 | -0.4 | -1.7 | -0.4 | 0.4 | -1.7 | -1.3 |
| | (%) | -28.3% | -53.7% | -12.0% | -53.8% | -13.9% | 13.0% | -52.3% | -42.5% |
| 广西 | 1.5 | -0.1 | 0.2 | 0.7 | -0.3 | -0.6 | 1.5 | -0.3 | -0.6 |
| | (%) | -9.5% | 10.6% | 50.4% | -17.1% | -37.6% | 98.0% | -22.7% | -42.2% |
| 海南 | 2.1 | -0.4 | -1.1 | -0.2 | -0.3 | -0.4 | 0.7 | -1.2 | -0.5 |
| | (%) | -16.9% | -55.3% | -9.0% | -16.1% | -21.3% | 34.0% | -59.9% | -25.4% |
| 云南 | 2.1 | 0.6 | -1.1 | 0.7 | 0.4 | -0.1 | 0.1 | -1.1 | -0.3 |
| | (%) | 26.1% | -51.0% | 32.7% | 19.4% | -4.8% | 4.1% | -53.1% | -14.5% |
| 平均 | 4.7 | -0.1 | -0.6 | 1.8 | -0.9 | -1.5 | 0.8 | -1.3 | -2.2 |
| | (%) | -2.5% | -15.0% | 44.2% | -21.3% | -36.5% | 20.2% | -33.4% | -54.9% |

**6.3.3 气候变化情景下臭氧对双季早稻产量和经济损失的影响**

根据全国各省份双季早稻的平均相对产量损失，结合表2.5中各省份双季早稻的实际生产产量，利用公式2.20和2.21，基于水稻的最低市场收购价格即可算出因 $O_3$ 污染造成的水稻产量损失和相应经济损失。本小节使用表2.5中 2019 年双季早稻产量数据作为 2060 年双季早稻的产量，但随着未来粮食需求的增长，双季早稻产量可能会被低估，因此本小结只能对双季早稻产量损失及经济损失进行保守估计。由公式2.21可知作物的实际产量损失，不

仅取决于由于作物相对产量损失，还取决于作物的实际产量强度。因此，一些作物相对产量损失较低的省份，如果实际作物产量强度比较大，作物产量损失也可能很高；反之，一些作物相对产量损失较高的省份，如果作物实际产量强度比较小，其产量损失也不会很高。

从表6.6可以看出，当前因 $O_3$ 污染造成的双季早稻减产量为 126.8 万吨，相应的经济损失为 515.1 百万美元。其中双季早稻产量损失和经济损失最高的省份为江西 (41.7 万吨，16.9 百万美元)，其次是湖南 (36.6 万吨，14.9 百万美元)，广东 (15.9 万吨，64.7 百万美元)，损失最低的省份为云南 (0.47 万吨，0.19 百万美元)。

如图6.16所示，在 EI BAU 和 RCP6.0 BAU 两个路径下，双季早稻减产量最高，分别为 170.5 万吨和 169.7 万吨，相应的经济损失分别 692.3 百万美元和 641.3 百万美元。湖南和江西在不同方案下双季早稻减产量和相应的经济损失均是都是最高的。湖北的双季早稻产量是广西的五分之一 (表2.5)，但是减产量却相当，这表明湖北的 $O_3$ 污染比广西严重。

在 MET6.0 路径下，浙江、安徽和云南双季早稻减产量分别增加了 0.84 万吨、2.95 万吨和 0.13 万吨，相应的经济损失分别增加了 37.2 百万美元、12.0 百万美元和 0.5 百万美元。在 MET4.5 路径下，浙江、安徽和云南双季早稻减产量分别增加了 1.6 万吨、0.3 万吨和 0.7 万吨，相应的经济损失分别增加了 24.4 百万美元、1.2 百万美元和 3.0 百万美元。江西在两个 MET 方案下早稻减产量减少均是最多，分别为 15.4 万吨 (MET6.0) 和 12.4 万吨 (MET4.5)，经济损失减少了 62.5 百万美元和 50.3 百万美元。在 EI BAU 路径下，除了广东和海南双季早稻减产量有轻微减少 (-2.0 万吨，-0.1 万吨)，其余省份减产量均是增加，并导致全国双季早稻减产量增加了 43.6 万吨，相应的经济损失增加了 177.2 百万美元。在 EI ECP 路径下，浙江、安徽和云南双季早稻减产量略有增加，其他省份减产量均为减少。而 EI BHE 路径下，所有省份双季早稻减产量均为减少。同时，江西在这两个 EI 方案下减产量减少最多，分别为 22.6 万吨 (EI ECP) 和 19.5 万吨 (EI BHE)，经济损失减少了 91.6 百万美元和 79.1 百万美元。

整体而言，在气象和排放源共同影响下，预计到 2060 年在 RCP6.0 BAU 路径下，双季早稻减产量增加了 42.9 万吨，经济损失增加了 174.0 百万美元；在 RCP4.5 ECP 路径下，双季早稻减产量减少了 55.9 万吨，经济损失减少了 226.9 百万美元；在 RCP4.5 BHE 路径下，双季早稻减产量减少最多，为 78.8 万吨，经济损失减少了 319.9 百万美元。

[Figure]

图 6.16: 各省份未来水稻产量损失和经济损失情况

表 6.6: Base 方案下各省份双季早稻产量损失和经济损失情况统计及各 Case 相对于 Base 的差值

| 省份 | 类别 | Base | MET | | EI | | | RCP | | |
|------|------|------|------|------|------|------|------|--------|--------|--------|
| | | | 6.0 | 4.5 | BAU | ECP | BHE | 6.0BAU | 4.5ECP | 4.5BHE |
| 浙江 | CPL | 2.1 | 0.9 | 1.6 | 2.2 | 0.3 | -0.8 | 1.0 | 0.7 | -1.3 |
| | ECL | 8.6 | 3.4 | 6.5 | 8.9 | 1.1 | -3.4 | 4.1 | 2.6 | -5.3 |
| 安徽 | CPL | 8.3 | 3.0 | -0.2 | 10.1 | 2.8 | -3.1 | -0.7 | -1.6 | -3.8 |
| | ECL | 33.8 | 12.0 | -0.9 | 41.0 | 11.5 | -12.6 | -2.8 | -6.6 | -15.4 |
| 福建 | CPL | 1.9 | -0.4 | 0.3 | 0.8 | -0.8 | -0.8 | 0.7 | -0.3 | -1.2 |
| | ECL | 7.7 | -1.5 | 1.2 | 3.2 | -3.0 | -3.2 | 2.7 | -1.2 | -4.9 |
| 江西 | CPL | 41.7 | -15.4 | -12.4 | 8.0 | -22.6 | -19.5 | 19.1 | -22.8 | -28.3 |
| | ECL | 169.2 | -62.5 | -50.3 | 32.3 | -91.6 | -79.1 | 77.4 | -92.4 | -115.1 |
| 湖北 | CPL | 11.7 | -1.0 | -4.0 | 4.3 | -4.1 | -4.9 | 0.3 | -5.5 | -7.6 |
| | ECL | 47.3 | -4.1 | -16.4 | 17.3 | -16.8 | -19.8 | 1.3 | -22.2 | -30.7 |
| 湖南 | CPL | 36.6 | -4.1 | -5.7 | 16.8 | -18.5 | -18.9 | 13.0 | -15.2 | -26.4 |
| | ECL | 148.5 | -16.7 | -23.1 | 68.2 | -75.3 | -76.6 | 52.7 | -61.9 | -107.1 |
| 广东 | CPL | 15.9 | -4.6 | -8.7 | -1.96 | -8.7 | -2.27 | 2.15 | -8.46 | -6.89 |
| | ECL | 64.7 | -18.8 | -35.3 | -8.0 | -35.3 | -9.2 | 8.7 | -34.3 | -28.0 |
| 广西 | CPL | 6.8 | -0.7 | 0.7 | 3.5 | -1.2 | -2.6 | 6.9 | -1.6 | -2.9 |
| | ECL | 27.6 | -2.7 | 3.0 | 14.2 | -4.8 | -10.5 | 27.8 | -6.3 | -11.7 |
| 海南 | CPL | 1.5 | -0.3 | -0.8 | -0.1 | -0.2 | -0.3 | 0.5 | -0.9 | -0.4 |
| | ECL | 5.9 | -1.0 | -3.3 | -0.5 | -1.0 | -1.3 | 2.1 | -3.6 | -1.5 |
| 云南 | CPL | 0.5 | 0.1 | -0.2 | 0.2 | 0.1 | -0.02 | 0.02 | -0.3 | -0.1 |
| | ECL | 1.9 | 0.5 | -1.0 | 0.6 | 0.4 | -0.1 | 0.1 | -1.0 | -0.3 |
| 全国 | CPL | 126.8 | -22.5 | -29.4 | 43.7 | -52.9 | -53.1 | 42.9 | -55.9 | -78.8 |
| | ECL | 515.1 | -91.4 | -119.5 | 177.2 | -214.8 | -215.7 | 174.0 | -226.9 | -319.9 |

**6.4 本章小结**

本章基于改进的 WRF-Chem 模式，选用 2060 年碳中和背景下 DPEC 三个未来情景数据 (SSP4-60-BAU、SSP2-45-ECP 和 SSP2-45-BHE)，分别为当前污染控制政策、强化污染控制政策和最佳污染控制政策，选取 7 月为研究时段，设置了九组敏感性实验，考虑气象条件和人为排放源以及两个气候因子共同作用下我国臭氧干沉降格局的影响，进而预估在碳中和背景下未来 $O_3$ 胁迫对作物产量及经济损失的影响。

人为排放源是影响 $O_3$ 浓度生成的主控因子，不同排放情景下 (EI BAU、EI ECP、EI BHE) 人为排放源的变化导致了中国地区 $O_3$ 浓度平均下降了 2.4 $\mu g \cdot m^{-3}$ (3.0%)、4.0 $\mu g \cdot m^{-3}$ (5.2%) 和 7.8 $\mu g \cdot m^{-3}$ (10.0%)。不同气象条件下 (MET 6.0、MET4.5)，气象要素的改变，使中国地区 $O_3$ 浓度平均下降 0.8 $\mu g \cdot m^{-3}$ (0.5%) 和 1.3 $\mu g \cdot m^{-3}$ (2.6%)。在气候变化两个因子 (RCP6.0 BAU、RCP4.5 ECP、RCP4.5 ECP) 共同作用下，$O_3$ 浓度分别下降 2.8 $\mu g \cdot m^{-3}$ (2.1%)、4.6 $\mu g \cdot m^{-3}$ (4.5%) 和 8.7 $\mu g \cdot m^{-3}$ (12.0%)。

气象因素是影响 $V_d(O_3)$ 生成的主控因子，气象条件的改变导致了中国地区 $V_d(O_3)$ 平均上升了 0.02 $cm \cdot s^{-1}$ (4.1%) 和 0.004 $cm \cdot s^{-1}$ (1.2%)。人为排放源的改变，使得中国地区 $V_d(O_3)$ 平均变化了 -0.003 $cm \cdot s^{-1}$ (-0.3%)、0.001 $cm \cdot s^{-1}$ (0.1%) 和 0.001 $cm \cdot s^{-1}$ (0.1%)。在气候变化两个因子共同作用下 $V_d(O_3)$ 分别上升了 0.01 $cm \cdot s^{-1}$ (4.0%)、0.008 $cm \cdot s^{-1}$ (2.0%) 和 0.01 $cm \cdot s^{-1}$ (2.3%)。

与 $O_3$ 浓度的变化类似，人为排放源是导致中国地区 $O_3$ddep 变化的主控因子；整体而言，两个气候变化因子的共同作用使中国地区 $O_3$ddep 分别下降了 2.9%、6.5% 和 8.4%。

当前的地表 $O_3$ 浓度水平，$O_3$ 污染导致双季早稻的平均相对产量损失为了 4.7%，减产量为 126.38 万吨，相应的经济损失为 515.07 百万美元。到 2060 年，不同气候变化背景下全国双季早稻平均相对产量损失相较于当前 $O_3$ 浓度水平分别变化了 0.9% (RCP6.0 BAU)、-1.6% (RCP4.5 ECP) 和 -2.6% (RCP4.5 BHE)。双季早稻减产量分别变化了 42.9 万吨、-55.9 万吨和 -78.8 万吨；相应的经济损失变化分别为 174.0 百万美元、-226.9 百万美元和 -319.9 百万美元。未来如果继续执行当前的污染控制政策 (RCP6.0 BAU)，$O_3$ 污染会对作物造成更高的伤害和经济损失；执行最佳污染控制政策 (RCP4.5 BHE)，双季早稻的减产量会减少一半以上，经济损失也会大幅度减少。

**第七章 论文总结与展望**

**7.1 研究成果的总结**

本文以中国地区为研究区域,基于数值模拟的方法,结合典型亚热带森林和农田下垫面的观测资料,综合利用干沉降机理诊断模式 Noah-MP-WDDM 和区域空气质量模式 WRF-Chem,建立了 Ball-Berry 气孔导度机制与 $O_3$ 化学沉降模块的程序接口。同时,构建了 $O_3$ 损伤植被的参数化方案,将臭氧累积吸收通量对植被光合速率和叶片气孔导度的影响过程考虑进区域数值模式中,对区域模式中的 $O_3$ 干沉降机制进行了改进,实现了在 WRF-Chem 中 $O_3$ 与植被的双向反馈和 $O_3$ 累积作用的表征。在此基础上,探讨了改进机制对区域植被生理过程,气象要素以及臭氧干沉降过程的影响,以提升区域尺度 $O_3$ 干沉降的模拟效果。运用改进后的模式进一步分析和揭示了我国典型下垫面 $O_3$ 干沉降过程变化特征及未来气候变化背景下潜在的生态环境效应。取得的主要的研究结论如下:

1. 典型下垫面 $V_d(O_3)$ 的观测

典型亚热带森林下垫面冠层外和冠层内观测的日间 $V_d(O_3)$ 分别为 0.75 cm·s$^{-1}$ 和 0.30 cm·s$^{-1}$。农田下垫面观测的日间 $V_d(O_3)$ 为 0.45 cm·s$^{-1}$。尽管不同下垫面的 $V_d(O_3)$ 均呈现白天干沉降速率较高,夜晚较低的日内变化特征。但由于 $V_d(O_3)$ 的变化很大程度上受植被生理活性和局地气象要素的控制,因此不同下垫面的 $V_d(O_3)$ 存在大小差异。

2. 臭氧干沉降机制的评估与改进

通过单点干沉降模式的模拟试验发现,不同气孔导度机制是短期内影响 $V_d(O_3)$ 日变化过程的关键因素。在典型亚热带森林下垫面中,Jarvis 气孔导度机制估算的 $g_s$ 呈双峰型变化,而 Ball-Berry 气孔机制估算的 $g_s$ 呈单峰型趋势;在农田下垫面中,两种气孔导度机制估算的 $g_s$ 日变化趋势基本一致。Ball-Berry 气孔导度机制对典型亚热带森林和农田下垫面的 $V_d(O_3)$ 模拟效果均优于 Jarvis 气孔导度机制。此外,$R_c$ 是影响白天 $V_d(O_3)$ 变化的主导因子,夜间则是由 $R_a$ 引起的湍流混合作用主导。

在区域模式模拟方面,基于单点模式上的评估,在 WRF-Chem 模式中开发了 Ball-Berry 气孔导度机制由物理模块中陆面模式 Noah-MP 输出到 $O_3$ 化学沉降模块的程序接口,使得 Wesely 干沉降机制可灵活调用 Ball-Berry 气孔导度机制进行气孔导度的计算。同时,构建了 $O_3$ 损伤植被的参数化方案,将臭氧累积吸收通量对植被光合速率和叶片气孔导度的影

响过程考虑进区域数值模式中，实现了在 WRF-Chem 中 $O_3$ 与植被的双向反馈和 $O_3$ 累积作用的表征。基于改进的区域大气化学传输模式 WRF-Chem 对我国 $O_3$ 干沉降过程开展模拟。结果表明，模式改进后对典型亚热带森林和农田下垫面上 $O_3$ 浓度的模拟精度分别提高了 10.0% 和 8.8%；对 $V_d(O_3)$ 模拟精度分别提高了 51.9% 和 28.6%。对于 $O_3$ddep，其在农田下垫面的改进效果不明显，仅提升了 0.3%；而对典型亚热带森林下垫面，模式改进后 $O_3$ddep 模拟精度显著提高了 16.8%。此外，模式改进还引起区域植被生理过程和气象要素的一系列改变，其中使气孔阻抗平均增加了 5.6%，光合速率降低了 6.4%。气孔阻抗和光合速率的变化进一步导致叶面积指数和总初级生产力平均降低了 4.1% 和 9.5%。蒸腾速率的降低导致潜热通量平均下降了 6.3%，显热通量增加了 8.5%，进而使 2m 温度升高，并造成了异戊二烯排放的增加。植被生理过程和气象要素的变化促使区域 $O_3$ 浓度平均上升了 3.2 $\mu g \cdot m^{-3}$，$V_d(O_3)$ 平均下降了 0.06 $cm \cdot s^{-1}$，$O_3$ddep 平均下降了 0.9 $kg \cdot ha^{-1} \cdot month^{-1}$。

3. 典型下垫面臭氧干沉降过程影响因子的量化

2019 年中国 $O_3$ 干沉降过程的模拟结果表明，农田、森林、草地和城市下垫面 $O_3$ 沉降通量分别为 596.3 $kg \cdot km^{-2} \cdot month^{-1}$、555.7 $kg \cdot km^{-2} \cdot month^{-1}$、528.9 $kg \cdot km^{-2} \cdot month^{-1}$ 和 323.9 $kg \cdot km^{-2} \cdot month^{-1}$。基于过程诊断，本研究量化了 $O_3$ 浓度和 $V_d(O_3)$ 对 $O_3$ddep 的相对贡献，并进一步识别了影响 $O_3$ 浓度和 $V_d(O_3)$ 的关键大气过程。分析发现，$O_3$ 浓度和 $V_d(O_3)$ 对 $O_3$ddep 的相对贡献分别为 34.4% 和 63.8%。其中，影响 $O_3$ 浓度变化的关键过程存在显著的昼夜差异，白天主要由垂直混合、化学和干沉降过程主导，贡献率分别为 33.6%，29.7% 和 19.5%，而夜间 $O_3$ 变化主要是化学过程的贡献，占比达到 45.6%。对于 $V_d(O_3)$，$R_a$ 和 $R_c$ 主导了其日变化，相对贡献分别为 53.2% 和 43.4%。

4. 臭氧干沉降过程对气候变化的响应及气候变化下作物产量及经济损失评估

基于控制试验探究了当前与未来气候变化情景下 (RCP6.0 BAU，RCP4.5 ECP 和 RCP4.5 BHE) 人为排放和气象要素对 $O_3$ 干沉降过程的影响。结果表明，人为排放是影响 $O_3$ 浓度和 $O_3$ddep 变化的主控因子，其次是气象要素的改变，在人为排放和气象变化的共同作用下 $O_3$ 浓度分别下降 3.7%、5.8% 和 11.6%，$O_3$ddep 的分别下降了为 2.9%、6.5% 和 8.4%。而气象要素则是影响 $V_d(O_3)$ 的主控因子，其与人为排放变化的共同作用导致 $V_d(O_3)$ 分别上升了 4.3%，2.0% 和 2.3%。

当前地表 $O_3$ 污染导致双季早稻的减产量达到 126.4 万吨，并造成 515.1 百万美元的经济损失。预计到 2060 年，如果继续执行当前的污染控制政策 (RCP6.0 BAU)，$O_3$ 污染会对作物造成更高的减产量 (169.3 万吨) 和经济损失 (689.1 百万美元)；执行最佳污染控制政策 (RCP4.5 BHE)，双季早稻的减产量 (47.6 万吨) 会减少一半以上，经济损失 (195.2 百万美元) 也会大幅度减少。

**7.2 研究主要创新点**

本研究的主要创新点主要有以下几点：

1. 区域模式 WRF-Chem 中实现了臭氧-植被的双向反馈，弥补了区域大气化学传输模式中未考虑臭氧损伤植被的缺陷，使得对臭氧浓度的模拟偏差由 31.5% 降低至 19.3%，臭氧干沉降速率的模拟偏差由 57.1% 降低至 16.8%，臭氧干沉降通量的模拟偏差由 37.6% 降低至 21.3%。

2. 定量评估了区域尺度臭氧与植被的相互反馈机制对植被生理过程、气象和臭氧干沉降过程的影响，厘清了典型下垫面影响区域臭氧干沉降过程的关键因子。

3. 基于改进模型，提升了对作物相对产量损失的预测能力，为区域上尺度评估臭氧的生态风险评估提供了一种更为可靠的新方法，并为今后的区域大气污染对作物影响的评估系统的建立提供技术基础。

**7.3 研究不足与展望**

尽管本研究对 WRF-Chem 模式的臭氧干沉降机制进行了改进，量化了对植被生理过程，气象和臭氧干沉降过程的影响，进一步分析和揭示了我国典型下垫面臭氧干沉降过程及未来气候变化背景下潜在的生态环境效应。但是本研究的工作依然存在一些问题和不足之处，主要包括以下三个方面：

1. 在观测技术方面，缺乏全国范围内不同下垫面臭氧干沉降速率观测数据，难以对区域模式模拟效果进行综合评估；

2. 在研究方法方面，开展的观测时间较短，且不同下垫面的观测时间不统一，缺乏不同季节的观测数据，可能会导致本研究构建的臭氧干沉降机制输入参数普适性不足；

3. 在科学问题方面，本研究对臭氧干沉降机制的非气孔阻抗的改进考虑不全面，未能在区域尺度上定量区分气孔和非气孔阻抗对臭氧干沉降的影响。

针对以上不足，未来关于臭氧干沉降过程的研究可在以下几个方面进行研究：

1. 收集更多的臭氧干沉降的观测资料对模拟结果进行评估；

2. 开展不同下垫面臭氧干沉降过程的长期观测，为构建臭氧干沉降机制提供更全面的输入参数；

3. 针对臭氧干沉降机制的非气孔阻抗进行改进，进而在区域尺度上定量区分气孔和非气孔阻抗对臭氧干沉降格局的影响，完善模式对臭氧干沉降的模拟性能。

**参考文献**

[1] D. D. Parrish, K. S. Law, J. Staehelin, et al. Lower tropospheric ozone at northern midlatitudes: Changing seasonal cycle[J]. Geophysical Research Letters, 2013, 40(8): 1631-1636.

[2] IPCC. Climate change 2007 - the physical science basis[M]. Cambridge University Press, 2007.

[3] D. Fowler, M. Amann, R. Anderson, et al. Ground-level ozone in the 21st century: future trends, impacts and policy implications[M]. The Royal Society, 2008.

[4] H. Liu, S. Liu, B. Xue, et al. Ground-level ozone pollution and its health impacts in China [J]. Atmospheric Environment, 2018, 173: 223-230.

[5] 中国大气臭氧污染防治蓝皮书 (2020 年)[M]. 2020.

[6] J. Lelieveld, F. J. Dentener. What controls tropospheric ozone?[J]. Journal of Geophysical Research: Atmospheres, 2000, 105(D3): 3531-3551.

[7] P. S. Monks, A. T. Archibald, A. Colette, et al. Tropospheric ozone and its precursors from the urban to the global scale from air quality to short-lived climate forcer[J]. Atmospheric Chemistry and Physics, 2015, 15(15): 8889-8973.

[8] M. Coyle, E. Nemitz, R. Storeton-West, et al. Measurements of ozone deposition to a potato canopy[J]. Agricultural and Forest Meteorology, 2009, 149(3): 655-666.

[9] M. Lin, L. W. Horowitz, R. Payton, et al. US surface ozone trends and extremes from 1980 to 2014: quantifying the rolesof rising Asian emissions, domestic controls, wildfires, and climate[J]. Atmospheric Chemistry and Physics, 2017, 17(4): 2943-2970.

[10] J. J. Guo, A. M. Fiore, L. T. Murray, et al. Average versus high surface ozone levels over the continental USA: model bias, background influences, and interannual variability[J]. Atmospheric Chemistry and Physics, 2018, 18(16): 12123-12140.

[11] T. R. Loveland, B. C. Reed, J. F. Brown, et al. Development of a global land cover characteristics database and IGBP DISCover from 1 km AVHRR data[J]. International Journal of Remote Sensing, 2000, 21(6-7): 1303-1330.

[12] P. J. Young, A. T. Archibald, K. W. Bowman, et al. Pre-industrial to end 21st century projections of tropospheric ozone from the Atmospheric Chemistry and Climate Model

Intercomparison Project (ACCMIP)[J]. Atmospheric Chemistry and Physics, 2013, 13(4): 2063-2090.

[13] D. S. Stevenson, F. J. Dentener, M. G. Schultz, et al. Multimodel ensemble simulations of present-day and near-future tropospheric ozone[J]. Journal of Geophysical Research: Atmospheres, 2006, 111(D8).

[14] E. L. Fiscus, F. L. Booker, K. O. Burkey. Crop responses to ozone: uptake, modes of action, carbon assimilation and partitioning[J]. Plant, Cell & Environment, 2005, 28(8): 997-1011.

[15] E. A. Ainsworth, C. R. Yendrek, S. Sitch, et al. The Effects of Tropospheric Ozone on Net Primary Productivity and Implications for Climate Change[J]. Annual Review of Plant Biology, 2012, 63(1): 637-661.

[16] Z. Feng, J. Sun, W. Wan, et al. Evidence of widespread ozone-induced visible injury on plants in Beijing, China[J]. Environmental Pollution, 2014, 193: 296-301.

[17] K. Vandermeiren, M. D. Bock, N. Horemans, et al. Ozone effects on yield quality of spring oilseed rape and broccoli[J]. Atmospheric Environment, 2012, 47: 76-83.

[18] G. Mills, F. Hayes, D. Simpson, et al. Evidence of widespread effects of ozone on crops and (semi-) natural vegetation in Europe (1990-2006) in relation to AOT40-and flux-based risk maps[J]. Global Change Biology, 2011, 17(1): 592-613.

[19] 曹嘉晨, 郑有飞, 赵辉, 等. 地表臭氧浓度升高对冬小麦和大豆生长和产量的影响[J]. 生态毒理学报, 2017, 12(2): 129-136.

[20] 徐静馨, 郑有飞, 赵辉, 等. 陆地生态系统臭氧干沉降的观测和模拟研究进展[J]. 生态毒理学报, 2017, 12(6): 57-68.

[21] X. Yue, N. Unger. Ozone vegetation damage effects on gross primary productivity in the United States[J]. Atmospheric Chemistry and Physics, 2014, 14(17): 9137-9153.

[22] Y. Hoshika, G. Katata, M. Deushi, et al. Ozone-induced stomatal sluggishness changes carbon and water balance of temperate deciduous forests[J]. Scientific Reports, 2015, 5(1): 1-8.

[23] D. Lombardozzi, S. Levis, G. Bonan, et al. The influence of chronic ozone exposure on global carbon and water cycles[J]. Journal of Climate, 2015, 28(1): 292-305.

[24] M. Franz, D. Simpson, A. Arneth, et al. Development and evaluation of an ozone deposition scheme for coupling to a terrestrial biosphere model[J]. Biogeosciences, 2017, 14(1): 45-71.

[25]  M. Sadiq, A. P. K. Tai, D. Lombardozzi, et al. Effects of ozone-vegetation coupling on surface ozone air quality via biogeochemical and meteorological feedbacks[J]. Atmospheric Chemistry and Physics, 2017, 17(4): 3055-3066.

[26]  S. R. Arnold, D. Lombardozzi, J F. Lamarque, et al. Simulated Global Climate Response to Tropospheric Ozone-Induced Changes in Plant Transpiration[J]. Geophysical Research Letters, 2018, 45(23): 13070-13079.

[27]  R. J. Oliver, L. M. Mercado, S. Sitch, et al. Large but decreasing effect of ozone on the European carbon sink[J]. Biogeosciences, 2018, 15(13): 4245-4269.

[28]  I. Super, J. Vilà-Guerau de Arellano, M. C. Krol. Cumulative ozone effect on canopy stomatal resistance and the impact on boundary layer dynamics and $CO_2$ assimilation at the diurnal scale: A case study for grassland in the Netherlands[J]. Journal of Geophysical Research: Biogeosciences, 2015, 120(7): 1348-1365.

[29]  J. Li, A. Mahalov, P. Hyde. Simulating the impacts of chronic ozone exposure on plant conductance and photosynthesis, and on the regional hydroclimate using WRF/Chem[J]. Environmental Research Letters, 2016, 11(11): 114017.

[30]  J. Li, A. Mahalov, P. Hyde. Simulating the effects of chronic ozone exposure on hydrometeorology and crop productivity using a fully coupled crop, meteorology and air quality modeling system[J]. Agricultural and Forest Meteorology, 2018, 260-261: 287-299.

[31]  S. Sitch, P. Cox, W. Collins, et al. Indirect radiative forcing of climate change through ozone effects on the land-carbon sink[J]. Nature, 2007, 448(7155): 791-794.

[32]  M. M. Kvalevåg, G. Myhre. The effect of carbon-nitrogen coupling on the reduced land carbon sink caused by tropospheric ozone[J]. Geophysical Research Letters, 2013, 40(12): 3227-3231.

[33]  S. S. Zhou, A. P. Tai, S. Sun, et al. Coupling between surface ozone and leaf area index in a chemical transport model: strength of feedback and implications for ozone air quality and vegetation health[J]. Atmospheric Chemistry and Physics, 2018, 18(19): 14133-14148.

[34]  O. E. Clifton, A. M. Fiore, W. J. Massman, et al. Dry Deposition of Ozone Over Land: Processes, Measurement, and Modeling[J]. Reviews of Geophysics, 2020, 58(1): e2019RG000670.

[35]  E. Potier, B. Loubet, B. Durand, et al. Chemical reaction rates of ozone in water infusions of wheat, beech, oak and pine leaves of different ages[J]. Atmospheric Environment, 2017,

151: 176-187.

[36]  D. Fowler, C. Flechard, J. N. Cape, et al. Measurements of ozone deposition to vegetation quantifying the flux, the stomatal and non-stomatal components[J]. Water, Air, and Soil Pollution, 2001, 130(1): 63-74.

[37]  W. Massman. Toward an ozone standard to protect vegetation based on effective dose: a review of deposition resistances and a possible metric[J]. Atmospheric Environment, 2004, 38(15): 2323-2337.

[38]  D. Fowler, K. Pilegaard, M. Sutton, et al. Atmospheric composition change: Ecosystems‐Atmosphere interactions[J]. Atmospheric Environment, 2009, 43(33): 5193-5267.

[39]  S. Fares, M. McKay, R. Holzinger, et al. Ozone fluxes in a Pinus ponderosa ecosystem are dominated by non-stomatal processes: Evidence from long-term continuous measurements [J]. Agricultural and Forest Meteorology, 2010, 150(3): 420-431.

[40]  O. E. Clifton, A. M. Fiore, J. Munger, et al. Spatiotemporal controls on observed daytime ozone deposition velocity over northeastern US forests during summer[J]. Journal of Geophysical Research: Atmospheres, 2019, 124(10): 5612-5628.

[41]  Ü. Rannik, N. Altimir, I. Mammarella, et al. Ozone deposition into a boreal forest over a decade of observations: evaluating deposition partitioning and driving variables[J]. Atmospheric Chemistry and Physics, 2012, 12(24): 12165-12182.

[42]  L. P. Wright, L. Zhang, F. J. Marsik. Overview of mercury dry deposition, litterfall, and throughfall studies[J]. Atmospheric Chemistry and Physics, 2016, 16(21): 13399-13416.

[43]  S. M. Mohan. An overview of particulate dry deposition: measuring methods, deposition velocity and controlling factors[J]. International Journal of Environmental Science and Technology, 2016, 13(1): 387-402.

[44]  M. Wesely, B. Lesht. Comparison of RADM dry deposition algorithms with a site-specific method for inferring dry deposition[J]. Water, Air, and Soil Pollution, 1989, 44(3): 273-293.

[45]  S. J. Silva, C. L. Heald. Investigating Dry Deposition of Ozone to Vegetation[J]. Journal of Geophysical Research: Atmospheres, 2018, 123(1): 559-573.

[46]  M. Val Martin, C. L. Heald, S. R. Arnold. Coupling dry deposition to vegetation phenology in the Community Earth System Model: Implications for the simulation of surface $O_3$[J]. Geophysical Research Letters, 2014, 41(8): 2988-2996.

[47] M. Wesely, B. Hicks. A review of the current status of knowledge on dry deposition[J]. Atmospheric Environment, 2000, 34(12): 2261-2282.

[48] L. Zhang, J. R. Brook, R. Vet. A revised parameterization for gaseous dry deposition in air-quality models[J]. Atmospheric Chemistry and Physics, 2003, 3(6): 2067-2082.

[49] U. Charusombat, D. Niyogi, A. Kumar, et al. Evaluating a new deposition velocity module in the Noah land-surface model[J]. Boundary-Layer Meteorology, 2010, 137(2): 271-290.

[50] J. Pleim, L. Ran. Surface Flux Modeling for Air Quality Applications[J]. Atmosphere, 2011, 2(3): 271-302.

[51] M. J. Hollaway, S. R. Arnold, W. J. Collins, et al. Sensitivity of midnineteenth century tropospheric ozone to atmospheric chemistry-vegetation interactions[J]. Journal of Geophysical Research: Atmospheres, 2016, 122(4): 2452-2473.

[52] L. Ran, J. Pleim, C. Song, et al. A photosynthesis-based two-leaf canopy stomatal conductance model for meteorology and air quality modeling with WRF/CMAQ PX LSM[J]. Journal of Geophysical Research: Atmospheres, 2017, 122(3): 1930-1952.

[53] D. Helmig, J. Ortega, T. Duhl, et al. Sesquiterpene Emissions from Pine Trees − Identifications, Emission Rates and Flux Estimates for the Contiguous United States[J]. Environmental Science & Technology, 2007, 41(5): 1545-1553.

[54] O. Wild. Modelling the global tropospheric ozone budget: exploring the variability in current models[J]. Atmospheric Chemistry and Physics, 2007, 7(10): 2643-2660.

[55] M. Vieno, A. J. Dore, D. S. Stevenson, et al. Modelling surface ozone during the 2003 heat-wave in the UK[J]. Atmospheric Chemistry and Physics, 2010, 10(16): 7963-7978.

[56] L. D. Emberson, N. Kitwiroon, S. Beevers, et al. Scorched Earth: how will changes in the strength of the vegetation sink to ozone deposition affect human health and ecosystems? [J]. Atmospheric Chemistry and Physics, 2013, 13(14): 6741-6755.

[57] M. M. Bela, K. M. Longo, S. R. Freitas, et al. Ozone production and transport over the Amazon Basin during the dry-to-wet and wet-to-dry transition seasons[J]. Atmospheric Chemistry and Physics, 2015, 15(2): 757-782.

[58] L. Huang, E. C. McDonald-Buller, G. McGaughey, et al. The impact of drought on ozone dry deposition over eastern Texas[J]. Atmospheric Environment, 2016, 127: 176-186.

[59] A. V. Beddows, N. Kitwiroon, M. L. Williams, et al. Emulation and sensitivity analysis of

the community multiscale air quality model for a UK ozone pollution episode[J]. Environmental Science & Technology, 2017, 51(11): 6229-6236.

[60] A. Anav, C. Proietti, L. Menut, et al. Sensitivity of stomatal conductance to soil moisture: implications fortropospheric ozone[J]. Atmospheric Chemistry and Physics, 2018, 18(8): 5747-5763.

[61] P. C. Campbell, J. O. Bash, T. L. Spero. Updates to the Noah Land Surface Model in WRF-CMAQ to Improve Simulated Meteorology, Air Quality, and Deposition[J]. Journal of Advances in Modeling Earth Systems, 2019, 11(1): 231-256.

[62] S. Falk, A. Søvde Haslerud. Update and evaluation of the ozone dry deposition in Oslo CTM3 v1.0[J]. Geoscientific Model Development, 2019, 12(11): 4705-4728.

[63] A. Y. H. Wong, J. A. Geddes, A. P. K. Tai, et al. Importance of dry deposition parameterization choice in global simulationsof surface ozone[J]. Atmospheric Chemistry and Physics, 2019, 19(22): 14365-14385.

[64] M. Lin, S. Malyshev, E. Shevliakova, et al. Sensitivity of Ozone Dry Deposition to Ecosystem-Atmosphere Interactions: A Critical Appraisal of Observations and Simulations [J]. Global Biogeochemical Cycles, 2019, 33(10): 1264-1288.

[65] C. Hogrefe, P. Liu, G. Pouliot, et al. Impacts of different characterizations of large-scale background on simulated regional-scale ozone over the continental United States[J]. Atmospheric Chemistry and Physics, 2018, 18(5): 3839-3864.

[66] S. C. Kavassalis, J. G. Murphy. Understanding ozone-meteorology correlations: A role for dry deposition[J]. Geophysical Research Letters, 2017, 44(6): 2922-2931.

[67] S. J. Silva, C. L. Heald, S. Ravela, et al. A Deep Learning Parameterization for Ozone Dry Deposition Velocities[J]. Geophysical Research Letters, 2019, 46(2): 983-989.

[68] K. R. Travis, D. J. Jacob. Systematic bias in evaluating chemical transport models with maximum daily8 h average (MDA8) surface ozone for air quality applications: a casestudy with GEOS-Chem v9.02[J]. Geoscientific Model Development, 2019, 12(8): 3641-3648.

[69] C. R. Flechard, E. Nemitz, R. I. Smith, et al. Dry deposition of reactive nitrogen to European ecosystems: a comparison of inferential models across the NitroEurope network[J]. Atmospheric Chemistry and Physics, 2011, 11(6): 2703-2728.

[70] Z. Wu, X. Wang, F. Chen, et al. Evaluating the calculated dry deposition velocities of reactive nitrogen oxides and ozone from two community models over a temperate deciduous

forest[J]. Atmospheric Environment, 2011, 45(16): 2663-2674.

[71] Z. Wu, X. Wang, A. A. Turnipseed, et al. Evaluation and improvements of two commu-nity models in simulating dry deposition velocities for peroxyacetyl nitrate (PAN) over a coniferous forest[J]. Journal of Geophysical Research: Atmospheres, 2012, 117(D4).

[72] C. Hardacre, O. Wild, L. Emberson. An evaluation of ozone dry deposition in global scale chemistry climate models[J]. Atmospheric Chemistry and Physics, 2015, 15(11): 6419-6436.

[73] D. Helmig, E. K. Lang, L. Bariteau, et al. Atmosphere-ocean ozone fluxes during the Tex-AQS 2006, STRATUS 2006, GOMECC 2007, GasEx 2008, and AMMA 2008 cruises[J]. Journal of Geophysical Research: Atmospheres, 2012, 117(D4).

[74] G. M. Wolfe, T. F. Hanisco, H. L. Arkinson, et al. Quantifying sources and sinks of reactive gases in the lower atmosphere using airborne flux observations[J]. Geophysical Research Letters, 2015, 42(19): 8231-8240.

[75] L. Tong, X. Wang, C. Geng, et al. Diurnal and phenological variations of $O_3$ and $CO_2$ fluxes of rice canopy exposed to different $O_3$ concentrations[J]. Atmospheric Environment, 2011, 45(31): 5621-5631.

[76] N. Altimir, T. Vesala, P. Keronen, et al. Methodology for direct field measurements of ozone flux to foliage with shoot chambers[J]. Atmospheric Environment, 2002, 36(1): 19-29.

[77] B. Almand-Hunter, J. Walker, N. Masson, et al. Development and validation of inexpen-sive, automated, dynamic flux chambers[J]. Atmospheric Measurement Techniques, 2015, 8(1): 267-280.

[78] I. Fumagalli, C. Gruening, R. Marzuoli, et al. Long-term measurements of NOx and $O_3$ soil fluxes in a temperate deciduous forest[J]. Agricultural and Forest Meteorology, 2016, 228: 205-216.

[79] P. Hari, E. Korpilahti, T. Pohja, et al. A field system for measuring the gas exchange of forest trees[J]. Silva Fennica, 1990, 24(1): 21-27.

[80] J. Fuentes, G. Den Hartog, H. Neumann, et al. Measurements and modelling of ozone de-position to wet foliage[G]. in: Air pollutants and the leaf cuticle. Springer, 1994: 239-253.

[81] L. Granat, A. Richter. Dry deposition to pine of sulphur dioxide and ozone at low concen-tration[J]. Atmospheric Environment, 1995, 29(14): 1677-1683.

[82] 袁月. 裸土臭氧干沉降通量观测与模拟[D]. 南京: 南京信息工程大学, 2017.

[83] A. Gut, S. Van Dijk, M. Scheibe, et al. NO emission from an Amazonian rain forest soil: Continuous measurements of NO flux and soil concentration[J]. Journal of Geophysical Research: Atmospheres, 2002, 107(D20): LBA-24.

[84] L. Horváth, E. Führer, K. Lajtha. Nitric oxide and nitrous oxide emission from Hungarian forest soils; linked with atmospheric N-deposition[J]. Atmospheric Environment, 2006, 40(40): 7786-7795.

[85] S. B. Verma, D. D. Baldocchi, D. E. Anderson, et al. Eddy fluxes of $CO_2$, water vapor, and sensible heat over a deciduous forest[J]. Boundary-Layer Meteorology, 1986, 36(1): 71-91.

[86] D. D. Baldocchi. Assessing the eddy covariance technique for evaluating carbon dioxide exchange rates of ecosystems: past, present and future[J]. Global Change Biology, 2003, 9(4): 479-492.

[87] 徐亚彬, 宋博, 任妙春, 等. 长白山森林生态系统二氧化碳通量与涡动相关研究[J]. 北京农业, 2012, 27: 100-101.

[88] 张海宏, 李林, 周秉荣, 等. 青藏高原高寒湿地 $CO_2$ 通量特征及影响因子分析[J]. 冰川冻土, 2017, 39(1): 54-60.

[89] D. Baldocchi. A multi-layer model for estimating sulfur dioxide deposition to a deciduous oak forest canopy[J]. Atmospheric Environment, 1988, 22(5): 869-884.

[90] J. W. Erisman, G. P. Draaijers. Atmospheric deposition: in relation to acidification and eutrophication[M]. Elsevier, 1995.

[91] Z. Wu, L. Zhang, X. Wang, et al. A modified micrometeorological gradient method for estimating $O_3$ dry depositions over a forest canopy[J]. Atmospheric Chemistry and Physics, 2015, 15(13): 7487-7496.

[92] J C. Mayer, A. Bargsten, U. Rummel, et al. Distributed Modified Bowen Ratio method for surface layer fluxes of reactive and non-reactive trace gases[J]. Agricultural and forest meteorology, 2011, 151(6): 655-668.

[93] S. Toet, J A. Subke, D. D'Haese, et al. A new stable isotope approach identifies the fate of ozone in plant-soil systems[J]. New Phytologist, 2009, 182(1): 85-90.

[94] J A. Subke, S. Toet, D. D'Haese, et al. A new method for using $^{18}O$ to trace ozone deposition [J]. Rapid Communications in Mass Spectrometry, 2009, 23(7): 980-984.

[95] T. P. Meyers, D. D. Baldocchi. Current micrometeorological flux methodologies with applications in agriculture[J]. Micrometeorology in Agricultural Systems, 2005, 47: 381-396.

[96] Z. Wu, D. B. Schwede, R. Vet, et al. Evaluation and intercomparison of five North American dry deposition algorithms at a mixed forest site[J]. Journal of Advances in Modeling Earth Systems, 2018, 10(7): 1571-1586.

[97] S M. Fan, S. C. Wofsy, P. S. Bakwin, et al. Atmosphere-biosphere exchange of $CO_2$ and $O_3$ in the central Amazon Forest[J]. Journal of Geophysical Research: Atmospheres, 1990, 95(D10): 16851-16864.

[98] U. Rummel, C. Ammann, G. Kirkman, et al. Seasonal variation of ozone deposition to a tropical rain forest in southwest Amazonia[J]. Atmospheric Chemistry and Physics, 2007, 7(20): 5415-5435.

[99] S. Cieslik. Ozone fluxes over various plant ecosystems in Italy: A review[J]. Environmental Pollution, 2009, 157(5): 1487-1496.

[100] S. Fares, R. Weber, J H. Park, et al. Ozone deposition to an orange orchard: Partitioning between stomatal and non-stomatal sinks[J]. Environmental Pollution, 2012, 169: 258-266.

[101] G. Gerosa, M. Vitale, A. Finco, et al. Ozone uptake by an evergreen Mediterranean Forest (Quercus ilex) in Italy. Part I: Micrometeorological flux measurements and flux partitioning [J]. Atmospheric Environment, 2005, 39(18): 3255-3266.

[102] A. A. Turnipseed, S. P. Burns, D. J. Moore, et al. Controls over ozone deposition to a high elevation subalpine forest[J]. Agricultural and Forest Meteorology, 2009, 149(9): 1447-1459.

[103] P. G. Jarvis, J. L. Monteith, P. E. Weatherley. The interpretation of the variations in leaf water potential and stomatal conductance found in canopies in the field[J]. Philosophical Transactions of the Royal Society of London. B, Biological Sciences, 1976, 273(927): 593-610.

[104] 朱治林, 孙晓敏, 董云社, 等. 鲁西北平原玉米地涡度相关臭氧通量日变化特征[J]. 中国科学: 地球科学, 2014, 44(2): 292-301.

[105] 朱治林, 孙晓敏, 于贵瑞, 等. 陆地生态系统臭氧通量观测和气孔吸收估算研究进展[J]. 生态学报, 2014, 34(21): 10.

[106] G. Gerosa, R. Marzuoli, S. Cieslik, et al. Stomatal ozone fluxes over a barley field in Italy. "Effective exposure" as a possible link between exposure- and flux-based approaches[J].

Atmospheric Environment, 2004, 38(15): 2421-2432.

[107]  E. Lamaud, B. Loubet, M. Irvine, et al. Partitioning of ozone deposition over a developed maize crop between stomatal and non-stomatal uptakes, using eddy-covariance flux measurements and modelling[J]. Agricultural and Forest Meteorology, 2009, 149(9): 1385-1396.

[108]  K. Matsuda, I. Watanabe, V. Wingpud, et al. Ozone dry deposition above a tropical forest in the dry season in northern Thailand[J]. Atmospheric Environment, 2005, 39(14): 2571-2577.

[109]  K. Matsuda, I. Watanabe, V. Wingpud, et al. Deposition velocity of $O_3$ and $SO_2$ in the dry and wet season above a tropical forest in northern Thailand[J]. Atmospheric Environment, 2006, 40(39): 7557-7564.

[110]  D. Fowler, E. Nemitz, P. Misztal, et al. Effects of land use on surface-atmosphere exchanges of trace gases and energy in Borneo: comparing fluxes over oil palm plantations and a rainforest[J]. Philosophical Transactions of the Royal Society B: Biological Sciences, 2011, 366(1582): 3196-3209.

[111]  M. Kurpius, M. McKay, A. Goldstein. Annual ozone deposition to a Sierra Nevada ponderosa pine plantation[J]. Atmospheric Environment, 2002, 36(28): 4503-4515.

[112]  S. Fares, A. Goldstein, F. Loreto. Determinants of ozone fluxes and metrics for ozone risk assessment in plants[J]. Journal of Experimental Botany, 2010, 61(3): 629-633.

[113]  R. Park, S. K. Hong, H A. Kwon, et al. An evaluation of ozone dry deposition simulations in East Asia[J]. Atmospheric Chemistry and Physics, 2014, 14(15): 7929-7940.

[114]  J. Padro. Summary of ozone dry deposition velocity measurements and model estimates over vineyard, cotton, grass and deciduous forest in summer[J]. Atmospheric Environment, 1996, 30(13): 2363-2369.

[115]  Z. Wu, R. Staebler, R. Vet, et al. Dry deposition of $O_3$ and $SO_2$ estimated from gradient measurements above a temperate mixed forest[J]. Environmental Pollution, 2016, 210: 202-210.

[116]  P. L. Finkelstein, T. G. Ellestad, J. F. Clarke, et al. Ozone and sulfur dioxide dry deposition to forests: Observations and model evaluation[J]. Journal of Geophysical Research: Atmospheres, 2000, 105(D12): 15365-15377.

[117] J. W. Munger, S. C. Wofsy, P. S. Bakwin, et al. Atmospheric deposition of reactive nitrogen oxides and ozone in a temperate deciduous forest and a subarctic woodland: 1. Measurements and mechanisms[J]. Journal of Geophysical Research: Atmospheres, 1996, 101(D7): 12639-12657.

[118] T. N. Mikkelsen, H. Ro-Poulsen, M. Hovmand, et al. Five-year measurements of ozone fluxes to a Danish Norway spruce canopy[J]. Atmospheric Environment, 2004, 38(15): 2361-2371.

[119] M. Zapletal, P. Cudlín, P. Chroust, et al. Ozone flux over a Norway spruce forest and correlation with net ecosystem production[J]. Environmental Pollution, 2011, 159(5): 1024-1034.

[120] E. Lamaud, A. Carrara, Y. Brunet, et al. Ozone fluxes above and within a pine forest canopy in dry and wet conditions[J]. Atmospheric Environment, 2002, 36(1): 77-88.

[121] S. Fares, F. Savi, J. Muller, et al. Simultaneous measurements of above and below canopy ozone fluxes help partitioning ozone deposition between its various sinks in a Mediterranean Oak Forest[J]. Agricultural and Forest Meteorology, 2014, 198: 181-191.

[122] A. Finco, M. Coyle, E. Nemitz, et al. Characterization of ozone deposition to a mixed oak-hornbeam forest - flux measurements at five levels above and inside the canopy and their interactions with nitric oxide[J]. Atmospheric Chemistry and Physics, 2018, 18(24): 17945-17961.

[123] P. Stella, E. Personne, B. Loubet, et al. Predicting and partitioning ozone fluxes to maize crops from sowing to harvest: the Surfatm-$O_3$ model[J]. Biogeosciences, 2011, 8(10): 2869-2886.

[124] T. P. Meyers, P. Finkelstein, J. Clarke, et al. A multilayer model for inferring dry deposition using standard meteorological measurements[J]. Journal of Geophysical Research: Atmospheres, 1998, 103(D17): 22645-22661.

[125] 魏莉. 冬小麦和玉米臭氧沉降特征的观测研究对比[D]. 南京: 南京信息工程大学, 2019.

[126] 徐静馨, 郑有飞, 麦博儒, 等. 麦田 $O_3$ 干沉降过程及不同沉降通道分配的模拟[J]. 中国环境科学, 2018, 38(2): 455-470.

[127] 刘俊, 郑有飞, 赵辉. 水稻田臭氧干沉降日变化特征研究[J]. 生态环境学报, 2017(11): 119-125.

[128] 李硕, 郑有飞, 吴荣军, 等. 冬麦田臭氧干沉降过程的观测[J]. 应用生态学报, 2016, 27(6): 1811-1819.

[129] A. Sorimachi, K. Sakamoto, H. Ishihara, et al. Measurements of sulfur dioxide and ozone dry deposition over short vegetation in northern China-a preliminary study[J]. Atmospheric Environment, 2003, 37(22): 3157-3166.

[130] 潘小乐, 王自发, 王喜全, 等. 秋季在北京城郊草地下垫面上的一次臭氧干沉降观测试验[J]. 大气科学, 2010, 34(001): 120-130.

[131] P. Stella, B. Loubet, C. de Berranger, et al. Soil ozone deposition: Dependence of soil resistance to soil texture[J]. Atmospheric Environment, 2019, 199: 202-209.

[132] 赵雄飞, 王体健, 黄满堂, 等. 大气污染物干沉降速度和通量的计算方法比较-以南京仙林地区为例[J]. 装备环境工程, 2019, v.16(06): 139-147.

[133] D. Szinyei. Modelling and evaluation of ozone dry deposition[D]. PhD thesis. Berlin: Freie Universität Berlin, 2015.

[134] J Y. Juang, G. G. Katul, M. B. Siqueira, et al. Investigating a hierarchy of Eulerian closure models for scalar transfer inside forested canopies[J]. Boundary-Layer Meteorology, 2008, 128(1): 1-32.

[135] S. Launiainen, G. Katul, T. Grönholm, et al. Partitioning ozone fluxes between canopy and forest floor by measurements and a multi-layer model[J]. Agricultural and Forest Meteorology, 2013, 173: 85-99.

[136] L. N. Ganzeveld, J. Lelieveld, F. J. Dentener, et al. Atmosphere-biosphere trace gas exchanges simulated with a single-column model[J]. Journal of Geophysical Research: Atmospheres, 2002, 107(D16): ACH 8-1-ACH 8-21.

[137] J. H. Duyzer, J. R. Dorsey, M. W. Gallagher, et al. Oxidized nitrogen and ozone interaction with forests. II: Multi-layer process-oriented modelling results and a sensitivity study for Douglas fir[J]. Quarterly Journal of the Royal Meteorological Society, 2004, 130(600): 1957-1971.

[138] G. M. Wolfe, J. A. Thornton, M. McKay, et al. Forest-atmosphere exchange of ozone: sensitivity to very reactive biogenic VOC emissions and implications for in-canopy photochemistry[J]. Atmospheric Chemistry and Physics, 2011, 11(15): 7875-7891.

[139] E. Potier, J. Ogée, J. Jouanguy, et al. Multilayer modelling of ozone fluxes on winter wheat reveals large deposition on wet senescing leaves[J]. Agricultural and Forest Meteorology,

2015, 211-212: 58-71.

[140]  P. Zhou, L. Ganzeveld, Ü. Rannik, et al. Simulating ozone dry deposition at a boreal forest with a multi-layer canopy deposition model[J]. Atmospheric Chemistry and Physics, 2017, 17(2): 1361-1379.

[141]  S. Fares, A. Alivernini, A. Conte, et al. Ozone and particle fluxes in a Mediterranean forest predicted by the AIRTREE model[J]. Science of The Total Environment, 2019, 682: 494-504.

[142]  K. Ashworth, S. H. Chung, R. J. Griffin, et al. FORest Canopy Atmosphere Transfer (FOR-CAsT) 1.0: a 1-D model of biosphere‐atmosphere chemical exchange[J]. Geoscientific Model Development, 2015, 8(11): 3765-3784.

[143]  E. G. Patton, P. P. Sullivan, R. H. Shaw, et al. Atmospheric Stability Influences on Coupled Boundary Layer and Canopy Turbulence[J]. Journal of the Atmospheric Sciences, 2016, 73(4): 1621-1647.

[144]  K Y. Chang, K. T. Paw U, S H. Chen. Canopy profile sensitivity on surface layer simulations evaluated by a multiple canopy layer higher order closure land surface model[J]. Agricultural and Forest Meteorology, 2018, 252: 192-207.

[145]  L. Zhang, J. R. Brook, R. Vet. On ozone dry deposition-with emphasis on non-stomatal uptake and wet canopies[J]. Atmospheric Environment, 2002, 36(30): 4787-4799.

[146]  S. Bassin, P. Calanca, T. Weidinger, et al. Modeling seasonal ozone fluxes to grassland and wheat: model improvement, testing, and application[J]. Atmospheric Environment, 2004, 38(15): 2349-2359.

[147]  A. Tuzet, A. Perrier, B. Loubet, et al. Modelling ozone deposition fluxes: The relative roles of deposition and detoxification processes[J]. Agricultural and Forest Meteorology, 2011, 151(4): 480-492.

[148]  W. Massman. A review of the molecular diffusivities of $H_2O$, $CO_2$, $CH_4$, CO, $O_3$, $SO_2$, $NH_3$, $N_2O$, NO, and $NO_2$ in air, $O_2$ and $N_2$ near STP[J]. Atmospheric Environment, 1998, 32(6): 1111-1127.

[149]  J. Uddling, R. Matyssek, J. B. Pettersson, et al. To what extent do molecular collisions arising from water vapour efflux impede stomatal $O_3$ influx?[J]. Environmental Pollution, 2012, 170: 39-42.

[150] I. Hassan, M. Ashmore, J. Bell. Effects of O₃ on the stomatal behaviour of Egyptian varieties of radish (Raphanus sativus L. cv. Baladey) and turnip (Brassica rapa L. cv. Sultani) [J]. New Phytologist, 1994, 128(2): 243-249.

[151] F. Manes, E. Donato, M. Vitale. Physiological response of Pinus halepensis needles under ozone and water stress conditions[J]. Physiologia Plantarum, 2001, 113(2): 249-257.

[152] G. Mills, F. Hayes, S. Wilkinson, et al. Chronic exposure to increasing background ozone impairs stomatal functioning in grassland species[J]. Global Change Biology, 2009, 15(6): 1522-1533.

[153] D. Lombardozzi, J. P. Sparks, G. Bonan. Integrating O₃ influences on terrestrial processes: photosynthetic and stomatal response data available for regional and global modeling[J]. Biogeosciences, 2013, 10(11): 6815-6831.

[154] V. Calatayud, J. Cerveró, M. J. Sanz. Foliar, physiologial and growth responses of four maple species exposed to ozone[J]. Water, Air, and Soil Pollution, 2007, 185(1): 239-254.

[155] K. Herbinger, C. Then, K. Haberer, et al. Gas exchange and antioxidative compounds in young beech trees under free-air ozone exposure and comparisons to adult trees[J]. Plant Biology, 2007, 9(02): 288-297.

[156] E. Paoletti, N. E. Grulke. Does living in elevated CO₂ ameliorate tree response to ozone? A review on stomatal responses[J]. Environmental Pollution, 2005, 137(3): 483-493.

[157] G. Mills, A. Buse, B. Gimeno, et al. A synthesis of AOT40-based response functions and critical levels of ozone for agricultural and horticultural crops[J]. Atmospheric Environment, 2007, 41(12): 2630-2643.

[158] G. Torsethaugen, E. J. Pell, S. M. Assmann. Ozone inhibits guard cell K+ channels implicated in stomatal opening[J]. Proceedings of the National Academy of Sciences, 1999, 96(23): 13577-13582.

[159] E. Paoletti, N. E. Grulke. Ozone exposure and stomatal sluggishness in different plant physiognomic classes[J]. Environmental Pollution, 2010, 158(8): 2664-2671.

[160] S. B. McLaughlin, S. D. Wullschleger, G. Sun, et al. Interactive effects of ozone and climate on water use, soil moisture content and streamflow in a southern Appalachian forest in the USA[J]. New Phytologist, 2007, 174(1): 125-136.

[161] R. Rai, M. Agrawal. Impact of tropospheric ozone on crop plants[J]. Proceedings of the

National Academy of Sciences, India Section B: Biological Sciences, 2012, 82(2): 241-257.

[162] W. Ren, H. Tian, M. Liu, et al. Effects of tropospheric ozone pollution on net primary productivity and carbon storage in terrestrial ecosystems of China[J]. Journal of Geophysical Research: Atmospheres, 2007, 112(D22).

[163] D. Lombardozzi, J. P. Sparks, G. Bonan, et al. Ozone exposure causes a decoupling of conductance and photosynthesis: implications for the Ball-Berry stomatal conductance model [J]. Oecologia, 2012, 169(3): 651-659.

[164] R. Matyssek, G. Wieser, A. Nunn, et al. Comparison between AOT40 and ozone uptake in forest trees of different species, age and site conditions[J]. Atmospheric Environment, 2004, 38(15): 2271-2281.

[165] L. Ganzeveld, C. Ammann, B. Loubet. Modelling Atmosphere-Biosphere Exchange of Ozone and Nitrogen Oxides[M]. in: Review and Integration of Biosphere-Atmosphere Modelling of Reactive Trace Gases and Volatile Aerosols. Dordrecht: Springer Netherlands, 2015: 85-105.

[166] J. Xu, Y. Zheng, B. Mai, et al. Characteristics and partitioning of ozone dry deposition measured by eddy-covariance technology in a winter wheat field[J]. Chinese Journal of Plant Ecology, 2017, 41(6): 670.

[167] N. Altimir, P. Kolari, J P. Tuovinen, et al. Foliage surface ozone deposition: a role for surface moisture?[J]. Biogeosciences, 2006, 3(2): 209-228.

[168] M. Coyle, G. Mills, D. Fowler, et al. Ozone Umbrella: Effects of ground-level ozone on (upland) vegetation in the UK. Final report[M]. 2006.

[169] L. D. Yee, G. Isaacman-VanWertz, R. A. Wernis, et al. Observations of sesquiterpenes and their oxidation products in central Amazonia during the wet and dry seasons[J]. Atmospheric Chemistry and Physics, 2018, 18(14): 10433-10457.

[170] M. R. Kurpius, A. H. Goldstein. Gas-phase chemistry dominates $O_3$ loss to a forest, implying a source of aerosols and hydroxyl radicals to the atmosphere[J]. Geophysical Research Letters, 2003, 30(7).

[171] D. Helmig, J. Ortega, A. Guenther, et al. Sesquiterpene emissions from loblolly pine and their potential contribution to biogenic aerosol formation in the Southeastern US[J]. Atmospheric Environment, 2006, 40(22): 4150-4157.

[172] K. Jardine, A. Yañez Serrano, A. Arneth, et al. Within-canopy sesquiterpene ozonolysis in Amazonia[J]. Journal of Geophysical Research: Atmospheres, 2011, 116(D19).

[173] A. Goldstein, M. McKay, M. Kurpius, et al. Forest thinning experiment confirms ozone deposition to forest canopy is dominated by reaction with biogenic VOCs[J]. Geophysical Research Letters, 2004, 31(22).

[174] N. Bouvier-Brown, A. H. Goldstein, J. Gilman, et al. In-situ ambient quantification of monoterpenes, sesquiterpenes, and related oxygenated compounds during BEARPEX 2007: implications for gas-and particle-phase chemistry[J]. Atmospheric Chemistry and Physics, 2009, 9(15): 5505-5518.

[175] E. Bourtsoukidis, T. Behrendt, A. M. Yáñez-Serrano, et al. Strong sesquiterpene emissions from Amazonian soils[J]. Nature Communications, 2018, 9(1): 1-11.

[176] S. Schobesberger, F. D. Lopez-Hilfiker, D. Taipale, et al. High upward fluxes of formic acid from a boreal forest canopy[J]. Geophysical Research Letters, 2016, 43(17): 9342-9351.

[177] H. D. Alwe, D. B. Millet, X. Chen, et al. Oxidation of volatile organic compounds as the major source of formic acid in a mixed forest canopy[J]. Geophysical Research Letters, 2019, 46(5): 2940-2948.

[178] D. Farmer, J. Kimmel, G. Phillips, et al. Eddy covariance measurements with high-resolution time-of-flight aerosol mass spectrometry: a new approach to chemically resolved aerosol fluxes[J]. Atmospheric Measurement Techniques, 2011, 4(6): 1275-1289.

[179] G. Wolfe, J. Thornton. The chemistry of atmosphere-forest exchange model-Part 1: Model description and characterization[J]. Atmospheric Chemistry and Physics, 2011, 11(1): 77-101.

[180] 龚道程. 南岭高山森林大气挥发性有机物的源汇机制研究[D]. 广州: 暨南大学, 2019.

[181] G. Wolfe, J. Thornton, R. Yatavelli, et al. Eddy covariance fluxes of acyl peroxy nitrates (PAN, PPN and MPAN) above a Ponderosa pine forest[J]. Atmospheric Chemistry and Physics, 2009, 9(2): 615-634.

[182] A. Sorimachi, K. Sakamoto. Laboratory measurement of the dry deposition of sulfur dioxide onto northern Chinese soil samples[J]. Atmospheric Environment, 2007, 41(13): 2862-2869.

[183] H. Güsten, G. Heinrich, E. Mönnich, et al. On-line measurements of ozone surface fluxes: Part II. Surface-level ozone fluxes onto the Sahara desert[J]. Atmospheric Environment,

1996, 30(6): 911-918.

[184]   T. P. Meyers, D. D. Baldocchi. Trace gas exchange above the floor of a deciduous forest: 2. SO$_2$ and O$_3$ deposition[J]. Journal of Geophysical Research: Atmospheres, 1993, 98(D7): 12631-12638.

[185]   W. Van Pul, A. Jacobs. The conductance of a maize crop and the underlying soil to ozone under various environmental conditions[J]. Boundary-Layer Meteorology, 1994, 69(1): 83-99.

[186]   P. Stella, B. Loubet, E. Lamaud, et al. Ozone deposition onto bare soil: A new parameterisation[J]. Agricultural and Forest Meteorology, 2011, 151(6): 669-681.

[187]   J. Neirynck, B. Gielen, I. Janssens, et al. Insights into ozone deposition patterns from decade-long ozone flux measurements over a mixed temperate forest[J]. Journal of Environmental Monitoring, 2012, 14(6): 1684-1695.

[188]   Z. Liu, Y. Pan, T. Song, et al. Eddy covariance measurements of ozone flux above and below a southern subtropical forest canopy[J]. Science of The Total Environment, 2021, 791: 148338.

[189]   K. Toyota, A. P. Dastoor, A. Ryzhkov. Parameterization of gaseous dry deposition in atmospheric chemistry models: Sensitivity to aerodynamic resistance formulations under statically stable conditions[J]. Atmospheric Environment, 2016, 147: 409-422.

[190]   C. Q. Dias-Júnior, N. L. Dias, R. M. N. dos Santos, et al. Is there a classical inertial sublayer over the Amazon forest?[J]. Geophysical Research Letters, 2019, 46(10): 5614-5622.

[191]   A. Bryan, S. Bertman, M. Carroll, et al. In-canopy gas-phase chemistry during CABINEX 2009: sensitivity of a 1-D canopy model to vertical mixing and isoprene chemistry[J]. Atmospheric Chemistry and Physics, 2012, 12(18): 8829-8849.

[192]   L. Emberson, D. Simpson, J. Tuovinen, et al. Towards a model of ozone deposition and stomatal uptake over Europe[M]. MSC-W, 2000.

[193]   F. Paulot, S. Malyshev, T. Nguyen, et al. Representing sub-grid scale variations in nitrogen deposition associated with land use in a global Earth system model: implications for present and future nitrogen deposition fluxes over North America[J]. Atmospheric Chemistry and Physics, 2018, 18(24): 17963-17978.

[194]   D. Simpson, J P. Tuovinen, L. Emberson, et al. Characteristics of an ozone deposition module[J]. Water, Air and Soil Pollution: Focus, 2001, 1(5): 253-262.

[195]  D. S. Niyogi, S. Raman, K. Alapaty. Comparison of four different stomatal resistance schemes using FIFE data. Part II: Analysis of terrestrial biospheric–atmospheric interactions[J]. Journal of Applied Meteorology, 1998, 37(10): 1301-1320.

[196]  L. Ganzeveld, J. Lelieveld. Dry deposition parameterization in a chemistry general circulation model and its influence on the distribution of reactive trace gases[J]. Journal of Geophysical Research: Atmospheres, 1995, 100(D10): 20999-21012.

[197]  Y. Wang, D. J. Jacob, J. A. Logan. Global simulation of tropospheric $O_3$-NOx-hydrocarbon chemistry: 1. Model formulation[J]. Journal of Geophysical Research: Atmospheres, 1998, 103(D9): 10713-10725.

[198]  D. J. Jacob, S. C. Wofsy. Budgets of reactive nitrogen, hydrocarbons, and ozone over the Amazon forest during the wet season[J]. Journal of Geophysical Research: Atmospheres, 1990, 95(D10): 16737-16754.

[199]  D. Schwede, L. Zhang, R. Vet, et al. An intercomparison of the deposition models used in the CASTNET and CAPMoN networks[J]. Atmospheric Environment, 2011, 45(6): 1337-1346.

[200]  P. Sellers, D. Randall, G. Collatz, et al. A revised land surface parameterization (SiB2) for atmospheric GCMs. Part I: Model formulation[J]. Journal of Climate, 1996, 9(4): 676-705.

[201]  G. Bonan. Climate Change and Terrestrial Ecosystem Modeling[M]. Cambridge University Press, 2019.

[202]  范嘉智, 王丹, 胡亚林, 等. 最优气孔行为理论和气孔导度模拟[J]. 植物生态学报, 2016, 40(6): 631.

[203]  李永秀, 娄运生, 张富存. 冬小麦气孔导度模型的比较[J]. 中国农业气象, 2011, 32(01): 106.

[204]  牛海山, 旭日, 张志诚, 等. 羊草气孔导度的 Jarvis-类模型[J]. 生态学杂志, 2005, 24(11): 1287-1290.

[205]  P. Büker, Z. Feng, J. Uddling, et al. New flux based dose–response relationships for ozone for European forest tree species[J]. Environmental Pollution, 2015, 206: 163-174.

[206]  L. Emberson, M. Ashmore, D. Simpson, et al. Modelling and mapping ozone deposition in Europe[J]. Water, Air, and Soil Pollution, 2001, 130(1): 577-582.

[207]  L. D. Emberson, P. Büker, M. R. Ashmore. Assessing the risk caused by ground level ozone

to European forest trees: A case study in pine, beech and oak across different climate regions [J]. Environmental Pollution, 2007, 147(3): 454-466.

[208] S. Elvira, V. Bermejo, E. Manrique, et al. On the response of two populations of Quercus coccifera to ozone and its relationship with ozone uptake[J]. Atmospheric Environment, 2004, 38(15): 2305-2311.

[209] G. L. Miner, W. L. Bauerle, D. D. Baldocchi. Estimating the sensitivity of stomatal conductance to photosynthesis: a review[J]. Plant, Cell & Environment, 2017, 40(7): 1214-1238.

[210] Ball, JT, Woodrow, et al. A Model Predicting Stomatal Conductance and its Contribution to the Control of Photosynthesis under Different Environmental Conditions[M]. Springer Netherlands, 1987.

[211] R. Leuning. A critical appraisal of a combined stomatal-photosynthesis model for C3 plants [J]. Plant, Cell & Environment, 1995, 18(4): 339-355.

[212] 叶听听. 我国对流层臭氧污染对农作物生产力的影响研究[D]. 南京: 南京大学, 2017.

[213] P. J. Franks, G. B. Bonan, J. A. Berry, et al. Comparing optimal and empirical stomatal conductance models for application in Earth system models[J]. Global Change Biology, 2018, 24(12): 5708-5723.

[214] B. E. Medlyn, R. A. Duursma, D. Eamus, et al. Reconciling the optimal and empirical approaches to modelling stomatal conductance[J]. Global Change Biology, 2011, 17(6): 2134-2144.

[215] Y. Lei, X. Yue, H. Liao, et al. Implementation of Yale Interactive terrestrial Biosphere model v1. 0 into GEOS-Chem v12. 0.0: a tool for biosphere–chemistry interactions[J]. Geoscientific Model Development, 2020, 13(3): 1137-1153.

[216] L. Misson, J. A. Panek, A. H. Goldstein. A comparison of three approaches to modeling leaf gas exchange in annually drought-stressed ponderosa pine forests[J]. Tree Physiology, 2004, 24(5): 529-541.

[remaining 24,189 characters of this post omitted]

---

## Editor Decision (ED2)

Reviewer: The authors stated that they used Lombardozzi et al. (2013)'s parameterizations for their study (L209). I am confused from where in Lombardozzi et al. (2013) the authors obtained their ap, ac, bp, and bc for the 6 vegetation types in their Table 2. In their results from "the exposed to charcoal-filtered air with medium or high confidence in cumulative O3 uptake (CUO) calculations", Lomdardozzi et al. (2013) showed no significance in the linearly regressed equations of photosynthesis in % of control vs. CUO for all plant types except crops and showed no significance in the linearly regressed equations of conductance in % control vs. CUS for all plant types except temperate evergreen trees (L2013's Tables 2&3). In their results from "ambient air" data, Lomdardozzi et al. (2013) showed no significance in the linearly regressed equations of photosynthesis in % of control vs. CUO and conductance in % control vs. CUO for all plant types except "temperature deciduous trees" (L2013's Tables B1&B2).

The values the authors used that I recognized, albeit not the ones intended for their purposes in this reviewer's opinion, were 2 orders of magnitude smaller than those in Lombardozzi et al. (2013). This reviewer was taken by surprise by the authors' statement that most of their plant types had "time-independent" sensitivity to CUO since ac and ap values were zero. First, I did not see zero values for ac and ap in Lombardozzi et al. (2013); instead, L2013 showed no significance in regression for most plants as stated above. Second, if what the authors stated were true, it'd totally defeat the purpose of that epic study of Lombardozzi et al. (2013)'s. In short, it was very confusing how and where the authors got the values in their Table 2 from.

*Response:* We are sorry for the confusion. The parameters we employed for L2013 scheme in our paper were originally adopted from Lombardozzi et al. (2013), which were provided in the unit of percentage but converted to the fraction in our study. As a result, the values in our paper are 2 orders of magnitude smaller than those in Lombardozzi et al. (2013). The specific values of ac and ap were set to zero in the Table 1 of Lombardozzi et al. (2015) based on the conclusions of Lombardozzi et al. (2013), as we presented below.

To clarify the source of parameter settings, we added a footnote to Table 2: "a The data source is Lombardozzi et al. (2015). Due to the data limit, we apply the same sensitivity parameters for EBF, DBF, and SHR."

**This reviewer found the difference between Lombardozzi et al. (2013) and Lombardozzi et al. (2015) to be confusing, since the latter simply presented a table of coefficients "based on Lombardozzi et al. (2013)" without pointing out and reconciling the differences. The present study's authors may believe that it should not be their responsibility to reconcile such differences, and they simply applied Lombardozzi et al. (2015)'s values. The fact that previous studies applied those values without questioning does not justify the inconsistencies. Notwithstanding, I believe that the authors' making it perfectly clear that those ac, ap, bc, bp values were actually from Lombardozzi et al. (2015) could probably help draw the community's attention to such confusing discrepancies. Therefore, I appreciate the authors' addition of such information.**

Further, Lombardozzi et al. (2013) emphasized "chronic ozone exposure" throughout their work, and thus they included the studies that used experimental periods longer than 7 days. That means that the parameterizations derived from L2013 would be only applicable for calculations over periods > 7 days. Hence, the question is: how could the authors' calculations for times shorter than that be valid?

*Response:* As mentioned by the reviewer, L2013 would only be applicable for calculations over periods > 7 days. In this study, we conducted four consecutive months of simulations with the first month excluded from the analysis as the spin-up. Hence, all of our simulations were longer than periods > 7 days and valid for the further analyses.

**The authors did not understand my comment. In their response to my 1st round of review, they stated, "The leaf-level CUO (mmol m$^{-2}$) is calculated by accumulating stomatal O$_3$ fluxes of Equation 4 from the start of the growing season to the specific time step". That means that the authors integrated Eq. 4 from the very first timestep, which I assume would be about 160 seconds, up to each ensuing timestep. Logically, all the simulations before the 8$^{th}$ simulation day should not be using Eqs. 5 and 6 to calculate O$_3$ damage ratios, simply because the duration was too short for the equations to be applicable. This logically led to the fact that the integrated stomatal ozone flux amounts > 7 days were in fact built upon erroneous initial values. That is why I've been skeptical of the applicability of L2013 in their modeling coupling exercise from the very beginning.**

Since S2007 calculated instantaneous effects while L2013 the effect of CUO, it is critical to know what exactly was presented in Figures 2 and 3. The author just stated "O3 damage", but they had 3 months simulations. The two figures must be showing post processed values. So, what exactly was shown in those figures? This question points to the comparability of those two figures and consequently their main findings.

*Response:* Figures 2 and 3 showed the three-month averages of O$_3$ vegetation damage. In the revised paper, we added month-to-month variations of O$_3$ vegetation damage in Figure S1 and S2 to clarify. For L2013 scheme, the O$_3$ damage to photosynthesis of sunlit and shaded leaves increases month by month with the increase of CUO, reaching a maximum in August. In contrast, For S2007 scheme, the O$_3$ damage peaks in July due to the highest O$_3$ concentrations. We modified the sentence as follows: "The S2007 scheme is dependent on instantaneous O$_3$ uptake, which peaks in July when both O$_3$ concentrations and stomatal conductance are high (Figures S1 and S2)."(Lines 300-302). For L2013: "The O$_3$ damage to photosynthesis of sunlit and shaded leaves increases month by month, reaching the maximum in August (Figures S1 and S2)." (Lines 307-309).

**Comparing the three-month averages of O$_3$ damage using S2007 and L2013 does not make sense to me. S2007 calculates instantaneous values, while L2013 simulates incremental ozone damage. The three-month average of S2007-calculated ozone damage shows the average ozone damage resulting from the amount of ozone exposure within that hour. The three-month average of L2013-calculated ozone damage shows the ozone damage due to ozone exposure averaged from over time periods from one week to three months. In this reviewer's opinion, they're comparing two completely different parameters!**

---

## Author Response (AR3)

We thank very much for the helpful comments and suggestions from the reviewer, which help us improve our manuscript. The comments were carefully considered and revisions have been made in response to suggestions. This round of author responses to review comments are shown in blue text.

Reviewer: The authors stated that they used Lombardozzi et al. (2013)'s parameterizations for their study (L209). I am confused from where in Lombardozzi et al. (2013) the authors obtained their ap, ac, bp, and bc for the 6 vegetation types in their Table 2. In their results from "the exposed to charcoal-filtered air with medium or high confidence in cumulative O3 uptake (CUO) calculations", Lomdardozzi et al. (2013) showed no significance in the linearly regressed equations of photosynthesis in % of control vs. CUO for all plant types except crops and showed no significance in the linearly regressed equations of conductance in % control vs. CUS for all plant types except temperate evergreen trees (L2013's Tables 2&3). In their results from "ambient air" data, Lomdardozzi et al. (2013) showed no significance in the linearly regressed equations of photosynthesis in % of control vs. CUO and conductance in % control vs. CUO for all plant types except "temperature deciduous trees" (L2013's Tables B1&B2).

The values the authors used that I recognized, albeit not the ones intended for their purposes in this reviewer's opinion, were 2 orders of magnitude smaller than those in Lombardozzi et al. (2013). This reviewer was taken by surprise by the authors' statement that most of their plant types had "time-independent" sensitivity to CUO since ac and ap values were zero. First, I did not see zero values for ac and ap in Lombardozzi et al. (2013); instead, L2013 showed no significance in regression for most plants as stated above. Second, if what the authors stated were true, it'd totally defeat the purpose of that epic study of Lombardozzi et al. (2013)'s. In short, it was very confusing how and where the authors got the values in their Table 2 from.

*Response:* We are sorry for the confusion. The parameters we employed for L2013 scheme in our paper were originally adopted from Lombardozzi et al. (2013), which were provided in the unit of percentage but converted to the fraction in our study. As a result, the values in our paper are 2 orders of magnitude smaller than those in Lombardozzi et al. (2013). The specific values of ac and ap were set to zero in the Table 1 of Lombardozzi et al. (2015) based on the conclusions of Lombardozzi et al. (2013), as we presented below.

To clarify the source of parameter settings, we added a footnote to Table 2: "[a] The data source is Lombardozzi et al. (2015). Due to the data limit, we apply the same sensitivity parameters for EBF, DBF, and SHR."

TABLE 1. Values used to parameterize plant functional types in CLM. Slopes (per $mmol\,m^{-2}$) and intercepts (unitless) are based on values presented in Lombardozzi et al. (2013).

| Plant group | Photosynthesis | | Conductance | |
|---|---|---|---|---|
| | Slope ($a_p$) | Intercept ($b_p$) | Slope ($a_c$) | Intercept ($b_c$) |
| O₃ response | | | | |
| Broadleaf | 0 | 0.8752 | 0 | 0.9125 |
| Needleleaf | 0 | 0.839 | 0.0048 | 0.7823 |
| Crop and grass | −0.0009 | 0.8021 | 0 | 0.7511 |
| | | Sensitivity simulations | | |
| High vulnerability | | | | |
| Broadleaf | 0 | 0.8502 | 0 | 0.89 |
| Needleleaf | −0.038 | 1.083 | −0.0144 | 0.8874 |
| Crop and grass | −0.0007 | 0.8564 | 0 | 0.7074 |
| Low vulnerability | | | | |
| Broadleaf | 0 | 0.9798 | 0 | 0.9425 |
| Needleleaf | 0 | 0.8595 | 0.0067 | 0.7574 |
| Crop and grass | 0 | 0.7159 | 0.0229 | 0.4621 |
| Fixed decrease | | | | |
| Broadleaf | 0 | 0.8752 | 0 | 0.9125 |
| Needleleaf | 0 | 0.839 | 0 | 0.8645 |
| Crop and grass | 0 | 0.7722 | 0 | 0.7511 |
| Single plant type | | | | |
| All plant types | −0.00098 | 0.8434 | 0 | 0.8444 |

Table R1 The source of slopes and intercepts (Lombardozzi et al. 2015)

**Table 2.** Slopes and intercepts used for L2013 O₃ damage scheme [a].

| PFTs | $a_p$ (mmol m⁻²) | $b_p$ | $a_c$ (mmol m⁻²) | $b_c$ |
|---|---|---|---|---|
| EBF | 0 | 0.8752 | 0 | 0.9125 |
| NF | 0 | 0.839 | 0.0048 | 0.7823 |
| DBF | 0 | 0.8752 | 0 | 0.9125 |
| SHR | 0 | 0.8752 | 0 | 0.9125 |
| GRA | -0.0009 | 0.8021 | 0 | 0.7511 |
| CRO | -0.0009 | 0.8021 | 0 | 0.7511 |

[a] The data source is Lombardozzi et al. (2015). Due to the data limit, we apply the same sensitivity parameters for EBF, DBF, and SHR.

We have the same concern as the reviewer that the L2013 scheme may not reasonably reflect the vegetation responses to CUO. However all the previous researches applied L2013 scheme to explore the climatic feedback of O₃-vegetation interactions (e.g., Sadiq et al. 2017; Zhu et al. 2022; Jin et al. 2023). In this study, we used both L2013 and S2007 schemes to assess and compare the climatic feedback due to O₃ vegetation damage so as to understand the uncertainties due to the differences in schemes. We also discussed the possible limitations of the L2013 scheme in the text: "The L2013 scheme considered the decoupling between photosynthesis and stomatal conductance. However, we found this scheme showed no significant different changes for sunlit and shaded

leaves. In addition, the calculation of CUO heavily relied on the $O_3$ threshold and accumulation period, leading to varied responses among different studies using the same scheme. Furthermore, the slopes of $O_3$ sensitivity in L2013 scheme were set to zero for some PFTs, leading to constant damages independent of CUO." (Lines 466-472)

Further, Lombardozzi et al. (2013) emphasized "chronic ozone exposure" throughout their work, and thus they included the studies that used experimental periods longer than 7 days. That means that the parameterizations derived from L2013 would be only applicable for calculations over periods > 7 days. Hence, the question is: how could the authors' calculations for times shorter than that be valid?

*Response:* As mentioned by the reviewer, L2013 would only be applicable for calculations over periods > 7 days. In this study, we conducted four consecutive months of simulations with the first month excluded from the analysis as the spin-up. Hence, all of our simulations were longer than periods > 7 days and valid for the further analyses.

Since S2007 calculated instantaneous effects while L2013 the effect of CUO, it is critical to know what exactly was presented in Figures 2 and 3. The author just stated "O3 damage", but they had 3 months simulations. The two figures must be showing post processed values. So, what exactly was shown in those figures? This question points to the comparability of those two figures and consequently their main findings.

*Response:* Figures 2 and 3 showed the three-month averages of $O_3$ vegetation damage. In the revised paper, we added month-to-month variations of $O_3$ vegetation damage in Figure S1 and S2 to clarify. For L2013 scheme, the $O_3$ damage to photosynthesis of sunlit and shaded leaves increases month by month with the increase of CUO, reaching a maximum in August. In contrast, For S2007 scheme, the $O_3$ damage peaks in July due to the highest $O_3$ concentrations. We modified the sentence as follows: "The S2007 scheme is dependent on instantaneous $O_3$ uptake, which peaks in July when both $O_3$ concentrations and stomatal conductance are high (Figures S1 and S2)."(Lines 300-302). For L2013: "The $O_3$ damage to photosynthesis of sunlit and shaded leaves increases month by month, reaching the maximum in August (Figures S1 and S2)." (Lines 307-309).

[Figure]

**Figure S1** Offline O$_3$ damage (%) to the summertime photosynthesis of sunlit leaves in (a-c) June, (d-f) July, and (g-i) August for different O$_3$ damage schemes and sensitivities. The area-weighted percentage changes are shown in the lower left corner.

[Figure]

**Figure S2** The same as Figure S1 but for the changes in photosynthesis of shaded leaves.

**Reference:**

Lombardozzi, D., Levis, S., Bonan, G., Hess, P. G., and Sparks, J. P.: The influence of chronic ozone exposure on global carbon and water cycles, J. Climate, 28, 292–305, https://doi.org/10.1175/JCLI-D-14-00223.1, 2015.

Jin, Z., Yan, D., Zhang, Z., Li, M., Wang, T., Huang, X., et al. (2023). Effects of elevated ozone exposure on regional meteorology and air quality in China through ozone-vegetation coupling. Journal of Geophysical Research: Atmospheres, 128, e2022JD038119. https://doi.org/10.1029/2022JD038119

Sadiq, M., Tai, A. P. K., Lombardozzi, D., and Val Martin, M.: Effects of ozone–vegetation coupling on surface ozone air quality via biogeochemical and meteorological feedbacks, Atmos. Chem. Phys., 17, 3055–3066, https://doi.org/10.5194/acp-17-3055-2017, 2017.

Zhu, J., Tai, A. P. K., and Yim, S. H. L.: Effects of ozone-vegetation interactions on meteorology and air quality in China using a two-way coupled land-atmosphere model, Atmos. Chem. Phys., 22, 765-782, https://doi.org/10.5194/acp-22-765-2022, 2022.

---

## Author Response (AR4)

We thank very much for the helpful comments and suggestions from the reviewer, which help us improve our manuscript. This round of author responses to review comments are shown in red text.

This reviewer found the difference between Lombardozzi et al. (2013) and Lombardozzi et al. (2015) to be confusing, since the latter simply presented a table of coefficients "based on Lombardozzi et al. (2013)" without pointing out and reconciling the differences. The present study's authors may believe that it should not be their responsibility to reconcile such differences, and they simply applied Lombardozzi et al. (2015)'s values. The fact that previous studies applied those values without questioning does not justify the inconsistencies. Notwithstanding, I believe that the authors' making it perfectly clear that those ac, ap, bc, bp values were actually from Lombardozzi et al. (2015) could probably help draw the community's attention to such confusing discrepancies. Therefore, I appreciate the authors' addition of such information.

*Response*: Thank you for your understanding. We have made it clear in the text that the coefficients we used for L2013 scheme were adopted from Lombardozzi et al. (2015).

The authors did not understand my comment. In their response to my 1st round of review, they stated, "The leaf-level CUO (mmol m-2) is calculated by accumulating stomatal O3 fluxes of Equation 4 from the start of the growing season to the specific time step". That means that the authors integrated Eq. 4 from the very first timestep, which I assume would be about 160 seconds, up to each ensuing timestep. Logically, all the simulations before the 8th simulation day should not be using Eqs. 5 and 6 to calculate O3 damage ratios, simply because the duration was too short for the equations to be applicable. This logically led to the fact that the integrated stomatal ozone flux amounts > 7 days were in fact built upon erroneous initial values. That is why I've been skeptical of the applicability of L2013 in their modeling coupling exercise from the very beginning.

*Response*: The reviewer may have some misunderstandings of the calculation of CUO. The criterion of "> 7 days" was used only in the selection of valid observational samples, because it allowed certain period of $O_3$ fumigation so that the symptoms of damages were more significant in the statistics. As for the calculation of CUO, it should be accumulated at the very beginning of the growing season. For example, the words from Lombardozzi et al. (2015) stated that: "CUO only accumulates during the growing season". We checked through L2015 paper and did not find any restrictions of CUO calculation for the first 7 days. In our simulation, we accumulated CUO form May 1st but removed the results of the first month, assuming that the $O_3$ fumigation was not long enough to derive the damages. We applied Equations (5) and (6) for the CUO of June-August, consistent with the way how these equations were derived.

Comparing the three-month averages of O3 damage using S2007 and L2013 does not make sense to me. S2007 calculates instantaneous values, while L2013 simulates

incremental ozone damage. The three-month average of S2007-calculated ozone damage shows the average ozone damage resulting from the amount of ozone exposure within that hour. The three-month average of L2013-calculated ozone damage shows the ozone damage due to ozone exposure averaged from over time periods from one week to three months. In this reviewer's opinion, they're comparing two completely different parameters!

*Response*: We agree that L2013 and S2007 schemes were based on different mechanisms. The L2013 depends on the accumulated $O_3$ fluxes while S2007 relied on the instantaneous $O_3$ fluxes. It was not our focus (and of course out of our capability) to determine which scheme was correct, because both of them had been derived or validated against many observations and had been widely used in previous modeling researches. The novelty of this study was to show how different these two schemes, and the consequent climatic feedbacks due to these discrepancies in $O_3$ damage schemes. From this perspective, we have to retain the original equations and procedures of calculations from L2013/L2015 and S2007 without artificial alterations.

**Reference:**

Lombardozzi, D., Sparks, J. P., and Bonan, G.: Integrating O3 influences on terrestrial processes: photosynthetic and stomatal response data available for regional and global modeling, Biogeosciences, 10, 6815–6831, https://doi:10.5194/bg-10-6815-2013, 2013.

Lombardozzi, D., Levis, S., Bonan, G., Hess, P. G., and Sparks, J. P.: The influence of chronic ozone exposure on global carbon and water cycles, J. Climate, 28, 292–305, https://doi.org/10.1175/JCLI-D-14-00223.1, 2015.

Sitch, S., Cox, P. M., Collins, W. J., and Huntingford, C.: Indirect radiative forcing of climate change through ozone effects on the land-carbon sink, Nature, 448, 791–794, https://doi.org/10.1038/nature06059, 2007.